# LEARNING TO RECALL WITH TRANSFORMERS BEYOND ORTHOGONAL EMBEDDINGS

**Nuri Mert Vural**[1,2,*] **Alberto Bietti**[3] **Mahdi Soltanolkotabi**[5] **Denny Wu**[3,4]

[1]University of Toronto, [2]Vector Institute, [3]Flatiron Institute
[4]New York University, [5]University of Southern California

vural@cs.toronto.edu, abietti@flatironinstitute.org
soltanol@usc.edu, dennywu@nyu.edu

## ABSTRACT

Modern large language models (LLMs) excel at tasks that require storing and retrieving knowledge, such as factual recall and question answering. Transformers are central to this capability because they can encode information during training and retrieve it at inference. Existing theoretical analyses typically study transformers under idealized assumptions such as infinite data or orthogonal embeddings. In realistic settings, however, models are trained on finite datasets with non-orthogonal (random) embeddings. We address this gap by analyzing a single-layer transformer with random embeddings trained with (empirical) gradient descent on a simple token-retrieval task, where the model must identify an informative token within a length-$L$ sequence and learn a one-to-one mapping from tokens to labels. Our analysis tracks the "early phase" of gradient descent and yields explicit formulas for the model's storage capacity—revealing a multiplicative dependence between sample size $N$, embedding dimension $d$, and sequence length $L$. We validate these scalings numerically and further complement them with a lower bound for the underlying statistical problem, demonstrating that this multiplicative scaling is intrinsic under non-orthogonal embeddings. Code to reproduce all experiments is publicly available.[1]

## 1 INTRODUCTION

Large language models (LLMs) routinely answer knowledge questions with little or no external context, indicating that substantial factual information is stored in parameters and can be retrieved by suitable prompts Petroni et al. (2019); Jiang et al. (2020); Roberts et al. (2020). A deeper theoretical understanding of how such parametric memories are learned and accessed is increasingly important: it can guide scaling choices (e.g., trading off memory capacity against compute budgets, Carlini et al. (2022); Allen-Zhu & Li (2024)) and clarify failure modes (e.g., hallucination, Zucchet et al. (2025); Huang et al. (2025)). Motivated by empirical results documenting the prevalence of parametric factual recall and its scaling with model size Allen-Zhu & Li (2024); Morris et al. (2025), recent theoretical works have begun to analyze the capacity and learning dynamics of transformers on controlled factual-recall tasks Cabannes et al. (2024a); Nichani et al. (2025).

Many theoretical studies of transformer optimization work in population-dynamics settings and adopt simplifying assumptions, such as treating token embeddings as orthogonal or one-hot vectors (see, e.g., Tian et al. (2023b); Chen et al. (2024); Ghosal et al. (2024)). While these choices do not always reflect practical applications, they make the mathematics, particularly gradient calculations, more tractable, and population analyses of this kind do not characterize the statistical or computational complexity of gradient-based learning. In factual-recall setups, strictly orthogonal embeddings are known not to be capacity-optimal, whereas random or non-orthogonal embeddings (i.e., *superposition*) enable near-optimal factual storage Nichani et al. (2025). At the same time, removing the orthogonality assumption introduces token interference that leads to intricate optimization

---

[*]Work done while interning at the Flatiron Institute.
[1]Code available at https://github.com/nurimertvural/learning-to-recall-experiments.

behavior (e.g., oscillatory trajectories Cabannes et al. (2024b)), and in practice, superposition-based, memory-efficient solutions can also be more difficult to train Elhage et al. (2022), which highlight a fundamental trade-off between optimization and statistical efficiency versus storage capacity.

Motivated by the above gaps, we aim to address the following question.

*Can we characterize the optimization and sample complexity of a transformer with non-orthogonal embeddings trained by gradient descent in the learning of a factual recall task?*

## 1.1 OUR CONTRIBUTIONS

In this paper, we analyze gradient-based learning of a single-layer transformer with an attention+MLP block and random embeddings on a synthetic task inspired by Nichani et al. (2025): the model must retrieve an informative token from a context containing many noisy tokens via attention, then map it to the correct label via factual recall. To mitigate the complex optimization dynamics arising from non-orthogonal embeddings, we follow Bietti et al. (2023); Oymak et al. (2023) and consider a simplified training regime involving only a few gradient steps with finite samples on the attention and value matrices. This perspective effectively zooms in the "early phase" of the training as commonly studied in the feature-learning literature Ba et al. (2022); Damian et al. (2022); Dandi et al. (2023); Vural & Erdogdu (2024); Wang et al. (2025).

Our analysis provides a fine-grained characterization of how vocabulary size $V$, sample size $N$, embedding dimension $d$, sequence length $L$, and MLP width $m$ interact to permit successful gradient-based learning of the recall mechanism. Our main result states that

- The success of learning depends on $(V, N, d, L, m)$ in a *multiplicative* manner: learning becomes easier as $(N, d, m)$ increase, which reflects the benefits of more data, higher-dimensional (and thus more orthogonal) embeddings, and larger MLP width; whereas learning becomes harder as $(V, L)$ increase; that is, the task becomes more difficult with a larger vocabulary or longer sequences. This multiplicative relation is visualized in Figure 1a, where we examine how the parameter size $m \times d$ depends on the vocabulary size $V$ for different sequence lengths $L$. The full phase diagram corresponding to this relation, which formalizes Figure 1a, is shown in Figure 1b.

- Consequently, while optimal capacity and sample complexity can be achieved jointly for short sequences, successful learning on long sequences requires either a larger embedding dimension (thus sacrificing capacity) or larger sample sizes (worsening statistical complexity).

The multiplicative rate above formalizes the "tradeoff" intuition that smaller embedding dimension $d$ — which increases superposition and thereby improves storage capacity — simultaneously yields a harder learning problem, as reflected in the required sample size. We complement this with a statistical lower bound showing that the trade-off is inherent for any estimator that accesses only gradient information from the initialized transformer. Finally, although our theory is derived for a specific three-step training algorithm, we empirically observe qualitatively similar multiplicative scaling when the transformer is optimized by gradient descent to low empirical risk.

## 1.2 RELATED WORK

**Learning dynamics of transformers.** A growing line of work analyzes how transformers acquire specific behaviors from gradient-based training. Much of this literature imposes population-level assumptions and orthogonal/one-hot embeddings to make gradients tractable, often on discrete synthetic tasks Li et al. (2023); Bietti et al. (2023); Tian et al. (2023a); Nichani et al. (2024); Chen et al. (2024); Ghosal et al. (2024); Chen et al. (2025); Wang et al. (2025). Several works study few-step training regimes as a lens on the "early phase" of feature learning Bietti et al. (2023); Wang et al. (2025). Beyond discrete settings, related analyses investigate attention learning for continuous inputs and sparse-signal retrieval Oymak et al. (2023); Marion et al. (2025); Duranthon et al. (2026). A complementary thread focuses on the emergence of in-context learning and induction mechanisms: single- and two-layer attention trained on linear-regression or Markov data provably implements gradient-descent-like updates and generalized induction heads Von Oswald et al. (2023); Zhang et al. (2024); Chen et al. (2024); Nichani et al. (2024). These results typically rely on simplified settings and do not address storage capacity. In contrast, our work analyzes finite-sample training with non-orthogonal embeddings in an attention+MLP architecture with a particular focus on factual recall.

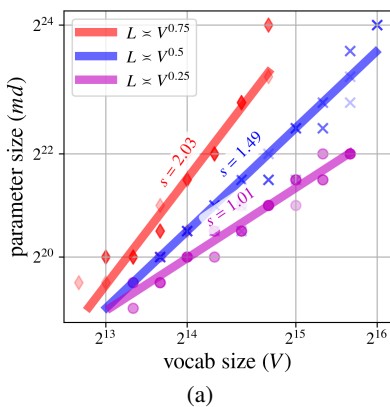
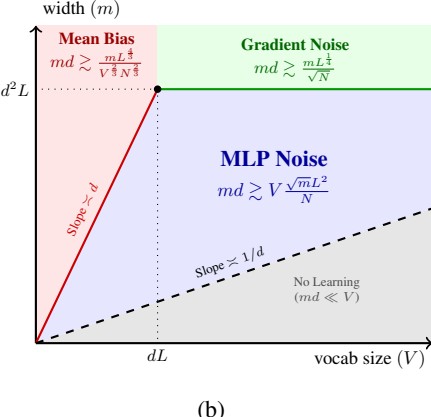

(a)          (b)

Figure 1: (a) Empirical scaling of the parameter size required for a GD-trained one-layer transformer to learn factual recall, where we use $m \asymp d^2$ (see Section 4.2 for details). For small $L$, the trained model achieves the optimal capacity $V \asymp md$ (purple line). As the sequence length $L$ increases, the scaling changes, suggesting a multiplicative rate (blue and red lines). (b) Phase diagram for the theoretical scaling of the parameter count given in Corollary 2. Each region corresponds to a regime where a particular noise term in Theorem 1 is dominant. The parameter-size condition ($md$) in each region is given in Corollary 2.

**Associative memories and storage capacity.** Classical associative memories (Hopfield-type models) study recall of vector patterns and established foundational capacity results Hopfield (1982); Amit et al. (1985); McEliece et al. (1988); Krotov & Hopfield (2016); Demircigil et al. (2017); Ramsauer et al. (2020); Schlag et al. (2021). Recent works adapt associative-memory viewpoints to transformers, modeling inner weights as superpositions of outer products and deriving scaling laws and optimization behaviors Bietti et al. (2023); Cabannes et al. (2024a;b). In factual recall specifically, random (non-orthogonal) embeddings enable near-parameter-count storage, whereas strictly orthogonal embeddings are not capacity-optimal Nichani et al. (2025). Various empirical works have studied the mechanisms and scaling behaviors of LLMs in factual association tasks Petroni et al. (2019); Jiang et al. (2020); Geva et al. (2020); Allen-Zhu & Li (2024). We provide a theoretical analysis of such mechanisms and quantify how vocabulary size, sequence length, embedding dimension, and MLP width jointly govern learning efficiency. Our work operates in a setting similar to Nichani et al. (2025) but allows finite samples and explicitly considers gradient descent dynamics. Our result is similar to the finite-sample results in Oymak et al. (2023), where the required sample size grows with the dimensionality and sparsity level of informative tokens, while we allow non-orthogonal embeddings and show optimal capacity as in Nichani et al. (2025) under certain conditions.

## 2 PROBLEM SETTING

Our goal is to understand the capacity of transformers trained on finite data with non-orthogonal embeddings, in a setting where the relevant information is hidden in a potentially large sequence of non-informative noisy tokens. The attention operation should then identify the relevant token, while the subsequent linear or MLP block can then recall the correct label via an associative memory mechanism. This is similar to the factual recall task studied by Nichani et al. (2025), with simplifications that make the analysis more tractable, as detailed below.

**Notation.** $\sigma$ denotes the softmax function. $\mathbb{1}_V := (1, \ldots, 1)^\top \in \mathbb{R}^V$ is the $V$-dimensional all-ones vector; $e_i$ is the one-hot vector with a 1 in the $i$-th position (dimension understood from context). We use $\gtrsim$ (resp. $\lesssim$) to mean "$\geq$" (resp. "$\leq$") up to polylogarithmic factors in $V$: $f_V \gtrsim g_V \iff f_V \geq \text{poly}(\log V) g_V$ and $f_V \lesssim g_V \iff f_V \leq \text{poly}(\log V) g_V$, for some fixed polynomial. Lastly, $\|\cdot\|_2$ denotes the Euclidean norm for vectors and the operator (spectral) norm for matrices.

**Problem setup.** Let the input/output tokens take values from a finite alphabet $[V] := \{1, \cdots, V\}$. For notational convenience, we represent the alphabet by the one-hot vocabulary $\mathcal{V} = \{e_1, \cdots, e_V\}$. Each example in the data consists of a length-$L$ input sequence $\boldsymbol{X} = [\boldsymbol{x}_1, \ldots, \boldsymbol{x}_L] \in \mathcal{V}^L$ and a label $\boldsymbol{p} \in \mathcal{V}$ generated as follows:

- *Input* tokens are sampled independently and uniformly: $[\boldsymbol{x}_1, \ldots, \boldsymbol{x}_L] \sim \text{Unif}(\mathcal{V}^L)$.
- *Informative position* is a random index $\ell \sim \text{Unif}([L])$ independent of $\boldsymbol{X}$.

- *Ground-truth function* is a permutation matrix $\mathbf{\Pi}_* \in \{0,1\}^{V \times V}$. Labels are generated as the permuted informative token, $\boldsymbol{p} = \mathbf{\Pi}_* \boldsymbol{x}_\ell$, while the remaining tokens are non-informative.

The goal is to identify the correct token position $\ell$ and learn the target function (permutation) $\mathbf{\Pi}_*$.

**Transformer architecture.** We consider a basic transformer block which first maps input tokens into a $d$-dimensional embedding space where $d < V$. The embedding layer is parameterized by $(\boldsymbol{Z}_{\text{in}}, \boldsymbol{Z}_{\text{out}}, \boldsymbol{z}_{\text{trig}}, \boldsymbol{z}_{\text{EOS}}) \in \mathbb{R}^{d \times V} \times \mathbb{R}^{d \times V} \times \mathbb{R}^d \times \mathbb{R}^d$, where

- The input tokens are embedded by the columns of the matrix $\boldsymbol{Z}_{\text{in}} \in \mathbb{R}^{d \times V}$.
- Output tokens are associated with unembedding vectors, which are collected in $\boldsymbol{Z}_{\text{out}} \in \mathbb{R}^{d \times V}$.
- $\boldsymbol{z}_{\text{trig}}$ is a trigger vector that marks the informative token.
- $\boldsymbol{z}_{\text{EOS}}$ is the special embedding vector that marks the end-of-sequence.

Given the embedding parameters, we define the self-attention head, parameterized by the key-query matrix $\boldsymbol{W}_{\text{KQ}} \in \mathbb{R}^{d \times d}$, which operates on the embedded sequence of inputs $\boldsymbol{Z}_{\text{in}}\boldsymbol{X} \in \mathbb{R}^{d \times L}$:

$$\text{attn}(\boldsymbol{X}; \boldsymbol{W}_{\text{KQ}}) \coloneqq \boldsymbol{Z}_{\text{in}}\boldsymbol{X}\sigma\Big((\boldsymbol{z}_{\text{trig}}\boldsymbol{e}_\ell^\top + \boldsymbol{Z}_{\text{in}}\boldsymbol{X})^\top \boldsymbol{W}_{\text{KQ}}\boldsymbol{z}_{\text{EOS}}\Big). \tag{2.1}$$

The trigger embedding $\boldsymbol{z}_{\text{trig}}$ is used to "mark" the informative token with a special direction, mimicking the behavior of previous transformer layers that may learn to flag particular tokens by adding to its residual stream[2] (note that the number of trainable parameters inside softmax can be reduced to $d$ by collapsing $\boldsymbol{W}_{\text{KQ}}\boldsymbol{z}_{\text{EOS}}$ into a vector). We consider two different learning models: an *Attention-only* model and a width-$m$, two-layer network model *Attention-MLP*, defined as:

$$\hat{\boldsymbol{p}}(\boldsymbol{X}; \boldsymbol{V}, \boldsymbol{W}_{\text{KQ}}) = \begin{cases} \sigma\Big(\boldsymbol{Z}_{\text{out}}^\top \boldsymbol{V}\,\text{attn}(\boldsymbol{X}; \boldsymbol{W}_{\text{KQ}})\Big), & \text{Attention only} \\ \sigma\Big(\boldsymbol{Z}_{\text{out}}^\top \boldsymbol{V}\phi(\boldsymbol{W}_{\text{in}}\text{attn}(\boldsymbol{X}; \boldsymbol{W}_{\text{KQ}}))\Big), & \text{Attention-MLP} \end{cases} \tag{2.2}$$

where $\boldsymbol{V} \in \mathbb{R}^{d \times d}$ for the *Attention-only* and $\boldsymbol{V} \in \mathbb{R}^{d \times m}$, $\boldsymbol{W}_{\text{in}} \in \mathbb{R}^{m \times d}$ for the *Attention-MLP* model. Note that compared with *Attention-only* model, the *Attention-MLP* model contains an additional set of trainable parameters and nonlinear activation function $\phi$ before the value matrix. Similar to in Nichani et al. (2025), the MLP allows using a smaller embedding dimension $d$ while keeping the capacity large by increasing width $m$.

For the *Attention-MLP*, we keep $\boldsymbol{W}_{\text{in}}$ fixed at its random initialization. The trainable parameters for both of our models are $(\boldsymbol{V}, \boldsymbol{W}_{\text{KQ}})$. We use cross-entropy loss to train our model:

$$\mathcal{L}\big((\boldsymbol{V}, \boldsymbol{W}_{\text{KQ}}), (\boldsymbol{X}, \boldsymbol{p})\big) = -\textstyle\sum_{i=1}^V p_i \log \hat{p}_i.$$

**Training algorithm.** Following Oymak et al. (2023), we consider a 3-step gradient-based algorithm with dataset $\{(\boldsymbol{X}_i, \boldsymbol{p}_i)\}_{i=1}^N$ with a sample size of $N$. We initialize our parameters as $\boldsymbol{V}^{(0)} = 0$, $\boldsymbol{W}_{\text{KQ}}^{(0)} = 0$ and use the learning rates $\eta, \gamma > 0$:

$$\boldsymbol{V}^{(1)} = \boldsymbol{V}^{(0)} - \eta \cdot \tfrac{1}{N}\textstyle\sum_{i=1}^N \nabla_{\boldsymbol{V}}\mathcal{L}\big((\boldsymbol{V}^{(0)}, \boldsymbol{W}_{\text{KQ}}^{(0)}); (\boldsymbol{X}_i, \boldsymbol{p}_i)\big) \tag{2.3}$$

$$\boldsymbol{W}_{\text{KQ}}^{(1)} = \boldsymbol{W}_{\text{KQ}}^{(0)} - \gamma \cdot \tfrac{1}{N}\textstyle\sum_{i=1}^N \nabla_{\boldsymbol{W}_{\text{KQ}}}\mathcal{L}\big((\boldsymbol{V}^{(1)}, \boldsymbol{W}_{\text{KQ}}^{(0)}); (\boldsymbol{X}_i, \boldsymbol{p}_i)\big) \tag{2.4}$$

$$\boldsymbol{V}^{(2)} = \boldsymbol{V}^{(1)} - \gamma \cdot \tfrac{1}{N}\textstyle\sum_{i=1}^N \nabla_{\boldsymbol{V}}\mathcal{L}\big((\boldsymbol{V}^{(1)}, \boldsymbol{W}_{\text{KQ}}^{(1)}); (\boldsymbol{X}_i, \boldsymbol{p}_i)\big). \tag{2.5}$$

**Network prediction and storage.** Given our model and training method, we use argmax decoding at inference and define the test accuracy as

$$\text{Accuracy} \coloneqq \mathbb{P}_{(\boldsymbol{X}, \boldsymbol{p})}\big[\boldsymbol{p} = \boldsymbol{e}_{\text{pred}(\boldsymbol{X})}\big], \quad \text{where} \quad \text{pred}(\boldsymbol{X}) \coloneqq \arg\max_{j \in [V]} \hat{p}_j(\boldsymbol{X}; \boldsymbol{V}^{(2)}, \boldsymbol{W}_{\text{KQ}}^{(1)}),$$

where $\hat{p}(\boldsymbol{X}; \boldsymbol{V}^{(2)}, \boldsymbol{W}_{\text{KQ}}^{(1)})$ is the network output defined in (2.2). In what follows, we characterize conditions under which the model stores the informative tokens asymptotically, i.e., $\text{Accuracy} \to 1$ as $V \to \infty$, in terms of the relevant parameters $(V, N, d, L, m)$.

---

[2]The "trigger" terminology is borrowed from Bietti et al. (2023), where a special previous token "triggers" a retrieval operation in the context of induction heads. Our setup resembles learning only the "induction head" layer assuming the first "previous token head" layer is already in place. The triggers often appear to be single directions in interpretability literature, see, e.g., the "X in opposite of X" feature in Kamath et al. (2025).

## 3 MAIN RESULTS

We first present our general theorem on learnability via gradient descent, and then specialize into different regimes to derive more interpretable scaling behaviors in Section 4. We provide a proof sketch in Section 5, and defer the full proof to Appendix C.

### 3.1 TECHNICAL ASSUMPTIONS

We first state generic assumptions that apply to both the *Attention-only* and *Attention-MLP* models.

**Assumption 1.**

- ***Parameter range:*** *Let $L = V^c$ for $c \in (0, 1)$, $\Omega(V \log V) \le N = o(VL)$, and $V \ge \Omega(1)$.*

- ***Learning rate:*** *We use a sufficiently small learning rate $\eta$ for the initial step (2.3), and sufficiently large learning rate $\gamma$ for the remaining steps (2.4)-(2.5) that satisfy Assumption 4.*

- ***Embeddings:*** *Let $\boldsymbol{Z}_{\text{in}}, \boldsymbol{Z}_{\text{out}} \in \mathbb{R}^{d \times V}$ be independent Gaussian matrices, and let $\boldsymbol{z}_{\text{trig}}, \boldsymbol{z}_{\text{EOS}} \in \mathbb{R}^d$ be independent Gaussian vectors, all with i.i.d. entries distributed as $\mathcal{N}(0, 1/d)$.*

We assume $c \in (0, 1)$ since in many practical pretraining setups, the context length is smaller than the vocabulary size, and the condition $L \ll V$ simplifies several terms in the proofs. The lower bound $N \gtrsim V \log V$ is required so that each element from the alphabet of size $V$ is seen at least once with high probability. The learning rates follow prior analyses (Oymak et al., 2023; Nichani et al., 2024): a small $\eta$ ensures that the network's predictions remain close to uniform after the first step, whereas a large $\gamma$ is needed to push the attention scores and predictions toward one-hot vectors.

In addition to the above assumptions, we require the transformer model to have sufficient capacity to reach perfect test accuracy. Such conditions are characterized by Nichani et al. (2025). For the *Attention-only* model, we have the following condition (see (Nichani et al., 2025, Theorem 3)).

**Assumption 2** (Attention-only)**.** *For the* Attention-only *model, we require $d^2 \gtrsim V$.*

With a nonlinear MLP layer, a smaller embedding dimension can suffice if the width is large enough. Hence for *Attention-MLP* we require the following condition.

**Assumption 3** (Attention-MLP)**.** *For the* Attention-MLP *model, we assume that*

- ***Polynomial activation:*** *$\phi : \mathbb{R} \to \mathbb{R}$ satisfies $\phi(0), \phi'(0), \phi''(0) \ne 0$.*

- ***MLP width:*** *$md \gtrsim V$ and $d \gtrsim V^{\frac{1}{k_\star + 1}}$, where $k_\star$ denotes the smallest nonzero Hermite mode of $\phi$, i.e., $k_\star := \min\{k > 0 : \mathbb{E}_{Z \sim \mathcal{N}(0,1)}[\phi(Z)h_k(Z)] \ne 0\}$ where $h_k$ is the $k^{th}$ Hermite polynomial.*

- ***Initialization:*** *$\boldsymbol{W}_{\text{in}} \in \mathbb{R}^{m \times d}$ are fixed with entries i.i.d. distributed as $\mathcal{N}(0, 1)$.*

The nonlinear MLP layer allows us to compensate for the embedding dimension and go beyond the $d^2 \gtrsim V$ lower bound required by the *Attention-only* model (Assumption 2). Note that $md \gtrsim V$ is a necessary condition for capacity as shown in (Nichani et al., 2025). The additional requirements imposed on the polynomial activation function appear to be artifacts of our three-step GD analysis, and we conjecture that they could be relaxed when considering a longer training horizon.

### 3.2 LEARNABILITY STATEMENT

Now we are ready to present our main theorem on the complexity of learning the factual recall task. Specifically, the transformer learns the desired mechanism when the signal term dominates the noise and bias terms as stated below.

**Theorem 1.** *Let Assumptions 1 and 3 hold for* Attention-MLP*, and 1 and 2 hold for* Attention-only*. The* Attention-MLP *model achieves* Accuracy $= 1 - o_V(1)$ *with probability* $1 - o_V(1)$ *whenever*

$$\underbrace{\frac{1}{VL^2}}_{Signal} \gtrsim \underbrace{\frac{1}{N\sqrt{L}d(d \wedge L)}}_{Gradient\ noise} + \underbrace{\frac{1}{N\sqrt{V}d(d \wedge L)}}_{Mean\ bias} + \underbrace{\frac{1}{Nd\sqrt{m}}}_{MLP\ noise}. \tag{3.1}$$

*For the* Attention-only *model, the same holds with the last MLP noise term removed.*

Theorem 1 characterizes learnability as a function of $(V, N, d, L, m)$ and identifies the following terms that impact the gradient signal-to-noise ratio:

1. *Signal* measures the alignment between the key–query weights $\boldsymbol{W}_{\mathrm{KQ}}^{(1)}$ and the trigger $\boldsymbol{z}_{\mathrm{trig}}$.

2. *Gradient noise* is due to the concentration error in the update of $\boldsymbol{W}_{\mathrm{KQ}}^{(1)}$.

3. *Mean bias* arises from the nonzero mean of token vectors $\{\boldsymbol{X}_i\}_{i=1}^N$.

4. *MLP noise* reflects the randomness in the MLP weight matrix $\boldsymbol{W}_{\mathrm{in}}$ in *Attention–MLP*.

We make the following observations.

- **Multiplicative scaling.** Note that the parameters $(V, N, d, L, m)$ interact in a multiplicative fashion. For example, the noise and bias terms in (3.1) all decay with $(N \times d)$, suggesting that increasing the embedding dimensions $d$ can lower the statistical complexity of learning the correct recall mechanism. While the full 5-parameter trade-off can be opaque, in Section 4 we focus on specific regimes that lead to simplification of the scaling relationship and validate the rate empirically.

- **Optimal storage & sample complexity.** Recall that the capacity-optimal construction for the factual recall task requires $md \gtrsim V$ parameters (or $d^2 \gtrsim V$ for *Attention–only*); and as discussed earlier, a sample size $N \asymp V \log V$ is necessary to observe all distinct tokens. (3.1) implies that in the small-$L$ regime, the optimized transformer achieve optimal capacity and sample complexity simultaneously. For longer sequences, however, these two conditions may not be achieved at the same time, i.e., one must increase either the network width or sample size beyond optimality to learn the task — this confirms the empirical observation in Figure 1.

### 3.3 STATISTICAL LOWER BOUND

Theorem 1 provides an upper bound (i.e., sufficient condition) on the model and sample size for learning factual recall under a 3-gradient-step optimization procedure. We complement this sufficient condition with a lower bound indicating that the multiplicative dependence on the problem parameters is partly statistical; that is, the scaling behavior will be observed in any model satisfying the broader conditions stated below. Our lower bound applies to statistical methods that can query the dataset through the attention outputs at initialization, $\boldsymbol{h}_i := \mathrm{attn}(\boldsymbol{X}_i, \boldsymbol{W}_{\mathrm{KQ}}^{(0)})$. In particular, we consider queries of the form $\boldsymbol{h}_i$ as the gradient with respect to the key–query matrix $\boldsymbol{W}_{\mathrm{KQ}}$ depends on $\{\boldsymbol{h}_i, \boldsymbol{h}_i \boldsymbol{h}_i^\top\}_{i=1}^N$ (see (B.1)). The statement is given below:

**Theorem 2** (Informal). *Any method that relies on the noisy version of the queries $\{\boldsymbol{h}_i,\ \boldsymbol{h}_i\boldsymbol{h}_i^\top\}_{i=1}^N$ fails, i.e.,* Accuracy $\not\to 1$ *with finite probability, if* $N \lesssim V \min\{1, L/d^2\}$.

The complete statement of Theorem 2 is deferred to Theorem 4 in Appendix E. We observe that the lower bound does not exactly match our upper bound in Theorem 1, as *Signal* $\lesssim$ *Gradient Noise* in (3.1) is stronger than the stated lower bound. This being said, Theorem 2 also confirms the multiplicative scaling, hence suggesting the trade-off between capacity and sample efficiency is present in a boarder class of learning algorithms. A stronger computational lower bound for transformers and gradient-based optimization is an interesting problem we leave for future work.

## 4 IMPLICATIONS AND EMPIRICAL VERIFICATIONS

In this section, we leverage our main theorem to obtain more concrete scalings between parameters, and present empirical evidence on the derived multiplicative rate.

### 4.1 ATTENTION-ONLY MODEL

We start with the *Attention-only model* which gives a simpler phase diagram.

**Corollary 1.** *For the* Attention-only *model, the bottleneck term in* (3.1) *is the* Mean bias *term. Therefore, Theorem 1 is equivalent to the parameter size requirement* $d^2 \gtrsim \max\{V, V^{2/3}L^{8/3}/N^{4/3}\}$.

We make the following observations:

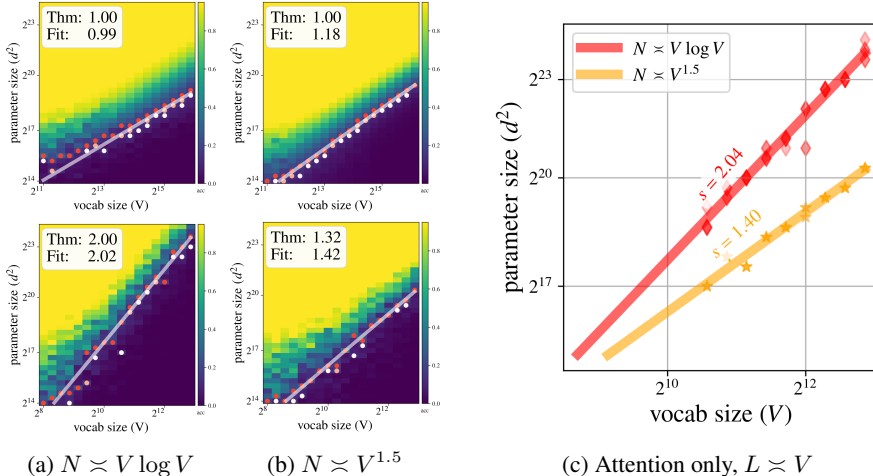

(a) $N \asymp V \log V$     (b) $N \asymp V^{1.5}$     (c) Attention only, $L \asymp V$

Figure 2: Empirical scaling of parameter size via three-step GD for the *Attention-only* model. In (a) and (b), top-left and top-right use $L \asymp V^{0.5}$; bottom-left and bottom-right use $L \asymp V$. In (c), we compare the parameter counts from (a) and (b) for the $L \asymp V$ case under two sample-size regimes, $N \asymp V \log V$ and $N \asymp V^{1.5}$. *Line fitting:* We identify in the heatmaps the smallest embedding dimension that achieves accuracies $\{0.1, 0.125, 0.15\}$ and perform a least squares fit. The slopes of the fitted lines and their theoretical counterparts are reported on the heatmaps.

- The condition in Corollary 1 is the maximum of two terms, where $d^2 \gtrsim V$ is due to the capacity requirement in Assumption 2, whereas the second term ensures *Signal* $\gtrsim$ *Mean bias* and implies a multiplicative scaling between the sample size $N$ and embedding dimension $d$ (i.e., increasing one of the parameters can compensate for the other).

- Note that the *Mean bias* term arises from a nonzero token mean, which can potentially be alleviated by centering the tokens, for instance through an appropriate normalization layer. Exploring the effect of applying normalization in this model is an interesting direction for future work.

**Empirical Findings.** We run the three-step gradient descent algorithm on an *Attention-only* model over varying $V$ and $d$, and report the accuracies in the heatmaps (Figure 2). The plots are in log-log scale; therefore, the slopes give the exponent $s$ in $d \asymp V^s$. As shown in the top row of Figures 2a-2b, the slope for relatively small $L$ (where $L \asymp V^{0.5}$) matches the optimal capacity condition $d^2 \asymp V$. By contrast, when the context window is larger ($L \asymp V$), the requirement becomes $d \asymp V$, which is also reflected in the experimental results, as observed in the bottom panel of Figure 2a.

In Figure 2b we run experiments with increasing sample size to observe the multiplicative trade-off. As seen in the bottom figure of Figure 2b, increasing the sample size from $V \log V$ to $V^{1.5}$ reduces the exponent of the parameter size from 2.02 to 1.42 (the theoretical value is $s = 1.32$). Finally, the learnability thresholds for $L \asymp V$ in Figures 2a and 2b are plotted together in Figure 2c, to illustrate that increasing the sample size can compensate for the number of parameters in the network.

## 4.2 ATTENTION-MLP MODEL

For the attention-MLP model, the nonlinear MLP layer introduces additional phases as stated below.

**Corollary 2.** *For the* Attention-MLP *model, Theorem 1 translates to* $md \gtrsim V$ *and*

$$\text{Signal} \gtrsim \begin{cases} \text{MLP noise,} & m = o(d^2 L) \text{ and } m = o(dV) \\ \text{Gradient noise,} & V \gtrsim dL \text{ and } m \gtrsim d^2 L \\ \text{Mean bias,} & V = o(dL) \text{ and } m \gtrsim dV, \end{cases}$$

*where*

- Signal $\gtrsim$ MLP noise *is equivalent to* $md \gtrsim V \frac{\sqrt{m} L^2}{N}$.

- Signal $\gtrsim$ Gradient noise *is equivalent to* $md \gtrsim V \frac{m L^{\frac{1}{4}}}{\sqrt{N}}$

- Signal $\gtrsim$ Mean bias *is equivalent to* $md \gtrsim \frac{m L^{\frac{4}{3}} V^{\frac{1}{3}}}{N^{\frac{2}{3}}}$.

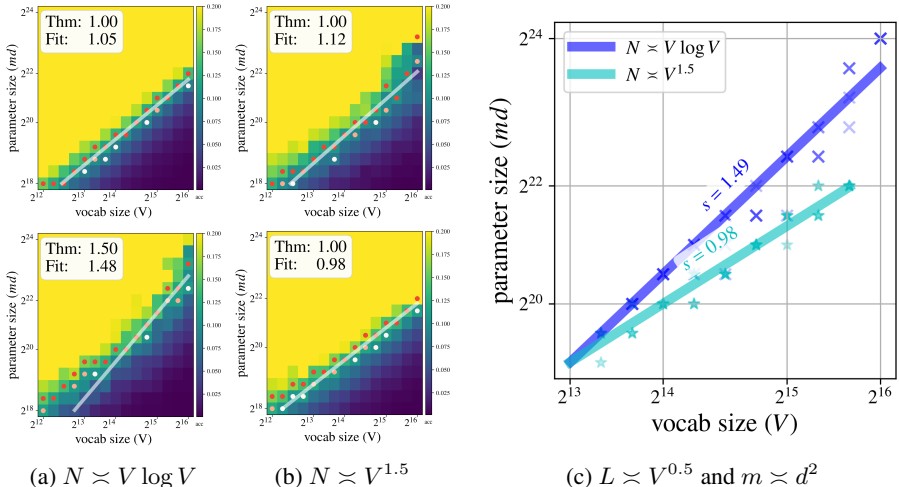

(a) $N \asymp V \log V$      (b) $N \asymp V^{1.5}$      (c) $L \asymp V^{0.5}$ and $m \asymp d^2$

Figure 3: Empirical scaling of parameter size for the *Attention-MLP* model under two sample size regimes, $N \asymp V \log V$ and $N \asymp V^{1.5}$. In (a) and (b), top-row uses $L \asymp V^{0.25}$; bottom-row uses $L \asymp V^{0.5}$. In (c), we compare the parameter counts from (a) and (b) for the $L \asymp V^{0.5}$ case under both sample-size regimes.

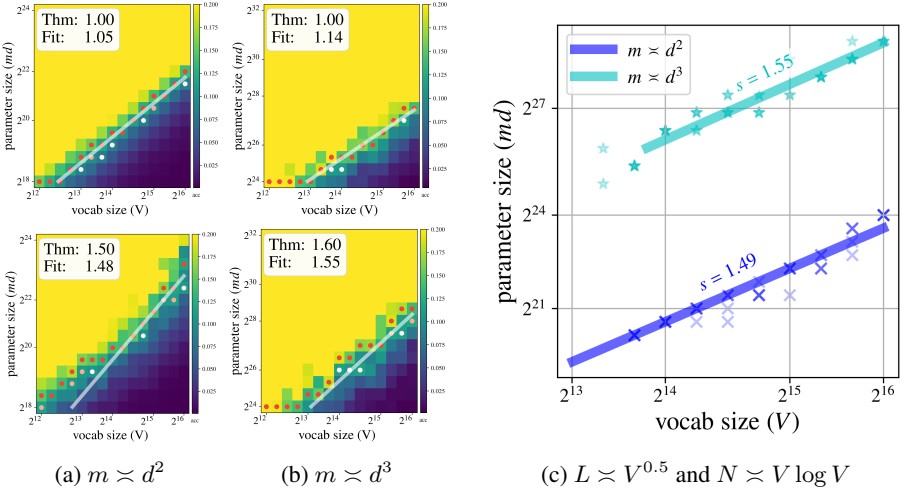

(a) $m \asymp d^2$      (b) $m \asymp d^3$      (c) $L \asymp V^{0.5}$ and $N \asymp V \log V$

Figure 4: Empirical scaling of embedding parameter size for the *Attention-MLP* model under two width regimes, $m \asymp d^2$ and $m \asymp d^3$. In (a) and (b), top-row uses $L \asymp V^{0.25}$; bottom-row uses $L \asymp V^{0.5}$. In (c), we compare the parameter counts from (a) and (b) for the $L \asymp V^{0.5}$ case under both width regimes.

The phase diagram for the Attention-MLP model is visualized in Figure 1b. Compared to the *Attention-only* case, it exhibits additional regimes because we can trade off $m$ and $d$ and thus use a smaller embedding dimension; this can lead to different dominant terms in the gradient. In particular, since large $L$ and $d$ entail a larger magnitude of the *Mean bias* (as in the *Attention-only* setting), increasing the MLP width $m$ and thereby reducing the required embedding dimension $d$ may suppress this bias term.

**Empirical Findings.** We run the 3-step gradient descent algorithm on an Attention-MLP network over varying $V$ and $d$ and plot the accuracies in Figures 3 and 4. We take the nonlinearity to be the mixture of two Hermite polynomials $\phi = 0.7h_2 + 0.3h_3$, satisfying the conditions in Assumption 3. We run experiments with width $m \asymp d^2$ and $m \asymp d^3$. Due to the prohibitive cost of increasing the width further, we restrict ourselves to the *MLP noise*-dominated region.

In Figure 1a, we plot the scaling of the number of parameters ($md$) as a function of vocabulary size $V$ for different sequence-length regimes in $L$. We observe that $L \asymp V^{0.25}$ requires $md \asymp V$, which is the optimal capacity, as predicted by our theory. As $L$ increases, we need more parameters to

achieve the same capacity, as observed in the $L \asymp V^{0.5}$ and $L \asymp V^{0.75}$ cases in Figure 1, where the slopes agree with our theoretical predictions as well (see also Figures 3a and 3b).

We further test the effect of sample size in Figure 3, where we use $L \asymp V^{0.5}$ and $m \asymp d^2$. We plot both heat maps in Figures 3a and 3b, and the fitted lines for $L \asymp V^{0.5}$ together in Figure 3c. Note that we state the plot in terms of parameter count, which scales as $md \asymp d^3$, so the slopes from the heat map are scaled accordingly. We observe that increasing $N$ from $N \asymp V \log V$ to $N \asymp V^{1.5}$ reduces the network size to the optimal level, aligning with our theoretical prediction. The heatmap versions of these experiments are shown in Figures 3a and 3b.

Lastly, we probe the width scaling by keeping the sample size $N \asymp V \log V$ and $L \asymp V^{0.5}$ fixed in Figure 4. Here, we observe that we can reduce the embedding-dimension requirement by increasing $m$ in Theorem 1, although it increases the total parameter count overall, as seen in Figures 4b and 4c, since width must grow proportionally more than $d$ to achieve the same accuracy. This is also consistent with our result.

### 4.3 Beyond Early Phase of Training

While our theoretical analysis focuses on a particular three-gradient-step training procedure, we empirically observe qualitatively similar multiplicative scalings when the transformer model is optimized beyond the "early phase". Specifically, we train our *Attention-only* model using Adam (Kingma & Ba, 2015) with mini-batch gradients. In the experiments, we use layer normalization in both the attention and output layers and set the learning rate to 0.005. We use a batch size of $\lfloor N/2 \rfloor$ (except in the last experiment, where we use $\lfloor N/16 \rfloor$), and run the training for 16 epochs. We highlight the following observations:

- *Capacity improvement with multi-pass training.* In the top row of Figure 5, we plot the heatmaps for $L \asymp V$ and $N \asymp V \log V$. In early training the slope is suboptimal; notably, by the end of Epoch 1 it closely aligns with our theoretical prediction. Moreover, training the network additional epochs improves the capacity condition to a near-optimal level, as shown in Figures 5c and 5d.

- *Effect of sample size.* In the bottom row of Figure 5, we plot the heatmaps for $L \asymp V$ and $N \asymp V^{1.5}$. We observe a similar trajectory in capacity, while the overall capacities improve compared to the small-sample regime, showing the multiplicative dependence on sample size $N$.

- *Effect of sequence length.* In Figure 6, we plot the heatmaps for $L \asymp V^{0.85}$ and $N \asymp V \log V$. We observe improvements in capacity over multiple epochs, while the capacity is larger than in the $L \asymp V$ setting at every stage of training, which shows the effect of the sequence length $L$.

- *Effect of batch size.* In Figure 7, we repeat the experiments from this section using the same learning rate and architecture but with a smaller batch size $\lfloor N/16 \rfloor$. As before, we consider $L \asymp V$ in two sample-size regimes, $N \asymp V \log V$ and $N \asymp V^{1.5}$. We observe behavior similar to the larger batch size setting, but with improved slopes in Figure 7. This suggests that smaller batch sizes may improve capacity in practice.

Overall, these experiments suggest that the multiplicative relation between the hyperparameters remains throughout training. However, the exponents depend on the iteration number and batch size. Understanding how capacity evolves during training remains an interesting open question.

## 5 Proof Overview

In this section, we outline the main ideas behind the proof of Theorem 1. For ease of exposition, we consider the Attention-only model under population dynamics with orthogonal embeddings.

The main idea of the proof is the following: We observe that the recall task is achieved with near-perfect accuracy if and only if the attention mechanism can distinguish informative tokens. Once this occurs, the remaining task reduces to learning a linearly separable problem, which is well understood. Therefore, the proof focuses on the attention scores in (2.1) and characterizes the conditions under which the mechanism selects the informative tokens.

The pre-softmax scores evaluated on a fresh sequence $\boldsymbol{X}_{\text{in}}$, with the key-query matrix given by the first gradient-descent iterate $\boldsymbol{W}_{\text{KQ}}^{(1)}$, are given by

$$\text{scores} := \left( \boldsymbol{z}_{\text{trig}} \boldsymbol{e}_\ell^\top + \boldsymbol{Z}_{\text{in}} \boldsymbol{X}_{\text{in}} \right)^\top \boldsymbol{W}_{\text{KQ}}^{(1)} \boldsymbol{z}_{\text{EOS}}. \tag{5.1}$$

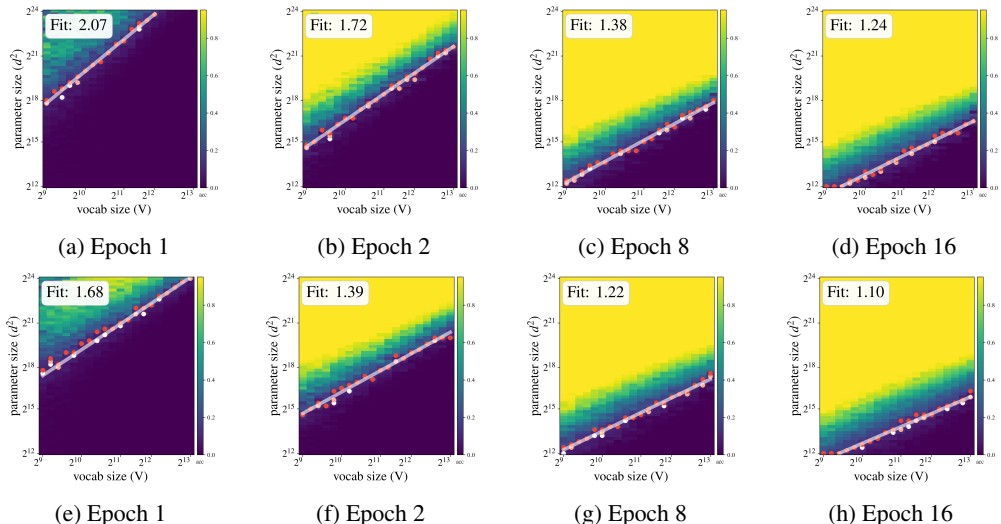

(a) Epoch 1      (b) Epoch 2      (c) Epoch 8      (d) Epoch 16

(e) Epoch 1      (f) Epoch 2      (g) Epoch 8      (h) Epoch 16

Figure 5: Empirical scaling of the parameter size for the *Attention-only* model under two sample size regimes. *Top row (a–d):* $N \asymp V \log V$. *Bottom row (e–h):* $N \asymp V^{1.5}$. The model uses $L \asymp V$ and is trained using Adam over 16 epochs. We observe that the capacity improves as the number of epochs increases.

By substituting the exact expression for $\boldsymbol{W}_{\mathrm{KQ}}^{(1)}$ into (5.1), we analyze scores. For intuition, we present the simplified expression below (see (B.1) for the full expression):

$$\text{scores} \approx \gamma \|\boldsymbol{z}_{\mathrm{trig}}\|_2^2 \boldsymbol{e}_\ell \underbrace{\left( \frac{1}{NL} \sum_{i=1}^N \boldsymbol{x}_{i,\ell}^\top \boldsymbol{Z}_{\mathrm{in}}^\top (\boldsymbol{V}^{(1)})^\top \boldsymbol{Z}_{\mathrm{out}}(\boldsymbol{p}_i - \tfrac{1}{V}\mathbb{1}_V) \right)}_{\text{Informative}} \tag{5.2}$$

$$+ \gamma \boldsymbol{X}_{\mathrm{in}}^\top \boldsymbol{Z}_{\mathrm{in}}^\top \underbrace{\left( \frac{1}{NL} \sum_{i=1}^N \boldsymbol{Z}_{\mathrm{in}} \boldsymbol{X}_i \boldsymbol{X}_i^\top \boldsymbol{Z}_{\mathrm{in}}^\top (\boldsymbol{V}^{(1)})^\top \boldsymbol{Z}_{\mathrm{out}}(\boldsymbol{p}_i - \tfrac{1}{V}\mathbb{1}_V) \right)}_{\text{Non-informative}}. \tag{5.3}$$

Here $\boldsymbol{V}^{(1)}$ denotes the first iterate of the value matrix defined in (2.3). The *informative term* in (5.2) captures the alignment between the trigger vector in the fresh input and the one encoded in the learned weights $\boldsymbol{W}_{\mathrm{KQ}}^{(1)}$, and therefore contains the position information of the informative token. By contrast, the *non-informative term* in (5.3) reflects correlations between tokens and does not contain information about the token position. Thus, the proof reduces to characterizing the conditions under which the informative term in (5.2) dominates the non-informative term. We present the proof sketch for the finite-sample setting with non-orthogonal embeddings, as well as how each noise term in Theorem 1 arises, in Appendix B.1.

## 6 CONCLUSION

In this paper, we derived precise asymptotic rates for learning with gradient descent on transformers trained on a simple recall task with random embeddings and finite samples. Our analysis and experiments reveal a rich picture of multiplicative scalings between various problem parameters, showing that parameter count is not the only important factor controlling capacity when learning with finite samples on large noisy sequences. Our results suggest that finer control of the data distribution may be necessary for learning efficiently at optimal capacity, for instance by ensuring sequences are less noisy and more informative, hoping that the discovered mechanisms are robust to harder settings. This is reminiscent of the procedures used for long context extension in LLMs, where most of training happens on shorter sequences, but the final models are extended to work with very long sequences, and empirically do well on retrieval tasks such as "needle-in-a-haystack" (e.g., Gemini Team, 2024), which resembles our theoretical setup. Analyzing similar scalings in more structured data distributions and architectures is thus an interesting avenue for future work.

ACKNOWLEDGMENT

The work of M. Soltanolkotabi was partially supported by AWS credits through an Amazon Faculty Research Award, a NAIRR Pilot Award, and generous funding by Coefficient Giving, and the USC-Capital One Center for Responsible AI and Decision Making in Finance (CREDIF) Fellowship. M. Soltanolkotabi is also supported by the Packard Fellowship in Science and Engineering, a Sloan Research Fellowship in Mathematics, NSF CAREER Award #1846369, DARPA FastNICS program, NSF CIF Awards #1813877 and #2008443, and NIH Award DP2LM014564-01. This collaboration began during the "Modern Paradigms in Generalization" and "Special Year on Large Language Models and Transformers, Part 1" programs at the Simons Institute for the Theory of Computing, Berkeley in 2024.

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

CONTENTS

## A ADDITIONAL EXPERIMENTS

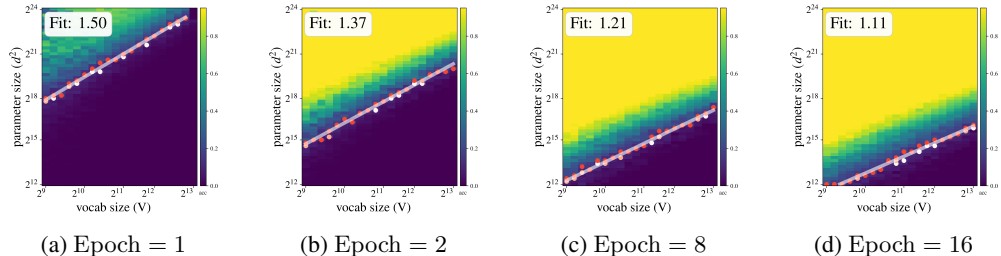

(a) Epoch = 1     (b) Epoch = 2     (c) Epoch = 8     (d) Epoch = 16

Figure 6: Empirical scaling of the parameter size for the *Attention-only* model with $N \asymp V \log V$ and $L \asymp V^{0.85}$. The model is trained using Adam over 16 epochs. The slopes are smaller than in Figure 5, which is consistent with the shorter sequence length.

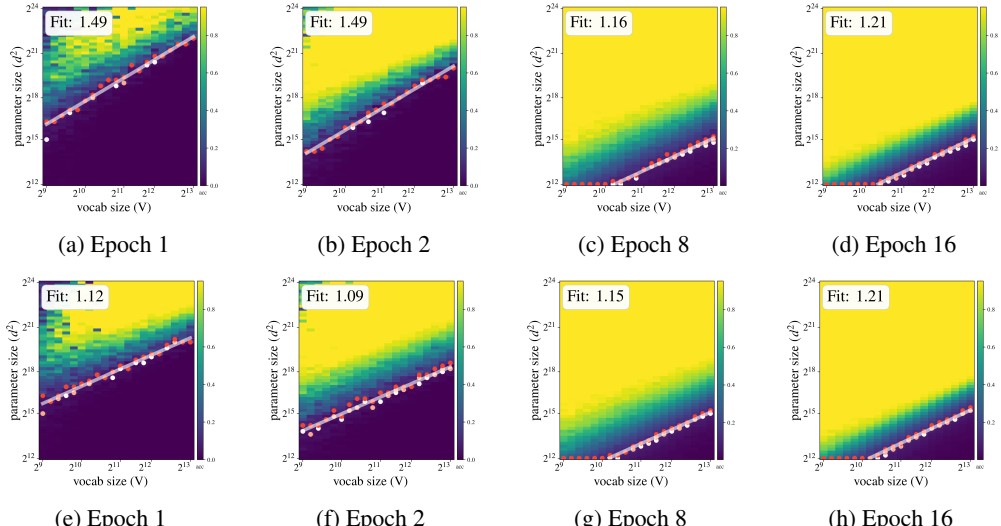

(a) Epoch 1     (b) Epoch 2     (c) Epoch 8     (d) Epoch 16

(e) Epoch 1     (f) Epoch 2     (g) Epoch 8     (h) Epoch 16

Figure 7: Empirical scaling of the parameter size for the *Attention-only* model ($L \asymp V$), trained with Adam using a batch size of $\lfloor N/16 \rfloor$. *Top row (a–d):* $N \asymp V \log V$. *Bottom row (e–h):* $N \asymp V^{1.5}$.

## B PRELIMINARIES FOR APPENDIX

**Additional Notation.** For a vector $\boldsymbol{x} \in \mathbb{R}^V$, we use $\mathrm{diag}(\boldsymbol{x}) \in \mathbb{R}^{V \times V}$ denotes the diagonal matrix which has the same diagonal entries with $\boldsymbol{x}$, while for a matrix $\boldsymbol{A}$, $\mathrm{diag}(\boldsymbol{A}) \in \mathbb{R}^V$ denotes the column vector whose elements coincide with the diagonal entries of $\boldsymbol{A}$. For a random variable $\boldsymbol{w}$, $\mathbb{E}_{\boldsymbol{w}}[\cdot]$ denotes taking expectation with respect to $\boldsymbol{w}$ and keeping the remaining independent terms fixed. Similarly, we use $\mathbb{E}[\cdot|\boldsymbol{w}]$ for conditional expectation, conditioned on $\boldsymbol{w}$. We use $\mathbb{1}_{\mathrm{Event}}$ as an indicator function, which takes values $\{0, 1\}$ depending on the event holds or not. We use $C$ to denote any constant in the upper-bound, which might depend on $\phi$. We use $\mathrm{poly}_{p,q}(N, d, V, L)$ denotes a polynomial function of $(N, d, V, L)$ whose degree depends on $(p, q)$ polynomially. For vectors $\boldsymbol{w}, \hat{\boldsymbol{w}}$ and a scaler variable $\eta > 0$, we use $\hat{\boldsymbol{w}} = \hat{\boldsymbol{w}} + O(\eta)$ to denote $\|\hat{\boldsymbol{w}} - \boldsymbol{w}\|_\infty = O(\eta)$.

Since we do not use positional encoding in the model, without loss of generality we can fix the informative index $\ell = 1$. We define the sequence of non-informative tokens as $\boldsymbol{N}_i := [\boldsymbol{x}_{i,2}, \cdots, \boldsymbol{x}_{i,L}]^\top$. We will denote the rows of $\boldsymbol{W}_{\mathrm{in}}$ with $\{\boldsymbol{w}_k\}_{k=1}^m$. For compact representation the attention with the trigger, we define

$$\mathsf{Z}_{\mathrm{in}} =: \begin{bmatrix} \boldsymbol{Z}_{\mathrm{in}} & \boldsymbol{z}_{\mathrm{trig}} \end{bmatrix} \text{ and } \mathsf{X}_i =: \begin{bmatrix} \boldsymbol{x}_{i,1}^\top & 1 \\ \boldsymbol{N}_i & 0 \end{bmatrix} \in \mathbb{R}^{L \times (V+1)}$$

With this notation, we can write the iterates in three-step GD. Let

$$\hat{\boldsymbol{p}}_{t,i} := \hat{\boldsymbol{p}}(\boldsymbol{X}_i; \boldsymbol{V}^{(t)}, \boldsymbol{W}_{\mathrm{KQ}}^{(0)}), \quad \text{and} \quad \boldsymbol{\alpha}_{0,i} := \sigma\Big(\mathsf{X}_i \mathsf{Z}_{\mathrm{in}}^\top \boldsymbol{W}_{\mathrm{KQ}}^{(0)} \boldsymbol{z}_{\mathrm{EOS}}\Big).$$

We have

$$\boldsymbol{V}^{(1)} = \boldsymbol{Z}_{\mathrm{out}}\Big(\frac{\eta}{N}\sum_{i=1}^N (\boldsymbol{p}_i - \hat{\boldsymbol{p}}_{0,i})\phi\big(\boldsymbol{\alpha}_{0,i}^\top \boldsymbol{X}_i \boldsymbol{Z}_{\mathrm{in}}^\top \boldsymbol{W}_{\mathrm{in}}^\top\big)\Big)$$

$$\boldsymbol{W}_{\mathrm{KQ}}^{(1)} = \mathsf{Z}_{\mathrm{in}} \frac{\gamma}{N}\sum_{i=1}^N \mathsf{X}_i^\top\big(\mathrm{diag}(\boldsymbol{\alpha}_{0,i}) - \boldsymbol{\alpha}_{0,i}\boldsymbol{\alpha}_{0,i}^\top\big)\boldsymbol{X}_i \boldsymbol{Z}_{\mathrm{in}}^\top \boldsymbol{W}_{\mathrm{in}}^\top$$

$$\times \mathrm{diag}\Big(\phi'\big(\boldsymbol{W}_{\mathrm{in}}\boldsymbol{Z}_{\mathrm{in}}\boldsymbol{X}_i^\top \boldsymbol{\alpha}_{0,i}\big)\Big)(\boldsymbol{V}^{(1)})^\top \boldsymbol{Z}_{\mathrm{out}}(\boldsymbol{p}_i - \hat{\boldsymbol{p}}_{i,1})\boldsymbol{z}_{\mathrm{EOS}}^\top. \quad \text{(B.1)}$$

For notational convenience, we define the noise due to finite width as (which we defined equivalently in (B.11))

$$\mathrm{FW}(\boldsymbol{W}_{\mathrm{in}}; \boldsymbol{Z}_{\mathrm{in}}, \boldsymbol{X}_i, \boldsymbol{X}_j) := \frac{1}{m}\Big(\boldsymbol{W}_{\mathrm{in}}^\top \mathrm{diag}\Big(\phi'\big(\tfrac{1}{L}\boldsymbol{W}_{\mathrm{in}}\boldsymbol{Z}_{\mathrm{in}}\boldsymbol{X}_i^\top \mathbb{1}_L\big)\Big)\phi\big(\tfrac{1}{L}\boldsymbol{W}_{\mathrm{in}}\boldsymbol{Z}_{\mathrm{in}}\boldsymbol{X}_j^\top \mathbb{1}_L\big)$$

$$- \mathbb{E}_{\boldsymbol{W}_{\mathrm{in}}}\Big[\boldsymbol{W}_{\mathrm{in}}^\top \mathrm{diag}\Big(\phi'\big(\tfrac{1}{L}\boldsymbol{W}_{\mathrm{in}}\boldsymbol{Z}_{\mathrm{in}}\boldsymbol{X}_i^\top \mathbb{1}_L\big)\Big)\phi\big(\tfrac{1}{L}\boldsymbol{W}_{\mathrm{in}}\boldsymbol{Z}_{\mathrm{in}}\boldsymbol{X}_j^\top \mathbb{1}_L\big)\Big]\Big).$$
(B.2)

For the terms arising in the expected value term in (B.2), we define

- $\alpha_{ij} := \mathbb{E}_{\boldsymbol{w}}\Big[\phi'\big(\tfrac{1}{L}\boldsymbol{w}^\top \boldsymbol{Z}_{\mathrm{in}}\boldsymbol{X}_i^\top \mathbb{1}_L\big)\phi'\big(\tfrac{1}{L}\boldsymbol{w}^\top \boldsymbol{Z}_{\mathrm{in}}\boldsymbol{X}_j^\top \mathbb{1}_L\big)\Big]$,

- $\beta_{ij} := \mathbb{E}_{\boldsymbol{w}}\Big[\phi''\big(\tfrac{1}{L}\boldsymbol{w}^\top \boldsymbol{Z}_{\mathrm{in}}\boldsymbol{X}_i^\top \mathbb{1}_L\big)\phi\big(\tfrac{1}{L}\boldsymbol{w}^\top \boldsymbol{Z}_{\mathrm{in}}\boldsymbol{X}_j^\top \mathbb{1}_L\big)\Big]$.

Moreover, we make the following definitions to simplify the notation in the following:

$$\boldsymbol{A}_{1,ir} := \boldsymbol{Z}_{\mathrm{in}}\Big(\frac{1}{LN}\sum_{j=1}^N \alpha_{ij}(\boldsymbol{x}_j - \tfrac{1}{V}\mathbb{1}_V)(\boldsymbol{x}_j - \tfrac{1}{V}\mathbb{1}_V)^\top\Big)$$

$$\times \Big(\frac{1}{LN}\sum_{j=1}^N \alpha_{rj}(\boldsymbol{x}_j - \tfrac{1}{V}\mathbb{1}_V)(\boldsymbol{x}_j - \tfrac{1}{V}\mathbb{1}_V)^\top\Big)\boldsymbol{Z}_{\mathrm{in}}^\top \quad \text{(B.3)}$$

$$\boldsymbol{A}_{2,ir} := \boldsymbol{Z}_{\mathrm{in}}\Big(\frac{1}{LN}\sum_{j=1}^N \alpha_{ij}(\boldsymbol{N}_j^\top - \tfrac{1}{V}\mathbb{1}_V \mathbb{1}_{L-1}^\top)\mathbb{1}_{L-1}(\boldsymbol{x}_j - \tfrac{1}{V}\mathbb{1}_V)^\top\Big)$$

$$\times \Big(\frac{1}{LN}\sum_{j=1}^N \alpha_{rj}(\boldsymbol{x}_j - \tfrac{1}{V}\mathbb{1}_V)\mathbb{1}_{L-1}^\top(\boldsymbol{N}_j^\top - \tfrac{1}{V}\mathbb{1}_V \mathbb{1}_{L-1}^\top)^\top\Big)\boldsymbol{Z}_{\mathrm{in}}^\top$$
(B.4)

$$\boldsymbol{A}_{3,ir} := \frac{1}{L^2 V^2}\Big(\frac{1}{N}\sum_{j=1}^N \alpha_{ij}(\boldsymbol{x}_j - \tfrac{1}{V}\mathbb{1}_V)\Big)^\top\Big(\frac{1}{N}\sum_{j=1}^N \alpha_{rj}(\boldsymbol{x}_j - \tfrac{1}{V}\mathbb{1}_V)\Big)\boldsymbol{Z}_{\mathrm{in}}\mathbb{1}_V \mathbb{1}_V^\top \boldsymbol{Z}_{\mathrm{in}}^\top$$

and

$$\boldsymbol{S}_1 := \Big(\frac{1}{LN}\sum_{j=1}^N (\boldsymbol{x}_j - \tfrac{1}{V}\mathbb{1}_V)(\boldsymbol{x}_j - \tfrac{1}{V}\mathbb{1}_V)^\top\Big)\Big(\frac{1}{LN}\sum_{j=1}^N (\boldsymbol{x}_j - \tfrac{1}{V}\mathbb{1}_V)(\boldsymbol{x}_j - \tfrac{1}{V}\mathbb{1}_V)^\top\Big) \quad \text{(B.5)}$$

$$\boldsymbol{S}_2 := \Big(\frac{1}{LN}\sum_{j=1}^N (\boldsymbol{N}_j^\top - \tfrac{1}{V}\mathbb{1}_V \mathbb{1}_{L-1}^\top)\mathbb{1}_{L-1}(\boldsymbol{x}_j - \tfrac{1}{V}\mathbb{1}_V)^\top\Big)$$

$$\times \Big(\frac{1}{LN}\sum_{j=1}^N (\boldsymbol{x}_j - \tfrac{1}{V}\mathbb{1}_V)\mathbb{1}_{L-1}^\top(\boldsymbol{N}_j^\top - \tfrac{1}{V}\mathbb{1}_V \mathbb{1}_{L-1}^\top)^\top\Big) \quad \text{(B.6)}$$

$$\boldsymbol{S}_3 := \frac{1}{L^2 V^2}\Big(\frac{1}{N}\sum_{j=1}^N (\boldsymbol{x}_j - \tfrac{1}{V}\mathbb{1}_V)\Big)^\top\Big(\frac{1}{N}\sum_{j=1}^N (\boldsymbol{x}_j - \tfrac{1}{V}\mathbb{1}_V)\Big)\mathbb{1}_V \mathbb{1}_V^\top. \quad \text{(B.7)}$$

### B.1 PROOF OVERVIEW

In this part, we consider (5.2)-(5.3) with empirical loss and non-orthogonal embeddings. For notational convenience, without loss of generality, we assume $\ell = 1$ and $\mathbf{\Pi}_* = \boldsymbol{I}_V$ (accordingly, $\boldsymbol{p}_i = \boldsymbol{x}_{i,1}$). Our goal is to show how the Signal, Gradient noise, Mean bias, and MLP noise terms arise from the dynamics of the first gradient step.

The analysis proceeds in two steps. First, we show that the first iterate of the output-layer weights $\boldsymbol{V}^{(1)}$ admits a natural decomposition into mean, bias, and noise components. We then show how this decomposition gives rise to the terms appearing in Theorem 1.

#### B.1.1 DECOMPOSITION OF THE VALUE MATRIX

Both the informative and non-informative terms depend on $\boldsymbol{V}^{(1)}$. We show that it can be decomposed as

$$\boldsymbol{V}^{(1)} = \boldsymbol{Z}_{\text{out}}\Big(\frac{1}{NL}\sum_{i=1}^{N}(\boldsymbol{x}_{i,1} - \tfrac{1}{V}\mathbb{1}_V)(\boldsymbol{X}_i\mathbb{1}_L)^\top\Big)\boldsymbol{Z}_{\text{in}}^\top \tag{B.8}$$

$$= \boldsymbol{Z}_{\text{out}}\Big(\underbrace{\frac{1}{VL}(\boldsymbol{I}_V - \tfrac{1}{V}\mathbb{1}_V\mathbb{1}_V^\top)}_{\text{Mean}} + \underbrace{\frac{1}{VN}\sum_{i=1}^{N}(\boldsymbol{x}_{i,1} - \tfrac{1}{V}\mathbb{1}_V)\mathbb{1}_V^\top}_{\text{Bias}} + \underbrace{\frac{1}{\sqrt{LVN}}\boldsymbol{\Xi}}_{\text{Noise}}\Big)\boldsymbol{Z}_{\text{in}}^\top \tag{B.9}$$

where the noise term is given by

$$\boldsymbol{\Xi} := \sqrt{\frac{V}{LN}}\Big(\sum_{i=1}^{N}(\boldsymbol{x}_{i,1} - \tfrac{1}{V}\mathbb{1}_V)(\boldsymbol{X}_i\mathbb{1}_L - \tfrac{L}{V}\mathbb{1}_V)^\top - \tfrac{1}{V}(\boldsymbol{I}_V - \tfrac{1}{V}\mathbb{1}_V\mathbb{1}_V^\top)\Big).$$

Here, the bias term arises from aggregating tokens at initialization; specifically, the aggregate-token averages $\frac{1}{L}\boldsymbol{X}_i\mathbb{1}_L$ in (B.8) concentrate around their mean $\frac{1}{V}\mathbb{1}_V$ as $L$ grows, and this effect appears as the bias term. The noise term captures finite-sample fluctuations of tokens around this mean. In (B.9), we explicitly factor out the typical operator-norm scaling $1/\sqrt{VLN}$ from the noise term so that the remaining matrix $\boldsymbol{\Xi}$ has constant norm on average, i.e., $\mathbb{E}[\|\boldsymbol{\Xi}\|_2^2] \asymp 1$.

#### B.1.2 CHARACTERIZATION OF NOISE TERMS

**Signal.** Using the mean component in (B.9), the informative term in (5.2) can be written as

$$\text{Informative} = \gamma\|\boldsymbol{z}_{\text{trig}}\|_2^2\boldsymbol{e}_1\Big(\frac{1}{NL}\sum_{i=1}^{N}\boldsymbol{x}_{i,1}^\top\boldsymbol{Z}_{\text{in}}^\top(\boldsymbol{V}^{(1)})^\top\boldsymbol{Z}_{\text{out}}(\boldsymbol{p}_i - \tfrac{1}{V}\mathbb{1}_V)\Big)$$

$$= \underbrace{\frac{\gamma\|\boldsymbol{z}_{\text{trig}}\|_2^2}{VL^2}\frac{1}{N}\sum_{i=1}^{N}\boldsymbol{x}_{i,1}^\top\boldsymbol{Z}_{\text{in}}^\top\boldsymbol{Z}_{\text{in}}(\boldsymbol{I}_V - \tfrac{1}{V}\mathbb{1}_V\mathbb{1}_V^\top)\boldsymbol{Z}_{\text{out}}^\top\boldsymbol{Z}_{\text{out}}(\boldsymbol{x}_{i,1} - \tfrac{1}{V}\mathbb{1}_V)}_{\text{Signal} \asymp \frac{1}{VL^2}}$$

$$+ \underbrace{\text{negligible terms}}_{= o(\frac{1}{VL^2})}.$$

The first term is due to the mean component; the negligible terms are due to the bias and noise in (B.9). Standard concentration arguments for Gaussian matrices can be used to show that the leading term scales as $\frac{1}{VL^2}$, which gives us the *Signal* term in (3.1). The detailed derivations are provided in Section D.1.1.

**Gradient Noise and Mean Bias.** For ease of presentation, we focus on the large-$L$ regime where we can use the following approximation due to concentration

$$\frac{1}{L}\boldsymbol{Z}_{\text{in}}\boldsymbol{X}_i\boldsymbol{X}_i^\top\boldsymbol{Z}_{\text{in}}^\top \approx \frac{1}{d}\boldsymbol{I}_d. \tag{B.10}$$

Let $\boldsymbol{x}_{\text{in}}$ denote an arbitrary row of $\boldsymbol{X}_{\text{in}}$. Using (B.10), we can approximate the non-informative with

$$\text{Non-informative} \approx \underbrace{\frac{1}{d\sqrt{LNV}}\boldsymbol{x}_{\text{in}}^{\top}\boldsymbol{Z}_{\text{in}}^{\top}\boldsymbol{Z}_{\text{in}}\boldsymbol{\Xi}\boldsymbol{Z}_{\text{out}}^{\top}\boldsymbol{Z}_{\text{out}}\frac{1}{N}\sum_{i=1}^{N}(\boldsymbol{x}_{i,1} - \frac{1}{V}\mathbb{1}_V)}_{\text{Gradient noise}}$$

$$+ \underbrace{\frac{1}{Vd}\boldsymbol{x}_{\text{in}}^{\top}\boldsymbol{Z}_{\text{in}}^{\top}\boldsymbol{Z}_{\text{in}}\mathbb{1}_V \Big\|\boldsymbol{Z}_{\text{out}}\frac{1}{N}\sum_{i=1}^{N}(\boldsymbol{x}_{i,1} - \frac{1}{V}\mathbb{1}_V)\Big\|_2^2}_{\text{Mean bias}} + \text{negligible terms.}$$

The first term arises from the noise component $\boldsymbol{\Xi}$ and determines the scaling of the *Gradient noise* term. The second term comes from the bias component and yields the *Mean bias* term in (3.1). We hide the contributions from the mean component in the negligible terms, since they are smaller in magnitude. The fluctuations of each term can be bounded as stated in Theorem 1 using standard concentration arguments. The detailed derivations are provided in Section D.1.2.

**MLP noise.** In this part, we consider the *Attention-MLP* model. The scores in (5.1) can be defined in the same way for this case as well. Let $\{\boldsymbol{w}_k\}_{k=1}^m$ denote the rows of $\boldsymbol{W}_{\text{in}}$, where $\boldsymbol{w}_k \sim \mathcal{N}(0, \boldsymbol{I}_d)$. For illustration, we work in the large-$L$ regime and adopt the approximation in (B.10).

We define the MLP-noise term as the deviation of the scores from their expectation with respect to the randomness in $\boldsymbol{W}_{\text{in}}$:

$$\text{MLP-noise} := \text{scores} - \mathbb{E}_{\boldsymbol{W}_{\text{in}}}[\text{scores}].$$

Under the large-$L$ assumption in (B.10), the scores admit the approximation (see (C.1) for the full form)

$$\text{MLP-noise} \approx \boldsymbol{x}_{\text{in}}^{\top}\boldsymbol{Z}_{\text{in}}^{\top}\frac{1}{N^2 d}\sum_{i,j=1}^{N}\text{FW}(\boldsymbol{W}_{\text{in}}; \boldsymbol{Z}_{\text{in}}, \boldsymbol{X}_i, \boldsymbol{X}_j)(\boldsymbol{x}_{i,1} - \frac{1}{V}\mathbb{1}_V)\boldsymbol{Z}_{\text{out}}^{\top}\boldsymbol{Z}_{\text{out}}(\boldsymbol{x}_{j,1} - \frac{1}{V}\mathbb{1}_V).$$

Here $\text{FW}(\boldsymbol{W}_{\text{in}}; \boldsymbol{Z}_{\text{in}}, \boldsymbol{X}_i, \boldsymbol{X}_j)$ denotes the noise induced by the finite width of $\boldsymbol{W}_{\text{in}}$, defined as

$$\text{FW}(\boldsymbol{W}_{\text{in}}; \boldsymbol{Z}_{\text{in}}, \boldsymbol{X}_i, \boldsymbol{X}_j) := \frac{1}{m}\sum_{k=1}^{m}\boldsymbol{w}_k\phi'\Big(\frac{1}{L}\boldsymbol{w}_k^{\top}\boldsymbol{Z}_{\text{in}}\boldsymbol{X}_i\mathbb{1}_L\Big)\phi\Big(\frac{1}{L}\boldsymbol{w}_k^{\top}\boldsymbol{Z}_{\text{in}}\boldsymbol{X}_j\mathbb{1}_L\Big)$$

$$- \mathbb{E}\Big[\boldsymbol{w}_k\phi'\Big(\frac{1}{L}\boldsymbol{w}_k^{\top}\boldsymbol{Z}_{\text{in}}\boldsymbol{X}_i\mathbb{1}_L\Big)\phi\Big(\frac{1}{L}\boldsymbol{w}_k^{\top}\boldsymbol{Z}_{\text{in}}\boldsymbol{X}_j\mathbb{1}_L\Big)\Big]. \qquad (B.11)$$

For large $L$, standard concentration arguments imply that $\big\|\frac{1}{L}\boldsymbol{w}_k^{\top}\boldsymbol{Z}_{\text{in}}\boldsymbol{X}_i\mathbb{1}_L\big\|_2 \approx L^{-1/2} \to 0$. Hence

$$\phi'\Big(\frac{1}{L}\boldsymbol{w}_k^{\top}\boldsymbol{Z}_{\text{in}}\boldsymbol{X}_i\mathbb{1}_L\Big)\phi\Big(\frac{1}{L}\boldsymbol{w}_k^{\top}\boldsymbol{Z}_{\text{in}}\boldsymbol{X}_j\mathbb{1}_L\Big) \to \underbrace{\phi(0)\phi'(0)}_{\text{nonzero constant}},$$

where Assumption 4 ensures $\phi(0)\phi'(0) \neq 0$. Since $\mathbb{E}[\boldsymbol{w}_k] = 0$, replacing the $\phi$-dependent factors by this constant yields, we have

$$\text{FW}(\boldsymbol{W}_{\text{in}}; \boldsymbol{Z}_{\text{in}}, \boldsymbol{X}_i, \boldsymbol{X}_j) \to \frac{\phi(0)\phi'(0)}{m}\sum_{k=1}^{m}\boldsymbol{w}_k.$$

Substituting this into the expression above gives

$$\text{MLP noise} \approx \frac{\phi(0)\phi'(0)}{dm}\underbrace{\sum_{k=1}^{m}\boldsymbol{x}_{\text{input}}^{\top}\boldsymbol{Z}_{\text{in}}^{\top}\boldsymbol{w}_k}_{\asymp \tilde{O}\left(\frac{1}{d\sqrt{m}}\right)}\underbrace{\Big\|\boldsymbol{Z}_{\text{out}}\frac{1}{N}\sum_{i=1}^{N}(\boldsymbol{x}_{i,1} - \frac{1}{V}\mathbb{1}_V)\Big\|_2^2}_{\asymp \tilde{O}\left(\frac{1}{N}\right)}.$$

Here, the terms can be bounded as in the displayed equation using standard concentration arguments, which yield the scaling of the *MLP noise* term in (3.1). The detailed derivations are provided in Section D.3.

## B.2 Preliminary Results: Characterization of Good Events

We start with characterizing "good events" which we will use in the proof of Theorem 1.

**Lemma 1.** *We consider $l \in \mathbb{N}$, and $V^3 \gg N \gg V \gg L$ and $L \asymp V^{\epsilon_1}$, and $d \asymp V^{\epsilon_2}$ for some $\epsilon_1, \epsilon_2 \in (0,1)$. For the following we define, $m_{ij} := (1 - 1/V)\delta_{ij} + \frac{L}{V}$. We define the following events:*

*(E1) Let $\boldsymbol{z}_k := \boldsymbol{Z}_{\text{in}}\boldsymbol{e}_k$ and $\mathsf{z}_k := (\boldsymbol{z}_k + \mathbb{1}_{l=1}\mathsf{z}_{\text{trig}})$. We have*

*(E1.1) $\frac{1}{V}\|\boldsymbol{Z}_{\text{in}}\boldsymbol{Z}_{\text{in}}^\top\|_2 \leq \frac{2}{d}$ and $\max_{k \leq V}\|\boldsymbol{z}_k\|_2 \vee \|\mathsf{z}_{\text{trig}}\|_2 \leq 2$ and $\max_{j \neq k}|\langle \boldsymbol{z}_j, \mathsf{z}_k\rangle| \leq \frac{\log V}{\sqrt{d}}$.*

*(E1.2) $\frac{1}{\sqrt{V}}\|\boldsymbol{Z}_{\text{in}}\mathbb{1}_V\|_2 \leq 2$ and $\frac{1}{\sqrt{V}}\|\boldsymbol{Z}_{\text{in}}^\top\boldsymbol{Z}_{\text{in}}\mathbb{1}_V\|_\infty \leq \frac{\log V}{\sqrt{d}}$*

*(E1.3) $\left|\mathsf{z}_k^\top\boldsymbol{Z}_{\text{in}}\mathbb{1}_V\right| \leq 2\log V\sqrt{\frac{V}{d}}$ and $\left|\mathsf{z}_k^\top\boldsymbol{Z}_{\text{in}}\boldsymbol{Z}_{\text{in}}^\top\boldsymbol{Z}_{\text{in}}\mathbb{1}_V\right| \leq C_K\log V\left(\frac{V}{d}\right)^{\frac{3}{2}}$ and $\left|\mathsf{z}_k^\top\boldsymbol{Z}_{\text{in}}\text{diag}\left(\boldsymbol{Z}_{\text{in}}^\top\boldsymbol{Z}_{\text{in}}\right)\right| \leq C_K\log V\sqrt{\frac{V}{d}}$*

*(E1.4) For all $i \in [N]$, $|\mathsf{z}_k^\top\boldsymbol{Z}_{\text{in}}\boldsymbol{X}_i^\top\mathbb{1}_L| \leq \boldsymbol{e}_k^\top\boldsymbol{X}_i^\top\mathbb{1}_L + C_K\log V\frac{\|\boldsymbol{X}_i^\top\mathbb{1}_L\|_2}{\sqrt{d}}$*

*(E1.5) For all $i \in [N]$, $|\mathbb{1}_V^\top\boldsymbol{Z}_{\text{in}}^\top\boldsymbol{Z}_{\text{in}}\boldsymbol{X}_i^\top\mathbb{1}_L| \leq L + C_K\log V\|\boldsymbol{X}_i^\top\mathbb{1}_L\|_2\sqrt{\frac{V}{d}}$.*

*(E1.6) For all $i \in [N]$, $\left|\mathsf{z}_k^\top\boldsymbol{Z}_{\text{in}}\boldsymbol{Z}_{\text{in}}^\top\boldsymbol{Z}_{\text{in}}\boldsymbol{X}_i^\top\mathbb{1}_L\right| \leq \frac{2V}{d}\left(\boldsymbol{e}_k^\top\boldsymbol{X}_i^\top\mathbb{1}_L + C_K\log V\frac{\|\boldsymbol{X}_i^\top\mathbb{1}_L\|_2}{\sqrt{d}}\right)$.*

*(E1.7) For all $i, j \in [N]$, $\left|\frac{1}{L}\mathbb{1}_L^\top\boldsymbol{X}_j\boldsymbol{Z}_{\text{in}}^\top\boldsymbol{Z}_{\text{in}}\boldsymbol{X}_i^\top\mathbb{1}_L - m_{ij}\right| \leq \left|\frac{1}{L}\mathbb{1}_L\boldsymbol{X}_j^\top\boldsymbol{X}_i^\top\mathbb{1}_L - m_{ij}\right| + C_K\frac{\|\boldsymbol{X}_i^\top\mathbb{1}_L\|_2\|\boldsymbol{X}_j^\top\mathbb{1}_L\|_2}{L}\frac{\log V}{\sqrt{d}}$*

*(E1.8) For all $i \in [N]$, $\|\boldsymbol{Z}_{\text{in}}\boldsymbol{N}_i^\top\boldsymbol{N}_i\boldsymbol{Z}_{\text{in}}^\top\mathsf{z}_k\|_2 \leq C_K\left(\boldsymbol{e}_k^\top\boldsymbol{N}_i^\top\mathbb{1}_{L-1} + \frac{L}{d} + \log^6 V\frac{\|\boldsymbol{N}_i^\top\mathbb{1}_{L-1}\|_2}{\sqrt{d}}\right)$.*

*(E2) We have*

*(E2.1) For all $i, j \in [N]$, $|\frac{1}{L}\mathbb{1}_L^\top\boldsymbol{X}_j\boldsymbol{X}_i^\top\mathbb{1}_L - m_{ij}| \leq C_K\frac{\log^2 V}{\sqrt{V \wedge L}}$,*

*(E2.2) For all $i \in [N]$, $\|\boldsymbol{X}_i^\top\mathbb{1}_L\|_\infty \leq \log L$ and $\|\boldsymbol{X}_i^\top\mathbb{1}_L\|_0 \geq \frac{L}{2}$*

*(E2.3) $\left|\|\frac{1}{N}\sum_{i=1}^N\boldsymbol{x}_i\|_2 - \frac{1}{N} - \frac{1}{V}\right| \leq C_K\frac{\log^2 N}{N\sqrt{V}}$ and $\left|\|\frac{1}{N}\sum_{i=1}^N\boldsymbol{x}_i - \frac{1}{V}\mathbb{1}_V\|_2 - \frac{1}{N}\right| \leq C_K\frac{\log^2 N}{N\sqrt{V}}$ and $\|\frac{1}{N}\sum_{i=1}^N\boldsymbol{x}_i - \frac{1}{V}\mathbb{1}_V\|_\infty \leq \frac{(e+1)}{V}$*

*(E2.4) $\sum_{i,j=1}^N|\mathbb{1}_{\boldsymbol{x}_i=\boldsymbol{x}_j} - \frac{1}{V}| \leq \frac{4N^2}{V}$ and $\sum_{i,j=1}^N(\mathbb{1}_{\boldsymbol{x}_i=\boldsymbol{x}_j} - \frac{1}{V}) \leq \frac{4N^2}{V}$ and for any $k \in [V]$, $\left|\sum_{i,j=1}^N|\mathbb{1}_{\boldsymbol{x}_j=\boldsymbol{e}_k} - \frac{1}{V}|(\mathbb{1}_{\boldsymbol{x}_i=\boldsymbol{e}_k} - \frac{1}{V})\right| \leq \frac{CN^2}{V^2}$.*

*(E2.5) $\|\boldsymbol{S}_1\|_2 \leq \frac{e}{L^2V^2}$ and $|\text{tr}(\boldsymbol{S}_1) - (1 - \frac{1}{V})\frac{1}{L^2}\left(\frac{1}{N} + \frac{1}{V}\right)| \leq \frac{C_K\log^2 V}{L^2N\sqrt{V}}$*

*(E2.6) $\|\boldsymbol{S}_2\|_2 \leq \frac{C_K\log^2 V}{NLV}$ and $|\text{tr}(\boldsymbol{S}_2) - (1 - \frac{1}{V})^2\frac{L-1}{L^2N}| \leq \frac{C_K\log^3 V}{N\sqrt{LV}}$*

*(E2.7) $\frac{-C_K\log^2 V}{N\sqrt{V}}\frac{1}{V^2L^2}\mathbb{1}_V\mathbb{1}_V^\top \preceq \boldsymbol{S}_3 - \frac{1}{N}\frac{1}{V^2L^2}\mathbb{1}_V\mathbb{1}_V^\top \preceq \frac{C_K\log^2 V}{N\sqrt{V}}\frac{1}{V^2L^2}\mathbb{1}_V\mathbb{1}_V^\top$*

*(E2.8) $\left\|\frac{1}{NL}\sum_{j=1}^N(\boldsymbol{N}_j^\top - \frac{1}{V}\mathbb{1}_{L-1})\mathbb{1}_{L-1}\mathbb{1}_{L-1}^\top(\boldsymbol{N}_j^\top - \frac{1}{V}\mathbb{1}_{L-1})^\top\right\|_2 = \frac{1}{V} \pm + \frac{C_K\log^2 V}{\sqrt{NV}}$.*

*For any $K > 0$, there exists a universal constant $C_K > 0$ depending only on $K$ such that*

$$\mathbb{P}[(E1)|\{\boldsymbol{X}_i\}_{i=1}^N] \geq 1 - \frac{1}{V^K} \text{ and } \mathbb{P}[(E2)] \geq 1 - \frac{1}{V^K}.$$

*Proof.* For (E1):

- By Proposition 4, we have $\|\frac{1}{V}\boldsymbol{Z}_{\text{in}}\boldsymbol{Z}_{\text{in}}^\top - \frac{1}{d}\boldsymbol{I}_d\|_2 \leq \frac{2\log V}{\sqrt{Vd}}$ and by Proposition 6, we have $\max_{k \leq V}\|\boldsymbol{z}_k\|_2 \vee \|\mathsf{z}_{\text{trig}}\|_2 \leq 2$ with probability at least $1 - CVd\exp(-c\log^2 V)$.

- By Proposition 6, $\frac{1}{\sqrt{V}}\|\boldsymbol{Z}_{\text{in}}\mathbb{1}_V\|_2 \leq 2$ and $\frac{1}{\sqrt{V}}\|\boldsymbol{Z}_{\text{in}}^\top\boldsymbol{Z}_{\text{in}}\mathbb{1}_V\|_\infty \leq \frac{2\log V}{\sqrt{d}}$ with probability at least $1 - CVd\exp(-c\log^2 V)$.

- By Propositions 6 and 7, we have $\frac{1}{\sqrt{V}}\left|z_k^\top Z_{\text{in}} \mathbb{1}_V\right| \leq \frac{2\log V}{\sqrt{d}}$ and $\left|z_k^\top Z_{\text{in}} Z_{\text{in}}^\top Z_{\text{in}} \mathbb{1}_V\right| \leq C_K \log V \left(\frac{V}{d}\right)^{\frac{3}{2}}$ with probability at least $1 - CVd\exp(-c\log^2 V)$. Moreover

$$\frac{1}{V}\left|z_k^\top Z_{\text{in}}\text{diag}\left(Z_{\text{in}}^\top Z_{\text{in}}\right)\right| = \frac{1}{V}\sum_{\substack{i=1\\i\neq k}}^{V}\|z_i\|_2^2\langle z_i, z_k\rangle + \frac{\mathbb{1}_{l=1}}{V}\sum_{\substack{i=1\\i\neq k}}^{V}\|z_i\|_2^2\langle z_i, z_{\text{trig}}\rangle + \underbrace{\frac{1}{V}z_k^\top z_k}_{\in\frac{1}{V}[-C_K, C_K]},$$

where we used previous items to bound the last term. For $i \neq k$, by using Lemma 3, we have for $p \leq \frac{d}{6}$,

$$\mathbb{E}[\|z_i\|_2^{4p}|\langle z_i, z_k\rangle|^{2p}] \leq d^{-p}\,\mathbb{E}[\|z_i\|_2^{6p}](2p)^p \leq d^{-p}2^p p^p\frac{d(d+2)\cdots(d+6p-2)}{d^{3p}} \leq d^{-p}2^{4p}p^p.$$

Therefore,

$$\mathbb{E}[\|z_i\|_2^{4p}|\langle z_i, z_k\rangle|^{2p}]^{\frac{1}{2p}} \leq 4d^{-1/2}\sqrt{p}.$$

By Proposition 15, we have for $2 \leq p \leq \frac{d}{6}$,

$$\mathbb{E}\left[\left|\frac{1}{V}z_k^\top Z_{\text{in}}\text{diag}\left(Z_{\text{in}}^\top Z_{\text{in}}\right)\right|^{2p}\right]^{\frac{1}{2p}} \leq Cd^{-1/2}\left[\sqrt{\frac{p}{V}} + V^{\frac{1}{p}}\frac{p^{3/2}}{V}\right]$$

By using $p = \log V$, we have the bound in the statement with probability $1 - \frac{1}{V^K}$.

- By Proposition 6 with probability at least $1 - \frac{1}{V^K}$

$$|z_k^\top Z_{\text{in}}X_i^\top \mathbb{1}_L| \leq |e_k^\top Z_{\text{in}}^\top Z_{\text{in}}X_i^\top \mathbb{1}_L| + \mathbb{1}_{l=1}|z_{\text{trig}}^\top Z_{\text{in}}X_i^\top \mathbb{1}_L|$$

$$\leq e_k^\top X_i^\top \mathbb{1}_L + C_K\log V\frac{\|X_i^\top \mathbb{1}_L\|_2}{\sqrt{d}}.$$

By the union bound, the item follows.

- By Proposition 6 with probability at least $1 - \frac{1}{V^K}$,

$$|\mathbb{1}_V^\top Z_{\text{in}}^\top Z_{\text{in}}X_i^\top \mathbb{1}_L| \leq \mathbb{1}_V^\top X_i^\top \mathbb{1}_L + C_K\log V\|X_i^\top \mathbb{1}_L\|_2\sqrt{\frac{V}{d}}$$

$$= L + C_K\log V\|X_i^\top \mathbb{1}_L\|_2\sqrt{\frac{V}{d}}.$$

- By Proposition 7, with probability at least $1 - CN\exp(-c\log^2 V)$, we have $\left|z_k^\top Z_{\text{in}}Z_{\text{in}}^\top Z_{\text{in}}X_i^\top \mathbb{1}_L\right| \leq \frac{2V}{d}\left(e_k^\top X_i^\top \mathbb{1}_L + C_K\log V\frac{\|X_i^\top \mathbb{1}_L\|_2}{\sqrt{d}}\right)$ for all $i \in [N]$.

- By Proposition 6, with probability at least $1 - \frac{1}{V^K}$

$$\frac{1}{L}\mathbb{1}_L^\top X_j Z_{\text{in}}^\top Z_{\text{in}}X_i^\top \mathbb{1}_L - m_{ij} = \frac{1}{L}\mathbb{1}_L^\top X_j X_i^\top \mathbb{1}_L - m_{ij} \pm C_K\frac{\|X_j^\top \mathbb{1}_L\|_2\|X_i^\top \mathbb{1}_L\|_2}{L}\frac{\log V}{\sqrt{d}}.$$

- For the last item, let $n_k := \mathbb{1}_{L-1}^\top N_i e_k$. We have

$$Z_{\text{in}}N_i^\top N_i Z_{\text{in}}^\top z_k = n_k\left(\|z_k\|_2^2 + \mathbb{1}_{l=1}z_k^\top z_{\text{trig}} - \frac{1}{d}\right)z_k + \frac{L}{d}z_k + \sum_{\substack{j=1\\j\neq k}}^{V}n_j\left(z_j z_j^\top - \frac{1}{d}I_d\right)z_k.$$

By Proposition 12, we have

$$\mathbb{E}\left[\left\|\sum_{\substack{j=1\\j\neq k}}^{V}n_j\left(z_j z_j^\top - \frac{1}{d}I_d\right)z_k\right\|_2^{2p}\right]^{\frac{1}{p}} \leq C(p-1)^6\,\mathbb{E}\left[\left\|\sum_{\substack{j=1\\j\neq k}}^{V}n_j\left(z_j z_j^\top - \frac{1}{d}I_d\right)z_k\right\|_2^2\right]$$

$$\leq \frac{C}{d}(p-1)^6 \|\boldsymbol{N}_i^\top \mathbb{1}_{L-1}\|_2^2.$$

Therefore, with probability $1 - \frac{1}{V^K}$, we have

$$\|\boldsymbol{Z}_{\text{in}}\boldsymbol{N}_i^\top \boldsymbol{N}_i \boldsymbol{Z}_{\text{in}}^\top \boldsymbol{z}_k\|_2 \leq C_K\Big(n_k + \frac{L}{d} + \log^6 V \frac{\|\boldsymbol{N}_i^\top \mathbb{1}_{L-1}\|_2}{\sqrt{d}}\Big).$$

For (E2):

- By Proposition 9, we have the first item with probability $1 - \frac{N^2}{V^K}$.

- By Corollary 3, we have $\max_{i \in [N]} \|\boldsymbol{X}_i^\top \mathbb{1}_L\|_\infty \leq \log L$ with probability $1 - \frac{N}{V^K}$ for large enough $L$. For the second part, we define $n_k := \mathbb{1}_L^\top \boldsymbol{X}_i \boldsymbol{e}_k$. We observe that

$$\mathbb{E}[\|\boldsymbol{X}_i^\top \mathbb{1}_L\|_0] = \sum_{k=1}^V \mathbb{P}[n_k > 0] = V\Big(1 - (1 - \frac{1}{V})^L\Big) = L\Big(1 - \frac{L}{2V} + o(L/V)\Big).$$

By McDiarmid inequality, we have

$$\mathbb{P}\Big[\Big|\|\boldsymbol{X}_i^\top \mathbb{1}_L\|_0 - L\Big(1 - \frac{L}{2V} + o(L/V)\Big)\Big| > \sqrt{L}\log V\Big] \leq 2\exp(-2\log^2 V),$$

which gives the result.

- Let $\boldsymbol{n} := \sum_{i=1}^N \boldsymbol{x}_i$, where $\mathbb{E}[\boldsymbol{n}] = \frac{N}{V}\mathbb{1}_V$. By Proposition 9 with probability $1 - \frac{1}{V^K}$, we have

$$\Big|\|\frac{1}{N}\boldsymbol{n} - \frac{1}{V}\mathbb{1}_V\|_2^2 - (1 - \frac{1}{V})\frac{1}{N}\Big| = \Big|\|\frac{1}{N}\boldsymbol{n}\|_2^2 - (1 - \frac{1}{V})\frac{1}{N} - \frac{1}{V}\Big| \leq C_K \frac{\log^2 V}{N\sqrt{V}}.$$

Lastly, by Corollary 3, we have $\|\frac{1}{N}\boldsymbol{n} - \frac{1}{V}\mathbb{1}_V\|_\infty \leq \frac{(e+1)}{V}$.

- We have

$$\sum_{i,j=1}^N |\mathbb{1}_{\boldsymbol{x}_i=\boldsymbol{x}_j} - \frac{1}{V}| = \Big(\sum_{i,j=1}^N |\mathbb{1}_{\boldsymbol{x}_i=\boldsymbol{x}_j} - \frac{1}{V}| - \frac{2}{V}(1 - \frac{1}{V})\Big) + \frac{2N^2}{V}(1 - \frac{1}{V})$$

$$= (1 - \frac{2}{V})\sum_{i,j=1}^N (\mathbb{1}_{\boldsymbol{x}_i=\boldsymbol{x}_j} - \frac{1}{V}) + \frac{2N^2}{V}(1 - \frac{1}{V})$$

$$= (1 - \frac{2}{V})\Big\|\sum_{i=1}^N (\boldsymbol{x}_i - \frac{1}{V}\mathbb{1}_V)\Big\|_2^2 + \frac{2N^2}{V}(1 - \frac{1}{V})$$

By the previous item, the statement follows. Moreover,

$$\sum_{i,j=1}^N \big(|\mathbb{1}_{\boldsymbol{x}_j=\boldsymbol{e}_k} - \frac{1}{V}| \pm \frac{2}{V}(1 - \frac{1}{V})\big)(\mathbb{1}_{\boldsymbol{x}_i=\boldsymbol{e}_k} - \frac{1}{V})$$

$$= (1 - \frac{2}{V})\Big(\sum_{i=1}^N (\mathbb{1}_{\boldsymbol{x}_i=\boldsymbol{e}_k} - \frac{1}{V})\Big)^2 + \frac{2N}{V}(1 - \frac{1}{V})\sum_{i=1}^N (\mathbb{1}_{\boldsymbol{x}_i=\boldsymbol{e}_k} - \frac{1}{V})$$

$$= N^2(1 - \frac{2}{V})\Big\langle \boldsymbol{e}_k, \frac{1}{N}\sum_{i=1}^N (\boldsymbol{x}_i - \frac{1}{V}\mathbb{1}_V)\Big\rangle^2 + \frac{2N^2}{V}(1 - \frac{1}{V})\Big\langle \boldsymbol{e}_k, \frac{1}{N}\sum_{i=1}^N (\boldsymbol{x}_i - \frac{1}{V}\mathbb{1}_V)\Big\rangle$$

$$\leq \frac{CN^2}{V^2}.$$

- The events for $\boldsymbol{S}_1$, $\boldsymbol{S}_2$ and $\boldsymbol{S}_3$ follows Proposition 10.

- (E2.8) follows the second item in Proposition 9.

$\square$

**Proposition 1.** *We consider the parameter regime in Lemma 1. Let $\bar{\phi} := \sup_{k_1,k_2\geq 1}|\phi^{(k_1)}(0)\phi^{(k_2)}(0)|$. The intersection of (E1) and (E2) implies the following events:*

*(R1)* For all $i,j \in [N]$, $|\frac{1}{L}\mathbb{1}_L \boldsymbol{X}_j^\top \boldsymbol{Z}_{\mathrm{in}}^\top \boldsymbol{Z}_{\mathrm{in}} \boldsymbol{X}_i^\top \mathbb{1}_L - m_{ij}| \leq C_K\big(\frac{\log V}{\sqrt{d}} + \frac{\log^2 V}{\sqrt{V}\wedge L}\big)$,

*(R2)* $\sup_{i,j}|\alpha_{ij} - \phi'(0)^2| \vee |\beta_{ij} - \phi''(0)\phi(0)| \leq \frac{\bar{\phi}}{L}\big(m_{ij} + C_K\frac{\log V}{\sqrt{d}} + C_K\frac{\log^2 V}{\sqrt{V}\wedge L}\big)$

*(R3)* Let $\Delta_{*,ir} := \boldsymbol{A}_{*,ir} - \phi'(0)^4 \boldsymbol{Z}_{\mathrm{in}} \boldsymbol{S}_* \boldsymbol{Z}_{\mathrm{in}}^\top$ for $* \in \{1,2,3\}$. We have

- $\sup_{i,r\in[N]}\|\Delta_{1,ir}\|_2 \leq C_K\phi'(0)^2\big(\frac{1}{NdL^3} + \frac{1}{VdL^2}\frac{1}{V\wedge L^2\wedge L\sqrt{d}}\big)$.
- $\sup_{i,r\in[N]}\|\Delta_{2,ir}\|_2 \leq \frac{C_K\sqrt{V}}{d\sqrt{NL}}\big(\frac{1}{NL^{\frac{3}{2}}} + \frac{1}{V\sqrt{L}}\frac{1}{V\wedge L^2\wedge L\sqrt{d}}\big)$.
- We have $\Delta_{3,ir} = \frac{\bar{\Delta}_{3,ir}}{V^2 L^2}\boldsymbol{Z}_{\mathrm{in}}\mathbb{1}_V \mathbb{1}_V^\top \boldsymbol{Z}_{\mathrm{in}}^\top$ such that

$$\sup_{i,r\in[N]}|\bar{\Delta}_{3,ir}| \leq \frac{C_K\phi'(0)^2}{N}\Big(\frac{1}{NL} + \frac{1}{\sqrt{N}}\frac{1}{V\wedge L^2\wedge L\sqrt{d}}\Big) + \Big(\frac{1}{NL} + \frac{1}{\sqrt{N}}\frac{1}{V\wedge L^2\wedge L\sqrt{d}}\Big)^2.$$

*(R4)* For all $i,r \in [N]$,

- We have

$$\Big\|\boldsymbol{A}_{1,ir} - \frac{\phi'(0)^4}{d}\Big(\frac{1}{N} + \big(1 - \frac{1}{V}\big)\frac{1}{V}\Big)\boldsymbol{I}_d\Big\|_2 \leq C_K\phi'(0)^2\Big(\frac{1}{NdL^3} + \frac{1}{VdL^2}\frac{1}{V\wedge L^2\wedge L\sqrt{d}}\Big)$$
$$+ C_K\phi'(0)^4\Big(\frac{\log V}{L^2 V^{3/2}\sqrt{d}} + \frac{\log^2 V}{L^2 N\sqrt{V}d}\Big).$$

- We have

$$\Big\|\boldsymbol{A}_{2,ir} - \frac{\phi'(0)^4}{d}\big(1 - \frac{1}{V}\big)^2\frac{L-1}{L^2 N}\boldsymbol{I}_d\Big\|_2 \leq \frac{C_K\sqrt{V}}{d\sqrt{NL}}\Big(\frac{1}{NL^{\frac{3}{2}}} + \frac{1}{V\sqrt{L}}\frac{1}{V\wedge L^2\wedge L\sqrt{d}}\Big)$$
$$+ C_K\phi'(0)^4\Big(\frac{\log V}{NL\sqrt{Vd}} + \frac{\log^3 V}{N\sqrt{LVd}}\Big).$$

- We have $\boldsymbol{A}_{3,ir} - \frac{\phi'(0)^4}{N}\frac{1}{V^2 L^2}\boldsymbol{Z}_{\mathrm{in}}\mathbb{1}_V\mathbb{1}_V^\top \boldsymbol{Z}_{\mathrm{in}}^\top =: \frac{\tilde{\Delta}_{3,ir}}{V^2 L^2}\boldsymbol{Z}_{\mathrm{in}}\mathbb{1}_V\mathbb{1}_V^\top \boldsymbol{Z}_{\mathrm{in}}^\top$ such that

$$|\tilde{\Delta}_{3,ir}| \leq \frac{C_K\phi'(0)^4 \log^2 V}{N\sqrt{V}}$$
$$+ \frac{C_K\phi'(0)^2}{N}\Big(\frac{1}{NL} + \frac{1}{\sqrt{N}}\frac{1}{V\wedge L^2\wedge L\sqrt{d}}\Big) + \Big(\frac{1}{NL} + \frac{1}{\sqrt{N}}\frac{1}{V\wedge L^2\wedge L\sqrt{d}}\Big)^2.$$

*Proof.* We have the following arguments.

- By (E1.7) and (E2.1), we have (R1).

- For (R2), we assume (R1) hold. Let $w_i := \frac{1}{L}\boldsymbol{w}^\top \boldsymbol{Z}_{\mathrm{in}}\boldsymbol{X}_i^\top \mathbb{1}_L / \|\frac{1}{L}\boldsymbol{Z}_{\mathrm{in}}\boldsymbol{X}_i^\top \mathbb{1}_L\|_2$. We write

$$\Big|\mathbb{E}_{\boldsymbol{w}}\big[\phi'\big(\|\tfrac{1}{L}\boldsymbol{Z}_{\mathrm{in}}\boldsymbol{X}_i^\top \mathbb{1}_L\|_2 w_i\big)\phi'\big(\|\tfrac{1}{L}\boldsymbol{Z}_{\mathrm{in}}\boldsymbol{X}_i^\top \mathbb{1}_L\|_2 w_j\big)\big] - \phi'(0)^2\Big|$$
$$\overset{(a)}{=} \Big|\sum_{u,v=1}^{p_\star}\|\tfrac{1}{L}\boldsymbol{Z}_{\mathrm{in}}\boldsymbol{X}_i^\top \mathbb{1}_L\|_2^u \|\tfrac{1}{L}\boldsymbol{Z}_{\mathrm{in}}\boldsymbol{X}_j^\top \mathbb{1}_L\|_2^v \frac{\mathbb{E}_{\boldsymbol{w}}\big[w_i^u w_j^v\big]}{u!v!}\phi^{(u+1)}(0)\phi^{(v+1)}(0)\Big|$$
$$\overset{(b)}{=} \Big|\frac{1}{L^2}\mathbb{1}_L^\top \boldsymbol{X}_j \boldsymbol{Z}_{\mathrm{in}}^\top \boldsymbol{Z}_{\mathrm{in}}\boldsymbol{X}_i^\top \mathbb{1}_L \phi^{(2)}(0)\phi^{(2)}(0)$$
$$+ \sum_{\substack{u,v=1 \\ u+v \text{ is even} \\ u+v>2}}^{p_\star}\|\tfrac{1}{L}\boldsymbol{Z}_{\mathrm{in}}\boldsymbol{X}_i^\top \mathbb{1}_L\|_2^u \|\tfrac{1}{L}\boldsymbol{Z}_{\mathrm{in}}\boldsymbol{X}_j^\top \mathbb{1}_L\|_2^v \frac{\mathbb{E}_{\boldsymbol{w}}\big[w_i^u w_j^v\big]}{u!v!}\phi^{(u+1)}(0)\phi^{(v+1)}(0)\Big|$$

$$\overset{(c)}{\le} \frac{\bar{\phi}}{L}\Big(m_{ij} + C_K \frac{\log V}{\sqrt{d}} + C_K \frac{\log^2 V}{\sqrt{V} \wedge L}\Big) + O\Big(\frac{1}{L^2}\Big),$$

where we used Taylor expansion of $\phi$ and $\mathbb{E}_{\boldsymbol{w}}\big[\boldsymbol{w}^\top \boldsymbol{Z}_{\mathrm{in}} \boldsymbol{X}_i^\top \mathbb{1}_L\big] = 0$ in (a), $\mathbb{E}[Z_1^u Z_2^v] = 0$ if $u + v$ is odd for jointly Gaussian $(Z_1, Z_2)$ in (b), and (R1) in (c). Similarly,

$$\big| \mathbb{E}\big[\phi''\big(\|\tfrac{1}{L}\boldsymbol{Z}_{\mathrm{in}}\boldsymbol{X}_i^\top \mathbb{1}_L\|_2 w_i\big)\phi\big(\|\tfrac{1}{L}\boldsymbol{Z}_{\mathrm{in}}\boldsymbol{X}_i^\top \mathbb{1}_L\|_2 w_j\big)\big] - \phi(0)\phi^{(2)}(0) \big|$$

$$= \Big| \sum_{u,v=1}^{k_2} \|\tfrac{1}{L}\boldsymbol{Z}_{\mathrm{in}}\boldsymbol{X}_i^\top \mathbb{1}_L\|_2^u \|\tfrac{1}{L}\boldsymbol{Z}_{\mathrm{in}}\boldsymbol{X}_j^\top \mathbb{1}_L\|_2^v \frac{\mathbb{E}\big[w_i^u w_j^v\big]}{u!v!}\phi^{(u+2)}(0)\phi^{(v)}(0) \Big|$$

$$= \Big| \frac{1}{L^2} \mathbb{1}_L^\top \boldsymbol{X}_j \boldsymbol{Z}_{\mathrm{in}}^\top \boldsymbol{Z}_{\mathrm{in}} \boldsymbol{X}_i^\top \mathbb{1}_L \phi(0)\phi^{(2)}(0)$$

$$+ \sum_{\substack{u,v=1 \\ u+v \text{ is even} \\ u+v>2}}^{k_2} \|\tfrac{1}{L}\boldsymbol{Z}_{\mathrm{in}}\boldsymbol{X}_i^\top \mathbb{1}_L\|_2^u \|\tfrac{1}{L}\boldsymbol{Z}_{\mathrm{in}}\boldsymbol{X}_j^\top \mathbb{1}_L\|_2^v \frac{\mathbb{E}\big[w_i^u w_j^v\big]}{u!v!}\phi^{(u+2)}(0)\phi^{(v)}(0) \Big|$$

$$\le \frac{\bar{\phi}}{L}\Big(m_{ij} + CK\frac{\log V}{\sqrt{d}} + CK\frac{\log^2 V}{\sqrt{V}\wedge L}\Big) + O\Big(\frac{1}{L^2}\Big).$$

- For (R3), we define

$$\bar{\Delta}_{1,ir} := \Big(\frac{1}{LN}\sum_{j=1}^N (\alpha_{ij} - \phi'(0)^2)(\boldsymbol{x}_j - \tfrac{1}{V}\mathbb{1}_V)(\boldsymbol{x}_j - \tfrac{1}{V}\mathbb{1}_V)^\top\Big)$$

$$\times \Big(\frac{1}{LN}\sum_{j=1}^N \phi'(0)^2(\boldsymbol{x}_j - \tfrac{1}{V}\mathbb{1}_V)(\boldsymbol{x}_j - \tfrac{1}{V}\mathbb{1}_V)^\top\Big)$$

$$+ \Big(\frac{1}{LN}\sum_{j=1}^N \phi'(0)^2(\boldsymbol{x}_j - \tfrac{1}{V}\mathbb{1}_V)(\boldsymbol{x}_j - \tfrac{1}{V}\mathbb{1}_V)^\top\Big)$$

$$\times \Big(\frac{1}{LN}\sum_{j=1}^N (\alpha_{rj} - \phi'(0)^2)(\boldsymbol{x}_j - \tfrac{1}{V}\mathbb{1}_V)(\boldsymbol{x}_j - \tfrac{1}{V}\mathbb{1}_V)^\top\Big)$$

$$+ \Big(\frac{1}{LN}\sum_{j=1}^N (\alpha_{ij} - \phi'(0)^2)(\boldsymbol{x}_j - \tfrac{1}{V}\mathbb{1}_V)(\boldsymbol{x}_j - \tfrac{1}{V}\mathbb{1}_V)^\top\Big)$$

$$\times \Big(\frac{1}{LN}\sum_{j=1}^N (\alpha_{rj} - \phi'(0)^2)(\boldsymbol{x}_j - \tfrac{1}{V}\mathbb{1}_V)(\boldsymbol{x}_j - \tfrac{1}{V}\mathbb{1}_V)^\top\Big)$$

We have

$$\|\bar{\Delta}_{1,ir}\|_2 \le \frac{C\phi'(0)^2 \sup_i|\alpha_{ii} - \phi'(0)^2|}{LN}\|\boldsymbol{S}_1\|_2^{\frac{1}{2}} + \phi'(0)^2 \sup_{i\neq j}|\alpha_{ij} - \phi'(0)^2|\|\boldsymbol{S}_1\|_2$$

$$\overset{(d)}{\le} C\phi'(0)^2\Big(\frac{1}{NVL^3} + \frac{1}{V^2L^2}\frac{1}{V\wedge L^2 \wedge L\sqrt{d}}\Big),$$

where we used (R2) and (E2.5) in (d). By (E1.1), we have

$$\|\Delta_{1,ir}\|_2 = \|\boldsymbol{Z}_{\mathrm{in}}\bar{\Delta}_{1,ir}\boldsymbol{Z}_{\mathrm{in}}^\top\|_2 \le C\phi'(0)^2\Big(\frac{1}{NdL^3} + \frac{1}{VdL^2}\frac{1}{V\wedge L^2 \wedge L\sqrt{d}}\Big).$$

Moreover, we define

$$\bar{\Delta}_{2,ir} := \Big(\frac{1}{NL}\sum_{j=1}^N (\alpha_{ij} - \phi'(0)^2)(\boldsymbol{N}_j^\top - \tfrac{1}{V}\mathbb{1}_V \mathbb{1}_{L-1}^\top)\mathbb{1}_{L-1}(\boldsymbol{x}_j - \tfrac{1}{V}\mathbb{1}_V)^\top\Big)$$

$$\times \Big(\frac{1}{NL}\sum_{j=1}^{N}\phi'(0)^2(\boldsymbol{x}_j - \tfrac{1}{V}\mathbb{1}_V)\mathbb{1}_{L-1}^{\top}(\boldsymbol{N}_j^{\top} - \tfrac{1}{V}\mathbb{1}_V\mathbb{1}_{L-1}^{\top})^{\top}\Big)$$

$$+\Big(\frac{1}{NL}\sum_{j=1}^{N}\phi'(0)^2(\boldsymbol{N}_j^{\top} - \tfrac{1}{V}\mathbb{1}_V\mathbb{1}_{L-1}^{\top})\mathbb{1}_{L-1}(\boldsymbol{x}_j - \tfrac{1}{V}\mathbb{1}_V)^{\top}\Big)$$

$$\times \Big(\frac{1}{NL}\sum_{j=1}^{N}(\alpha_{rj} - \phi'(0)^2)(\boldsymbol{x}_j - \tfrac{1}{V}\mathbb{1}_V)\mathbb{1}_{L-1}^{\top}(\boldsymbol{N}_j^{\top} - \tfrac{1}{V}\mathbb{1}_V\mathbb{1}_{L-1}^{\top})^{\top}\Big)$$

$$+\Big(\frac{1}{NL}\sum_{j=1}^{N}(\alpha_{ij} - \phi'(0)^2)(\boldsymbol{N}_j^{\top} - \tfrac{1}{V}\mathbb{1}_V\mathbb{1}_{L-1}^{\top})\mathbb{1}_{L-1}(\boldsymbol{x}_j - \tfrac{1}{V}\mathbb{1}_V)^{\top}\Big)$$

$$\times \Big(\frac{1}{NL}\sum_{j=1}^{N}(\alpha_{rj} - \phi'(0)^2)(\boldsymbol{x}_j - \tfrac{1}{V}\mathbb{1}_V)\mathbb{1}_{L-1}^{\top}(\boldsymbol{N}_j^{\top} - \tfrac{1}{V}\mathbb{1}_V\mathbb{1}_{L-1}^{\top})^{\top}\Big).$$

We have

$$\|\bar{\Delta}_{2,ir}\|_2 \le \phi'(0)^2\|\boldsymbol{S}_2\|_2^{\frac{1}{2}}\Big\|\frac{1}{NL}\sum_{j=1}^{N}(\alpha_{rj} - \phi'(0)^2)(\boldsymbol{x}_j - \tfrac{1}{V}\mathbb{1}_V)\mathbb{1}_{L-1}^{\top}(\boldsymbol{N}_j^{\top} - \tfrac{1}{V}\mathbb{1}_V\mathbb{1}_{L-1}^{\top})^{\top}\Big\|_2$$

$$+ \phi'(0)^2\|\boldsymbol{S}_2\|_2^{\frac{1}{2}}\Big\|\frac{1}{NL}\sum_{j=1}^{N}(\alpha_{ij} - \phi'(0)^2)(\boldsymbol{x}_j - \tfrac{1}{V}\mathbb{1}_V)\mathbb{1}_{L-1}^{\top}(\boldsymbol{N}_j^{\top} - \tfrac{1}{V}\mathbb{1}_V\mathbb{1}_{L-1}^{\top})^{\top}\Big\|_2$$

$$+ \Big\|\frac{1}{NL}\sum_{j=1}^{N}(\alpha_{rj} - \phi'(0)^2)(\boldsymbol{x}_j - \tfrac{1}{V}\mathbb{1}_V)\mathbb{1}_{L-1}^{\top}(\boldsymbol{N}_j^{\top} - \tfrac{1}{V}\mathbb{1}_V\mathbb{1}_{L-1}^{\top})^{\top}\Big\|_2$$

$$\times \Big\|\frac{1}{NL}\sum_{j=1}^{N}(\alpha_{ij} - \phi'(0)^2)(\boldsymbol{x}_j - \tfrac{1}{V}\mathbb{1}_V)\mathbb{1}_{L-1}^{\top}(\boldsymbol{N}_j^{\top} - \tfrac{1}{V}\mathbb{1}_V\mathbb{1}_{L-1}^{\top})^{\top}\Big\|_2.$$

We observe that

$$\Big\|\frac{1}{NL}\sum_{j=1}^{N}(\alpha_{ij} - \phi'(0)^2)(\boldsymbol{x}_j - \tfrac{1}{V}\mathbb{1}_V)\mathbb{1}_{L-1}^{\top}(\boldsymbol{N}_j^{\top} - \tfrac{1}{V}\mathbb{1}_V\mathbb{1}_{L-1}^{\top})^{\top}\Big\|_2$$

$$\le \Big\|\frac{1}{NL}(\alpha_{ii} - \phi'(0)^2)(\boldsymbol{x}_i - \tfrac{1}{V}\mathbb{1}_V)\mathbb{1}_{L-1}^{\top}(\boldsymbol{N}_i^{\top} - \tfrac{1}{V}\mathbb{1}_V\mathbb{1}_{L-1}^{\top})^{\top}\Big\|_2$$

$$+ \Big\|\frac{1}{NL}\sum_{\substack{j=1 \\ j\neq i}}^{N}(\alpha_{ij} - \phi'(0)^2)(\boldsymbol{x}_j - \tfrac{1}{V}\mathbb{1}_V)\mathbb{1}_{L-1}^{\top}(\boldsymbol{N}_j^{\top} - \tfrac{1}{V}\mathbb{1}_V\mathbb{1}_{L-1}^{\top})^{\top}\Big\|_2$$

$$\overset{(e)}{\le} \frac{C}{NL\sqrt{L}} + \frac{\sup_{i\neq j}|\alpha_{ij} - \phi'(0)^2|}{\sqrt{L}}\Big\|\frac{1}{N}\sum_{\substack{j=1 \\ j\neq i}}^{N}(\boldsymbol{x}_j - \tfrac{1}{V}\mathbb{1}_V)(\boldsymbol{x}_j - \tfrac{1}{V}\mathbb{1}_V)^{\top}\Big\|_2$$

$$+ \frac{\sup_{i\neq j}|\alpha_{ij} - \phi'(0)^2|}{\sqrt{L}}\Big\|\frac{1}{NL}\sum_{\substack{j=1 \\ j\neq i}}^{N}(\boldsymbol{N}_j^{\top} - \tfrac{1}{V}\mathbb{1}_{L-1})\mathbb{1}_{L-1}\mathbb{1}_{L-1}^{\top}(\boldsymbol{N}_j^{\top} - \tfrac{1}{V}\mathbb{1}_{L-1})^{\top}\Big\|_2$$

$$\overset{(f)}{\le} \frac{C}{NL\sqrt{L}} + \frac{C}{V\sqrt{L}}\frac{1}{V \wedge L^2 \wedge L\sqrt{d}}.$$

where we used (R2) and (E2.1) in (e), and (R2), (E2.5) and (E2.8) in (f). Then, by (E2.6), we have

$$\|\bar{\Delta}_{2,ir}\|_2 \le \frac{C}{\sqrt{NVL}}\Big(\frac{1}{NL^{\frac{3}{2}}} + \frac{1}{V\sqrt{L}}\frac{1}{V \wedge L^2 \wedge L\sqrt{d}}\Big) + C^2\Big(\frac{1}{NL^{\frac{3}{2}}} + \frac{1}{V\sqrt{L}}\frac{1}{V \wedge L^2 \wedge L\sqrt{d}}\Big)^2.$$

Therefore, by (E1.1)

$$\|\boldsymbol{\Delta}_{2,ir}\|_2 = \|\boldsymbol{Z}_{\mathrm{in}}\bar{\boldsymbol{\Delta}}_{2,ir}\boldsymbol{Z}_{\mathrm{in}}^\top\|_2 \le \frac{C\sqrt{V}}{d\sqrt{NL}}\Big(\frac{1}{NL^{\frac{3}{2}}} + \frac{1}{V\sqrt{L}}\frac{1}{V \wedge L^2 \wedge L\sqrt{d}}\Big).$$

Lastly, we define

$$\bar{\boldsymbol{\Delta}}_{3,ir} := \Big(\frac{1}{N}\sum_{j=1}^{N}(\alpha_{ij} - \phi'(0)^2)(\boldsymbol{x}_j - \tfrac{1}{V}\mathbb{1}_V)\Big)^\top \Big(\frac{1}{N}\sum_{j=1}^{N}\phi'(0)^2(\boldsymbol{x}_j - \tfrac{1}{V}\mathbb{1}_V)\Big)$$

$$+ \Big(\frac{1}{N}\sum_{j=1}^{N}\phi'(0)^2(\boldsymbol{x}_j - \tfrac{1}{V}\mathbb{1}_V)\Big)^\top \Big(\frac{1}{N}\sum_{j=1}^{N}(\alpha_{rj} - \phi'(0)^2)(\boldsymbol{x}_j - \tfrac{1}{V}\mathbb{1}_V)\Big)$$

$$+ \Big(\frac{1}{N}\sum_{j=1}^{N}(\alpha_{ij} - \phi'(0)^2)(\boldsymbol{x}_j - \tfrac{1}{V}\mathbb{1}_V)\Big)^\top \Big(\frac{1}{N}\sum_{j=1}^{N}(\alpha_{rj} - \phi'(0)^2)(\boldsymbol{x}_j - \tfrac{1}{V}\mathbb{1}_V)\Big)$$

We have

$$|\bar{\boldsymbol{\Delta}}_{3,ir}| \le \phi'(0)^2 \Big\|\frac{1}{N}\sum_{j=1}^{N}(\boldsymbol{x}_j - \tfrac{1}{V}\mathbb{1}_V)\Big\|_2 \Big\|\frac{1}{N}\sum_{j=1}^{N}(\alpha_{ij} - \phi'(0)^2)(\boldsymbol{x}_j - \tfrac{1}{V}\mathbb{1}_V)\Big\|_2$$

$$+ \phi'(0)^2 \Big\|\frac{1}{N}\sum_{j=1}^{N}(\boldsymbol{x}_j - \tfrac{1}{V}\mathbb{1}_V)\Big\|_2 \Big\|\frac{1}{N}\sum_{j=1}^{N}(\alpha_{rj} - \phi'(0)^2)(\boldsymbol{x}_j - \tfrac{1}{V}\mathbb{1}_V)\Big\|_2$$

$$+ \Big\|\frac{1}{N}\sum_{j=1}^{N}(\alpha_{ij} - \phi'(0)^2)(\boldsymbol{x}_j - \tfrac{1}{V}\mathbb{1}_V)\Big\|_2 \underbrace{\Big\|\frac{1}{N}\sum_{j=1}^{N}(\alpha_{rj} - \phi'(0)^2)(\boldsymbol{x}_j - \tfrac{1}{V}\mathbb{1}_V)\Big\|_2}_{\le \frac{C}{NL} + \frac{C}{\sqrt{N}}\frac{1}{V \wedge L^2 \wedge L\sqrt{d}}}$$

$$\le \frac{C\phi'(0)^2}{N}\Big(\frac{1}{NL} + \frac{1}{\sqrt{N}}\frac{1}{V \wedge L^2 \wedge L\sqrt{d}}\Big) + \Big(\frac{1}{NL} + \frac{1}{\sqrt{N}}\frac{1}{V \wedge L^2 \wedge L\sqrt{d}}\Big)^2 \quad \text{(B.12)}$$

We observe that $\boldsymbol{\Delta}_{3,ir} = \frac{\bar{\boldsymbol{\Delta}}_{3,ir}}{V^2 L^2}\boldsymbol{Z}_{\mathrm{in}}\mathbb{1}_V\mathbb{1}_V^\top\boldsymbol{Z}_{\mathrm{in}}^\top$, and by (B.12), the last result follows.

- For (R4), we assume (E1.1), (E1.2), (E2.3), and (E2.5)-(E2.7). We write

$$\boldsymbol{A}_{1,ir} - \frac{\phi'(0)^4}{d}\frac{(1 - \tfrac{1}{V})}{L^2}\Big(\frac{1}{N} + \frac{1}{V}\Big)\boldsymbol{I}_d$$

$$= \Delta_{1,ir} + \phi'(0)^4\Big(\boldsymbol{Z}_{\mathrm{in}}\boldsymbol{S}_1\boldsymbol{Z}_{\mathrm{in}} \pm \frac{\mathrm{tr}(\boldsymbol{S}_1)}{d}\boldsymbol{I}_d - \frac{1}{d}\frac{(1 - \tfrac{1}{V})}{L^2}\Big(\frac{1}{N} + \frac{1}{V}\Big)\boldsymbol{I}_d\Big).$$

We have

$$\Big\|\boldsymbol{A}_{1,ir} - \frac{\phi'(0)^4}{d}\frac{(1 - \tfrac{1}{V})}{L^2}\Big(\frac{1}{N} + \frac{1}{V}\Big)\boldsymbol{I}_d\Big\|_2$$

$$\le \|\Delta_{1,ir}\|_2 + 2\phi'(0)^4\log V\frac{\|\boldsymbol{S}_1\|_F}{\sqrt{d}} + \frac{|\mathrm{tr}(\boldsymbol{S}_1) - \frac{(1 - \tfrac{1}{V})}{L^2}(\frac{1}{N} + \frac{1}{V})|}{d}$$

$$\le C_K\phi'(0)^2\Big(\frac{1}{NdL^3} + \frac{1}{VdL^2}\frac{1}{V \wedge L^2 \wedge L\sqrt{d}}\Big) + C_K\phi'(0)^4\Big(\frac{\log V}{L^2 V^{3/2}\sqrt{d}} + \frac{\log^2 V}{L^2 N\sqrt{V}d}\Big).$$

Similarly,

$$\Big\|\boldsymbol{A}_{2,ir} - \frac{\phi'(0)^4}{d}(1 - \tfrac{1}{V})^2\frac{L-1}{L^2 N}\boldsymbol{I}_d\Big\|_2$$

$$\le \|\Delta_{2,ir}\|_2 + 2\phi'(0)^4\log V\frac{\|\boldsymbol{S}_2\|_F}{\sqrt{d}} + \frac{|\mathrm{tr}(\boldsymbol{S}_2) - (1 - \tfrac{1}{V})^2\frac{L-1}{L^2 N}|}{d}$$

$$\leq \frac{C_K\sqrt{V}}{d\sqrt{NL}}\Big(\frac{1}{NL^{\frac{3}{2}}} + \frac{1}{V\sqrt{L}}\frac{1}{V\wedge L^2 \wedge L\sqrt{d}}\Big) + C_K\phi'(0)^4\Big(\frac{\log V}{NL\sqrt{Vd}} + \frac{\log^3 V}{N\sqrt{LV}d}\Big).$$

Lastly,

$$\boldsymbol{A}_{3,ir} - \frac{\phi'(0)^4}{N}\frac{1}{V^2L^2}\boldsymbol{Z}_{\mathrm{in}}\mathbb{1}_V\mathbb{1}_V^\top\boldsymbol{Z}_{\mathrm{in}}^\top$$

$$= \Delta_{3,ir} + \phi'(0)^4\Big(\Big\|\frac{1}{N}\sum_{j=1}^N(\boldsymbol{x}_j - \frac{1}{V}\mathbb{1}_V)\Big\|_2^2 - \frac{1}{N}\Big)\frac{1}{V^2L^2}\boldsymbol{Z}_{\mathrm{in}}\mathbb{1}_V\mathbb{1}_V^\top\boldsymbol{Z}_{\mathrm{in}}^\top.$$

By (E2.3), we have

$$\frac{CK^2\log^2 V}{N\sqrt{V}}\frac{1}{V^2L^2}\boldsymbol{Z}_{\mathrm{in}}\mathbb{1}_V\mathbb{1}_V^\top\boldsymbol{Z}_{\mathrm{in}}^\top \preceq \Big(\Big\|\frac{1}{N}\sum_{j=1}^N(\boldsymbol{x}_j - \frac{1}{V}\mathbb{1}_V)\Big\|_2^2 - \frac{1}{N}\Big)\frac{1}{V^2L^2}\boldsymbol{Z}_{\mathrm{in}}\mathbb{1}_V\mathbb{1}_V^\top\boldsymbol{Z}_{\mathrm{in}}^\top$$

$$\preceq \frac{CK^2\log^2 V}{N\sqrt{V}}\frac{1}{V^2L^2}\boldsymbol{Z}_{\mathrm{in}}\mathbb{1}_V\mathbb{1}_V^\top\boldsymbol{Z}_{\mathrm{in}}^\top.$$

By (R3), the result follows.

$\square$

**Proposition 2.** *We recall that* $\mathsf{z}_k = \boldsymbol{z}_k + \mathbb{1}_{k=1}\mathsf{z}_{\mathrm{trig}}$. *Given that (E1) holds, the following statements hold:*

*(P1) We have for* $i \neq j$ *and any* $k \in [V]$,

$$\Big|\mathbb{E}\big[(\mathbb{1}_{\boldsymbol{x}_i=\boldsymbol{x}_j} - \tfrac{1}{V})\mathsf{z}_k^\top\boldsymbol{Z}_{\mathrm{in}}\boldsymbol{X}_i^\top\boldsymbol{X}_i\boldsymbol{Z}_{\mathrm{in}}^\top\boldsymbol{Z}_{\mathrm{in}}\boldsymbol{X}_j^\top\boldsymbol{X}_j\boldsymbol{Z}_{\mathrm{in}}^\top\mathsf{z}_k|\mathsf{Z}_{\mathrm{in}}\big]\Big| \leq \frac{C}{Vd}.$$

*(P2) For any* $k \in [V]$,

$$\mathbb{E}[\|\boldsymbol{Z}_{\mathrm{in}}\boldsymbol{X}_i^\top\boldsymbol{X}_i\boldsymbol{Z}_{\mathrm{in}}^\top\mathsf{z}_k\|_2^2|\mathsf{Z}_{\mathrm{in}}] \leq C\Big(\frac{L}{d} + \frac{L^2}{d^2}\Big).$$

*(P3) We have for* $i \neq j$ *and any* $k \in [V]$,

$$\Big|\mathbb{E}\big[(\mathbb{1}_{\boldsymbol{x}_i=\boldsymbol{x}_j} - \tfrac{1}{V})\mathsf{z}_k^\top\boldsymbol{Z}_{\mathrm{in}}\boldsymbol{X}_i^\top\boldsymbol{X}_i\boldsymbol{Z}_{\mathrm{in}}^\top\boldsymbol{Z}_{\mathrm{in}}\mathbb{1}_V\mathbb{1}_V^\top\boldsymbol{Z}_{\mathrm{in}}^\top\boldsymbol{Z}_{\mathrm{in}}\boldsymbol{X}_r^\top\boldsymbol{X}_r\boldsymbol{Z}_{\mathrm{in}}^\top\mathsf{z}_k|\mathsf{Z}_{\mathrm{in}}\big]\Big| \leq \frac{C\log^2 V}{d^2}.$$

*(P4) For any* $k \in [V]$,

$$\mathbb{E}\big[\big(\mathbb{1}_V^\top\boldsymbol{Z}_{\mathrm{in}}^\top\boldsymbol{Z}_{\mathrm{in}}\boldsymbol{X}_i^\top\boldsymbol{X}_i\boldsymbol{Z}_{\mathrm{in}}^\top\mathsf{z}_k\big)^2|\mathsf{Z}_{\mathrm{in}}\big] \leq C\frac{V\log^2 V}{d}\Big(\frac{L}{d} + \frac{L^2}{d^2}\Big).$$

*(P5) For notational convenience, let*

$$\varsigma := \mathsf{z}_k^\top\boldsymbol{Z}_{\mathrm{in}}\Big(\boldsymbol{X}_i^\top\boldsymbol{X}_i - \frac{L}{V}\boldsymbol{I}_V\Big)\boldsymbol{Z}_{\mathrm{in}}^\top\boldsymbol{Z}_{\mathrm{in}}\big(\boldsymbol{X}_i^\top - \frac{1}{V}\mathbb{1}_V\mathbb{1}_L^\top\big)\mathbb{1}_L.$$

*For any* $i, k \in [V]$,

$$|\mathbb{E}[\varsigma|\mathsf{Z}_{\mathrm{in}}]| \leq \frac{CL\log V}{\sqrt{Vd}} \quad and \quad \mathbb{E}[\varsigma^2|\mathsf{Z}_{\mathrm{in}}] \leq C\log^2 V\Big(\frac{L}{d} + \frac{L^2}{d^2}\Big).$$

*Proof.* For the first item, we have

$$\mathbb{E}\big[(\mathbb{1}_{\boldsymbol{x}_i=\boldsymbol{x}_j} - \tfrac{1}{V})\mathsf{z}_k^\top\boldsymbol{Z}_{\mathrm{in}}\boldsymbol{X}_i^\top\boldsymbol{X}_i\boldsymbol{Z}_{\mathrm{in}}^\top\boldsymbol{Z}_{\mathrm{in}}\boldsymbol{X}_j^\top\boldsymbol{X}_j\boldsymbol{Z}_{\mathrm{in}}^\top\mathsf{z}_k|\mathsf{Z}_{\mathrm{in}}\big]$$

$$\overset{(a)}{=} \mathbb{E}\big[(\mathbb{1}_{\boldsymbol{x}_i=\boldsymbol{x}_j} - \tfrac{1}{V})\mathsf{z}_k^\top\boldsymbol{Z}_{\mathrm{in}}\boldsymbol{x}_i\boldsymbol{x}_i^\top\boldsymbol{Z}_{\mathrm{in}}^\top\boldsymbol{Z}_{\mathrm{in}}\boldsymbol{x}_j\boldsymbol{x}_j^\top\boldsymbol{Z}_{\mathrm{in}}^\top\mathsf{z}_k|\mathsf{Z}_{\mathrm{in}}\big]$$

$$= \frac{1}{V}\mathbb{E}\big[\mathsf{z}_k^\top\boldsymbol{Z}_{\mathrm{in}}\boldsymbol{x}_i\boldsymbol{x}_i^\top\boldsymbol{Z}_{\mathrm{in}}^\top\boldsymbol{Z}_{\mathrm{in}}\boldsymbol{x}_i\boldsymbol{x}_i^\top\boldsymbol{Z}_{\mathrm{in}}^\top\mathsf{z}_k|\mathsf{Z}_{\mathrm{in}}\big] - \frac{1}{V^3}\mathsf{z}_k^\top\boldsymbol{Z}_{\mathrm{in}}\boldsymbol{Z}_{\mathrm{in}}^\top\boldsymbol{Z}_{\mathrm{in}}\boldsymbol{Z}_{\mathrm{in}}^\top\mathsf{z}_k$$

$$\leq \frac{C}{Vd},$$

where we used the independence of the rows of $\boldsymbol{X}$ in (a). For the second item, we write

$$\mathbb{E}[\mathsf{z}_k^\top \boldsymbol{Z}_{\mathrm{in}} \boldsymbol{X}_i^\top \boldsymbol{X}_i \boldsymbol{Z}_{\mathrm{in}}^\top \boldsymbol{Z}_{\mathrm{in}} \boldsymbol{X}_i^\top \boldsymbol{X}_i \boldsymbol{Z}_{\mathrm{in}}^\top \mathsf{z}_k | \mathsf{Z}_{\mathrm{in}}] = \frac{L}{V} \mathsf{z}_k^\top \boldsymbol{Z}_{\mathrm{in}} \mathrm{diag}(\boldsymbol{Z}_{\mathrm{in}}^\top \boldsymbol{Z}_{\mathrm{in}}) \boldsymbol{Z}_{\mathrm{in}}^\top \mathsf{z}_k$$
$$+ \frac{L(L-1)}{V^2} \mathsf{z}_k^\top \boldsymbol{Z}_{\mathrm{in}} \boldsymbol{Z}_{\mathrm{in}}^\top \boldsymbol{Z}_{\mathrm{in}} \boldsymbol{Z}_{\mathrm{in}}^\top \mathsf{z}_k^\top$$
$$\leq C\Big(\frac{L}{d} + \frac{L^2}{d^2}\Big).$$

For the third item, we have

$$\mathbb{E}\Big[(\mathbb{1}_{\boldsymbol{x}_i=\boldsymbol{x}_j} - \tfrac{1}{V}) \mathsf{z}_k^\top \boldsymbol{Z}_{\mathrm{in}} \boldsymbol{X}_i^\top \boldsymbol{X}_i \boldsymbol{Z}_{\mathrm{in}}^\top \boldsymbol{Z}_{\mathrm{in}} \mathbb{1}_V \mathbb{1}_V^\top \boldsymbol{Z}_{\mathrm{in}}^\top \boldsymbol{Z}_{\mathrm{in}} \boldsymbol{X}_j^\top \boldsymbol{X}_j \boldsymbol{Z}_{\mathrm{in}}^\top \mathsf{z}_k | \mathsf{Z}_{\mathrm{in}}\Big]$$
$$= \mathbb{E}\Big[(\mathbb{1}_{\boldsymbol{x}_i=\boldsymbol{x}_j} - \tfrac{1}{V}) \mathsf{z}_k^\top \boldsymbol{Z}_{\mathrm{in}} \boldsymbol{x}_i \boldsymbol{x}_i^\top \boldsymbol{Z}_{\mathrm{in}}^\top \boldsymbol{Z}_{\mathrm{in}} \mathbb{1}_V \mathbb{1}_V^\top \boldsymbol{Z}_{\mathrm{in}}^\top \boldsymbol{Z}_{\mathrm{in}} \boldsymbol{x}_j \boldsymbol{x}_j^\top \boldsymbol{Z}_{\mathrm{in}}^\top \mathsf{z}_k | \mathsf{Z}_{\mathrm{in}}\Big]$$
$$= \frac{1}{V} \mathbb{E}\big[\mathsf{z}_k^\top \boldsymbol{Z}_{\mathrm{in}} \boldsymbol{x}_i \boldsymbol{x}_i^\top \boldsymbol{Z}_{\mathrm{in}}^\top \boldsymbol{Z}_{\mathrm{in}} \mathbb{1}_V \mathbb{1}_V^\top \boldsymbol{Z}_{\mathrm{in}}^\top \boldsymbol{Z}_{\mathrm{in}} \boldsymbol{x}_i \boldsymbol{x}_i^\top \boldsymbol{Z}_{\mathrm{in}}^\top \mathsf{z}_k | \mathsf{Z}_{\mathrm{in}}\big]$$
$$- \frac{1}{V^3} \mathsf{z}_k^\top \boldsymbol{Z}_{\mathrm{in}} \boldsymbol{Z}_{\mathrm{in}}^\top \boldsymbol{Z}_{\mathrm{in}} \mathbb{1}_V \mathbb{1}_V^\top \boldsymbol{Z}_{\mathrm{in}}^\top \boldsymbol{Z}_{\mathrm{in}} \boldsymbol{Z}_{\mathrm{in}}^\top \mathsf{z}_k$$
$$\leq C\frac{\log^2 V}{d^2}.$$

For the fourth item, we have

$$\mathbb{E}\big[\mathsf{z}_k^\top \boldsymbol{Z}_{\mathrm{in}} \boldsymbol{X}_i^\top \boldsymbol{X}_i \boldsymbol{Z}_{\mathrm{in}}^\top \boldsymbol{Z}_{\mathrm{in}} \mathbb{1}_V \mathbb{1}_V^\top \boldsymbol{Z}_{\mathrm{in}}^\top \boldsymbol{Z}_{\mathrm{in}} \boldsymbol{X}_i^\top \boldsymbol{X}_i \boldsymbol{Z}_{\mathrm{in}}^\top \mathsf{z}_k | \mathsf{Z}_{\mathrm{in}}\big]$$
$$= L \mathbb{E}\big[\mathsf{z}_k^\top \boldsymbol{Z}_{\mathrm{in}} \boldsymbol{x}_i \boldsymbol{x}_i^\top \boldsymbol{Z}_{\mathrm{in}}^\top \boldsymbol{Z}_{\mathrm{in}} \mathbb{1}_V \mathbb{1}_V^\top \boldsymbol{Z}_{\mathrm{in}}^\top \boldsymbol{Z}_{\mathrm{in}} \boldsymbol{x}_i \boldsymbol{x}_i^\top \boldsymbol{Z}_{\mathrm{in}}^\top \mathsf{z}_k | \mathsf{Z}_{\mathrm{in}}\big]$$
$$+ \frac{L(L-1)}{V^2} \mathsf{z}_k^\top \boldsymbol{Z}_{\mathrm{in}} \boldsymbol{Z}_{\mathrm{in}}^\top \boldsymbol{Z}_{\mathrm{in}} \mathbb{1}_V \mathbb{1}_V^\top \boldsymbol{Z}_{\mathrm{in}}^\top \boldsymbol{Z}_{\mathrm{in}} \boldsymbol{Z}_{\mathrm{in}}^\top \mathsf{z}_k$$
$$\leq C\Big(\frac{LV\log^2 V}{d^2} + \frac{L^2 V \log^2 V}{d^3}\Big).$$

For the fifth item, we have

$$\mathbb{E}\Big[\mathsf{z}_k^\top \boldsymbol{Z}_{\mathrm{in}} \Big(\boldsymbol{X}_i^\top \boldsymbol{X}_i - \tfrac{L}{V}\boldsymbol{I}_V\Big) \boldsymbol{Z}_{\mathrm{in}}^\top \boldsymbol{Z}_{\mathrm{in}} \boldsymbol{X}_i^\top \mathbb{1}_L | \mathsf{Z}_{\mathrm{in}}\Big]$$
$$= \mathbb{E}\Big[\mathsf{z}_k^\top \boldsymbol{Z}_{\mathrm{in}} \boldsymbol{X}_i^\top \boldsymbol{X}_i \boldsymbol{Z}_{\mathrm{in}}^\top \boldsymbol{Z}_{\mathrm{in}} \boldsymbol{X}_i^\top \boldsymbol{X}_i | \mathsf{Z}_{\mathrm{in}}\Big] \mathbb{1}_V - \frac{L^2}{V^2} \mathsf{z}_k^\top \boldsymbol{Z}_{\mathrm{in}} \boldsymbol{Z}_{\mathrm{in}}^\top \boldsymbol{Z}_{\mathrm{in}} \mathbb{1}_V$$
$$= L\mathsf{z}_k^\top \boldsymbol{Z}_{\mathrm{in}} \mathbb{E}\Big[\boldsymbol{x}_i \boldsymbol{x}_i^\top \boldsymbol{Z}_{\mathrm{in}}^\top \boldsymbol{Z}_{\mathrm{in}} \boldsymbol{x}_i \boldsymbol{x}_i^\top | \mathsf{Z}_{\mathrm{in}}\Big] \mathbb{1}_V - \frac{L^2}{V^2} \mathsf{z}_k^\top \boldsymbol{Z}_{\mathrm{in}} \boldsymbol{Z}_{\mathrm{in}}^\top \boldsymbol{Z}_{\mathrm{in}} \mathbb{1}_V \text{(B.13)}$$

By (E1), we have

$$|(\text{B.13})| \leq \frac{CL\log V}{\sqrt{Vd}}.$$

For the second part, let $n_j := \boldsymbol{e}_j^\top \boldsymbol{X}_i \mathbb{1}_L$. We have

$$\mathsf{z}_k^\top \boldsymbol{Z}_{\mathrm{in}} \big(\boldsymbol{X}_i^\top \boldsymbol{X}_i - \tfrac{L}{V}\boldsymbol{I}_V\big) \boldsymbol{Z}_{\mathrm{in}}^\top \boldsymbol{Z}_{\mathrm{in}} \big(\boldsymbol{X}_i^\top - \tfrac{1}{V}\mathbb{1}_V \mathbb{1}_L^\top\big) \mathbb{1}_L$$
$$= \sum_{j=1}^V (n_j - \tfrac{L}{V}) \mathsf{z}_k^\top \boldsymbol{z}_j \boldsymbol{z}_j^\top \Big(\sum_{l=1}^V (n_l - \tfrac{L}{V}) \boldsymbol{z}_l\Big)$$
$$= \sum_{j=1}^V \sum_{l=1}^V (n_j - \tfrac{L}{V})(n_l - \tfrac{L}{V}) \mathsf{z}_k^\top \boldsymbol{z}_i \boldsymbol{z}_i^\top \boldsymbol{z}_j. \tag{B.14}$$

Let $\boldsymbol{S} = (s_{jl})_{jl\in[V]}$ such that $s_{jl} := \frac{1}{2}\big(\mathsf{z}_k^\top \boldsymbol{z}_j \boldsymbol{z}_j^\top \boldsymbol{z}_l + \mathsf{z}_k^\top \boldsymbol{z}_l \boldsymbol{z}_l^\top \boldsymbol{z}_j\big)$. We will use the third item in Proposition 8 to bound second moment of (B.14). We bound each term separately below.

- We have

$$
\begin{aligned}
\left| \mathrm{tr}\big( (\boldsymbol{I}_V - \tfrac{1}{V} \mathbb{1}_V \mathbb{1}_V^\top) \boldsymbol{S} \big) \right| &= \left| \mathrm{tr}(\boldsymbol{S}) - \tfrac{1}{V} \mathbb{1}_V^\top \boldsymbol{S} \mathbb{1}_V \right| \\
&= V \left| \mathsf{z}_k^\top \boldsymbol{Z}_{\mathrm{in}} \, \mathbb{E}[\boldsymbol{x}_1 \boldsymbol{x}_1^\top \boldsymbol{Z}_{\mathrm{in}}^\top \boldsymbol{Z}_{\mathrm{in}} \boldsymbol{x}_1] - \tfrac{1}{V} \mathsf{z}_k^\top \boldsymbol{Z}_{\mathrm{in}} \boldsymbol{Z}_{\mathrm{in}}^\top \boldsymbol{Z}_{\mathrm{in}} \mathbb{1}_V \right| \\
&\leq \frac{C \log V \sqrt{V}}{\sqrt{d}}.
\end{aligned}
$$

- Moreover,

$$
\mathrm{tr}\big( (\boldsymbol{I}_V - \tfrac{1}{V} \mathbb{1}_V \mathbb{1}_V^\top) \boldsymbol{S} (\boldsymbol{I}_V - \tfrac{1}{V} \mathbb{1}_V \mathbb{1}_V^\top) \boldsymbol{S} \big) = \mathrm{tr}(\boldsymbol{S}^2) - \frac{2}{V} \| \boldsymbol{S} \mathbb{1}_V \|_2^2 + \frac{1}{V^2} \big( \mathbb{1}_V^\top \boldsymbol{S} \mathbb{1}_V \big)^2.
$$

We have $\mathrm{tr}(\boldsymbol{S}^2) \leq \frac{C V^2 \log^2 V}{d^2}$ and

$$
\boldsymbol{e}_i^\top \boldsymbol{S} \mathbb{1}_V = \frac{1}{2} \sum_{l=1}^V \mathsf{z}_k^\top \boldsymbol{z}_j \boldsymbol{z}_j^\top \boldsymbol{z}_l + \frac{1}{2} \sum_{l=1}^V \mathsf{z}_k^\top \boldsymbol{z}_l \boldsymbol{z}_l^\top \boldsymbol{z}_j = \mathsf{z}_k^\top \boldsymbol{z}_j \boldsymbol{z}_j^\top \boldsymbol{Z}_{\mathrm{in}} \mathbb{1}_V + \mathsf{z}_k^\top \boldsymbol{Z}_{\mathrm{in}} \boldsymbol{Z}_{\mathrm{in}}^\top \boldsymbol{z}_j.
$$

Therefore,

$$
\left| \mathrm{tr}\big( (\boldsymbol{I}_V - \tfrac{1}{V} \mathbb{1}_V \mathbb{1}_V^\top) \boldsymbol{S} (\boldsymbol{I}_V - \tfrac{1}{V} \mathbb{1}_V \mathbb{1}_V^\top) \boldsymbol{S} \big) \right| \leq \frac{C V^2 \log^2 V}{d^2}.
$$

- Moreover, $\| \mathrm{diag}(\boldsymbol{S}) \|_2^2 \leq \frac{C V \log^2 V}{d}$.

Therefore, by Proposition 8, we have

$$
\mathbb{E}\left[ \left( \mathsf{z}_k^\top \boldsymbol{Z}_{\mathrm{in}} \big( \boldsymbol{X}_i^\top \boldsymbol{X}_i - \tfrac{L}{V} \boldsymbol{I}_V \big) \boldsymbol{Z}_{\mathrm{in}}^\top \boldsymbol{Z}_{\mathrm{in}} \big( \boldsymbol{X}_i^\top - \tfrac{1}{V} \mathbb{1}_V \mathbb{1}_L^\top \big) \mathbb{1}_L \right)^2 \Big| \mathsf{Z}_{\mathrm{in}} \right] \leq C \log^2 V \left( \frac{L}{d} + \frac{L^2}{d^2} \right).
$$

$\square$

## C  Proof of Theorem 1

We consider the following technical assumptions for the subsequent proof.

**Assumption 4** (Technical conditions). *We work under the following conditions:*

- ***Permutation.*** *Without loss of generality, assume $\boldsymbol{\Pi} = \boldsymbol{I}_V$.*

- ***Learning rates.*** *Take $\eta = o_V(1)$, chosen sufficiently small so that we can write $\hat{\boldsymbol{p}}_1 = \tfrac{1}{V} \mathbb{1}_V + O(\eta)$.*

- ***Activation.*** *We consider a polynomial activation $\phi$ with a degree of $p_\star$ satisfying:*

  - $\phi(0), \phi'(0), \phi''(0) \neq 0$
  - *The smallest non-zero Hermite component of $\phi$ has index $q_\star$, i.e, $q^\star := \min\{k > 0 \,|\, \mathbb{E}[\phi(Z) H_{e_k}(Z)] \neq 0\}$, for $Z \sim N(0,1)$.*

Since the learning algorithm does not assume any structure in the ground-truth permutation, we may, without loss of generality, take it to be the identity. This simplifies the notation in the analysis below. The learning rate $\eta$ is chosen sufficiently small so that the network output remains close to its initialization $\tfrac{1}{V} \mathbb{1}_V$, which simplifies the analysis of the three-step gradient descent algorithm. The assumption on the activation function is technical and is needed for the analysis of the three-step gradient descent dynamics; however, we believe that such an assumption would not be necessary for general multi-step training.

## C.1 ATTENTION SCORES AND THEIR ASYMPTOTIC SCALING

Let $\mathsf{X}$ be an independent copy of input sequence. By using the technical condition above, we decompose the attention scores in to three terms $\boldsymbol{s}_1, \boldsymbol{s}_2, \boldsymbol{s}_3 \in \mathbb{R}^L$:

$$
\mathsf{X}\mathsf{Z}_{\mathrm{in}}^\top \boldsymbol{W}_{\mathrm{KQ}}^{(1)}
$$

$$
= \frac{\eta\gamma}{N^2 L^2} \mathsf{X}\mathsf{Z}_{\mathrm{in}}^\top \mathsf{Z}_{\mathrm{in}} \sum_{i,j=1}^N \alpha_{ij} \boldsymbol{X}_i^\top \left(\boldsymbol{I}_L - \tfrac{1}{L}\mathbb{1}_L \mathbb{1}_L^\top\right) \boldsymbol{X}_i \boldsymbol{Z}_{\mathrm{in}}^\top
$$

$$
\times \boldsymbol{Z}_{\mathrm{in}} \boldsymbol{X}_j^\top \mathbb{1}_L (\boldsymbol{x}_j - \tfrac{1}{V}\mathbb{1}_V)^\top \boldsymbol{Z}_{\mathrm{out}}^\top \boldsymbol{Z}_{\mathrm{out}} (\boldsymbol{x}_i - \tfrac{1}{V}\mathbb{1}_V)
$$

$$
+ \frac{\eta\gamma}{N^2 L^2} \mathsf{X}\mathsf{Z}_{\mathrm{in}}^\top \mathsf{Z}_{\mathrm{in}} \sum_{i,j=1}^N \beta_{ij} \boldsymbol{X}_i^\top \left(\boldsymbol{I}_L - \tfrac{1}{L}\mathbb{1}_L \mathbb{1}_L^\top\right) \boldsymbol{X}_i \boldsymbol{Z}_{\mathrm{in}}^\top
$$

$$
\times \boldsymbol{Z}_{\mathrm{in}} \boldsymbol{X}_i^\top \mathbb{1}_L (\boldsymbol{x}_j - \tfrac{1}{V}\mathbb{1}_V)^\top \boldsymbol{Z}_{\mathrm{out}}^\top \boldsymbol{Z}_{\mathrm{out}} (\boldsymbol{x}_i - \tfrac{1}{V}\mathbb{1}_V)
$$

$$
+ \frac{\eta\gamma}{N^2 L} \mathsf{X}\mathsf{Z}_{\mathrm{in}}^\top \mathsf{Z}_{\mathrm{in}} \sum_{i,j=1}^N \boldsymbol{X}_i^\top \left(\boldsymbol{I}_L - \tfrac{1}{L}\mathbb{1}_L \mathbb{1}_L^\top\right) \boldsymbol{X}_i \boldsymbol{Z}_{\mathrm{in}}^\top
$$

$$
\times \mathrm{FW}(\boldsymbol{W}_{\mathrm{in}}; \boldsymbol{Z}_{\mathrm{in}}, \boldsymbol{X}_i, \boldsymbol{X}_j)(\boldsymbol{x}_j - \tfrac{1}{V}\mathbb{1}_V)^\top \boldsymbol{Z}_{\mathrm{out}}^\top \boldsymbol{Z}_{\mathrm{out}} (\boldsymbol{x}_i - \tfrac{1}{V}\mathbb{1}_V)
$$

$$(\mathrm{C.1})$$

$$
+ O(\eta^2 \gamma \mathrm{poly}(d, N))
$$

$$
=: \eta\gamma\big(\boldsymbol{s}_1 + \boldsymbol{s}_2 + \boldsymbol{s}_3\big) + O(\eta^2 \gamma \mathrm{poly}(d, N)).
$$

The following theorem characterizes the scaling of each term. We recall that $\{\boldsymbol{e}_1, \cdots, \boldsymbol{e}_L\}$ denotes the standard basis vectors in $\mathbb{R}^L$.

**Theorem 3.** *With probability at least $1 - o_V(1)$, we have the following:*

- *For all $l \in [L]$, $\left|\langle \boldsymbol{e}_l, \boldsymbol{s}_1 \rangle - \frac{\mathbb{1}_{l=1}}{VL^2}\right| \lesssim \frac{\mathbb{1}_{l=1}}{\sqrt{NV}L^{3/2}d} + \frac{1}{N\sqrt{L}d(d \wedge L^2)^{1/2}(d \wedge L)^{1/2}}$,*

- $\|\boldsymbol{s}_2\|_\infty \lesssim \frac{1}{N\sqrt{L}d(L \wedge d)} + \frac{1}{NLd(L \wedge d)^{1/2}}$

- $\|\boldsymbol{s}_3\|_\infty \lesssim \frac{1}{Nd\sqrt{m}}$.

We first make an observation that we will frequently rely on in the following:

**Proposition 3.** *For any $p \in \mathbb{N}$, we have*

$$
\mathbb{E}[\|\boldsymbol{A}_{1,ir}\|_2^p] \vee \mathbb{E}[\|\boldsymbol{A}_{2,ir}\|_2^p] \vee \mathbb{E}[\|\boldsymbol{A}_{3,ir}\|_2^p] \le \mathrm{poly}_{p,p_\star}(d, V, L).
$$

*Proof.* By Proposition 13, we observe that $\alpha_{ij} \le \mathrm{poly}_{p_\star}(d, V, L)$. Therefore, we have

$$
\|\boldsymbol{A}_{1,ir}\|_2 \vee \|\boldsymbol{A}_{2,ir}\|_2 \vee \|\boldsymbol{A}_{3,ir}\|_2 \le \mathrm{poly}_{p_\star}(d, V, L)\|\mathsf{Z}_{\mathrm{in}}\mathsf{Z}_{\mathrm{in}}^\top\|_2,
$$

from which the result follows. $\qquad\square$

## D  PROOF OF THEOREM 3

We observe that

$$
\mathsf{X}\mathsf{Z}_{\mathrm{in}}^\top \mathsf{Z}_{\mathrm{in}} \boldsymbol{X}_i^\top = \left(\boldsymbol{Z}_{\mathrm{in}}\boldsymbol{X}^\top + \mathsf{Z}_{\mathrm{in}}\boldsymbol{e}_{V+1}\boldsymbol{e}_1^\top\right)^\top \left(\boldsymbol{Z}_{\mathrm{in}}\boldsymbol{X}_i^\top + \mathsf{Z}_{\mathrm{in}}\boldsymbol{e}_{V+1}\boldsymbol{e}_1^\top\right)
$$

$$
= \boldsymbol{X}\boldsymbol{Z}_{\mathrm{in}}^\top \boldsymbol{Z}_{\mathrm{in}}\boldsymbol{X}_i^\top + \boldsymbol{e}_1 \mathsf{z}_{\mathrm{trig}}^\top \boldsymbol{Z}_{\mathrm{in}}\boldsymbol{X}_i^\top + \boldsymbol{X}\boldsymbol{Z}_{\mathrm{in}}^\top \mathsf{z}_{\mathrm{trig}}\boldsymbol{e}_1^\top + \|\mathsf{z}_{\mathrm{trig}}\|_2^2 \boldsymbol{e}_1\boldsymbol{e}_1^\top. \quad (\mathrm{D.1})
$$

In the following, we will consider $\boldsymbol{x}_l = \boldsymbol{e}_k$, for a fixed $k \in [V]$. We will write

$$
\left(\mathsf{z}_{\mathrm{trig}}\boldsymbol{e}_1^\top + \boldsymbol{Z}_{\mathrm{in}}\boldsymbol{X}^\top\right)\boldsymbol{e}_l = \mathsf{z}_k + \mathbb{1}_{l=1}\mathsf{z}_{\mathrm{trig}} = \mathsf{z}_k, \quad (\mathrm{D.2})
$$

and

$$\left( e_1 z_{\text{trig}}^\top Z_{\text{in}} X^\top + \| z_{\text{trig}} \|_2^2 e_1 e_1^\top \right) e_l = \underbrace{\langle z_k, z_{\text{trig}} \rangle}_{=: \mu_{kl}} e_1 = \mu_{kl} e_1. \tag{D.3}$$

In the following, we will consider the event.

$$\texttt{Event} := (E1) \cap (E2).$$

## D.1 CONCENTRATION BOUND FOR $\mathbf{s_1}$

By (D.1)-(D.2)-(D.3), we can write that

$$\langle e_l, s_1 \rangle = \frac{1}{N^2 L^2} \sum_{i,j=1}^N \alpha_{ij} z_k^\top Z_{\text{in}} X_i^\top \left( I_L - \frac{1}{L} \mathbb{1}_L \mathbb{1}_L^\top \right) X_i Z_{\text{in}}^\top$$

$$\times Z_{\text{in}} X_j^\top \mathbb{1}_L (x_j - \tfrac{1}{V} \mathbb{1}_V)^\top Z_{\text{out}}^\top Z_{\text{out}} (x_i - \tfrac{1}{V} \mathbb{1}_V)$$

$$+ \frac{\mu_{kl}}{N^2 L^2} \sum_{i,j=1}^N \alpha_{ij} \left( e_1 - \frac{1}{L} \mathbb{1}_L \right)^\top X_i Z_{\text{in}}^\top Z_{\text{in}} X_j^\top \mathbb{1}_L (x_j - \tfrac{1}{V} \mathbb{1}_V)^\top Z_{\text{out}}^\top Z_{\text{out}} (x_i - \tfrac{1}{V} \mathbb{1}_V)$$

$$=: \vartheta + \varphi.$$

We will analyze $\vartheta$ and $\varphi$ separately. We define

$$B_i := B_{i,1} + B_{i,2} + B_{i,3},$$

where

$$B_{i,1} := \left( \frac{1}{NL} \sum_{j=1}^N \alpha_{ij} (x_j - \tfrac{1}{V} \mathbb{1}_V)(x_j - \tfrac{1}{V} \mathbb{1}_V)^\top \right)$$

$$B_{i,2} := \left( \frac{1}{NL} \sum_{j=1}^N \alpha_{ij} \left( N_j^\top - \tfrac{1}{V} \mathbb{1}_V \mathbb{1}_{L-1}^\top \right) \mathbb{1}_{L-1} (x_j - \tfrac{1}{V} \mathbb{1}_V)^\top \right)$$

$$B_{i,3} := \frac{1}{V} \mathbb{1}_V \left( \frac{1}{NL} \sum_{j=1}^N \alpha_{ij} (x_j - \tfrac{1}{V} \mathbb{1}_V)^\top \right).$$

### D.1.1 CONCENTRATION BOUND FOR $\vartheta$

We define

$$C_i := \frac{1}{L} (x_i - \tfrac{1}{V} \mathbb{1}_V) z_k^\top Z_{\text{in}} X_i^\top \left( I_L - \frac{1}{L} \mathbb{1}_L \mathbb{1}_L^\top \right) X_i Z_{\text{in}}^\top Z_{\text{in}}$$

$$= \underbrace{\frac{1}{L} (x_i - \tfrac{1}{V} \mathbb{1}_V) z_k^\top Z_{\text{in}} X_i^\top X_i Z_{\text{in}}^\top Z_{\text{in}}}_{:= C_{i,1}} - \underbrace{\frac{1}{L^2} (x_i - \tfrac{1}{V} \mathbb{1}_V) z_k^\top Z_{\text{in}} X_i^\top \mathbb{1}_L \mathbb{1}_L^\top X_i Z_{\text{in}}^\top Z_{\text{in}}}_{:= C_{i,2}}$$

By Chebyshev's inequality, with probability $1 - o_V(1)$,

$$\vartheta = \frac{1}{N} \sum_{i=1}^N \text{tr}(B_i Z_{\text{out}}^\top Z_{\text{out}} C_i) = \text{tr}\left( Z_{\text{out}} \frac{1}{N} \sum_{i=1}^N C_i B_i Z_{\text{out}}^\top \right)$$

$$= \underbrace{\text{tr}\left( \frac{1}{N} \sum_{i=1}^N C_i B_i \right)}_{\vartheta_1} \pm \underbrace{\frac{1}{\sqrt{d}} \left\| \frac{\log V}{N} \sum_{i=1}^N C_i B_i \right\|_F}_{\vartheta_2}.$$

**Bounding $\vartheta_2$:** We start with bounding $\vartheta_2$ term. We have

$$\vartheta_2 \leq \frac{1}{\sqrt{d}}\Big\|\frac{1}{N}\sum_{i=1}^{N}\boldsymbol{C}_i\boldsymbol{B}_{i,1}\Big\|_F + \frac{1}{\sqrt{d}}\Big\|\frac{1}{N}\sum_{i=1}^{N}\boldsymbol{C}_i\boldsymbol{B}_{i,2}\Big\|_F + \frac{1}{\sqrt{d}}\Big\|\frac{1}{N}\sum_{i=1}^{N}\boldsymbol{C}_i\boldsymbol{B}_{i,3}\Big\|_F.$$

We have

- $\left\|\frac{1}{N}\sum_{i=1}^{N}\boldsymbol{C}_i\boldsymbol{B}_{i,1}\right\|_F^2 \leq \frac{2}{N^2}\Big(\sum_{i,r=1}^{N}\operatorname{tr}(\boldsymbol{C}_{i,1}\boldsymbol{B}_{i,1}\boldsymbol{B}_{r,1}^\top\boldsymbol{C}_{r,1}^\top) + \operatorname{tr}(\boldsymbol{C}_{i,2}\boldsymbol{B}_{i,1}\boldsymbol{B}_{r,1}^\top\boldsymbol{C}_{r,2}^\top)\Big).$

- $\left\|\frac{1}{N}\sum_{i=1}^{N}\boldsymbol{C}_i\boldsymbol{B}_{i,2}\right\|_F^2 \leq \frac{2}{N^2}\Big(\sum_{i,r=1}^{N}\operatorname{tr}(\boldsymbol{C}_{i,2}\boldsymbol{B}_{i,2}\boldsymbol{B}_{r,2}^\top\boldsymbol{C}_{r,2}^\top) + \operatorname{tr}(\boldsymbol{C}_{i,2}\boldsymbol{B}_{i,2}\boldsymbol{B}_{r,2}^\top\boldsymbol{C}_{r,2}^\top)\Big).$

- $\left\|\frac{1}{N}\sum_{i=1}^{N}\boldsymbol{C}_i\boldsymbol{B}_{i,3}\right\|_F^2 \leq \frac{2}{N^2}\Big(\sum_{i,r=1}^{N}\operatorname{tr}(\boldsymbol{C}_{i,1}\boldsymbol{B}_{i,3}\boldsymbol{V}_{r,3}^\top\boldsymbol{C}_{r,1}^\top) + \operatorname{tr}(\boldsymbol{C}_{i,2}\boldsymbol{B}_{i,3}\boldsymbol{B}_{r,3}^\top\boldsymbol{C}_{r,2}^\top)\Big).$

We define the scalars

$$t_1 \coloneqq \frac{\phi'(0)^4}{d}\frac{(1-\frac{1}{V})}{L^2}\Big(\frac{1}{N}+\frac{1}{V}\Big), \qquad t_2 \coloneqq \frac{\phi'(0)^4}{d}(1-\frac{1}{V})^2\frac{L-1}{L^2N}, \qquad t_3 \coloneqq \frac{\phi'(0)^4}{NV^2L^2}.$$

First, we will bound the first two terms. Let $* \in \{1,2\}$.

*Bounding first two terms.* For $i \neq r$, by using the definition in $\boldsymbol{A}_{1,ir}$ and $\boldsymbol{A}_{2,ir}$ in (B.3)-(B.4), we have

$$\operatorname{tr}(\boldsymbol{C}_{i,1}\boldsymbol{B}_{i,*}\boldsymbol{B}_{r,*}^\top\boldsymbol{C}_{r,1}^\top) = \frac{1}{L^2}(\mathbb{1}_{\boldsymbol{x}_i=\boldsymbol{x}_r}-\tfrac{1}{V})\mathsf{z}_k^\top\boldsymbol{Z}_{\mathrm{in}}\boldsymbol{X}_i^\top\boldsymbol{X}_i\boldsymbol{Z}_{\mathrm{in}}^\top\boldsymbol{A}_{*,ir}\boldsymbol{Z}_{\mathrm{in}}\boldsymbol{X}_r^\top\boldsymbol{X}_r\boldsymbol{Z}_{\mathrm{in}}^\top\mathsf{z}_k$$

$$= \frac{t_*}{L^2}(\mathbb{1}_{\boldsymbol{x}_i=\boldsymbol{x}_r}-\tfrac{1}{V})\mathsf{z}_k^\top\boldsymbol{Z}_{\mathrm{in}}\boldsymbol{X}_i^\top\boldsymbol{X}_i\boldsymbol{Z}_{\mathrm{in}}^\top\boldsymbol{Z}_{\mathrm{in}}\boldsymbol{X}_r^\top\boldsymbol{X}_r\boldsymbol{Z}_{\mathrm{in}}^\top\mathsf{z}_k$$

$$+ \frac{1}{L^2}(\mathbb{1}_{\boldsymbol{x}_i=\boldsymbol{x}_r}-\tfrac{1}{V})\mathsf{z}_k^\top\boldsymbol{Z}_{\mathrm{in}}\boldsymbol{X}_i^\top\boldsymbol{X}_i\boldsymbol{Z}_{\mathrm{in}}^\top(\boldsymbol{A}_{*,ir}-t_*\boldsymbol{I}_d)\boldsymbol{Z}_{\mathrm{in}}\boldsymbol{X}_r^\top\boldsymbol{X}_r\boldsymbol{Z}_{\mathrm{in}}^\top\mathsf{z}_k$$

$$\leq \frac{t_*}{L^2}(\mathbb{1}_{\boldsymbol{x}_i=\boldsymbol{x}_r}-\tfrac{1}{V})\mathsf{z}_k^\top\boldsymbol{Z}_{\mathrm{in}}\boldsymbol{X}_i^\top\boldsymbol{X}_i\boldsymbol{Z}_{\mathrm{in}}^\top\boldsymbol{Z}_{\mathrm{in}}\boldsymbol{X}_r^\top\boldsymbol{X}_r\boldsymbol{Z}_{\mathrm{in}}^\top\mathsf{z}_k$$

$$+ \frac{1}{L^2}(\mathbb{1}_{\boldsymbol{x}_i=\boldsymbol{x}_r}-\tfrac{1}{V})\|\boldsymbol{A}_{*,ir}-t_*\boldsymbol{I}_d\|_2\|\boldsymbol{Z}_{\mathrm{in}}\boldsymbol{X}_r^\top\boldsymbol{X}_r\boldsymbol{Z}_{\mathrm{in}}^\top\mathsf{z}_k\|_2\|\boldsymbol{Z}_{\mathrm{in}}\boldsymbol{X}_i^\top\boldsymbol{X}_i\boldsymbol{Z}_{\mathrm{in}}^\top\mathsf{z}_k\|_2.$$

By (P1) in Proposition 2, (E1) implies

$$\frac{t_*}{L^2}\Big|\,\mathbb{E}\Big[(\mathbb{1}_{\boldsymbol{x}_i=\boldsymbol{x}_r}-\tfrac{1}{V})\mathsf{z}_k^\top\boldsymbol{Z}_{\mathrm{in}}\boldsymbol{X}_i^\top\boldsymbol{X}_i\boldsymbol{Z}_{\mathrm{in}}^\top\boldsymbol{Z}_{\mathrm{in}}\boldsymbol{X}_r^\top\boldsymbol{X}_r\boldsymbol{Z}_{\mathrm{in}}^\top\mathsf{z}_k\Big|\boldsymbol{Z}_{\mathrm{in}}\Big]\Big| \leq \frac{Ct_*}{L^2}\frac{1}{Vd}. \tag{D.4}$$

Moreover, by using (R4) and (P2), we have

$$\frac{1}{L^2}\,\mathbb{E}\Big[(\mathbb{1}_{\boldsymbol{x}_i=\boldsymbol{x}_r}-\tfrac{1}{V})\|\boldsymbol{A}_{*,ir}-t_*\boldsymbol{I}_d\|_2\|\boldsymbol{Z}_{\mathrm{in}}\boldsymbol{X}_r^\top\boldsymbol{X}_r\boldsymbol{Z}_{\mathrm{in}}^\top\mathsf{z}_k\|_2\|\boldsymbol{Z}_{\mathrm{in}}\boldsymbol{X}_i^\top\boldsymbol{X}_i\boldsymbol{Z}_{\mathrm{in}}^\top\mathsf{z}_k\|_2 \,\Big|\, \boldsymbol{Z}_{\mathrm{in}}\Big]$$

$$\leq \frac{C}{Vd(L\wedge d)}\begin{cases} \phi'(0)^2\Big(\frac{1}{NdL^3}+\frac{1}{VdL^2}\frac{1}{V\wedge L^2\wedge L\sqrt{d}}\Big) + \phi'(0)^4\Big(\frac{\log V}{L^2V^{3/2}\sqrt{d}}+\frac{\log^2 V}{L^2N\sqrt{V}d}\Big), & *=1 \\[2mm] \frac{\sqrt{V}}{d\sqrt{NL}}\Big(\frac{1}{NL^{\frac{3}{2}}}+\frac{1}{V\sqrt{L}}\frac{1}{V\wedge L^2\wedge L\sqrt{d}}\Big) + \phi'(0)^4\Big(\frac{\log V}{NL\sqrt{V}d}+\frac{\log^3 V}{N\sqrt{L}Vd}\Big), & *=2 \end{cases}$$

$$\leq \frac{C}{N^{3/2}\sqrt{V}d^2L^2}\frac{1}{L\wedge d} + \frac{C}{V^{3/2}\sqrt{N}Ld^2}\frac{1}{L\wedge d}\frac{1}{V\wedge L^2\wedge L\sqrt{d}} + \frac{C\log^3 V}{NV^{3/2}L^{1/2}d^{3/2}}\frac{1}{(L\wedge d)^{3/2}}. \tag{D.5}$$

On the other hand,

$$\operatorname{tr}(\boldsymbol{C}_{i,2}\boldsymbol{B}_{i,*}\boldsymbol{B}_{r,*}^\top\boldsymbol{C}_{r,2}^\top)$$

$$= \frac{1}{L^4}(\mathbb{1}_{\boldsymbol{x}_i=\boldsymbol{x}_r}-\tfrac{1}{V})\mathsf{z}_k^\top\boldsymbol{Z}_{\mathrm{in}}\boldsymbol{X}_i^\top\mathbb{1}_L\mathbb{1}_L^\top\boldsymbol{X}_i\boldsymbol{Z}_{\mathrm{in}}^\top\boldsymbol{A}_{*,ir}\boldsymbol{Z}_{\mathrm{in}}\boldsymbol{X}_r^\top\mathbb{1}_L\mathbb{1}_L^\top\boldsymbol{X}_r\boldsymbol{Z}_{\mathrm{in}}^\top\mathsf{z}_k$$

$$= \frac{t_*}{L^4}(\mathbb{1}_{\boldsymbol{x}_i=\boldsymbol{x}_r}-\tfrac{1}{V})\mathsf{z}_k^\top\boldsymbol{Z}_{\mathrm{in}}\boldsymbol{X}_i^\top\mathbb{1}_L\mathbb{1}_L^\top\boldsymbol{X}_i\boldsymbol{Z}_{\mathrm{in}}^\top\boldsymbol{Z}_{\mathrm{in}}\boldsymbol{X}_r^\top\mathbb{1}_L\mathbb{1}_L^\top\boldsymbol{X}_r\boldsymbol{Z}_{\mathrm{in}}^\top\mathsf{z}_k$$

$$+ \frac{1}{L^4}(\mathbb{1}_{\boldsymbol{x}_i = \boldsymbol{x}_r} - \tfrac{1}{V})\mathsf{z}_k^\top \boldsymbol{Z}_{\mathrm{in}} \boldsymbol{X}_i^\top \mathbb{1}_L \mathbb{1}_L^\top \boldsymbol{X}_i \boldsymbol{Z}_{\mathrm{in}}^\top (\boldsymbol{A}_{*,ir} - t_* \boldsymbol{I}_d) \boldsymbol{Z}_{\mathrm{in}} \boldsymbol{X}_r^\top \mathbb{1}_L \mathbb{1}_L^\top \boldsymbol{X}_r \boldsymbol{Z}_{\mathrm{in}}^\top \mathsf{z}_k$$

$$\le \frac{t_*}{L^4}(\mathbb{1}_{\boldsymbol{x}_i = \boldsymbol{x}_r} - \tfrac{1}{V})\mathsf{z}_k^\top \boldsymbol{Z}_{\mathrm{in}} \boldsymbol{X}_i^\top \mathbb{1}_L \mathbb{1}_L^\top \boldsymbol{X}_i \boldsymbol{Z}_{\mathrm{in}}^\top \boldsymbol{Z}_{\mathrm{in}} \boldsymbol{X}_r^\top \mathbb{1}_L \mathbb{1}_L^\top \boldsymbol{X}_r \boldsymbol{Z}_{\mathrm{in}}^\top \mathsf{z}_k$$

$$+ \frac{1}{L^4}(\mathbb{1}_{\boldsymbol{x}_i = \boldsymbol{x}_r} - \tfrac{1}{V})\|\boldsymbol{A}_{*,ir} - t_* \boldsymbol{I}_d\|_2 \|\boldsymbol{Z}_{\mathrm{in}} \boldsymbol{X}_r^\top \mathbb{1}_L \mathbb{1}_L^\top \boldsymbol{X}_r \boldsymbol{Z}_{\mathrm{in}}^\top \mathsf{z}_k\|_2 \|\boldsymbol{Z}_{\mathrm{in}} \boldsymbol{X}_i^\top \mathbb{1}_L \mathbb{1}_L^\top \boldsymbol{X}_i \boldsymbol{Z}_{\mathrm{in}}^\top \mathsf{z}_k\|_2.$$

By using (E1.4), (E2.2), and (R1), we have

$$\frac{t_*}{L^4} \mathbb{E}\left[(\mathbb{1}_{\boldsymbol{x}_i = \boldsymbol{x}_r} - \tfrac{1}{V})\mathsf{z}_k^\top \boldsymbol{Z}_{\mathrm{in}} \boldsymbol{X}_i^\top \mathbb{1}_L \mathbb{1}_L^\top \boldsymbol{X}_i \boldsymbol{Z}_{\mathrm{in}}^\top \boldsymbol{Z}_{\mathrm{in}} \boldsymbol{X}_r^\top \mathbb{1}_L \mathbb{1}_L^\top \boldsymbol{X}_r \boldsymbol{Z}_{\mathrm{in}}^\top \mathsf{z}_k | \mathsf{Z}_{\mathrm{in}}\right]$$

$$\le \frac{Ct_*}{VL} \frac{\log^2 V}{V \wedge L^2 \wedge L\sqrt{d}} \frac{1}{L \wedge d}.$$

Moreover, by using (E1.4), (E2.2), (R1), and (R4)

$$\frac{1}{L^4} \mathbb{E}\left[(\mathbb{1}_{\boldsymbol{x}_i = \boldsymbol{x}_r} - \tfrac{1}{V})\|\boldsymbol{A}_{*,ir} - t_* \boldsymbol{I}_d\|_2 \|\boldsymbol{Z}_{\mathrm{in}} \boldsymbol{X}_r^\top \mathbb{1}_L \mathbb{1}_L^\top \boldsymbol{X}_r \boldsymbol{Z}_{\mathrm{in}}^\top \mathsf{z}_k\|_2 \|\boldsymbol{Z}_{\mathrm{in}} \boldsymbol{X}_i^\top \mathbb{1}_L \mathbb{1}_L^\top \boldsymbol{X}_i \boldsymbol{Z}_{\mathrm{in}}^\top \mathsf{z}_k\|_2 \;\Big|\; \mathsf{Z}_{\mathrm{in}}\right]$$

$$\le \frac{C}{VL^2(L \wedge d)} \begin{cases} \phi'(0)^2\left(\frac{1}{NdL^3} + \frac{1}{VdL^2}\frac{1}{V \wedge L^2 \wedge L\sqrt{d}}\right) + \phi'(0)^4\left(\frac{\log V}{L^2 V^{3/2}\sqrt{d}} + \frac{\log^2 V}{L^2 N\sqrt{V}d}\right), & * = 1 \\[2mm] \frac{\sqrt{V}}{d\sqrt{NL}}\left(\frac{1}{NL^{\frac{3}{2}}} + \frac{1}{V\sqrt{L}}\frac{1}{V \wedge L^2 \wedge L\sqrt{d}}\right) + \phi'(0)^4\left(\frac{\log V}{NL\sqrt{V}d} + \frac{\log^3 V}{N\sqrt{LV}d}\right), & * = 2 \end{cases}$$

$$\le \frac{C}{N^{3/2}\sqrt{V}dL^4}\frac{1}{L \wedge d} + \frac{C}{V^{3/2}\sqrt{N}L^3 d}\frac{1}{L \wedge d}\frac{1}{V \wedge L^2 \wedge L\sqrt{d}} + \frac{C\log^3 V}{NV^{3/2}L^{5/2}\sqrt{d}}\frac{1}{(L \wedge d)^{3/2}}.$$

$$\tag{D.6}$$

On the other hand, for $i = r$, by (R4),

$$\operatorname{tr}(\boldsymbol{C}_{i,1}\boldsymbol{B}_{i,*}^\top \boldsymbol{B}_{i,*}^\top \boldsymbol{C}_{i,1}^\top) + \operatorname{tr}(\boldsymbol{C}_{i,2}\boldsymbol{B}_{i,*}^\top \boldsymbol{B}_{i,*}^\top \boldsymbol{C}_{i,2}^\top)$$

$$= \frac{1}{L^2}(1 - \tfrac{1}{V})\mathsf{z}_k^\top \boldsymbol{Z}_{\mathrm{in}} \boldsymbol{X}_i^\top \boldsymbol{X}_i \boldsymbol{Z}_{\mathrm{in}}^\top \boldsymbol{A}_{*,ii} \boldsymbol{Z}_{\mathrm{in}} \boldsymbol{X}_i^\top \boldsymbol{X}_i \boldsymbol{Z}_{\mathrm{in}}^\top \mathsf{z}_k$$

$$+ \frac{(1 - \tfrac{1}{V})}{L^4}\mathsf{z}_k^\top \boldsymbol{Z}_{\mathrm{in}} \boldsymbol{X}_i^\top \mathbb{1}_L \mathbb{1}_L^\top \boldsymbol{X}_i \boldsymbol{Z}_{\mathrm{in}}^\top \boldsymbol{A}_{*,ii} \boldsymbol{Z}_{\mathrm{in}} \boldsymbol{X}_i^\top \mathbb{1}_L \mathbb{1}_L^\top \boldsymbol{X}_i \boldsymbol{Z}_{\mathrm{in}}^\top \mathsf{z}_k$$

$$\le \frac{t_*}{L^2}(1 - \tfrac{1}{V})\|\boldsymbol{Z}_{\mathrm{in}} \boldsymbol{X}_i^\top \boldsymbol{X}_i \boldsymbol{Z}_{\mathrm{in}}^\top \mathsf{z}_k\|_2^2 + \frac{t_*}{L^4}\|\boldsymbol{Z}_{\mathrm{in}} \boldsymbol{X}_i^\top \mathbb{1}_L \mathbb{1}_L^\top \boldsymbol{X}_i \boldsymbol{Z}_{\mathrm{in}}^\top \mathsf{z}_k\|_2^2.$$

By using (E1.4), (E2.2), (R1), and (P2)

$$\frac{t_*}{L^2} \mathbb{E}\left[\|\boldsymbol{Z}_{\mathrm{in}} \boldsymbol{X}_i^\top \boldsymbol{X}_i \boldsymbol{Z}_{\mathrm{in}}^\top \mathsf{z}_k\|_2^2 \Big| \mathsf{Z}_{\mathrm{in}}\right] + \frac{t_*}{L^4} \mathbb{E}\left[\|\boldsymbol{Z}_{\mathrm{in}} \boldsymbol{X}_i^\top \mathbb{1}_L \mathbb{1}_L^\top \boldsymbol{X}_i \boldsymbol{Z}_{\mathrm{in}}^\top \mathsf{z}_k\|_2^2 \Big| \mathsf{Z}_{\mathrm{in}}\right]$$

$$\le \frac{Ct_*}{L^2}\left(\frac{L}{d} + \frac{L^2}{d^2}\right) + \frac{Ct_*}{L^2}\frac{1}{L \wedge d}.$$

$$\tag{D.7}$$

Therefore, we have by (D.4)-(D.5)-(D.6)-(D.7) and using $N \ll VL$ and $L \ll V$, we have

$$\mathbb{E}\left[\left\|\frac{1}{N}\sum_{i=1}^N \boldsymbol{C}_i \boldsymbol{B}_{i,1}\right\|_F^2 \Big| \mathsf{Z}_{\mathrm{in}}\right] + \mathbb{E}\left[\left\|\frac{1}{N}\sum_{i=1}^N \boldsymbol{C}_i \boldsymbol{B}_{i,2}\right\|_F^2 \Big| \mathsf{Z}_{\mathrm{in}}\right]$$

$$\le \frac{C}{N^2 dL(d \wedge L^2)(d \wedge L)} + \frac{C}{N^{3/2}\sqrt{V}dL^2(d \wedge L^2)(L \wedge d)}$$

$$+ \frac{C}{V^{3/2}\sqrt{N}Ld(d \wedge L^2)(L \wedge d)}\frac{1}{V \wedge L^2 \wedge L\sqrt{d}} + \frac{C\log^3 V}{NV^{3/2}\sqrt{Ld}(d \wedge L^2)(L \wedge d)^{3/2}}$$

$$\le \frac{C}{N^2 dL(d \wedge L^2)(d \wedge L)} + \frac{C\log^3 V}{NV^{3/2}\sqrt{Ld}(d \wedge L^2)(L \wedge d)^{3/2}}.$$

$$\tag{D.8}$$

*Bounding the third term.* We have

$$\Big\| \frac{1}{N} \sum_{i=1}^N \boldsymbol{C}_i \boldsymbol{B}_{i,3} \Big\|_F^2 \le \frac{2}{N^2} \Big( \sum_{i,r=1}^N \mathrm{tr}(\boldsymbol{C}_{i,1} \boldsymbol{B}_{i,3} \boldsymbol{B}_{r,3}^\top \boldsymbol{C}_{r,1}^\top) + \mathrm{tr}(\boldsymbol{C}_{i,2} \boldsymbol{B}_{i,3} \boldsymbol{B}_{r,3}^\top \boldsymbol{C}_{r,2}^\top) \Big).$$

We recall the definition $\tilde{\Delta}_{3,ir}$ in (R4):

$$\tilde{\Delta}_{3,ir} = \Big( \frac{1}{N} \sum_{j=1}^N \alpha_{ij}(\boldsymbol{x}_j - \tfrac{1}{V} \mathbb{1}_V) \Big)^\top \Big( \frac{1}{N} \sum_{j=1}^N \alpha_{rj}(\boldsymbol{x}_j - \tfrac{1}{V} \mathbb{1}_V) \Big) - \frac{\phi'(0)^4}{N}.$$

We have for $i \ne r$,

$$\mathrm{tr}(\boldsymbol{C}_{i,1} \boldsymbol{B}_{i,3} \boldsymbol{B}_{r,3}^\top \boldsymbol{C}_{r,1}^\top) + \mathrm{tr}(\boldsymbol{C}_{i,2} \boldsymbol{B}_{i,3} \boldsymbol{B}_{r,3}^\top \boldsymbol{C}_{r,2}^\top)$$

$$= \frac{1}{L^2}(\mathbb{1}_{\boldsymbol{x}_i = \boldsymbol{x}_r} - \tfrac{1}{V}) \mathsf{z}_k^\top \boldsymbol{Z}_{\mathrm{in}} \boldsymbol{X}_i^\top \boldsymbol{X}_i \boldsymbol{Z}_{\mathrm{in}}^\top \boldsymbol{A}_{3,ir} \boldsymbol{Z}_{\mathrm{in}} \boldsymbol{X}_r^\top \boldsymbol{X}_r \boldsymbol{Z}_{\mathrm{in}}^\top \mathsf{z}_k$$

$$+ \frac{1}{L^4}(\mathbb{1}_{\boldsymbol{x}_i = \boldsymbol{x}_r} - \tfrac{1}{V}) \mathsf{z}_k^\top \boldsymbol{Z}_{\mathrm{in}} \boldsymbol{X}_i^\top \mathbb{1}_L \mathbb{1}_L^\top \boldsymbol{X}_i \boldsymbol{Z}_{\mathrm{in}}^\top \boldsymbol{A}_{3,ir} \boldsymbol{Z}_{\mathrm{in}} \boldsymbol{X}_r^\top \mathbb{1}_L \mathbb{1}_L^\top \boldsymbol{X}_r \boldsymbol{Z}_{\mathrm{in}}^\top \mathsf{z}_k$$

$$\le \frac{t_3}{L^2}(\mathbb{1}_{\boldsymbol{x}_i = \boldsymbol{x}_r} - \tfrac{1}{V}) \mathsf{z}_k^\top \boldsymbol{Z}_{\mathrm{in}} \boldsymbol{X}_i^\top \boldsymbol{X}_i \boldsymbol{Z}_{\mathrm{in}}^\top \boldsymbol{Z}_{\mathrm{in}} \mathbb{1}_V \mathbb{1}_V^\top \boldsymbol{Z}_{\mathrm{in}}^\top \boldsymbol{Z}_{\mathrm{in}} \boldsymbol{X}_r^\top \boldsymbol{X}_r \boldsymbol{Z}_{\mathrm{in}}^\top \mathsf{z}_k$$

$$+ \frac{t_3}{L^4}(\mathbb{1}_{\boldsymbol{x}_i = \boldsymbol{x}_r} - \tfrac{1}{V}) \mathsf{z}_k^\top \boldsymbol{Z}_{\mathrm{in}} \boldsymbol{X}_i^\top \mathbb{1}_L \mathbb{1}_L^\top \boldsymbol{X}_i \boldsymbol{Z}_{\mathrm{in}}^\top \boldsymbol{Z}_{\mathrm{in}} \mathbb{1}_V \mathbb{1}_V^\top \boldsymbol{Z}_{\mathrm{in}}^\top \boldsymbol{Z}_{\mathrm{in}} \boldsymbol{X}_r^\top \mathbb{1}_L \mathbb{1}_L^\top \boldsymbol{X}_r \boldsymbol{Z}_{\mathrm{in}}^\top \mathsf{z}_k$$

$$+ \frac{\tilde{\Delta}_{3,ir}}{V^2 L^4}(\mathbb{1}_{\boldsymbol{x}_i = \boldsymbol{x}_r} - \tfrac{1}{V}) \mathsf{z}_k^\top \boldsymbol{Z}_{\mathrm{in}} \boldsymbol{X}_i^\top \boldsymbol{X}_i \boldsymbol{Z}_{\mathrm{in}}^\top \mathbb{1}_V \mathbb{1}_V^\top \boldsymbol{Z}_{\mathrm{in}} \boldsymbol{X}_r^\top \boldsymbol{X}_r \boldsymbol{Z}_{\mathrm{in}}^\top \mathsf{z}_k$$

$$+ \frac{\tilde{\Delta}_{3,ir}}{V^2 L^6}(\mathbb{1}_{\boldsymbol{x}_i = \boldsymbol{x}_r} - \tfrac{1}{V}) \mathsf{z}_k^\top \boldsymbol{Z}_{\mathrm{in}} \boldsymbol{X}_i^\top \mathbb{1}_L \mathbb{1}_L^\top \boldsymbol{X}_i \boldsymbol{Z}_{\mathrm{in}}^\top \mathbb{1}_V \mathbb{1}_V^\top \boldsymbol{Z}_{\mathrm{in}} \boldsymbol{X}_r^\top \mathbb{1}_L \mathbb{1}_L^\top \boldsymbol{X}_r \boldsymbol{Z}_{\mathrm{in}}^\top \mathsf{z}_k$$

For the first term, by (P3),

$$\frac{t_3}{L^2} \mathbb{E}\Big[ (\mathbb{1}_{\boldsymbol{x}_i = \boldsymbol{x}_r} - \tfrac{1}{V}) \mathsf{z}_k^\top \boldsymbol{Z}_{\mathrm{in}} \boldsymbol{X}_i^\top \boldsymbol{X}_i \boldsymbol{Z}_{\mathrm{in}}^\top \boldsymbol{Z}_{\mathrm{in}} \mathbb{1}_V \mathbb{1}_V^\top \boldsymbol{Z}_{\mathrm{in}}^\top \boldsymbol{Z}_{\mathrm{in}} \boldsymbol{X}_r^\top \boldsymbol{X}_r \boldsymbol{Z}_{\mathrm{in}}^\top \mathsf{z}_k | \boldsymbol{Z}_{\mathrm{in}} \Big] \le \frac{C\phi'(0)^4 \log^2 V}{NV^2 L^4} \frac{1}{d^2}.$$

For the second term, by using (E1.4), (E1.5) and (E2.2)

$$\frac{t_3}{L^4} \mathbb{E}\Big[ (\mathbb{1}_{\boldsymbol{x}_i = \boldsymbol{x}_r} - \tfrac{1}{V}) \mathsf{z}_k^\top \boldsymbol{Z}_{\mathrm{in}} \boldsymbol{X}_i^\top \mathbb{1}_L \mathbb{1}_L^\top \boldsymbol{X}_i \boldsymbol{Z}_{\mathrm{in}}^\top \boldsymbol{Z}_{\mathrm{in}} \mathbb{1}_V \mathbb{1}_V^\top \boldsymbol{Z}_{\mathrm{in}}^\top \boldsymbol{Z}_{\mathrm{in}} \boldsymbol{X}_r^\top \mathbb{1}_L \mathbb{1}_L^\top \boldsymbol{X}_r \boldsymbol{Z}_{\mathrm{in}}^\top \mathsf{z}_k | \boldsymbol{Z}_{\mathrm{in}} \Big]$$

$$\le \frac{\phi'(0)^4}{NV^3 L^4} \frac{1}{L \wedge d}\Big(L \vee \frac{V}{d}\Big)$$

For the last two terms, by using (E1.1), (E1.4), (E2.2), (R1), (R4), and (P2),

$$\frac{1}{V^2 L^4} \mathbb{E}\Big[ (\mathbb{1}_{\boldsymbol{x}_i = \boldsymbol{x}_r} - \tfrac{1}{V}) |\tilde{\Delta}_{3,ir}| \| \boldsymbol{Z}_{\mathrm{in}} \boldsymbol{X}_i^\top \boldsymbol{X}_i \boldsymbol{Z}_{\mathrm{in}}^\top \mathsf{z}_k \|_2 \| \boldsymbol{Z}_{\mathrm{in}} \boldsymbol{X}_r^\top \boldsymbol{X}_r \boldsymbol{Z}_{\mathrm{in}}^\top \mathsf{z}_k \|_2 | \boldsymbol{Z}_{\mathrm{in}} \Big] +$$

$$\frac{1}{V^2 L^6} \mathbb{E}\Big[ (\mathbb{1}_{\boldsymbol{x}_i = \boldsymbol{x}_r} - \tfrac{1}{V}) |\tilde{\Delta}_{3,ir}| \| \boldsymbol{Z}_{\mathrm{in}}^\top \boldsymbol{Z}_{\mathrm{in}} \boldsymbol{X}_i^\top \mathbb{1}_L \mathbb{1}_L^\top \boldsymbol{X}_i \boldsymbol{Z}_{\mathrm{in}}^\top \mathsf{z}_k \|_2 \| \boldsymbol{Z}_{\mathrm{in}}^\top \boldsymbol{Z}_{\mathrm{in}} \boldsymbol{X}_r^\top \mathbb{1}_L \mathbb{1}_L^\top \boldsymbol{X}_r \boldsymbol{Z}_{\mathrm{in}}^\top \mathsf{z}_k \|_2 | \boldsymbol{Z}_{\mathrm{in}} \Big]$$

$$\le \frac{C}{V^3 L^4} \Big( \frac{L}{d} + \frac{L^2}{d^2} + \frac{V}{d^2} + \frac{V}{Ld} \Big) \Big( \frac{\phi'(0)^4 \log^2 V}{N\sqrt{V}} + \frac{\phi'(0)^2}{N}\Big( \frac{1}{NL} + \frac{1}{\sqrt{N}} \frac{1}{V \wedge L^2 \wedge L\sqrt{d}} \Big) \Big)$$

$$+ \frac{C}{V^3 L^4} \Big( \frac{L}{d} + \frac{L^2}{d^2} + \frac{V}{d^2} + \frac{V}{Ld} \Big) \Big( \frac{1}{NL} + \frac{1}{\sqrt{N}} \frac{1}{V \wedge L^2 \wedge L\sqrt{d}} \Big)^2$$

$$\le \frac{C}{NV^3 L^2 d(L \wedge d)} \frac{\phi'(0)^4 \log^2 V}{\sqrt{V} \wedge L^4 \wedge L^2 d} + \frac{C}{NV^2 L^4 d^2} \frac{\phi'(0)^4 \log^2 V}{\sqrt{V} \wedge L^4 \wedge L^2 d}. \tag{D.9}$$

For $i = r$, by using (R4),

$$\mathrm{tr}(\boldsymbol{C}_{i,1} \boldsymbol{B}_{i,3} \boldsymbol{B}_{i,3}^\top \boldsymbol{C}_{i,1}^\top) + \mathrm{tr}(\boldsymbol{C}_{i,2} \boldsymbol{B}_{i,3} \boldsymbol{B}_{i,3}^\top \boldsymbol{C}_{i,2}^\top)$$

$$= \frac{1}{L^2}(1 - \frac{1}{V}) \mathsf{z}_k^\top \boldsymbol{Z}_{\mathrm{in}} \boldsymbol{X}_i^\top \boldsymbol{X}_i \boldsymbol{Z}_{\mathrm{in}}^\top \boldsymbol{A}_{3,ir} \boldsymbol{Z}_{\mathrm{in}} \boldsymbol{X}_i^\top \boldsymbol{X}_i \boldsymbol{Z}_{\mathrm{in}}^\top \mathsf{z}_k$$

$$+ \frac{1}{L^4}\Big(1 - \frac{1}{V}\Big)\mathsf{z}_k^\top \mathbf{Z}_{\mathrm{in}} \mathbf{X}_i^\top \mathbb{1}_L \mathbb{1}_L^\top \mathbf{X}_i \mathbf{Z}_{\mathrm{in}}^\top \mathbf{A}_{3,ir} \mathbf{Z}_{\mathrm{in}} \mathbf{X}_i^\top \mathbb{1}_L \mathbb{1}_L^\top \mathbf{X}_i \mathbf{Z}_{\mathrm{in}}^\top \mathsf{z}_k$$

$$\leq \frac{2t_3}{L^2}|\mathbb{1}_V^\top \mathbf{Z}_{\mathrm{in}}^\top \mathbf{Z}_{\mathrm{in}} \mathbf{X}_i^\top \mathbf{X}_i \mathbf{Z}_{\mathrm{in}}^\top \mathsf{z}_k|^2 + \frac{2t_3}{L^4}|\mathbb{1}_V^\top \mathbf{Z}_{\mathrm{in}}^\top \mathbf{Z}_{\mathrm{in}} \mathbf{X}_i^\top \mathbb{1}_L \mathbb{1}_L^\top \mathbf{X}_i \mathbf{Z}_{\mathrm{in}}^\top \mathsf{z}_k|^2.$$

Then, by (P4), (E1.4), (E2.2), and (E1.5), we have

$$\frac{t_3}{L^2}\, \mathbb{E}\Big[ (\mathbb{1}_V^\top \mathbf{Z}_{\mathrm{in}}^\top \mathbf{Z}_{\mathrm{in}} \mathbf{X}_i^\top \mathbf{X}_i \mathbf{Z}_{\mathrm{in}}^\top \mathsf{z}_k)^2 | \mathsf{Z}_{\mathrm{in}}\Big] + \frac{t_3}{L^4}\, \mathbb{E}\Big[ (\mathbb{1}_V^\top \mathbf{Z}_{\mathrm{in}}^\top \mathbf{Z}_{\mathrm{in}} \mathbf{X}_i^\top \mathbb{1}_L \mathbb{1}_L^\top \mathbf{X}_i \mathbf{Z}_{\mathrm{in}}^\top \mathsf{z}_k)^2 | \mathsf{Z}_{\mathrm{in}}\Big]$$
$$\leq \frac{C\phi'(0)^4 \log^2 V}{NVd^2L^2}\frac{1}{L \wedge d}\Big(1 + \frac{d}{L^2} + \frac{d^2}{VL}\Big). \tag{D.10}$$

Therefore, by using (D.9)-(D.10) and using $L \ll V$ and $N \ll VL$, we have

$$\mathbb{E}\Big[\Big\| \frac{1}{N}\sum_{i=1}^N \mathbf{C}_i \mathbf{B}_{i,3}\Big\|_F^2 | \mathsf{Z}_{\mathrm{in}}\Big] \ll \frac{1}{N^2 dL(d \wedge L^2)(d \wedge L)}. \tag{D.11}$$

Therefore, by (D.8)-(D.11), we have

$$\vartheta_2 \leq \frac{C \log V}{N\sqrt{L}d(d \wedge L^2)^{1/2}(d \wedge L)^{1/2}} + \frac{C \log^{5/2} V}{\sqrt{N}(Vd)^{3/4}L^{1/4}(d \wedge L^2)^{1/2}(L \wedge d)^{3/4}}.$$

**Bounding $\vartheta_1$:** We have

$$\vartheta_1 = \underbrace{\mathrm{tr}\Big(\frac{1}{N}\sum_{i=1}^N \mathbf{C}_i \mathbf{B}_{i,1}\Big) + \mathrm{tr}\Big(\frac{1}{N}\sum_{i=1}^N \mathbf{C}_i \mathbf{B}_{i,2}\Big)}_{:=\vartheta_{11}} + \underbrace{\mathrm{tr}\Big(\frac{1}{N}\sum_{i=1}^N \mathbf{C}_i \mathbf{B}_{i,3}\Big)}_{:=\vartheta_{12}}.$$

We have

$$\vartheta_{11} = \frac{1}{N^2L^2}\sum_{i,j=1}^N \alpha_{ij}(\mathbb{1}_{\boldsymbol{x}_i = \boldsymbol{x}_j} - \tfrac{1}{V}\mathbb{1}_V)\mathsf{z}_k^\top \mathbf{Z}_{\mathrm{in}} \mathbf{X}_i^\top \big(\mathbf{I}_L - \tfrac{1}{L}\mathbb{1}_L \mathbb{1}_L^\top\big)\mathbf{X}_i \mathbf{Z}_{\mathrm{in}}^\top \mathbf{Z}_{\mathrm{in}}(\mathbf{X}_j^\top - \tfrac{1}{V}\mathbb{1}_V \mathbb{1}_L^\top)\mathbb{1}_L$$

$$= \frac{\phi'(0)^2}{N^2L^2}\sum_{j=1}^N \mathsf{z}_k^\top \mathbf{Z}_{\mathrm{in}}\Big(\sum_{\substack{i=1\\i\neq j}}^N (\mathbb{1}_{\boldsymbol{x}_i = \boldsymbol{x}_j} - \tfrac{1}{V}\mathbb{1}_V)\mathbf{X}_i^\top \big(\mathbf{I}_L - \tfrac{1}{L}\mathbb{1}_L \mathbb{1}_L^\top\big)\mathbf{X}_i\Big)\mathbf{Z}_{\mathrm{in}}^\top \mathbf{Z}_{\mathrm{in}}(\mathbf{X}_j^\top - \tfrac{1}{V}\mathbb{1}_V \mathbb{1}_L^\top)\mathbb{1}_L$$

$$+ \frac{(1 - \tfrac{1}{V})}{N^2L^2}\sum_{j=1}^N \alpha_{jj}\mathsf{z}_k^\top \mathbf{Z}_{\mathrm{in}} \mathbf{X}_j^\top \big(\mathbf{I}_L - \tfrac{1}{L}\mathbb{1}_L \mathbb{1}_L^\top\big)\mathbf{X}_j$$
$$\times \mathbf{Z}_{\mathrm{in}}^\top \mathbf{Z}_{\mathrm{in}}(\mathbf{X}_j^\top - \tfrac{1}{V}\mathbb{1}_V \mathbb{1}_L^\top)\mathbb{1}_L$$

$$+ \frac{1}{N^2L^2}\sum_{j=1}^N\sum_{\substack{i=1\\i\neq j}}^N (\alpha_{ij} - \phi'(0)^2)(\mathbb{1}_{\boldsymbol{x}_i = \boldsymbol{x}_j} - \tfrac{1}{V})\mathsf{z}_k^\top$$
$$\times \mathbf{Z}_{\mathrm{in}} \mathbf{X}_i^\top \big(\mathbf{I}_L - \tfrac{1}{L}\mathbb{1}_L \mathbb{1}_L^\top\big)\mathbf{X}_i \mathbf{Z}_{\mathrm{in}}^\top \mathbf{Z}_{\mathrm{in}}(\mathbf{X}_j^\top - \tfrac{1}{V}\mathbb{1}_V \mathbb{1}_L^\top)\mathbb{1}_L$$

$$=: \vartheta_{11a} + \vartheta_{11b} + \vartheta_{11c}.$$

We start with the last term. By using Hölder's inequality,

$$|\vartheta_{11c}| \leq \Big(\frac{1}{N^2L^2}\sum_{j=1}^N\sum_{\substack{i=1\\i\neq j}}^N |\mathbb{1}_{\boldsymbol{x}_i = \boldsymbol{x}_j} - \tfrac{1}{V}|\Big) \sup_{i\neq j\in[N]}|\alpha_{ij} - \phi'(0)^2|$$
$$\times \sup_{i\neq j\in[N]}|\mathsf{z}_k^\top \mathbf{Z}_{\mathrm{in}} \mathbf{X}_i^\top \big(\mathbf{I}_L - \tfrac{1}{L}\mathbb{1}_L \mathbb{1}_L^\top\big)\mathbf{X}_i \mathbf{Z}_{\mathrm{in}}^\top \mathbf{Z}_{\mathrm{in}}(\mathbf{X}_j^\top - \tfrac{1}{V}\mathbb{1}_V \mathbb{1}_L^\top)\mathbb{1}_L|$$

$$\leq \frac{C \log V}{V L \sqrt{d} \sqrt{L \wedge d}} \frac{1}{V \wedge L^2 \wedge L\sqrt{d}}. \tag{D.12}$$

where we used (E1.8), (E2.4), (E2.8), and (R2) in (D.12). Next, we consider $\vartheta_{11b}$:

$$|\vartheta_{11b}| = \frac{(1 - \frac{1}{V})}{N^2 L^2} \sum_{j=1}^{N} \mathsf{z}_k^\top \boldsymbol{Z}_{\text{in}} \Big( \alpha_{jj} \boldsymbol{X}_j^\top \boldsymbol{X}_j \boldsymbol{Z}_{\text{in}}^\top \boldsymbol{Z}_{\text{in}} \boldsymbol{X}_j^\top \mathbb{1}_L - \mathbb{E} \left[ \alpha_{jj} \boldsymbol{X}_j^\top \boldsymbol{X}_j \boldsymbol{Z}_{\text{in}}^\top \boldsymbol{Z}_{\text{in}} \boldsymbol{X}_j^\top \mathbb{1}_L \, \big| \mathsf{Z}_{\text{in}} \right] \Big)$$

$$+ \frac{(1 - \frac{1}{V})}{N L^2} \mathsf{z}_k^\top \boldsymbol{Z}_{\text{in}} \mathbb{E} \left[ \alpha_{11} \boldsymbol{X}_1^\top \boldsymbol{X}_1 \boldsymbol{Z}_{\text{in}}^\top \boldsymbol{Z}_{\text{in}} \boldsymbol{X}_1^\top \mathbb{1}_L \, \big| \mathsf{Z}_{\text{in}} \right]$$

$$- \frac{(1 - \frac{1}{V})}{N^2 L V} \sum_{j=1}^{N} \alpha_{jj} \mathsf{z}_k^\top \boldsymbol{Z}_{\text{in}} \boldsymbol{X}_j^\top \boldsymbol{X}_j \boldsymbol{Z}_{\text{in}}^\top \boldsymbol{Z}_{\text{in}} \mathbb{1}_V$$

$$- \frac{(1 - \frac{1}{V})}{N^2 L^3} \sum_{j=1}^{N} \alpha_{jj} \mathsf{z}_k^\top \boldsymbol{Z}_{\text{in}} \boldsymbol{X}_j^\top \mathbb{1}_L \mathbb{1}_L^\top \boldsymbol{X}_j \boldsymbol{Z}_{\text{in}}^\top \boldsymbol{Z}_{\text{in}} (\boldsymbol{X}_j^\top - \tfrac{1}{V} \mathbb{1}_V \mathbb{1}_L^\top) \mathbb{1}_L.$$

- For the first summand,

$$\mathbb{E} \Bigg[ \Bigg\{ \frac{(1 - \frac{1}{V})}{N^2 L^2} \sum_{j=1}^{N} \mathsf{z}_k^\top \boldsymbol{Z}_{\text{in}}$$

$$\times \Big( \alpha_{jj} \boldsymbol{X}_j^\top \boldsymbol{X}_j \boldsymbol{Z}_{\text{in}}^\top \boldsymbol{Z}_{\text{in}} \boldsymbol{X}_j^\top \mathbb{1}_L - \mathbb{E} \left[ \alpha_{jj} \boldsymbol{X}_j^\top \boldsymbol{X}_j \boldsymbol{Z}_{\text{in}}^\top \boldsymbol{Z}_{\text{in}} \boldsymbol{X}_j^\top \mathbb{1}_L \big| \mathsf{Z}_{\text{in}} \right] \Big) \Bigg\}^2 \big| \mathsf{Z}_{\text{in}} \Bigg]$$

$$\leq \frac{(1 - \frac{1}{V})^2}{N^3 L^4} \mathbb{E} \left[ \alpha_{jj}^2 \Big( \mathsf{z}_k^\top \boldsymbol{Z}_{\text{in}} \boldsymbol{X}_1^\top \boldsymbol{X}_1 \boldsymbol{Z}_{\text{in}}^\top \boldsymbol{Z}_{\text{in}} \boldsymbol{X}_1^\top \mathbb{1}_L \Big)^2 \big| \mathsf{Z}_{\text{in}} \right]$$

$$\leq \frac{C \phi'(0)^4}{N^3 L^3} \mathbb{E} \left[ \left\| \mathsf{z}_k^\top \boldsymbol{Z}_{\text{in}} \boldsymbol{X}_1^\top \boldsymbol{X}_1 \boldsymbol{Z}_{\text{in}}^\top \right\|_2^2 \big| \mathsf{Z}_{\text{in}} \right] \tag{D.13}$$

where we used (R1) in (D.13).
By Chebyshev's inequality and (P2), with probability $1 - o_V(1)$, we have

$$\mathsf{z}_k^\top \boldsymbol{Z}_{\text{in}} \frac{(1 - \frac{1}{V})}{N^2 L^2} \sum_{j=1}^{N} \Big( \alpha_{jj} \boldsymbol{X}_j^\top \boldsymbol{X}_j \boldsymbol{Z}_{\text{in}}^\top \boldsymbol{Z}_{\text{in}} \boldsymbol{X}_j^\top \mathbb{1}_L - \mathbb{E} \left[ \alpha_{jj} \boldsymbol{X}_j^\top \boldsymbol{X}_j \boldsymbol{Z}_{\text{in}}^\top \boldsymbol{Z}_{\text{in}} \boldsymbol{X}_j^\top \mathbb{1}_L \big| \mathsf{Z}_{\text{in}} \right] \Big)$$

$$\leq \frac{C \phi'(0)^2 \log V}{N^{\frac{3}{2}} \sqrt{Ld} \sqrt{L \wedge d}}.$$

- For the second summand,

$$\frac{(1 - \frac{1}{V})}{N L^2} \mathsf{z}_k^\top \boldsymbol{Z}_{\text{in}} \mathbb{E} \left[ \alpha_{11} \boldsymbol{X}_1^\top \boldsymbol{X}_1 \boldsymbol{Z}_{\text{in}}^\top \boldsymbol{Z}_{\text{in}} \boldsymbol{X}_1^\top \mathbb{1}_L \, \big| \mathsf{Z}_{\text{in}} \right]$$

$$= \frac{(1 - \frac{1}{V}) \phi'(0)^2}{N L^2} \mathsf{z}_k^\top \boldsymbol{Z}_{\text{in}} \mathbb{E} \left[ \boldsymbol{X}_1^\top \boldsymbol{X}_1 \boldsymbol{Z}_{\text{in}}^\top \boldsymbol{Z}_{\text{in}} \boldsymbol{X}_1^\top \boldsymbol{X}_1 \, \big| \mathsf{Z}_{\text{in}} \right] \mathbb{1}_V$$

$$+ \frac{(1 - \frac{1}{V})}{N L^2} \mathsf{z}_k^\top \boldsymbol{Z}_{\text{in}} \mathbb{E} \left[ (\alpha_{11} - \phi'(0)^2) \boldsymbol{X}_1^\top \boldsymbol{X}_1 \boldsymbol{Z}_{\text{in}}^\top \boldsymbol{Z}_{\text{in}} \boldsymbol{X}_1^\top \boldsymbol{X}_1 \, \big| \mathsf{Z}_{\text{in}} \right] \mathbb{1}_V$$

$$= \frac{(1 - \frac{1}{V}) \phi'(0)^2}{N L} \mathsf{z}_k^\top \boldsymbol{Z}_{\text{in}} \mathbb{E} \left[ \boldsymbol{x}_1 \boldsymbol{x}_1^\top \boldsymbol{Z}_{\text{in}}^\top \boldsymbol{Z}_{\text{in}} \boldsymbol{x}_1 \boldsymbol{x}_1^\top \, \big| \mathsf{Z}_{\text{in}} \right] \mathbb{1}_V + \frac{(1 - \frac{1}{V}) \phi'(0)^2}{N V^2} \mathsf{z}_k^\top \boldsymbol{Z}_{\text{in}} \boldsymbol{Z}_{\text{in}}^\top \boldsymbol{Z}_{\text{in}} \mathbb{1}_V \tag{D.14}$$

$$+ \frac{(1 - \frac{1}{V})}{N L V} \mathsf{z}_k^\top \boldsymbol{Z}_{\text{in}} \mathbb{E} \left[ (\alpha_{11} - \phi'(0)^2) \boldsymbol{Z}_{\text{in}}^\top \boldsymbol{Z}_{\text{in}} \boldsymbol{X}_1^\top \mathbb{1}_L \big| \mathsf{Z}_{\text{in}} \right] \tag{D.15}$$

$$+ \frac{(1 - \frac{1}{V})}{N L^2} \mathsf{z}_k^\top \boldsymbol{Z}_{\text{in}} \mathbb{E} \left[ (\alpha_{11} - \phi'(0)^2) (\boldsymbol{X}_1^\top \boldsymbol{X}_1 - \tfrac{L}{V} \boldsymbol{I}_V) \boldsymbol{Z}_{\text{in}}^\top \boldsymbol{Z}_{\text{in}} (\boldsymbol{X}_1^\top - \tfrac{1}{V} \mathbb{1}_V \mathbb{1}_L^\top) \mathbb{1}_L \big| \mathsf{Z}_{\text{in}} \right] \tag{D.16}$$

$$+ \frac{(1 - \frac{1}{V})}{N L V} \mathsf{z}_k^\top \boldsymbol{Z}_{\text{in}} \mathbb{E} \left[ (\alpha_{11} - \phi'(0)^2) (\boldsymbol{X}_1^\top \boldsymbol{X}_1 - \tfrac{L}{V} \boldsymbol{I}_V) \boldsymbol{Z}_{\text{in}}^\top \boldsymbol{Z}_{\text{in}} \mathbb{1}_V \big| \mathsf{Z}_{\text{in}} \right] \tag{D.17}$$

$$\leq C \log V \Big( \frac{1}{N\sqrt{V}d(L \wedge d)} + \frac{1}{NL^{3/2}\sqrt{d}(L \wedge d)} \Big).$$

where we use (E1.3) to bound (D.14); (E1.6), (R2) for (D.15); (P5), (R2) for (D.16); and (E1.8), (R2) for (D.17).

- For the third summand,

$$\frac{1}{N^2LV} \sum_{j=1}^{N} \alpha_{jj} \mathsf{z}_k^\top \mathbf{Z}_{\mathrm{in}} \mathbf{X}_j^\top \mathbf{X}_j \mathbf{Z}_{\mathrm{in}}^\top \mathbf{Z}_{\mathrm{in}} \mathbb{1}_V$$

$$= \frac{\phi'(0)^2}{NV^2} \mathsf{z}_k^\top \mathbf{Z}_{\mathrm{in}} \mathbf{Z}_{\mathrm{in}} \mathbf{Z}_{\mathrm{in}}^\top \mathbb{1}_V + \frac{\phi'(0)^2}{N^2LV} \sum_{j=1}^{N} \mathsf{z}_k^\top \mathbf{Z}_{\mathrm{in}} \Big( \mathbf{X}_j^\top \mathbf{X}_j - \tfrac{L}{V} \mathbf{I}_V \Big) \mathbf{Z}_{\mathrm{in}}^\top \mathbf{Z}_{\mathrm{in}} \mathbb{1}_V$$

$$+ \frac{1}{N^2LV} \sum_{j=1}^{N} (\alpha_{jj} - \phi'(0)^2) \mathsf{z}_k^\top \mathbf{Z}_{\mathrm{in}} \mathbf{X}_j^\top \mathbf{X}_j \mathbf{Z}_{\mathrm{in}}^\top \mathbf{Z}_{\mathrm{in}} \mathbb{1}_V.$$

The first term:

$$\Big| \frac{\phi'(0)^2}{NV^2} \mathsf{z}_k^\top \mathbf{Z}_{\mathrm{in}} \mathbf{Z}_{\mathrm{in}} \mathbf{Z}_{\mathrm{in}}^\top \mathbb{1}_V \Big| \leq \frac{C \log V}{N\sqrt{V}d^{\frac{3}{2}}}.$$

The second term: By using

$$\mathbb{E} \Big[ \Big( \sum_{j=1}^{N} \mathsf{z}_k^\top \mathbf{Z}_{\mathrm{in}} \Big( \mathbf{X}_j^\top \mathbf{X}_j - \tfrac{L}{V} \mathbf{I}_V \Big) \mathbf{Z}_{\mathrm{in}}^\top \mathbf{Z}_{\mathrm{in}} \mathbb{1}_V \Big)^2 | \mathbf{Z}_{\mathrm{in}} \Big]$$

$$= \sum_{j=1}^{N} \mathbb{E} \Big[ \Big( \mathsf{z}_k^\top \mathbf{Z}_{\mathrm{in}} \Big( \mathbf{X}_j^\top \mathbf{X}_j - \tfrac{L}{V} \mathbf{I}_V \Big) \mathbf{Z}_{\mathrm{in}}^\top \mathbf{Z}_{\mathrm{in}} \mathbb{1}_V \Big)^2 | \mathbf{Z}_{\mathrm{in}} \Big] = \frac{LVN}{d^2}.$$

Therefore, by Chebyshev's inequality, we have

$$\Big| \frac{\phi'(0)^2}{N^2LV} \sum_{j=1}^{N} \mathsf{z}_k^\top \mathbf{Z}_{\mathrm{in}} \Big( \mathbf{X}_j^\top \mathbf{X}_j - \tfrac{L}{V} \mathbf{I}_V \Big) \mathbf{Z}_{\mathrm{in}}^\top \mathbf{Z}_{\mathrm{in}} \mathbb{1}_V \Big| \leq \frac{\phi'(0)^2}{N^{3/2}\sqrt{V}Ld}.$$

Finally,

$$\Big| \frac{1}{N^2LV} \sum_{j=1}^{N} (\alpha_{jj} - \phi'(0)^2) \mathsf{z}_k^\top \mathbf{Z}_{\mathrm{in}} \mathbf{X}_j^\top \mathbf{X}_j \mathbf{Z}_{\mathrm{in}}^\top \mathbf{Z}_{\mathrm{in}} \mathbb{1}_V \Big|$$

$$\leq \frac{C}{N\sqrt{V}} \Big\| \frac{1}{NL} \sum_{j=1}^{N} (\alpha_{jj} - \phi'(0)^2) \mathbf{Z}_{\mathrm{in}} \mathbf{X}_j^\top \mathbf{X}_j \mathbf{Z}_{\mathrm{in}}^\top \Big\|_2$$

$$\leq \frac{C}{N\sqrt{V}Ld},$$

where we use (E2.8) and (E1.1) and (R2).

Therefore,

$$\Big| \frac{1}{N^2LV} \sum_{j=1}^{N} \alpha_{jj} \mathsf{z}_k^\top \mathbf{Z}_{\mathrm{in}} \mathbf{X}_j^\top \mathbf{X}_j \mathbf{Z}_{\mathrm{in}}^\top \mathbf{Z}_{\mathrm{in}} \mathbb{1}_V \Big| \leq C \log V \Big( \frac{1}{N\sqrt{V}d^{\frac{3}{2}}} + \frac{1}{N\sqrt{V}Ld} \Big).$$

- For the last summand,

$$\frac{(1 - \frac{1}{V})}{N^2L^3} \sum_{j=1}^{N} \alpha_{jj} \mathsf{z}_k^\top \mathbf{Z}_{\mathrm{in}} \mathbf{X}_j^\top \mathbb{1}_L \mathbb{1}_L^\top \mathbf{X}_j \mathbf{Z}_{\mathrm{in}}^\top \mathbf{Z}_{\mathrm{in}} (\mathbf{X}_j^\top - \tfrac{1}{V} \mathbb{1}_V \mathbb{1}_L^\top) \mathbb{1}_L$$

$$= \frac{(1 - \frac{1}{V})}{N^2L^3} \sum_{j=1}^{N} \mathsf{z}_k^\top \mathbf{Z}_{\mathrm{in}}$$

$$\times \Big( \alpha_{jj} \boldsymbol{X}_j^\top \mathbb{1}_L \mathbb{1}_L^\top \boldsymbol{X}_j \boldsymbol{Z}_{\mathrm{in}}^\top \boldsymbol{Z}_{\mathrm{in}} (\boldsymbol{X}_j^\top - \tfrac{1}{V} \mathbb{1}_V \mathbb{1}_L^\top) \mathbb{1}_L$$

$$- \mathbb{E}\big[ \alpha_{jj} \boldsymbol{X}_j^\top \mathbb{1}_L \mathbb{1}_L^\top \boldsymbol{X}_j \boldsymbol{Z}_{\mathrm{in}}^\top \boldsymbol{Z}_{\mathrm{in}} (\boldsymbol{X}_j^\top - \tfrac{1}{V} \mathbb{1}_V \mathbb{1}_L^\top) \mathbb{1}_L \,\big|\, \mathsf{Z}_{\mathrm{in}} \big] \Big)$$

$$+ \frac{(1-\tfrac{1}{V})\phi'(0)^2}{NL^3} \mathsf{z}_k^\top \boldsymbol{Z}_{\mathrm{in}} \mathbb{E}\big[ \boldsymbol{X}_1^\top \mathbb{1}_L \mathbb{1}_L^\top \boldsymbol{X}_1 \boldsymbol{Z}_{\mathrm{in}}^\top \boldsymbol{Z}_{\mathrm{in}} (\boldsymbol{X}_1^\top - \tfrac{1}{V} \mathbb{1}_V \mathbb{1}_L^\top) \mathbb{1}_L \,\big|\, \mathsf{Z}_{\mathrm{in}} \big]$$

$$+ \frac{(1-\tfrac{1}{V})}{NL^3} \mathsf{z}_k^\top \boldsymbol{Z}_{\mathrm{in}} \mathbb{E}\big[ (\alpha_{11} - \phi'(0)^2) \boldsymbol{X}_1^\top \mathbb{1}_L \mathbb{1}_L^\top \boldsymbol{X}_1 \boldsymbol{Z}_{\mathrm{in}}^\top \boldsymbol{Z}_{\mathrm{in}} (\boldsymbol{X}_1^\top - \tfrac{1}{V} \mathbb{1}_V \mathbb{1}_L^\top) \mathbb{1}_L \,\big|\, \mathsf{Z}_{\mathrm{in}} \big]. \text{(D.18)}$$

We have

$$\mathsf{z}_k^\top \boldsymbol{Z}_{\mathrm{in}} \mathbb{E}\Big[ \Big( \alpha_{jj} \boldsymbol{X}_1^\top \mathbb{1}_L \mathbb{1}_L^\top \boldsymbol{X}_1 \boldsymbol{Z}_{\mathrm{in}}^\top \boldsymbol{Z}_{\mathrm{in}} (\boldsymbol{X}_1^\top - \tfrac{1}{V} \mathbb{1}_V \mathbb{1}_L^\top) \mathbb{1}_L \Big)^2 \,\big|\, \mathsf{Z}_{\mathrm{in}} \Big] \boldsymbol{Z}_{\mathrm{in}}^\top \mathsf{z}_k$$

$$\le CL^2 \mathsf{z}_k^\top \boldsymbol{Z}_{\mathrm{in}} \mathbb{E}\Big[ \boldsymbol{X}_1^\top \mathbb{1}_L \mathbb{1}_L^\top \boldsymbol{X}_1 \Big] \boldsymbol{Z}_{\mathrm{in}}^\top \mathsf{z}_k \le CL^2 \Big( \frac{L}{d} + \frac{L^2}{Vd} \Big).$$

Moreover, by using Proposition 8

$$\mathbb{E}\big[ \boldsymbol{X}_1^\top \mathbb{1}_L \mathbb{1}_L^\top \boldsymbol{X}_1 \boldsymbol{Z}_{\mathrm{in}}^\top \boldsymbol{Z}_{\mathrm{in}} (\boldsymbol{X}_1^\top - \tfrac{1}{V} \mathbb{1}_V \mathbb{1}_L^\top) \mathbb{1}_L \,\big|\, \mathsf{Z}_{\mathrm{in}} \big]$$

$$= \mathbb{E}\big[ \boldsymbol{X}_1^\top \mathbb{1}_L \mathbb{1}_L^\top \boldsymbol{X}_1 \boldsymbol{Z}_{\mathrm{in}}^\top \boldsymbol{Z}_{\mathrm{in}} \boldsymbol{X}_1^\top \mathbb{1}_L \,\big|\, \mathsf{Z}_{\mathrm{in}} \big] - \frac{L}{V} \mathbb{E}\big[ \boldsymbol{X}_1^\top \mathbb{1}_L \mathbb{1}_L^\top \boldsymbol{X}_1 \boldsymbol{Z}_{\mathrm{in}}^\top \boldsymbol{Z}_{\mathrm{in}} \mathbb{1}_V \,\big|\, \mathsf{Z}_{\mathrm{in}} \big]$$

$$= L \mathbb{E}\big[ \boldsymbol{x}_1 \boldsymbol{x}_1^\top \boldsymbol{Z}_{\mathrm{in}}^\top \boldsymbol{Z}_{\mathrm{in}} \boldsymbol{x}_1 \,\big|\, \mathsf{Z}_{\mathrm{in}} \big] + \Big( \frac{L(L-1)}{V^2} \mathrm{tr}(\boldsymbol{Z}_{\mathrm{in}}^\top \boldsymbol{Z}_{\mathrm{in}}) - \frac{2L(L-1)}{V^3} \mathbb{1}_V^\top \boldsymbol{Z}_{\mathrm{in}}^\top \boldsymbol{Z}_{\mathrm{in}} \mathbb{1}_V \Big) \mathbb{1}_V$$

$$+ \frac{L(L-2)}{V^2} \boldsymbol{Z}_{\mathrm{in}}^\top \boldsymbol{Z}_{\mathrm{in}} \mathbb{1}_V.$$

Lastly,

$$\Big| \frac{1}{NL^3} \mathsf{z}_k^\top \boldsymbol{Z}_{\mathrm{in}} \mathbb{E}\big[ (\alpha_{11} - \phi'(0)^2) \boldsymbol{X}_1^\top \mathbb{1}_L \mathbb{1}_L^\top \boldsymbol{X}_1 \boldsymbol{Z}_{\mathrm{in}}^\top \boldsymbol{Z}_{\mathrm{in}} (\boldsymbol{X}_1^\top - \tfrac{1}{V} \mathbb{1}_V \mathbb{1}_L^\top) \mathbb{1}_L \,\big|\, \mathsf{Z}_{\mathrm{in}} \big] \Big|$$

$$\le \frac{1}{NL^3} \mathbb{E}\Big[ |\alpha_{11} - \phi'(0)^2| |\mathsf{z}_k^\top \boldsymbol{Z}_{\mathrm{in}} \boldsymbol{X}_1^\top \mathbb{1}_L| \, |\mathbb{1}_L^\top \boldsymbol{X}_1 \boldsymbol{Z}_{\mathrm{in}}^\top \boldsymbol{Z}_{\mathrm{in}} (\boldsymbol{X}_1^\top - \tfrac{1}{V} \mathbb{1}_V \mathbb{1}_L^\top) \mathbb{1}_L| \,\Big|\, \mathsf{Z}_{\mathrm{in}} \Big]$$

$$\le \frac{C \log V}{NL^{5/2}\sqrt{d}},$$

where we used (R1),(R2), (E1.4), (E1.5) for the last inequality.

Therefore, by Chebyshev's inequality, with probability $1 - o_V(1)$, we have

$$\text{(D.18)} \le C \log V \Big( \frac{1}{NL^2\sqrt{L \wedge d}} + \frac{1}{NL\sqrt{Vd}} \Big)$$

Therefore, we have

$$|\vartheta_{11b}| \le C \log V \Big( \frac{1}{N\sqrt{V}(L \wedge d)\sqrt{d}} + \frac{1}{NL^2\sqrt{L \wedge d}} \Big). \tag{D.19}$$

Finally, we consider $\vartheta_{11a}$:

$$\vartheta_{11a} = (1 - \tfrac{1}{L}) \frac{\phi'(0)^2}{NL} \sum_{j=1}^N \mathsf{z}_k^\top \boldsymbol{Z}_{\mathrm{in}} \Big( \frac{1}{NL} \sum_{\substack{i=1 \\ i \ne j}}^N (\mathbb{1}_{\boldsymbol{x}_i = \boldsymbol{x}_j} - \tfrac{1}{V}) \boldsymbol{x}_i \boldsymbol{x}_i^\top \Big) \boldsymbol{Z}_{\mathrm{in}}^\top \boldsymbol{Z}_{\mathrm{in}} (\boldsymbol{X}_j^\top - \tfrac{1}{V} \mathbb{1}_V \mathbb{1}_L^\top) \mathbb{1}_L$$

$$- \frac{\phi'(0)^2}{NL^2} \sum_{j=1}^N \mathsf{z}_k^\top \boldsymbol{Z}_{\mathrm{in}} \Big( \frac{1}{NL} \sum_{\substack{i=1 \\ i \ne j}}^N (\mathbb{1}_{\boldsymbol{x}_i = \boldsymbol{x}_j} - \tfrac{1}{V})(\boldsymbol{x}_i \mathbb{1}_L^\top \boldsymbol{X}_i + \boldsymbol{X}_i^\top \mathbb{1}_L \boldsymbol{x}_i^\top) \Big) \boldsymbol{Z}_{\mathrm{in}}^\top \boldsymbol{Z}_{\mathrm{in}} (\boldsymbol{X}_j^\top - \tfrac{1}{V} \mathbb{1}_V \mathbb{1}_L^\top) \mathbb{1}_L$$

$$+ \frac{\phi'(0)^2}{NL} \sum_{j=1}^N \mathsf{z}_k^\top \boldsymbol{Z}_{\mathrm{in}} \Big( \frac{1}{NL} \sum_{\substack{i=1 \\ i \ne j}}^N (\mathbb{1}_{\boldsymbol{x}_i = \boldsymbol{x}_j} - \tfrac{1}{V}) \boldsymbol{N}_i^\top \boldsymbol{N}_i \Big) \boldsymbol{Z}_{\mathrm{in}}^\top \boldsymbol{Z}_{\mathrm{in}} (\boldsymbol{x}_j - \tfrac{1}{V} \mathbb{1}_V)$$

$$+ \frac{\phi'(0)^2}{NL} \sum_{j=1}^N \mathsf{z}_k^\top \boldsymbol{Z}_{\mathrm{in}} \Big( \frac{1}{NL} \sum_{\substack{i=1 \\ i \ne j}}^N (\mathbb{1}_{\boldsymbol{x}_i = \boldsymbol{x}_j} - \tfrac{1}{V}) \boldsymbol{N}_i^\top \boldsymbol{N}_i \Big) \boldsymbol{Z}_{\mathrm{in}}^\top \boldsymbol{Z}_{\mathrm{in}} (\boldsymbol{N}_j^\top - \tfrac{1}{V} \mathbb{1}_V \mathbb{1}_{L-1}^\top) \mathbb{1}_{L-1}$$

$$- \frac{\phi'(0)^2}{NL^2} \sum_{j=1}^{N} \mathsf{z}_k^\top \boldsymbol{Z}_{\mathrm{in}} \Big( \frac{1}{NL} \sum_{\substack{i=1 \\ i \neq j}}^{N} (\mathbb{1}_{\boldsymbol{x}_i = \boldsymbol{x}_j} - \tfrac{1}{V}) \boldsymbol{N}_i^\top \mathbb{1}_L \mathbb{1}_L^\top \boldsymbol{N}_i \Big) \boldsymbol{Z}_{\mathrm{in}}^\top \boldsymbol{Z}_{\mathrm{in}} (\boldsymbol{N}_j^\top - \tfrac{1}{V} \mathbb{1}_V \mathbb{1}_{L-1}^\top) \mathbb{1}_{L-1}$$

$$=: \vartheta_{aa} + \vartheta_{ab} + \vartheta_{ac} + \vartheta_{ad} + \vartheta_{ae}.$$

For the first summand, we write

$$\vartheta_{aa} := \underbrace{(1 - \frac{1}{L}) \frac{\phi'(0)^2}{NL} \sum_{j=1}^{N} \mathsf{z}_k^\top \boldsymbol{Z}_{\mathrm{in}} \Big( \frac{1}{NL} \sum_{\substack{i=1 \\ i \neq j}}^{N} (\mathbb{1}_{\boldsymbol{x}_i = \boldsymbol{x}_j} - \tfrac{1}{V}) \boldsymbol{x}_i \boldsymbol{x}_i^\top \Big) \boldsymbol{Z}_{\mathrm{in}}^\top \boldsymbol{Z}_{\mathrm{in}} (\boldsymbol{x}_j - \tfrac{1}{V} \mathbb{1}_V)}_{:= \vartheta_{aa1}}$$

$$+ \underbrace{(1 - \frac{1}{L}) \frac{\phi'(0)^2}{NL} \sum_{j=1}^{N} \mathsf{z}_k^\top \boldsymbol{Z}_{\mathrm{in}} \Big( \frac{1}{NL} \sum_{\substack{i=1 \\ i \neq j}}^{N} (\mathbb{1}_{\boldsymbol{x}_i = \boldsymbol{x}_j} - \frac{1}{V}) \boldsymbol{x}_i \boldsymbol{x}_i^\top \Big) \boldsymbol{Z}_{\mathrm{in}}^\top \boldsymbol{Z}_{\mathrm{in}} (\boldsymbol{N}_j^\top - \frac{1}{V} \mathbb{1}_V \mathbb{1}_{L-1}^\top) \mathbb{1}_{L-1}}_{:= \vartheta_{aa2}} .$$

We have

$$|\vartheta_{aa1}| \leq \Big( \frac{\phi'(0)^2}{N^2 L^2} \sum_{j=1}^{N} \sum_{\substack{i=1 \\ i \neq j}}^{N} |\mathbb{1}_{\boldsymbol{x}_i = \boldsymbol{x}_j} - \tfrac{1}{V}| \Big) \sup_{i \neq j} \Big| \mathbb{1}_{\boldsymbol{x}_i \neq \boldsymbol{e}_k} \mathsf{z}_k^\top \boldsymbol{Z}_{\mathrm{in}} \boldsymbol{x}_i \boldsymbol{x}_i^\top \boldsymbol{Z}_{\mathrm{in}}^\top \boldsymbol{Z}_{\mathrm{in}} (\boldsymbol{x}_j - \tfrac{1}{V} \mathbb{1}_V) \Big|$$

$$+ \Big( \frac{\phi'(0)^2}{N^2 L^2} \sum_{j=1}^{N} \sum_{\substack{i=1 \\ i \neq j}}^{N} |\mathbb{1}_{\boldsymbol{x}_j = \boldsymbol{e}_k} - \tfrac{1}{V}| (\mathbb{1}_{\boldsymbol{x}_i = \boldsymbol{e}_k} - \tfrac{1}{V}) \Big) \sup_{j} |\mathsf{z}_k^\top \boldsymbol{Z}_{\mathrm{in}} \boldsymbol{e}_k \boldsymbol{e}_k^\top \boldsymbol{Z}_{\mathrm{in}}^\top \boldsymbol{Z}_{\mathrm{in}} (\boldsymbol{x}_j - \tfrac{1}{V} \mathbb{1}_V) \Big|$$

$$+ \Big( \frac{\phi'(0)^2}{N^2 V L^2} \sum_{j=1}^{N} \sum_{\substack{i=1 \\ i \neq j}}^{N} |\mathbb{1}_{\boldsymbol{x}_j = \boldsymbol{e}_k} - \tfrac{1}{V}| \Big) \sup_{j} |\mathsf{z}_k^\top \boldsymbol{Z}_{\mathrm{in}} \boldsymbol{e}_k \boldsymbol{e}_k^\top \boldsymbol{Z}_{\mathrm{in}}^\top \boldsymbol{Z}_{\mathrm{in}} (\boldsymbol{x}_j - \tfrac{1}{V} \mathbb{1}_V) \Big|$$

$$\leq \frac{C \log V}{V L^2 \sqrt{d}}.$$

where we use (E1.1) and (E2.4).

Moreover, let

$$\vartheta_{aa2} =: (1 - \frac{1}{L}) \frac{\phi'(0)^2}{NL} \sum_{j=1}^{N} \vartheta_{aa2,j}.$$

We have $\mathbb{E}[\vartheta_{aa2,j} | \mathsf{Z}_{\mathrm{in}}] = 0$ and $\mathbb{E}[\vartheta_{aa2,j} \vartheta_{aa2,j'} | \mathsf{Z}_{\mathrm{in}}] = 0$ for $j \neq j'$, and

$$\mathbb{E}[\vartheta_{aa2,j}^2 | \mathsf{Z}_{\mathrm{in}}]$$

$$\leq \frac{CL}{d} \mathbb{E} \Big[ \mathsf{z}_k^\top \boldsymbol{Z}_{\mathrm{in}} \Big( \frac{1}{NL} \sum_{\substack{i=1 \\ i \neq j}}^{N} \mathbb{1}_{\boldsymbol{x}_i = \boldsymbol{x}_j} \boldsymbol{x}_i \boldsymbol{x}_i^\top \Big) \boldsymbol{Z}_{\mathrm{in}}^\top \boldsymbol{Z}_{\mathrm{in}} \Big( \frac{1}{NL} \sum_{\substack{i=1 \\ i \neq j}}^{N} \mathbb{1}_{\boldsymbol{x}_i = \boldsymbol{x}_j} \boldsymbol{x}_i \boldsymbol{x}_i^\top \Big) \boldsymbol{Z}_{\mathrm{in}}^\top \mathsf{z}_k | \mathsf{Z}_{\mathrm{in}} \Big]$$

$$+ \frac{CL}{dV^2} \mathbb{E} \Big[ \mathsf{z}_k^\top \boldsymbol{Z}_{\mathrm{in}} \Big( \frac{1}{NL} \sum_{\substack{i=1 \\ i \neq j}}^{N} \boldsymbol{x}_i \boldsymbol{x}_i^\top \Big) \boldsymbol{Z}_{\mathrm{in}}^\top \boldsymbol{Z}_{\mathrm{in}} \Big( \frac{1}{NL} \sum_{\substack{i=1 \\ i \neq j}}^{N} \boldsymbol{x}_i \boldsymbol{x}_i^\top \Big) \boldsymbol{Z}_{\mathrm{in}}^\top \mathsf{z}_k | \mathsf{Z}_{\mathrm{in}} \Big] \leq \frac{C}{V^2 L d^2},$$

where we use (E2.8) and (E1.1).

Therefore, by Chebyshev's inequality with probability $1 - o_V(1)$, we have

$$|\vartheta_{aa2}| \leq \frac{C \log V}{\sqrt{N} V L^{3/2} d}.$$

Therefore,

$$|\vartheta_{aa}| \leq \frac{C \log V}{V L^2 \sqrt{d}} + \frac{C \log V}{\sqrt{N} V L^{3/2} d}.$$

Moreover, for the second term, we write

$$|\vartheta_{ab}| \le \left(\frac{\phi'(0)^2}{N^2 L^3} \sum_{j=1}^{N} \sum_{\substack{i=1 \\ i \ne j}}^{N} |\mathbb{1}_{\boldsymbol{x}_i = \boldsymbol{x}_j} - \tfrac{1}{V}|\right)$$

$$\times \sup_{i \ne j} |\mathbb{1}_{\boldsymbol{x}_i \ne \boldsymbol{e}_k} \mathsf{z}_k^\top \boldsymbol{Z}_{\mathrm{in}} (\boldsymbol{x}_i \mathbb{1}_L^\top \boldsymbol{X}_i + \boldsymbol{X}_i^\top \mathbb{1}_L \boldsymbol{x}_i^\top) \boldsymbol{Z}_{\mathrm{in}}^\top \boldsymbol{Z}_{\mathrm{in}} (\boldsymbol{X}_j^\top - \tfrac{1}{V} \mathbb{1}_V \mathbb{1}_L^\top) \mathbb{1}_L|$$

$$+ \left(\frac{\phi'(0)^2}{N^2 L^3} \sum_{j=1}^{N} \sum_{\substack{i=1 \\ i \ne j}}^{N} |\mathbb{1}_{\boldsymbol{x}_j = \boldsymbol{e}_k} - \tfrac{1}{V}|(\mathbb{1}_{\boldsymbol{x}_i = \boldsymbol{e}_k} - \tfrac{1}{V})\right)$$

$$\times \sup_{j} |\mathsf{z}_k^\top \boldsymbol{Z}_{\mathrm{in}} (\boldsymbol{e}_k \mathbb{1}_L^\top \boldsymbol{X}_i + \boldsymbol{X}_i^\top \mathbb{1}_L \boldsymbol{e}_k^\top) \boldsymbol{Z}_{\mathrm{in}}^\top \boldsymbol{Z}_{\mathrm{in}} (\boldsymbol{X}_j^\top - \tfrac{1}{V} \mathbb{1}_V \mathbb{1}_L^\top) \mathbb{1}_L|$$

$$+ \left(\frac{\phi'(0)^2}{N^2 V L^3} \sum_{j=1}^{N} \sum_{\substack{i=1 \\ i \ne j}}^{N} |\mathbb{1}_{\boldsymbol{x}_j = \boldsymbol{e}_k} - \tfrac{1}{V}|\right)$$

$$\times \sup_{j} |\mathsf{z}_k^\top \boldsymbol{Z}_{\mathrm{in}} (\boldsymbol{e}_k \mathbb{1}_L^\top \boldsymbol{X}_i + \boldsymbol{X}_i^\top \mathbb{1}_L \boldsymbol{e}_k^\top) \boldsymbol{Z}_{\mathrm{in}}^\top \boldsymbol{Z}_{\mathrm{in}} (\boldsymbol{X}_j^\top - \tfrac{1}{V} \mathbb{1}_V \mathbb{1}_L^\top) \mathbb{1}_L|$$

$$\le \frac{C \log V}{V L^2 d}.$$

where we used (E1.1), (E1.3), (E2.4) and (R1).

For the third term, we write

$$\vartheta_{ac} = \frac{\phi'(0)^2}{N L^2} \sum_{i=1}^{N} \mathsf{z}_k^\top \boldsymbol{Z}_{\mathrm{in}} \left(\boldsymbol{N}_i^\top \boldsymbol{N}_i - \frac{L-1}{V} \boldsymbol{I}_V\right) \boldsymbol{Z}_{\mathrm{in}}^\top \boldsymbol{Z}_{\mathrm{in}} \left(\frac{1}{N} \sum_{\substack{j=1 \\ j \ne i}}^{N} (\mathbb{1}_{\boldsymbol{x}_i = \boldsymbol{x}_j} - \tfrac{1}{V})(\boldsymbol{x}_j - \tfrac{1}{V} \mathbb{1}_V)\right)$$

$$+ \frac{\phi'(0)^2 (L-1)}{N L^2 V} \mathsf{z}_k^\top \boldsymbol{Z}_{\mathrm{in}} \boldsymbol{Z}_{\mathrm{in}}^\top \boldsymbol{Z}_{\mathrm{in}} \left(\frac{1}{N} \sum_{j=1}^{N} (\boldsymbol{x}_j - \tfrac{1}{V} \mathbb{1}_V)(\boldsymbol{x}_j - \tfrac{1}{V} \mathbb{1}_V)^\top\right) \sum_{i=1}^{N} (\boldsymbol{x}_i - \tfrac{1}{V} \mathbb{1}_V)$$

$$- \frac{\phi'(0)^2 (1 - \tfrac{1}{V})(L-1)}{N^2 L^2 V} \mathsf{z}_k^\top \boldsymbol{Z}_{\mathrm{in}} \boldsymbol{Z}_{\mathrm{in}}^\top \boldsymbol{Z}_{\mathrm{in}} \sum_{i=1}^{N} (\boldsymbol{x}_i - \tfrac{1}{V} \mathbb{1}_V)$$

$$=: \vartheta_{ac1} + \vartheta_{ac2} + \vartheta_{ac3}.$$

By using (E1.1), (E2.3), (E2.5),

$$\mathbb{E}[\vartheta_{ac2}^2 | \mathsf{Z}_{\mathrm{in}}] \le \frac{C}{N^2 L^2 V^3 d} \mathsf{z}_k^\top \boldsymbol{Z}_{\mathrm{in}} \boldsymbol{Z}_{\mathrm{in}}^\top \boldsymbol{Z}_{\mathrm{in}} \boldsymbol{Z}_{\mathrm{in}}^\top \mathsf{z}_k \le \frac{C}{N^2 L^2 V d^3}.$$

Moreover,

$$\mathbb{E}[\vartheta_{ac3}^2 | \mathsf{Z}_{\mathrm{in}}] \le \frac{C}{N^3 L^2 V^2 d} \mathsf{z}_k^\top \boldsymbol{Z}_{\mathrm{in}} \boldsymbol{Z}_{\mathrm{in}}^\top \boldsymbol{Z}_{\mathrm{in}} \boldsymbol{Z}_{\mathrm{in}}^\top \mathsf{z}_k \le \frac{C}{N^3 L^2 d^3}$$

Therefore,

$$|\vartheta_{ac2}| \le \frac{C \log V}{N \sqrt{V} L d^{\frac{3}{2}}}, \qquad |\vartheta_{ac3}| \le \frac{C \log V}{N \sqrt{N} L d^{\frac{3}{2}}}.$$

Moreover, we have

$$\mathbb{E}[\vartheta_{ac1}^2]$$

$$\le \frac{C}{N^2 V^2 L^4 d} \sum_{i=1}^{N} \mathbb{E}\left[\mathsf{z}_k^\top \boldsymbol{Z}_{\mathrm{in}} \left(\boldsymbol{N}_i^\top \boldsymbol{N}_i - \frac{L-1}{V} \boldsymbol{I}_V\right) \boldsymbol{Z}_{\mathrm{in}}^\top \boldsymbol{Z}_{\mathrm{in}} \left(\boldsymbol{N}_i^\top \boldsymbol{N}_i - \frac{L-1}{V} \boldsymbol{I}_V\right) \boldsymbol{Z}_{\mathrm{in}}^\top \mathsf{z}_k | \mathsf{Z}_{\mathrm{in}}\right]$$

$$\leq \frac{C}{N^2 V^2 L^4 d} \frac{L-1}{V} \sum_{i=1}^{N} \mathbb{E}\left[ z_k^\top \boldsymbol{Z}_{\text{in}} \boldsymbol{Z}_{\text{in}}^\top z_k | \mathsf{Z}_{\text{in}} \right]$$

$$- \frac{C}{N^2 V^2 L^4 d} \frac{L-1}{V^2} \sum_{i=1}^{N} \mathbb{E}\left[ z_k^\top \boldsymbol{Z}_{\text{in}} \boldsymbol{Z}_{\text{in}}^\top \boldsymbol{Z}_{\text{in}} \boldsymbol{Z}_{\text{in}}^\top z_k | \mathsf{Z}_{\text{in}} \right]$$

$$\leq \frac{C}{N V^2 L^3 d^2}.$$

Therefore, by Chebyshev's inequality, we have

$$|\vartheta_{ac1}| \leq \frac{C \log V}{\sqrt{N} V L^{\frac{3}{2}} d}$$

For the fourth term, we have $\mathbb{E}[\vartheta_{ad}|\mathsf{Z}_{\text{in}}] = 0$ and

$$\mathbb{E}[\vartheta_{ad}^2 | \mathsf{Z}_{\text{in}}]$$

$$= \frac{C}{N^4 L^4} \mathbb{E}\Big[\Big( \sum_{i,j=1}^{N} \mathbb{1}_{i \neq j} (\mathbb{1}_{\boldsymbol{x}_i = \boldsymbol{x}_j} - \tfrac{1}{V} \mathbb{1}_V) z_k^\top \boldsymbol{Z}_{\text{in}}$$

$$\times \boldsymbol{N}_i^\top \boldsymbol{N}_i \boldsymbol{Z}_{\text{in}}^\top \boldsymbol{Z}_{\text{in}} (\boldsymbol{N}_j^\top - \frac{1}{V} \mathbb{1}_V \mathbb{1}_{L-1}^\top) \mathbb{1}_{L-1} \Big)^2 | \mathsf{Z}_{\text{in}} \Big]$$

$$= \frac{C}{V N^4 L^4} \sum_{i,j=1}^{N} \mathbb{E}\left[ \Big( z_k^\top \boldsymbol{Z}_{\text{in}} \boldsymbol{N}_i^\top \boldsymbol{N}_i \boldsymbol{Z}_{\text{in}}^\top \boldsymbol{Z}_{\text{in}} (\boldsymbol{N}_j^\top - \frac{1}{V} \mathbb{1}_V \mathbb{1}_{L-1}^\top) \mathbb{1}_{L-1} \Big)^2 | \mathsf{Z}_{\text{in}} \right]$$

$$\leq \frac{C}{V N^4 L^3 d} \sum_{i,j=1}^{N} \mathbb{E}\left[ z_k^\top \boldsymbol{Z}_{\text{in}} \boldsymbol{N}_i^\top \boldsymbol{N}_i \boldsymbol{Z}_{\text{in}}^\top \boldsymbol{Z}_{\text{in}} \boldsymbol{N}_i^\top \boldsymbol{N}_i \boldsymbol{Z}_{\text{in}}^\top z_k | \mathsf{Z}_{\text{in}} \right]$$

$$\leq \frac{C}{V N^2 L d^2 (L \wedge d)}$$

where we used (P2) in the last step.

Therefore, by Chebyshev's inequality with probability $1 - o_V(1)$, we have

$$|\vartheta_{ad}| \leq \frac{C \log V}{N \sqrt{V L} d \sqrt{L \wedge d}}$$

For the last term, we have $\mathbb{E}[\vartheta_{ae}|\mathsf{Z}_{\text{in}}] = 0$ and

$$\mathbb{E}[\vartheta_{ae}^2 | \mathsf{Z}_{\text{in}}]$$

$$\leq \frac{C}{N^4 L^6} \mathbb{E}\Big[\Big( \sum_{i,j=1}^{N} \mathbb{1}_{i \neq j} (\mathbb{1}_{\boldsymbol{x}_i = \boldsymbol{x}_j} - \tfrac{1}{V}) z_k^\top \boldsymbol{Z}_{\text{in}}$$

$$\times \boldsymbol{N}_i^\top \mathbb{1}_L \mathbb{1}_L^\top \boldsymbol{N}_i \boldsymbol{Z}_{\text{in}}^\top \boldsymbol{Z}_{\text{in}} (\boldsymbol{N}_j^\top - \frac{1}{V} \mathbb{1}_V \mathbb{1}_{L-1}^\top) \mathbb{1}_{L-1} \Big)^2 | \mathsf{Z}_{\text{in}} \Big]$$

$$\leq \frac{C}{V N^4 L^6} \sum_{i,j=1}^{N} \mathbb{E}\left[ \Big( z_k^\top \boldsymbol{Z}_{\text{in}} \boldsymbol{N}_i^\top \mathbb{1}_L \mathbb{1}_L^\top \boldsymbol{N}_i \boldsymbol{Z}_{\text{in}}^\top \boldsymbol{Z}_{\text{in}} (\boldsymbol{N}_j^\top - \frac{1}{V} \mathbb{1}_V \mathbb{1}_{L-1}^\top) \mathbb{1}_{L-1} \Big)^2 | \mathsf{Z}_{\text{in}} \right]$$

$$\leq \frac{C}{N^4 L^4 d V} \sum_{i,j=1}^{N} \mathbb{E}\left[ z_k^\top \boldsymbol{Z}_{\text{in}} \boldsymbol{N}_i^\top \mathbb{1}_{L-1} \mathbb{1}_{L-1}^\top \boldsymbol{N}_i \boldsymbol{Z}_{\text{in}}^\top z_k | \mathsf{Z}_{\text{in}} \right]$$

$$\leq \frac{C}{N^2 L^4 d V} \Big( 1 + \frac{L}{d} \Big)$$

where we use (E1.4) in the last step.

Therefore, by Chebyshev's inequality with probability $1 - o_V(1)$, we have

$$|\vartheta_{ae}| \leq \frac{C \log V}{N \sqrt{V} L^{3/2} \sqrt{d} (L \wedge d)^{1/2}}.$$

Overall, we have

$$|\vartheta_{11a}| \leq C \log V \Big( \frac{1}{VL^2\sqrt{d}} + \frac{1}{\sqrt{N}V\sqrt{Ld}(L \wedge d)} \Big). \tag{D.20}$$

Therefore, by (D.12)-(D.19)-(D.20) and using $N \ll VL$ and $L \ll V$, we have

$$
\begin{aligned}
|\vartheta_{11}| &\leq C \log V \Big( \frac{1}{VL^2\sqrt{d}} + \frac{1}{\sqrt{N}V\sqrt{Ld}(L \wedge d)} \Big) \\
&\quad + C \log V \Big( \frac{1}{N\sqrt{V}(L \wedge d)\sqrt{d}} + \frac{1}{NL^2\sqrt{L \wedge d}} \Big) + \frac{C \log V}{VL\sqrt{d}\sqrt{L \wedge d}} \frac{1}{V \wedge L^2 \wedge L\sqrt{d}} \\
&\leq C \log V \Big( \frac{1}{N\sqrt{V}(L \wedge d)\sqrt{d}} + \frac{1}{VL^2(L \wedge d)^{1/2}} \Big). \tag{D.21}
\end{aligned}
$$

Finally,

$$
\begin{aligned}
\vartheta_{12} &= \frac{1}{N^2L^2V} \sum_{i,j=1}^{N} \alpha_{ij}(\mathbb{1}_{\boldsymbol{x}_i=\boldsymbol{x}_j} - \tfrac{1}{V})\mathsf{z}_k^\top \boldsymbol{Z}_{\text{in}} \boldsymbol{X}_i^\top \big( \boldsymbol{I}_L - \tfrac{1}{L}\mathbb{1}_L\mathbb{1}_L^\top \big) \boldsymbol{X}_i \boldsymbol{Z}_{\text{in}}^\top \boldsymbol{Z}_{\text{in}} \mathbb{1}_V \\
&= \frac{1}{N^2L^2V} \sum_{i,j=1}^{N} \alpha_{ij}(\mathbb{1}_{\boldsymbol{x}_i=\boldsymbol{x}_j} - \tfrac{1}{V})\mathsf{z}_k^\top \boldsymbol{Z}_{\text{in}} \Big( \boldsymbol{X}_i^\top \boldsymbol{X}_i - \tfrac{L}{V}\boldsymbol{I}_V \Big) \boldsymbol{Z}_{\text{in}}^\top \boldsymbol{Z}_{\text{in}} \mathbb{1}_V \\
&\quad + \frac{1}{N^2LV^2} \sum_{i,j=1}^{N} \alpha_{ij}(\mathbb{1}_{\boldsymbol{x}_i=\boldsymbol{x}_j} - \tfrac{1}{V})\mathsf{z}_k^\top \boldsymbol{Z}_{\text{in}} \boldsymbol{Z}_{\text{in}}^\top \boldsymbol{Z}_{\text{in}} \mathbb{1}_V \\
&\quad - \frac{1}{N^2L^3V} \sum_{i,j=1}^{N} \alpha_{ij}(\mathbb{1}_{\boldsymbol{x}_i=\boldsymbol{x}_j} - \tfrac{1}{V})\mathsf{z}_k^\top \boldsymbol{Z}_{\text{in}} \boldsymbol{X}_i^\top \mathbb{1}_L \mathbb{1}_L^\top \boldsymbol{X}_i \boldsymbol{Z}_{\text{in}}^\top \boldsymbol{Z}_{\text{in}} \mathbb{1}_V
\end{aligned}
$$

- For the first term,

$$
\begin{aligned}
&\frac{1}{N^2L^2V} \sum_{i,j=1}^{N} \alpha_{ij}(\mathbb{1}_{\boldsymbol{x}_i=\boldsymbol{x}_j} - \tfrac{1}{V})\mathsf{z}_k^\top \boldsymbol{Z}_{\text{in}} \big( \boldsymbol{X}_i^\top \boldsymbol{X}_i - \tfrac{L}{V}\boldsymbol{I}_V \big) \boldsymbol{Z}_{\text{in}}^\top \boldsymbol{Z}_{\text{in}} \mathbb{1}_V \\
&\leq \frac{1}{N^2L^2V} \Big( \sum_{i,j=1}^{N} |\mathbb{1}_{\boldsymbol{x}_i=\boldsymbol{x}_j} - \tfrac{1}{V}| \Big) \sup_{i,j} \Big| (\alpha_{ij} - \phi'(0)^2)\mathsf{z}_k^\top \boldsymbol{Z}_{\text{in}} \big( \boldsymbol{X}_i^\top \boldsymbol{X}_i - \tfrac{L}{V}\boldsymbol{I}_V \big) \boldsymbol{Z}_{\text{in}}^\top \boldsymbol{Z}_{\text{in}} \mathbb{1}_V \Big| \\
&\quad + \frac{\phi'(0)^2}{N^2L^2V} \Big( \sum_{i=1}^{N} \Big| \sum_{j=1}^{N} (\mathbb{1}_{\boldsymbol{x}_i=\boldsymbol{x}_j} - \tfrac{1}{V}) \Big| \Big) \sup_{i} \Big| (\mathsf{z}_k^\top \boldsymbol{Z}_{\text{in}} \big( \boldsymbol{X}_i^\top \boldsymbol{X}_i - \tfrac{L}{V}\boldsymbol{I}_V \big) \boldsymbol{Z}_{\text{in}}^\top \boldsymbol{Z}_{\text{in}} \mathbb{1}_V \Big| \\
&\leq \frac{1}{N^2L^2V} \Big( \sum_{i,j=1}^{N} |\mathbb{1}_{\boldsymbol{x}_i=\boldsymbol{x}_j} - \tfrac{1}{V}| \Big) \sup_{i,j} \Big| (\alpha_{ij} - \phi'(0)^2)\mathsf{z}_k^\top \boldsymbol{Z}_{\text{in}} \big( \boldsymbol{X}_i^\top \boldsymbol{X}_i - \tfrac{L}{V}\boldsymbol{I}_V \big) \boldsymbol{Z}_{\text{in}}^\top \boldsymbol{Z}_{\text{in}} \mathbb{1}_V \Big| \\
&\quad + \frac{\phi'(0)^2}{NL^2V} \Big\| \sum_{j=1}^{N} (\boldsymbol{x}_j - \tfrac{1}{V}\mathbb{1}_V) \Big\|_\infty \sup_{i} \Big| \mathsf{z}_k^\top \boldsymbol{Z}_{\text{in}} \big( \boldsymbol{X}_i^\top \boldsymbol{X}_i - \tfrac{L}{V}\boldsymbol{I}_V \big) \boldsymbol{Z}_{\text{in}}^\top \boldsymbol{Z}_{\text{in}} \mathbb{1}_V \Big| \\
&\lesssim \frac{1}{VL^2\sqrt{V}(L \wedge d)},
\end{aligned}
$$

where we (E1.2), (E1.8), (E2.3), and (R2).

- For the second term

$$
\begin{aligned}
&\frac{1}{N^2LV^2} \sum_{i=1}^{N} \sum_{j=1}^{N} \alpha_{ij}(\mathbb{1}_{\boldsymbol{x}_i=\boldsymbol{x}_j} - \frac{1}{V})\mathsf{z}_k^\top \boldsymbol{Z}_{\text{in}} \boldsymbol{Z}_{\text{in}}^\top \boldsymbol{Z}_{\text{in}} \mathbb{1}_V \\
&\leq \frac{1}{N^2LV^2} \Big( \sum_{i=1}^{N} \sum_{j=1}^{N} |\mathbb{1}_{\boldsymbol{x}_i=\boldsymbol{x}_j} - \frac{1}{V}| \Big) \sup_{i,j} |\alpha_{ij} \mathsf{z}_k^\top \boldsymbol{Z}_{\text{in}} \boldsymbol{Z}_{\text{in}}^\top \boldsymbol{Z}_{\text{in}} \mathbb{1}_V| \leq \frac{C \log V}{V^{3/2}Ld^{3/2}},
\end{aligned}
$$

where we used (E1.3), (E2.4), and (R2).

- For the third term,

$$\frac{1}{N^2 L^3 V} \sum_{i,j=1}^{N} \alpha_{ij} (\mathbb{1}_{\boldsymbol{x}_i = \boldsymbol{x}_j} - \tfrac{1}{V}) \mathbf{z}_k^\top \boldsymbol{Z}_{\text{in}} \boldsymbol{X}_i^\top \mathbb{1}_L \mathbb{1}_L^\top \boldsymbol{X}_i \boldsymbol{Z}_{\text{in}}^\top \boldsymbol{Z}_{\text{in}} \mathbb{1}_V$$

$$\leq \frac{1}{N^2 L^3 V} \Big( \sum_{i,j=1}^{N} |\mathbb{1}_{\boldsymbol{x}_i = \boldsymbol{x}_j} - \tfrac{1}{V} \mathbb{1}_V| \Big) \sup_{i,j} |\alpha_{ij} \mathbf{z}_k^\top \boldsymbol{Z}_{\text{in}} \boldsymbol{X}_i^\top \mathbb{1}_L \mathbb{1}_L^\top \boldsymbol{X}_i \boldsymbol{Z}_{\text{in}}^\top \boldsymbol{Z}_{\text{in}} \mathbb{1}_V|$$

$$\leq \frac{1}{V^2 L^2} \frac{\left( \sqrt{L} + \sqrt{\frac{V}{d}} \right)}{\sqrt{L \wedge d}}$$

where we used (E1.4), (E1.5), (E2.4), and (R2).

Therefore, by Chebyshev's inequality with probability $1 - o_V(1)$, we have

$$|\vartheta_{12}| \leq \frac{C \log^3 V}{N L^3 V d^2} + \frac{C \log^2 V}{V L \sqrt{V} d^{3/2}}. \tag{D.22}$$

By (D.21)-(D.22), overall we have

$$\vartheta_1 \leq C \log V \Big( \frac{1}{V L^2 \sqrt{d}} + \frac{1}{\sqrt{NV} L^{3/2} d} + \frac{1}{N \sqrt{VL} d \sqrt{L \wedge d}} + \frac{1}{N L d \sqrt{V} (L \wedge \sqrt{V})} \Big)$$

$$+ C \log V \Big( \frac{1}{N \sqrt{V} (L \wedge d) \sqrt{d}} + \frac{1}{N \sqrt{V} d^{\frac{3}{2}}} + \frac{1}{N L^2 \sqrt{L \wedge d}} \Big) + \frac{C \log V}{V \sqrt{Ld} (L \wedge d)} \frac{1}{V \wedge L^2 \wedge L \sqrt{d}}$$

$$\leq C \log V \Big( \frac{1}{N \sqrt{V} (L \wedge d) \sqrt{d}} + \frac{1}{\sqrt{NV} L^{3/2} d} + \frac{1}{V L^{3/2} d (L \wedge d)} + \frac{1}{V^2 \sqrt{Ld} (L \wedge d)} \Big)$$

$$+ \frac{C \log V}{V L^2 (L \wedge d)^{1/2}}.$$

### D.1.2 Concentration bound for $\varphi$

We recall that

$$\varphi = \frac{\mu_{kl}}{N^2 L^2} \sum_{i,j=1}^{N} \alpha_{ij} \big(\boldsymbol{e}_1 - \tfrac{1}{L} \mathbb{1}_L\big)^\top \boldsymbol{X}_i \boldsymbol{Z}_{\text{in}}^\top \boldsymbol{Z}_{\text{in}} \boldsymbol{X}_j^\top \mathbb{1}_L (\boldsymbol{x}_j - \tfrac{1}{V} \mathbb{1}_V)^\top \boldsymbol{Z}_{\text{out}}^\top \boldsymbol{Z}_{\text{out}} (\boldsymbol{x}_i - \tfrac{1}{V} \mathbb{1}_V)$$

In this part, we will focus on the term

$$\frac{1}{N^2 L^2} \sum_{i,j=1}^{N} \alpha_{ij} \big(\boldsymbol{e}_1 - \tfrac{1}{L} \mathbb{1}_L\big)^\top \boldsymbol{X}_i \boldsymbol{Z}_{\text{in}}^\top \boldsymbol{Z}_{\text{in}} \boldsymbol{X}_j^\top \mathbb{1}_L (\boldsymbol{x}_j - \tfrac{1}{V} \mathbb{1}_V)^\top \boldsymbol{Z}_{\text{out}}^\top \boldsymbol{Z}_{\text{out}} (\boldsymbol{x}_i - \tfrac{1}{V} \mathbb{1}_V)$$

$$= \frac{1}{N^2 L^2} \sum_{i,j=1}^{N} \alpha_{ij} \text{tr} \Big( \boldsymbol{Z}_{\text{in}}^\top \boldsymbol{Z}_{\text{in}} \boldsymbol{X}_j^\top \mathbb{1}_L (\boldsymbol{x}_j - \tfrac{1}{V} \mathbb{1}_V)^\top \boldsymbol{Z}_{\text{out}}^\top \boldsymbol{Z}_{\text{out}} (\boldsymbol{x}_i - \tfrac{1}{V} \mathbb{1}_V) \boldsymbol{x}_i^\top \Big)$$

$$- \frac{1}{N^2 L^3} \sum_{i,j=1}^{N} \alpha_{ij} \text{tr} \Big( \boldsymbol{Z}_{\text{in}}^\top \boldsymbol{Z}_{\text{in}} \boldsymbol{X}_j^\top \mathbb{1}_L (\boldsymbol{x}_j - \tfrac{1}{V} \mathbb{1}_V)^\top \boldsymbol{Z}_{\text{out}}^\top \boldsymbol{Z}_{\text{out}} (\boldsymbol{x}_i - \tfrac{1}{V} \mathbb{1}_V) \mathbb{1}_L^\top \boldsymbol{X}_i \Big)$$

$$=: \varphi_1 + \varphi_2.$$

For the first term, we write

$$\varphi_1 = \frac{\phi'(0)^2}{N^2 L^2} \sum_{i,j=1}^{N} \text{tr} \Big( \boldsymbol{Z}_{\text{out}} (\boldsymbol{x}_i - \tfrac{1}{V} \mathbb{1}_V) \boldsymbol{x}_i^\top \boldsymbol{Z}_{\text{in}}^\top \boldsymbol{Z}_{\text{in}} \boldsymbol{X}_j^\top \mathbb{1}_L (\boldsymbol{x}_j - \tfrac{1}{V} \mathbb{1}_V)^\top \boldsymbol{Z}_{\text{out}}^\top \Big)$$

$$+ \frac{1}{N^2 L^2} \sum_{i,j=1}^{N} \text{tr} \Big( \boldsymbol{Z}_{\text{out}} (\alpha_{ij} - \phi'(0)^2)(\boldsymbol{x}_i - \tfrac{1}{V} \mathbb{1}_V) \boldsymbol{x}_i^\top \boldsymbol{Z}_{\text{in}}^\top \boldsymbol{Z}_{\text{in}} \boldsymbol{X}_j^\top \mathbb{1}_L (\boldsymbol{x}_j - \tfrac{1}{V} \mathbb{1}_V)^\top \boldsymbol{Z}_{\text{out}}^\top \Big)$$

$$=: \varphi_{11} + \varphi_{12}$$

We start with the second term. By Proposition 5, we have

$$\varphi_{12} = \frac{1}{N^2 L^2} \sum_{i,j=1}^{N} (\alpha_{ij} - \phi'(0)^2)(\mathbb{1}_{\boldsymbol{x}_i = \boldsymbol{x}_j} - \frac{1}{V}) \boldsymbol{x}_i^\top \boldsymbol{Z}_{\text{in}}^\top \boldsymbol{Z}_{\text{in}} \boldsymbol{X}_j^\top \mathbb{1}_L$$

$$\pm \frac{\log^2 V}{N^2 L^2 \sqrt{d}} \Big\| \sum_{i,j=1}^{N} (\alpha_{ij} - \phi'(0)^2)(\boldsymbol{x}_i - \frac{1}{V}\mathbb{1}_V) \boldsymbol{x}_i^\top \boldsymbol{Z}_{\text{in}}^\top \boldsymbol{Z}_{\text{in}} \boldsymbol{X}_j^\top \mathbb{1}_L (\boldsymbol{x}_j - \frac{1}{V}\mathbb{1}_V)^\top \Big\|_F$$

Let $\boldsymbol{M} := \big\{ (\alpha_{ij} - \phi'(0)^2) \boldsymbol{x}_i^\top \boldsymbol{Z}_{\text{in}}^\top \boldsymbol{Z}_{\text{in}} \boldsymbol{X}_j^\top \mathbb{1}_L) \big\}_{i,j \in [N]}$. We have

$$\Big\| \sum_{i,j=1}^{N} (\alpha_{ij} - \phi'(0)^2)(\boldsymbol{x}_i - \frac{1}{V}\mathbb{1}_V) \boldsymbol{x}_i^\top \boldsymbol{Z}_{\text{in}}^\top \boldsymbol{Z}_{\text{in}} \boldsymbol{X}_j^\top \mathbb{1}_L (\boldsymbol{x}_j - \frac{1}{V}\mathbb{1}_V)^\top \Big\|_F$$

$$\leq \|\boldsymbol{M}\|_F \Big\| \sum_{i=1}^{N} (\boldsymbol{x}_i - \frac{1}{V}\mathbb{1}_V)(\boldsymbol{x}_i - \frac{1}{V}\mathbb{1}_V)^\top \Big\|_2 \leq \frac{N}{V} \|\boldsymbol{M}\|_F,$$

where we used (E2.5). Moroever,

$$\|\boldsymbol{S}\|_F^2 = \sum_{i,j=1}^{N} (\alpha_{ij} - \phi'(0)^2)^2 (\boldsymbol{x}_i^\top \boldsymbol{Z}_{\text{in}}^\top \boldsymbol{Z}_{\text{in}} \boldsymbol{X}_j^\top \mathbb{1}_L)^2$$

$$\leq \Big( \sum_{i \neq j=1}^{N} |\alpha_{ij} - \phi'(0)^2|^2 + \sum_{i=1}^{N} |\alpha_{ii} - \phi'(0)^2|^2 \Big) \sup_{i,j} |\boldsymbol{x}_i^\top \boldsymbol{Z}_{\text{in}}^\top \boldsymbol{Z}_{\text{in}} \boldsymbol{X}_j^\top \mathbb{1}_L|$$

$$\lesssim \Big( \frac{N^2}{(V \wedge L^2 \wedge L\sqrt{d})^2} + \frac{N}{L^2} \Big) \Big( 1 + \frac{L}{d} \Big),$$

where we used (E1.4) and (E2.4). Therefore,

$$\frac{1}{N^2 L^2 \sqrt{d}} \Big\| \sum_{i,j=1}^{N} (\alpha_{ij} - \phi'(0)^2)(\boldsymbol{x}_i - \frac{1}{V}\mathbb{1}_V) \boldsymbol{x}_i^\top \boldsymbol{Z}_{\text{in}}^\top \boldsymbol{Z}_{\text{in}} \boldsymbol{X}_j^\top \mathbb{1}_L (\boldsymbol{x}_j - \frac{1}{V}\mathbb{1}_V)^\top \Big\|_F$$

$$\leq \frac{1}{N V L^{3/2} \sqrt{d}(L \wedge d)^{1/2}} \Big( \frac{N}{V \wedge L^2 \wedge L\sqrt{d}} + \frac{\sqrt{N}}{L} \Big) \lesssim \frac{1}{V L^2 \sqrt{L \wedge d}}.$$

Moreover,

$$\frac{1}{N^2 L^2} \Big| \sum_{i,j=1}^{N} (\alpha_{ij} - \phi'(0)^2)(\mathbb{1}_{\boldsymbol{x}_i = \boldsymbol{x}_j} - \frac{1}{V}) \boldsymbol{x}_i^\top \boldsymbol{Z}_{\text{in}}^\top \boldsymbol{Z}_{\text{in}} \boldsymbol{X}_j^\top \mathbb{1}_L \Big|$$

$$\leq \frac{1}{N^2 L^2} \Big( \sum_{i,j=1}^{N} |\mathbb{1}_{\boldsymbol{x}_i = \boldsymbol{x}_j} - \frac{1}{V}| \Big) \sup_{i,j \in [N]} |(\alpha_{ij} - \phi'(0)^2) \boldsymbol{x}_i^\top \boldsymbol{Z}_{\text{in}}^\top \boldsymbol{Z}_{\text{in}} \boldsymbol{X}_j^\top \mathbb{1}_L| \lesssim \frac{1}{V L^2 \sqrt{L \wedge d}},$$

where we used (E1.4), (E2.4), (R2). Therefore, $|\varphi_{12}| \lesssim \frac{1}{V L^2 \sqrt{L \wedge d}}$.

Next, we consider $|\varphi_2|$. By Proposition 5,

$$\varphi_2 = \frac{\phi'(0)^2}{N^2 L^3} \sum_{i,j=1}^{N} (\mathbb{1}_{\boldsymbol{x}_i = \boldsymbol{x}_j} - \frac{1}{V}) \mathbb{1}_L^\top \boldsymbol{X}_i \boldsymbol{Z}_{\text{in}}^\top \boldsymbol{Z}_{\text{in}} \boldsymbol{X}_j^\top \mathbb{1}_L$$

$$\pm \frac{\phi'(0)^2}{\sqrt{d}} \frac{1}{N^2 L^3} \Big\| \sum_{i,j=1}^{N} (\boldsymbol{x}_i - \frac{1}{V}\mathbb{1}_V) \mathbb{1}_L^\top \boldsymbol{X}_i \boldsymbol{Z}_{\text{in}}^\top \boldsymbol{Z}_{\text{in}} \boldsymbol{X}_j^\top \mathbb{1}_L (\boldsymbol{x}_j - \frac{1}{V}\mathbb{1}_V)^\top \Big\|_F$$

$$+ \frac{1}{N^2 L^3} \sum_{i,j=1}^{N} (\alpha_{ij} - \phi'(0)^2)(\mathbb{1}_{\boldsymbol{x}_i = \boldsymbol{x}_j} - \frac{1}{V}) \mathbb{1}_L^\top \boldsymbol{X}_i \boldsymbol{Z}_{\text{in}}^\top \boldsymbol{Z}_{\text{in}} \boldsymbol{X}_j^\top \mathbb{1}_L$$

$$\pm \frac{1}{\sqrt{d}} \frac{1}{N^2 L^3} \Big\| \sum_{i,j=1}^{N} (\alpha_{ij} - \phi'(0)^2)(\boldsymbol{x}_i - \tfrac{1}{V}\mathbb{1}_V)\mathbb{1}_L^\top \boldsymbol{X}_i \boldsymbol{Z}_{\mathrm{in}}^\top \boldsymbol{Z}_{\mathrm{in}} \boldsymbol{X}_j^\top \mathbb{1}_L (\boldsymbol{x}_j - \tfrac{1}{V}\mathbb{1}_V)^\top \Big\|_F$$

$$=: \varphi_{21} + \varphi_{22} + \varphi_{23} + \varphi_{24}.$$

For $\varphi_{24}$, we define $\boldsymbol{M} := \big\{ (\alpha_{ij} - \phi'(0)^2)\mathbb{1}_L^\top \boldsymbol{X}_i \boldsymbol{Z}_{\mathrm{in}}^\top \boldsymbol{Z}_{\mathrm{in}} \boldsymbol{X}_j^\top \mathbb{1}_L \big\}_{i,j \in [N]}$. Similar to above, we have

$$\varphi_{24} \le \frac{1}{N V L^3 \sqrt{d}} \|\boldsymbol{M}\|_F.$$

We have

$$\|\boldsymbol{M}\|_F^2 = \sum_{i,j=1}^{N} (\alpha_{ij} - \phi'(0)^2)^2 (\boldsymbol{X}_i^\top \boldsymbol{Z}_{\mathrm{in}}^\top \boldsymbol{Z}_{\mathrm{in}} \boldsymbol{X}_j^\top \mathbb{1}_L)^2$$

$$\le \Big( \sum_{i \ne j=1}^{N} |\alpha_{ij} - \phi'(0)^2|^2 \Big) \sup_{i \ne j} |\mathbb{1}_L^\top \boldsymbol{X}_i^\top \boldsymbol{Z}_{\mathrm{in}}^\top \boldsymbol{Z}_{\mathrm{in}} \boldsymbol{X}_j^\top \mathbb{1}_L|$$

$$+ \Big( \sum_{i=1}^{N} |\alpha_{ii} - \phi'(0)^2|^2 \Big) \sup_i |\mathbb{1}_L^\top \boldsymbol{X}_i^\top \boldsymbol{Z}_{\mathrm{in}}^\top \boldsymbol{Z}_{\mathrm{in}} \boldsymbol{X}_i^\top \mathbb{1}_L|$$

$$\lesssim L \Big( \frac{N^2}{(V \wedge L^2 \wedge L\sqrt{d})^3} + \frac{N}{L^2} \Big),$$

where we used (R1), (R2). Therefore,

$$|\varphi_{24}| \le \frac{1}{N V L^2 \sqrt{d}} \Big( \frac{N}{(V \wedge L^2 \wedge L\sqrt{d})^{3/2}} + \frac{\sqrt{N}}{L} \Big) \lesssim \frac{1}{V L^2 \sqrt{L \wedge d}}.$$

For $\varphi_{23}$, we have

$$|\varphi_{23}| \le \frac{1}{N^2 L^3} \Big( \sum_{i \ne j=1}^{N} |\mathbb{1}_{\boldsymbol{x}_i = \boldsymbol{x}_j} - \tfrac{1}{V}| \Big) |\sup_{i \ne j} |(\alpha_{ij} - \phi'(0)^2)\mathbb{1}_L^\top \boldsymbol{X}_i \boldsymbol{Z}_{\mathrm{in}}^\top \boldsymbol{Z}_{\mathrm{in}} \boldsymbol{X}_j^\top \mathbb{1}_L|$$

$$+ \frac{1}{N L^3} \sup_i |(\alpha_{ii} - \phi'(0)^2)\mathbb{1}_L^\top \boldsymbol{X}_i \boldsymbol{Z}_{\mathrm{in}}^\top \boldsymbol{Z}_{\mathrm{in}} \boldsymbol{X}_i^\top \mathbb{1}_L| \lesssim \frac{1}{V L^2 \sqrt{L \wedge d}},$$

where we used (E2.4), (R1), (R2).

For the first two terms, we define

$$\boldsymbol{V}_0 := \frac{1}{NL} \sum_{j=1}^{N} \boldsymbol{X}_j^\top \mathbb{1}_L (\boldsymbol{x}_j - \tfrac{1}{V}\mathbb{1}_V)^\top, \qquad\qquad \boldsymbol{V}_{0,1} := \frac{1}{NL} \sum_{i=1}^{N} \boldsymbol{x}_i (\boldsymbol{x}_i - \tfrac{1}{V}\mathbb{1}_V)^\top$$

$$\boldsymbol{V}_{0,2} := \frac{1}{NL} \sum_{i=1}^{N} \big( \boldsymbol{N}_j^\top - \tfrac{1}{V}\mathbb{1}_{L-1}^\top \big) \mathbb{1}_{L-1} (\boldsymbol{x}_i - \tfrac{1}{V}\mathbb{1}_V)^\top, \quad \boldsymbol{V}_{0,3} := \frac{1}{V}\mathbb{1}_V \frac{1}{NL} \sum_{i=1}^{N} (\boldsymbol{x}_i - \tfrac{1}{V}\mathbb{1}_V)^\top.$$

We have by (E2.5)-(E2.7),

$$|\varphi_{22}| \le \frac{\phi'(0)^2}{\sqrt{d}} \frac{1}{L} \Big\| \boldsymbol{Z}_{\mathrm{in}} \boldsymbol{V}_0 \boldsymbol{V}_0^\top \boldsymbol{Z}_{\mathrm{in}}^\top \Big\|_F \le \frac{\phi'(0)^2}{N V L^2 \sqrt{d}} \Big\| \boldsymbol{Z}_{\mathrm{in}} \boldsymbol{Z}_{\mathrm{in}}^\top \Big\|_F \le \frac{2\phi'(0)^2}{L d L N} \lesssim \frac{1}{V L^2 d}.$$

Lastly,

$$|\varphi_{21}| = \frac{\phi'(0)^2}{L} \mathrm{tr}\Big( \boldsymbol{Z}_{\mathrm{in}} \boldsymbol{V}_0 \boldsymbol{V}_0^\top \boldsymbol{Z}_{\mathrm{in}}^\top \Big) \le \frac{\phi'(0)^2}{N V L^2 \sqrt{d}} \mathrm{tr}\Big( \boldsymbol{Z}_{\mathrm{in}} \boldsymbol{Z}_{\mathrm{in}}^\top \Big) \le \frac{2\phi'(0)^2}{N L^2 \sqrt{d}} \lesssim \frac{1}{V L^2 \sqrt{d}}.$$

Therefore, $|\varphi_{21}| \lesssim \frac{1}{V L^2 \sqrt{L \wedge d}}$.

Lastly, we consider $\varphi_{11}$. By Proposition 5, we have

$$|\varphi_{11}|$$
$$= \phi'(0)^2 \mathrm{tr}(\boldsymbol{V}_{0,1}^\top \boldsymbol{Z}_{\mathrm{in}}^\top \boldsymbol{Z}_{\mathrm{in}} \boldsymbol{V}_{0,1}) + \phi'(0)^2 \mathrm{tr}(\boldsymbol{V}_{0,1}^\top \boldsymbol{Z}_{\mathrm{in}}^\top \boldsymbol{Z}_{\mathrm{in}} \boldsymbol{V}_{0,2}) + \phi'(0)^2 L \mathrm{tr}(\boldsymbol{V}_{0,1}^\top \boldsymbol{Z}_{\mathrm{in}}^\top \boldsymbol{Z}_{\mathrm{in}} \boldsymbol{V}_{0,3})$$
$$\pm \frac{1}{\sqrt{d}} \Big\| \boldsymbol{V}_{0,1}^\top \boldsymbol{Z}_{\mathrm{in}}^\top \boldsymbol{Z}_{\mathrm{in}} \boldsymbol{V}_{0,1} \Big\|_F \pm \frac{1}{\sqrt{d}} \Big\| \boldsymbol{V}_{0,1}^\top \boldsymbol{Z}_{\mathrm{in}}^\top \boldsymbol{Z}_{\mathrm{in}} \boldsymbol{V}_{0,2} \Big\|_F \pm \frac{L}{\sqrt{d}} \Big\| \boldsymbol{V}_{0,1}^\top \boldsymbol{Z}_{\mathrm{in}}^\top \boldsymbol{Z}_{\mathrm{in}} \boldsymbol{V}_{0,3} \Big\|_F.$$

- For the first term, by (E2.5), we have

$$\mathrm{tr}(\boldsymbol{V}_{0,1}^\top \boldsymbol{Z}_{\mathrm{in}}^\top \boldsymbol{Z}_{\mathrm{in}} \boldsymbol{V}_{0,1}) = \mathrm{tr}(\boldsymbol{Z}_{\mathrm{in}} \boldsymbol{V}_{0,1} \boldsymbol{V}_{0,1}^\top \boldsymbol{Z}_{\mathrm{in}}^\top) \asymp \frac{1}{VL^2}$$

- For the second term,

$$\mathrm{tr}(\boldsymbol{V}_{0,1}^\top \boldsymbol{Z}_{\mathrm{in}}^\top \boldsymbol{Z}_{\mathrm{in}} \boldsymbol{V}_{0,2}) = \frac{1}{NL} \sum_{j=1}^N (\boldsymbol{x}_j - \tfrac{1}{V}\mathbb{1}_V)^\top \boldsymbol{V}_{0,1}^\top \boldsymbol{Z}_{\mathrm{in}}^\top \boldsymbol{Z}_{\mathrm{in}} (\boldsymbol{N}_j^\top - \tfrac{1}{V}\mathbb{1}_{L-1}^\top)\mathbb{1}_{L-1}$$

We have

$$\mathbb{E}\left[\left((\boldsymbol{x}_j - \tfrac{1}{V}\mathbb{1}_V)^\top \boldsymbol{V}_{0,1}^\top \boldsymbol{Z}_{\mathrm{in}}^\top \boldsymbol{Z}_{\mathrm{in}}(\boldsymbol{N}_j^\top - \tfrac{1}{V}\mathbb{1}_{L-1}^\top)\mathbb{1}_{L-1}\right)^2 \Big| \boldsymbol{Z}_{\mathrm{in}}\right] \lesssim \frac{1}{V^2 L d}$$

By Chebyshev's inequality,

$$\left|\mathrm{tr}(\boldsymbol{V}_{0,1}^\top \boldsymbol{Z}_{\mathrm{in}}^\top \boldsymbol{Z}_{\mathrm{in}} \boldsymbol{V}_{0,2})\right| \lesssim \frac{1}{\sqrt{N}VL^{3/2}\sqrt{d}}.$$

- The third summand: We have

$$(L-1)\mathrm{tr}(\boldsymbol{V}_{0,1}^\top \boldsymbol{Z}_{\mathrm{in}}^\top \boldsymbol{Z}_{\mathrm{in}} \boldsymbol{V}_{0,3}) \le L \|\boldsymbol{Z}_{\mathrm{in}}\boldsymbol{V}_{0,1}\|_2 \|\boldsymbol{Z}_{\mathrm{in}}\boldsymbol{V}_{0,3}\|_2$$

where we used that $\boldsymbol{Z}_{\mathrm{in}}\boldsymbol{V}_{0,3}$ is 1-rank. By (E2.5)-(E2.7)

$$L \|\boldsymbol{V}_{0,1}^\top \boldsymbol{Z}_{\mathrm{in}}^\top\|_2 \|\boldsymbol{Z}_{\mathrm{in}}\boldsymbol{V}_{0,3}\|_2 \lesssim \frac{L}{\sqrt{V}L\sqrt{d}} \frac{1}{\sqrt{V}LN} \le \frac{1}{NVL\sqrt{d}}.$$

- The fourth summand: We have by (E2.5)-(E2.7)

$$\frac{1}{\sqrt{d}}\left\|\boldsymbol{V}_{0,1}^\top \boldsymbol{Z}_{\mathrm{in}}^\top \boldsymbol{Z}_{\mathrm{in}} \boldsymbol{V}_{0,1}\right\|_F = \frac{1}{\sqrt{d}}\left\|\boldsymbol{Z}_{\mathrm{in}}\boldsymbol{V}_{0,1}\boldsymbol{V}_{0,1}^\top \boldsymbol{Z}_{\mathrm{in}}^\top\right\|_F \le \frac{C}{VL^2 d}$$

- The fifth summand: We have by (E2.5)-(E2.7)

$$\left\|\boldsymbol{V}_{0,1}^\top \boldsymbol{Z}_{\mathrm{in}}^\top \boldsymbol{Z}_{\mathrm{in}} \boldsymbol{V}_{0,2}\right\|_F^2 \le \mathrm{tr}\left(\boldsymbol{V}_{0,1}^\top \boldsymbol{Z}_{\mathrm{in}}^\top \boldsymbol{Z}_{\mathrm{in}} \boldsymbol{V}_{0,2}\boldsymbol{V}_{0,2}^\top \boldsymbol{Z}_{\mathrm{in}}^\top \boldsymbol{Z}_{\mathrm{in}} \boldsymbol{V}_{0,1}\right)$$

$$\le \frac{1}{NLd}\mathrm{tr}(\boldsymbol{Z}_{\mathrm{in}}\boldsymbol{V}_{0,1}\boldsymbol{V}_{0,1}^\top \boldsymbol{Z}_{\mathrm{in}}^\top) \le \frac{V}{NLd}\frac{1}{V^2 L^2} = \frac{1}{NVL^3 d}.$$

Therefore,

$$\frac{1}{\sqrt{d}}\left\|\boldsymbol{V}_{0,1}^\top \boldsymbol{Z}_{\mathrm{in}}^\top \boldsymbol{Z}_{\mathrm{in}} \boldsymbol{V}_{0,2}\right\|_F \le \frac{1}{\sqrt{NV}L^{3/2}d}$$

- The sixth summand:

$$(L-1)^2\left\|\boldsymbol{V}_{0,1}^\top \boldsymbol{Z}_{\mathrm{in}}^\top \boldsymbol{Z}_{\mathrm{in}} \boldsymbol{V}_{0,3}\right\|_F^2 \le \frac{1}{V^2 N}\mathbb{1}_V \boldsymbol{Z}_{\mathrm{in}}^\top \boldsymbol{Z}_{\mathrm{in}}\boldsymbol{V}_{0,1}\boldsymbol{V}_{0,1}^\top \boldsymbol{Z}_{\mathrm{in}}^\top \boldsymbol{Z}_{\mathrm{in}}\mathbb{1}_V \le \frac{1}{V^2 NL^2 d}$$

Therefore,

$$\frac{L-1}{\sqrt{d}}\left\|\boldsymbol{V}_{0,1}^\top \boldsymbol{Z}_{\mathrm{in}}^\top \boldsymbol{Z}_{\mathrm{in}} \boldsymbol{V}_{0,3}\right\|_F \le \frac{C}{VL\sqrt{N}d}.$$

Therefore, we have

$$\varphi = \mu_{kl}\left(\frac{1 \pm o_V(1)}{VL^2} \pm \frac{\tilde{O}(1)}{\sqrt{NV}L^{3/2}d}\right).$$

## D.2 Concentration bound for $\mathbf{s}_2$

In this section, we will use $\bar{\beta} := \phi''(0)\phi(0)$. We have

$$\boldsymbol{e}_l^\top \boldsymbol{s}_2 = \frac{1}{N^2 L^2}\sum_{i,j=1}^N \beta_{ij}\boldsymbol{z}_k^\top \boldsymbol{Z}_{\mathrm{in}}\boldsymbol{X}_i^\top \boldsymbol{X}_i \boldsymbol{Z}_{\mathrm{in}}^\top \boldsymbol{Z}_{\mathrm{in}}\boldsymbol{X}_i^\top \mathbb{1}_L (\boldsymbol{x}_i - \tfrac{1}{V}\mathbb{1}_V)^\top \boldsymbol{Z}_{\mathrm{out}}^\top \boldsymbol{Z}_{\mathrm{out}}(\boldsymbol{x}_j - \tfrac{1}{V}\mathbb{1}_V)$$

$$- \frac{1}{N^2 L^3}\sum_{i,j=1}^N \beta_{ij}\boldsymbol{z}_k^\top \boldsymbol{Z}_{\mathrm{in}}\boldsymbol{X}_i^\top \mathbb{1}_L \mathbb{1}_L^\top \boldsymbol{X}_i \boldsymbol{Z}_{\mathrm{in}}^\top \boldsymbol{Z}_{\mathrm{in}}\boldsymbol{X}_i^\top \mathbb{1}_L (\boldsymbol{x}_i - \tfrac{1}{V}\mathbb{1}_V)^\top \boldsymbol{Z}_{\mathrm{out}}^\top \boldsymbol{Z}_{\mathrm{out}}(\boldsymbol{x}_j - \tfrac{1}{V}\mathbb{1}_V)$$

$$+ \frac{\mu_{kl}}{N^2 L^2}\sum_{i,j=1}^N \beta_{ij}\left(\boldsymbol{e}_1 - \tfrac{1}{L}\mathbb{1}_L\right)^\top \boldsymbol{X}_i \boldsymbol{Z}_{\mathrm{in}}^\top \boldsymbol{Z}_{\mathrm{in}}\boldsymbol{X}_i^\top \mathbb{1}_L (\boldsymbol{x}_i - \tfrac{1}{V}\mathbb{1}_V)^\top \boldsymbol{Z}_{\mathrm{out}}^\top \boldsymbol{Z}_{\mathrm{out}}(\boldsymbol{x}_j - \tfrac{1}{V}\mathbb{1}_V)$$

$$=: \kappa + \text{negligible terma}.$$

### D.2.1 CONCENTRATION FOR $\kappa$

We will write $\kappa$ as follows:

$$\kappa = \frac{1}{N^2 L^2} \sum_{i,j=1}^{N} (\beta_{ij} - \bar{\beta}) \mathbf{z}_k^\top \boldsymbol{Z}_{\text{in}} \boldsymbol{X}_i^\top \boldsymbol{X}_i \boldsymbol{Z}_{\text{in}}^\top \boldsymbol{Z}_{\text{in}} \boldsymbol{X}_i^\top \mathbb{1}_L$$
$$\times (\boldsymbol{x}_i - \tfrac{1}{V}\mathbb{1}_V)^\top \boldsymbol{Z}_{\text{out}}^\top \boldsymbol{Z}_{\text{out}}(\boldsymbol{x}_j - \tfrac{1}{V}\mathbb{1}_V)$$
$$+ \frac{\bar{\beta}}{N^2 L^2} \sum_{i,j=1}^{N} \mathbf{z}_k^\top \boldsymbol{Z}_{\text{in}} \Big( \boldsymbol{x}_i \boldsymbol{x}_i^\top - \tfrac{1}{V}\boldsymbol{I}_V \Big) \boldsymbol{Z}_{\text{in}}^\top \boldsymbol{Z}_{\text{in}}(\boldsymbol{x}_i + \tfrac{L-1}{V}\mathbb{1}_V)$$
$$\times (\boldsymbol{x}_i - \tfrac{1}{V}\mathbb{1}_V)^\top \boldsymbol{Z}_{\text{out}}^\top \boldsymbol{Z}_{\text{out}}(\boldsymbol{x}_j - \tfrac{1}{V}\mathbb{1}_V)$$
$$+ \frac{\bar{\beta}}{N^2 L^2} \sum_{i,j=1}^{N} \mathbf{z}_k^\top \boldsymbol{Z}_{\text{in}} \Big( \boldsymbol{N}_i^\top \boldsymbol{N}_i - \tfrac{L-1}{V}\boldsymbol{I}_V \Big) \boldsymbol{Z}_{\text{in}}^\top \boldsymbol{Z}_{\text{in}}(\boldsymbol{x}_i + \tfrac{L-1}{V}\mathbb{1}_V)$$
$$\times (\boldsymbol{x}_i - \tfrac{1}{V}\mathbb{1}_V)^\top \boldsymbol{Z}_{\text{out}}^\top \boldsymbol{Z}_{\text{out}}(\boldsymbol{x}_j - \tfrac{1}{V}\mathbb{1}_V)$$
$$+ \frac{\bar{\beta}}{N^2 L^2} \sum_{i,j=1}^{N} \mathbf{z}_k^\top \boldsymbol{Z}_{\text{in}} \Big( \boldsymbol{x}_i \boldsymbol{x}_i^\top - \tfrac{1}{V}\boldsymbol{I}_V \Big) \boldsymbol{Z}_{\text{in}}^\top \boldsymbol{Z}_{\text{in}}\big( \boldsymbol{N}_i^\top - \tfrac{1}{V}\mathbb{1}_V \mathbb{1}_{L-1}^\top \big)\mathbb{1}_{L-1}$$
$$\times (\boldsymbol{x}_i - \tfrac{1}{V}\mathbb{1}_V)^\top \boldsymbol{Z}_{\text{out}}^\top \boldsymbol{Z}_{\text{out}}(\boldsymbol{x}_j - \tfrac{1}{V}\mathbb{1}_V)$$
$$+ \frac{\bar{\beta}}{N^2 L^2} \sum_{i,j=1}^{N} \mathbf{z}_k^\top \boldsymbol{Z}_{\text{in}} \Big( \boldsymbol{N}_i \boldsymbol{N}_i^\top - \tfrac{L-1}{V}\boldsymbol{I}_V \Big) \boldsymbol{Z}_{\text{in}}^\top \boldsymbol{Z}_{\text{in}}\big( \boldsymbol{N}_i^\top - \tfrac{1}{V}\mathbb{1}_V \mathbb{1}_{L-1}^\top \big)$$
$$\times (\boldsymbol{x}_i - \tfrac{1}{V}\mathbb{1}_V)^\top \boldsymbol{Z}_{\text{out}}^\top \boldsymbol{Z}_{\text{out}}(\boldsymbol{x}_j - \tfrac{1}{V}\mathbb{1}_V)$$
$$+ \frac{\bar{\beta}}{N^2 L V} \sum_{i,j=1}^{N} \mathbf{z}_k^\top \boldsymbol{Z}_{\text{in}} \boldsymbol{Z}_{\text{in}}^\top \boldsymbol{Z}_{\text{in}} \boldsymbol{X}_i^\top \mathbb{1}_L (\boldsymbol{x}_i - \tfrac{1}{V}\mathbb{1}_V)^\top \boldsymbol{Z}_{\text{out}}^\top \boldsymbol{Z}_{\text{out}}(\boldsymbol{x}_j - \tfrac{1}{V}\mathbb{1}_V)$$
$$=: \kappa_1 + \kappa_2 + \kappa_3 + \kappa_4 + \kappa_5 + \kappa_6.$$

By Proposition 5, we have

$$\kappa_1 = \frac{1}{N^2 L^2} \sum_{i,j=1}^{N} (\beta_{ij} - \bar{\beta})(\mathbb{1}_{\boldsymbol{x}_i = \boldsymbol{x}_j} - \tfrac{1}{V}) \mathbf{z}_k^\top \boldsymbol{Z}_{\text{in}} \boldsymbol{X}_i^\top \boldsymbol{X}_i \boldsymbol{Z}_{\text{in}}^\top \boldsymbol{Z}_{\text{in}} \boldsymbol{X}_i^\top \mathbb{1}_L$$
$$\pm \frac{\log^2 V}{N^2 L^2 \sqrt{d}} \Big\| \sum_{i,j=1}^{N} (\beta_{ij} - \bar{\beta})(\boldsymbol{x}_j - \tfrac{1}{V}\mathbb{1}_V) \mathbf{z}_k^\top \boldsymbol{Z}_{\text{in}} \boldsymbol{X}_i^\top \boldsymbol{X}_i \boldsymbol{Z}_{\text{in}}^\top \boldsymbol{Z}_{\text{in}} \boldsymbol{X}_i^\top \mathbb{1}_L (\boldsymbol{x}_i - \tfrac{1}{V}\mathbb{1}_V)^\top \Big\|_F$$
$$=: \kappa_{11} + \kappa_{12}.$$

We have

$$|\kappa_{11}| \le \frac{1}{N^2 L^2} \Big( \sum_{i=1}^{N} \Big( \sum_{j=1}^{N} (\beta_{ij} - \bar{\beta})(\mathbb{1}_{\boldsymbol{x}_i = \boldsymbol{x}_j} - \tfrac{1}{V}) \Big)^2 \Big)^{\frac{1}{2}} \Big( \sum_{i=1}^{N} (\mathbf{z}_k^\top \boldsymbol{Z}_{\text{in}} \boldsymbol{X}_i^\top \boldsymbol{X}_i \boldsymbol{Z}_{\text{in}}^\top \boldsymbol{Z}_{\text{in}} \boldsymbol{X}_i^\top \mathbb{1}_L)^2 \Big)^{\frac{1}{2}}$$
$$= \frac{1}{N^2 L V} \Big( \sum_{i=1}^{N} \Big( \sum_{j=1}^{N} (\beta_{ij} - \bar{\beta})(\mathbb{1}_{\boldsymbol{x}_i = \boldsymbol{x}_j} - \tfrac{1}{V}) \Big)^2 \Big)^{\frac{1}{2}} \Big( \sum_{i=1}^{N} (\mathbf{z}_k^\top \boldsymbol{Z}_{\text{in}} \boldsymbol{X}_i^\top \boldsymbol{X}_i \boldsymbol{Z}_{\text{in}}^\top \boldsymbol{Z}_{\text{in}} \mathbb{1}_V)^2 \Big)^{\frac{1}{2}}$$
$$+ \frac{1}{N^2 L^2} \Big( \sum_{i=1}^{N} \Big( \sum_{j=1}^{N} (\beta_{ij} - \bar{\beta})(\mathbb{1}_{\boldsymbol{x}_i = \boldsymbol{x}_j} - \tfrac{1}{V}) \Big)^2 \Big)^{\frac{1}{2}}$$
$$\times \Big( \sum_{i=1}^{N} (\mathbf{z}_k^\top \boldsymbol{Z}_{\text{in}}(\boldsymbol{X}_i^\top \boldsymbol{X}_i - \tfrac{L}{V}\boldsymbol{I}_V) \boldsymbol{Z}_{\text{in}}^\top \boldsymbol{Z}_{\text{in}}(\boldsymbol{X}_i^\top - \tfrac{1}{V}\mathbb{1}_V \mathbb{1}_L^\top)\mathbb{1}_L)^2 \Big)^{\frac{1}{2}}$$
$$+ \frac{1}{N^2 L V} \Big( \sum_{i=1}^{N} \Big( \sum_{j=1}^{N} (\beta_{ij} - \bar{\beta})(\mathbb{1}_{\boldsymbol{x}_i = \boldsymbol{x}_j} - \tfrac{1}{V}) \Big)^2 \Big)^{\frac{1}{2}}$$

$$\times \Big( \sum_{i=1}^{N} (\mathrm{z}_k^\top \boldsymbol{Z}_{\mathrm{in}} \boldsymbol{Z}_{\mathrm{in}}^\top \boldsymbol{Z}_{\mathrm{in}} (\boldsymbol{X}_i^\top - \tfrac{1}{V} \mathbb{1}_V \mathbb{1}_L^\top) \mathbb{1}_L)^2 \Big)^{\frac{1}{2}}$$

$$\lesssim \frac{1}{N^{3/2} L V} \Big( \frac{\sqrt{N}}{L} + \frac{N^{3/2}}{V} \frac{1}{V \wedge L^2 \wedge L\sqrt{d}} \Big) \Big( \frac{L}{d} \frac{\sqrt{V}}{\sqrt{L \wedge d}} + \frac{V\sqrt{L}}{d^{3/2}} \Big)$$

$$+ \frac{1}{N^{3/2} L^2} \Big( \frac{\sqrt{N}}{L} + \frac{N^{3/2}}{V} \frac{1}{V \wedge L^2 \wedge L\sqrt{d}} \Big) \frac{L}{\sqrt{d}\sqrt{L \wedge d}}$$

$$\lesssim \frac{1}{V L^2 \sqrt{d}(L \wedge d)^{\frac{1}{2}}} + \frac{1}{N L^{3/2} d \sqrt{L \wedge d}} + \frac{1}{V L^{1/2} d \sqrt{L \wedge d}} \frac{1}{V \wedge L^2 \wedge L\sqrt{d}},$$

where we use (P4)- (P5), and (E1.3)-(E1.5). Moreover,

$$|\kappa_{12}| \lesssim \frac{1}{N^{3/2} L^2 \sqrt{d}} \Big\| \sum_{i=1}^{N} (\boldsymbol{x}_i - \tfrac{1}{V} \mathbb{1}_V)(\boldsymbol{x}_i - \tfrac{1}{V} \mathbb{1}_V)^\top \Big\|_2$$

$$\times \Big( \frac{1}{N} \sum_{i=1}^{N} \sum_{j=1}^{N} (\beta_{ij} - \phi'(0)^2)^2 |\mathrm{z}_k^\top \boldsymbol{Z}_{\mathrm{in}} \boldsymbol{X}_i^\top \boldsymbol{X}_i \boldsymbol{Z}_{\mathrm{in}}^\top \boldsymbol{Z}_{\mathrm{in}} \boldsymbol{X}_i^\top \mathbb{1}_L|^2 \Big)^{\frac{1}{2}}$$

$$\lesssim \frac{1}{N^{3/2} L^2 \sqrt{d}} \frac{N}{V} \Big( \frac{\sqrt{N}}{V \wedge L^2 \wedge L\sqrt{d}} + \frac{1}{L} \Big) \Big( \frac{\sqrt{L}}{\sqrt{d}} + \frac{L^{3/2}}{d^{3/2}} \Big)$$

$$\lesssim \frac{1}{V \sqrt{L} d(L \wedge d)} \frac{1}{V \wedge L^2 \wedge L\sqrt{d}} + \frac{1}{\sqrt{N} V L^{3/2} d(L \wedge d)}.$$

Therefore,

$$|\kappa_1| \lesssim \frac{1}{V L^2 \sqrt{d}(L \wedge d)^{\frac{1}{2}}} + \frac{1}{N L^{3/2} d \sqrt{L \wedge d}} + \frac{1}{V L^{1/2} d \sqrt{L \wedge d}} \frac{1}{V \wedge L^2 \wedge L\sqrt{d}}.$$

By Proposition 5,

$$\kappa_2 = \frac{(1 - \tfrac{1}{V})}{N^2 L^2} \Big( \sum_{j=1}^{N} \boldsymbol{x}_j - \tfrac{1}{V} \mathbb{1}_V \Big)^\top$$

$$\times \sum_{i=1}^{N} (\boldsymbol{x}_i - \tfrac{1}{V} \mathbb{1}_V) \mathrm{z}_k^\top \boldsymbol{Z}_{\mathrm{in}} \Big( \boldsymbol{x}_i \boldsymbol{x}_i^\top - \tfrac{1}{V} \boldsymbol{I}_V \Big) \boldsymbol{Z}_{\mathrm{in}}^\top \boldsymbol{Z}_{\mathrm{in}} \Big( \boldsymbol{x}_i + \tfrac{L-1}{V} \mathbb{1}_V \Big)$$

$$\pm \frac{1}{N^2 L^2 \sqrt{d}} \Big\| \sum_{j=1}^{N} (\boldsymbol{x}_j - \tfrac{1}{V} \mathbb{1}_V) \Big\|_2$$

$$\times \Big\| \sum_{i=1}^{N} (\boldsymbol{x}_i - \tfrac{1}{V} \mathbb{1}_V) \mathrm{z}_k^\top \boldsymbol{Z}_{\mathrm{in}} \Big( \boldsymbol{x}_i \boldsymbol{x}_i^\top - \tfrac{1}{V} \boldsymbol{I}_V \Big) \boldsymbol{Z}_{\mathrm{in}}^\top \boldsymbol{Z}_{\mathrm{in}} \Big( \boldsymbol{x}_i + \tfrac{L-1}{V} \mathbb{1}_V \Big) \Big\|_2.$$

Let $n_i := |\{j \leq N | \boldsymbol{x}_j = \boldsymbol{e}_i\}|$. We have

$$\frac{1}{N^2 L^2} \Big\| \sum_{j=1}^{N} (\boldsymbol{x}_j - \tfrac{1}{V} \mathbb{1}_V) \Big\|_2 \Big\| \sum_{i=1}^{N} (\boldsymbol{x}_i - \tfrac{1}{V} \mathbb{1}_V) \mathrm{z}_k^\top \boldsymbol{Z}_{\mathrm{in}} \Big( \boldsymbol{x}_i \boldsymbol{x}_i^\top - \tfrac{1}{V} \boldsymbol{I}_V \Big) \boldsymbol{Z}_{\mathrm{in}}^\top \boldsymbol{Z}_{\mathrm{in}} \boldsymbol{x}_i \Big\|_2$$

$$+ \frac{L-1}{L} \frac{1}{N^2 L V} \Big\| \sum_{j=1}^{N} (\boldsymbol{x}_j - \tfrac{1}{V} \mathbb{1}_V) \Big\|_2 \Big\| \sum_{i=1}^{N} (\boldsymbol{x}_i - \tfrac{1}{V} \mathbb{1}_V) \mathrm{z}_k^\top \boldsymbol{Z}_{\mathrm{in}} \big( \boldsymbol{x}_i \boldsymbol{x}_i^\top - \tfrac{1}{V} \boldsymbol{I}_V \big) \boldsymbol{Z}_{\mathrm{in}}^\top \boldsymbol{Z}_{\mathrm{in}} \mathbb{1}_V \Big\|_2$$

$$\lesssim \frac{1}{N L^2} \Big( \frac{1}{N} \sum_{i=1}^{V} n_i^2 |\mathrm{z}_k^\top \boldsymbol{Z}_{\mathrm{in}} \big( \boldsymbol{e}_i \boldsymbol{e}_i^\top - \tfrac{1}{V} \boldsymbol{I}_V \big) \boldsymbol{Z}_{\mathrm{in}}^\top \boldsymbol{Z}_{\mathrm{in}} \boldsymbol{e}_i|^2 \Big)^{\frac{1}{2}}$$

$$+ \frac{1}{N^2 L V} \Big\| \sum_{i=1}^{N} (\boldsymbol{x}_i - \tfrac{1}{V} \mathbb{1}_V) \mathrm{z}_k^\top \boldsymbol{Z}_{\mathrm{in}} \big( \boldsymbol{x}_i \boldsymbol{x}_i^\top - \tfrac{1}{V} \boldsymbol{I}_V \big) \boldsymbol{Z}_{\mathrm{in}}^\top \boldsymbol{Z}_{\mathrm{in}} \mathbb{1}_V \Big\|_2$$

$$\lesssim \frac{1}{V L^2 \sqrt{d}} + \frac{1}{N^2 L V} \Big\| \sum_{i=1}^{N} (\boldsymbol{x}_i - \tfrac{1}{V} \mathbb{1}_V) \mathrm{z}_k^\top \boldsymbol{Z}_{\mathrm{in}} \big( \boldsymbol{x}_i \boldsymbol{x}_i^\top - \tfrac{1}{V} \boldsymbol{I}_V \big) \boldsymbol{Z}_{\mathrm{in}}^\top \boldsymbol{Z}_{\mathrm{in}} \mathbb{1}_V \Big\|_2,$$

where we used (E1.1) and (E2.3). Moreover,

$$\mathbb{E}\left[\left\|\frac{1}{N}\sum_{i=1}^{N}(\boldsymbol{x}_i - \tfrac{1}{V}\mathbb{1}_V)\mathsf{z}_k^\top \boldsymbol{Z}_{\text{in}}\big(\boldsymbol{x}_i\boldsymbol{x}_i^\top - \tfrac{1}{V}\boldsymbol{I}_V\big)\boldsymbol{Z}_{\text{in}}^\top \boldsymbol{Z}_{\text{in}}\mathbb{1}_V\right\|_2^2 \Big|\mathsf{Z}_{\text{in}}\right]$$

$$\leq \frac{1}{N}\,\mathbb{E}\left[\mathsf{z}_k^\top \boldsymbol{Z}_{\text{in}}\big(\boldsymbol{x}_i\boldsymbol{x}_i^\top - \tfrac{1}{V}\boldsymbol{I}_V\big)\boldsymbol{Z}_{\text{in}}^\top \boldsymbol{Z}_{\text{in}}\mathbb{1}_V \mathbb{1}_V^\top \boldsymbol{Z}_{\text{in}}^\top \boldsymbol{Z}_{\text{in}}\big(\boldsymbol{x}_i\boldsymbol{x}_i^\top - \tfrac{1}{V}\boldsymbol{I}_V\big)\boldsymbol{Z}_{\text{in}}^\top \mathsf{z}_k|\mathsf{Z}_{\text{in}}\right]$$

$$+ \frac{1}{N^2}\sum_{\substack{i\neq j=1}}^{N}\mathbb{E}\left[(\mathbb{1}_{\boldsymbol{x}_i=\boldsymbol{x}_j} - \tfrac{1}{V})\mathsf{z}_k^\top \boldsymbol{Z}_{\text{in}}\big(\boldsymbol{x}_i\boldsymbol{x}_i^\top - \tfrac{1}{V}\boldsymbol{I}_V\big)\boldsymbol{Z}_{\text{in}}^\top \boldsymbol{Z}_{\text{in}}\mathbb{1}_V \mathbb{1}_V^\top \boldsymbol{Z}_{\text{in}}^\top \boldsymbol{Z}_{\text{in}}\big(\boldsymbol{x}_j\boldsymbol{x}_j^\top - \tfrac{1}{V}\boldsymbol{I}_V\big)\boldsymbol{Z}_{\text{in}}^\top \mathsf{z}_k|\mathsf{Z}_{\text{in}}\right]$$

$$\leq \Big(\frac{1}{N} + \frac{N-1}{NV}\Big)\mathbb{E}\left[\mathsf{z}_k^\top \boldsymbol{Z}_{\text{in}}\big(\boldsymbol{x}_i\boldsymbol{x}_i^\top - \tfrac{1}{V}\boldsymbol{I}_V\big)\boldsymbol{Z}_{\text{in}}^\top \boldsymbol{Z}_{\text{in}}\mathbb{1}_V \mathbb{1}_V^\top \boldsymbol{Z}_{\text{in}}^\top \boldsymbol{Z}_{\text{in}}\big(\boldsymbol{x}_j\boldsymbol{x}_j^\top - \tfrac{1}{V}\boldsymbol{I}_V\big)\boldsymbol{Z}_{\text{in}}^\top \mathsf{z}_k|\mathsf{Z}_{\text{in}}\right]$$

$$\lesssim \Big(\frac{1}{N} + \frac{N-1}{NV}\Big)\frac{V}{d^2},$$

where we (E1.1)-(E1.3). Therefore, by Chebyshev's inequality, we have

$$|\kappa_2| \lesssim \frac{1}{VL^2\sqrt{d}} + \frac{1}{NLVd}$$

Moreover, by using Chebyshev's inequality

$$\kappa_3 = \frac{1}{N^2 L^2}\Big(\sum_{j=1}^{N}(\boldsymbol{x}_j - \tfrac{1}{V}\mathbb{1}_V)\Big)^\top$$

$$\times \Big(\sum_{i=1}^{N}(\boldsymbol{x}_i - \tfrac{1}{V}\mathbb{1}_V)\mathsf{z}_k^\top \boldsymbol{Z}_{\text{in}}\Big(\boldsymbol{N}_i^\top \boldsymbol{N}_i - \tfrac{L-1}{V}\boldsymbol{I}_V\Big)\boldsymbol{Z}_{\text{in}}^\top \boldsymbol{Z}_{\text{in}}\big(\boldsymbol{x}_i + \tfrac{L-1}{V}\mathbb{1}_V\big)\Big)$$

$$\pm \frac{1}{N^2 L^2\sqrt{d}}\Big\|\sum_{j=1}^{N}(\boldsymbol{x}_j - \tfrac{1}{V}\mathbb{1}_V)\Big\|_2$$

$$\times \Big\|\sum_{i=1}^{N}(\boldsymbol{x}_i - \tfrac{1}{V}\mathbb{1}_V)\mathsf{z}_k^\top \boldsymbol{Z}_{\text{in}}\Big(\boldsymbol{N}_i^\top \boldsymbol{N}_i - \tfrac{L-1}{V}\boldsymbol{I}_V\Big)\boldsymbol{Z}_{\text{in}}^\top \boldsymbol{Z}_{\text{in}}\big(\boldsymbol{x}_i + \tfrac{L-1}{V}\mathbb{1}_V\big)\Big\|_2.$$

We have

$$\mathbb{E}\left[\Big(\sum_{i,j=1}^{N}(\mathbb{1}_{\boldsymbol{x}_i=\boldsymbol{x}_j} - \tfrac{1}{V})\mathsf{z}_k^\top \boldsymbol{Z}_{\text{in}}\Big(\boldsymbol{N}_i^\top \boldsymbol{N}_i - \tfrac{L-1}{V}\boldsymbol{I}_V\Big)\boldsymbol{Z}_{\text{in}}^\top \boldsymbol{Z}_{\text{in}}\big(\boldsymbol{x}_i + \tfrac{L-1}{V}\mathbb{1}_V\big)\Big)^2\Big|\mathsf{Z}_{\text{in}}\right]$$

$$= \sum_{i=1}^{N}\mathbb{E}\left[\Big(\sum_{j=1}^{N}(\mathbb{1}_{\boldsymbol{x}_i=\boldsymbol{x}_j} - \tfrac{1}{V})\mathsf{z}_k^\top \boldsymbol{Z}_{\text{in}}\Big(\boldsymbol{N}_i^\top \boldsymbol{N}_i - \tfrac{L-1}{V}\boldsymbol{I}_V\Big)\boldsymbol{Z}_{\text{in}}^\top \boldsymbol{Z}_{\text{in}}\big(\boldsymbol{x}_i + \tfrac{L-1}{V}\mathbb{1}_V\big)\Big)^2\Big|\mathsf{Z}_{\text{in}}\right]$$

$$\lesssim \frac{N^2}{V}\mathbb{E}\left[\Big(\mathsf{z}_k^\top \boldsymbol{Z}_{\text{in}}\Big(\boldsymbol{N}_1^\top \boldsymbol{N}_1 - \tfrac{L-1}{V}\boldsymbol{I}_V\Big)\boldsymbol{Z}_{\text{in}}^\top \boldsymbol{Z}_{\text{in}}\big(\boldsymbol{x}_1 + \tfrac{L-1}{V}\mathbb{1}_V\big)\Big)^2\Big|\mathsf{Z}_{\text{in}}\right] \qquad \text{(D.23)}$$

We have

$$\mathbb{E}\left[\Big(\mathsf{z}_k^\top \boldsymbol{Z}_{\text{in}}\Big(\boldsymbol{N}_1^\top \boldsymbol{N}_1 - \tfrac{L-1}{V}\boldsymbol{I}_V\Big)\boldsymbol{Z}_{\text{in}}^\top \boldsymbol{Z}_{\text{in}}\big(\boldsymbol{x}_1 + \tfrac{L-1}{V}\mathbb{1}_V\big)\Big)^2\Big|\mathsf{Z}_{\text{in}}\right] \leq \frac{CL}{d^2}\Big(1 + \frac{L^2}{V}\Big).$$

where we used (E1.1)-(E1.3) and (E1.8). Therefore, we have (D.23) $\lesssim \frac{N^2 L}{V d^2}\big(1 + \frac{L^2}{V}\big)$ . Also,

$$\mathbb{E}\left[\Big\|\sum_{i=1}^{N}(\boldsymbol{x}_i - \tfrac{1}{V}\mathbb{1}_V)\mathsf{z}_k^\top \boldsymbol{Z}_{\text{in}}\Big(\boldsymbol{N}_i^\top \boldsymbol{N}_i - \tfrac{L-1}{V}\boldsymbol{I}_V\Big)\boldsymbol{Z}_{\text{in}}^\top \boldsymbol{Z}_{\text{in}}\big(\boldsymbol{x}_i + \tfrac{L-1}{V}\mathbb{1}_V\big)\Big\|_2^2\Big|\mathsf{Z}_{\text{in}}\right]$$

$$\leq \frac{CL}{d^2}\Big(1 + \frac{L^2}{V}\Big).$$

Therefore, we have

$$|\kappa_3| \lesssim \Big(\frac{1}{N\sqrt{V}L^{3/2}d} + \frac{1}{NL^{3/2}d^{3/2}} + \frac{1}{N\sqrt{V}\sqrt{L}d^{3/2}}\Big).$$

Moreover, by Chebyshev's inequality

$$\kappa_4 = \frac{1}{N^2 L^2} \Big( \sum_{j=1}^{N} (\boldsymbol{x}_j - \tfrac{1}{V} \mathbb{1}_V) \Big)^\top$$

$$\times \Big( \sum_{i=1}^{N} (\boldsymbol{x}_i - \tfrac{1}{V} \mathbb{1}_V) \mathbf{z}_k^\top \boldsymbol{Z}_{\mathrm{in}} \Big( \boldsymbol{x}_i \boldsymbol{x}_i^\top - \tfrac{1}{V} \boldsymbol{I}_V \Big) \boldsymbol{Z}_{\mathrm{in}}^\top \boldsymbol{Z}_{\mathrm{in}} \big( \boldsymbol{N}_i^\top - \tfrac{1}{V} \mathbb{1}_V \mathbb{1}_{L-1}^\top \big) \mathbb{1}_{L-1} \Big)$$

$$\pm \frac{1}{N^2 L^2 \sqrt{d}} \Big\| \sum_{j=1}^{N} (\boldsymbol{x}_j - \tfrac{1}{V} \mathbb{1}_V) \Big\|_2$$

$$\times \Big\| \sum_{i=1}^{N} (\boldsymbol{x}_i - \tfrac{1}{V} \mathbb{1}_V) \mathbf{z}_k^\top \boldsymbol{Z}_{\mathrm{in}} \Big( \boldsymbol{x}_i \boldsymbol{x}_i^\top - \tfrac{1}{V} \boldsymbol{I}_V \Big) \boldsymbol{Z}_{\mathrm{in}}^\top \boldsymbol{Z}_{\mathrm{in}} \big( \boldsymbol{N}_i^\top - \tfrac{1}{V} \mathbb{1}_V \mathbb{1}_{L-1}^\top \big) \mathbb{1}_{L-1} \Big\|_2.$$

We have

$$\mathbb{E} \left[ \Big( \sum_{i,j=1}^{N} (\mathbb{1}_{\boldsymbol{x}_i = \boldsymbol{x}_j} - \tfrac{1}{V}) \mathbf{z}_{\nu,\delta}^\top \boldsymbol{Z}_{\mathrm{in}} \Big( \boldsymbol{x}_i^\top \boldsymbol{x}_i - \tfrac{1}{V} \boldsymbol{I}_V \Big) \boldsymbol{Z}_{\mathrm{in}}^\top \boldsymbol{Z}_{\mathrm{in}} \big( \boldsymbol{N}_i^\top - \tfrac{1}{V} \mathbb{1}_V \mathbb{1}_{L-1}^\top \big) \mathbb{1}_{L-1} \Big)^2 \Big| \mathsf{Z}_{\mathrm{in}} \right]$$

$$= \sum_{i=1}^{N} \mathbb{E} \left[ \Big( \sum_{j=1}^{N} (\mathbb{1}_{\boldsymbol{x}_i = \boldsymbol{x}_j} - \tfrac{1}{V}) \mathbf{z}_k^\top \boldsymbol{Z}_{\mathrm{in}} \Big( \boldsymbol{x}_i \boldsymbol{x}_i^\top - \tfrac{1}{V} \boldsymbol{I}_V \Big) \boldsymbol{Z}_{\mathrm{in}}^\top \boldsymbol{Z}_{\mathrm{in}} \big( \boldsymbol{N}_i^\top - \tfrac{1}{V} \mathbb{1}_V \mathbb{1}_{L-1}^\top \big) \mathbb{1}_{L-1} \Big)^2 \Big| \mathsf{Z}_{\mathrm{in}} \right]$$

$$\lesssim \frac{N^2}{V} \mathbb{E} \left[ \Big( \mathbf{z}_k^\top \boldsymbol{Z}_{\mathrm{in}} \Big( \boldsymbol{x}_i \boldsymbol{x}_i^\top - \tfrac{1}{V} \boldsymbol{I}_V \Big) \boldsymbol{Z}_{\mathrm{in}}^\top \boldsymbol{Z}_{\mathrm{in}} \big( \boldsymbol{N}_i^\top - \tfrac{1}{V} \mathbb{1}_V \mathbb{1}_{L-1}^\top \big) \mathbb{1}_{L-1} \Big)^2 \Big| \mathsf{Z}_{\mathrm{in}} \right] \qquad \text{(D.24)}$$

We have

$$\mathbb{E} \left[ \Big( \mathbf{z}_k^\top \boldsymbol{Z}_{\mathrm{in}} \Big( \boldsymbol{x}_i \boldsymbol{x}_i^\top - \tfrac{1}{V} \boldsymbol{I}_V \Big) \boldsymbol{Z}_{\mathrm{in}}^\top \boldsymbol{Z}_{\mathrm{in}} \big( \boldsymbol{N}_i^\top - \tfrac{1}{V} \mathbb{1}_V \mathbb{1}_{L-1}^\top \big) \mathbb{1}_{L-1} \Big)^2 \Big| \mathsf{Z}_{\mathrm{in}} \right]$$

$$\leq \frac{CL}{d} \mathbf{z}_k^\top \boldsymbol{Z}_{\mathrm{in}} \mathbb{E} \left[ \Big( \boldsymbol{x}_i \boldsymbol{x}_i^\top - \tfrac{1}{V} \boldsymbol{I}_V \Big) \boldsymbol{Z}_{\mathrm{in}}^\top \boldsymbol{Z}_{\mathrm{in}} \Big( \boldsymbol{x}_i \boldsymbol{x}_i^\top - \tfrac{1}{V} \boldsymbol{I}_V \Big) \Big| \mathsf{Z}_{\mathrm{in}} \right] \boldsymbol{Z}_{\mathrm{in}}^\top \mathbf{z}_k \leq \frac{CL}{d^2}.$$

where we used Proposition 8. Therefore, (D.24) $\lesssim \frac{N^2 L}{V d^2}$. Also,

$$\mathbb{E} \left[ \Big\| \sum_{i=1}^{N} (\boldsymbol{x}_i - \tfrac{1}{V} \mathbb{1}_V) \mathbf{z}_k^\top \boldsymbol{Z}_{\mathrm{in}} \Big( \boldsymbol{x}_i \boldsymbol{x}_i^\top - \tfrac{L-1}{V} \boldsymbol{I}_V \Big) \boldsymbol{Z}_{\mathrm{in}}^\top \boldsymbol{Z}_{\mathrm{in}} \big( \boldsymbol{N}_i^\top - \tfrac{1}{V} \mathbb{1}_V \mathbb{1}_{L-1}^\top \big) \mathbb{1}_{L-1} \Big\|_2^2 \Big| \mathsf{Z}_{\mathrm{in}} \right]$$

$$= \frac{CL}{d} \sum_{i=1}^{N} \mathbf{z}_k^\top \boldsymbol{Z}_{\mathrm{in}} \mathbb{E} \left[ \Big( \boldsymbol{x}_i \boldsymbol{x}_i^\top - \tfrac{1}{V} \boldsymbol{I}_V \Big) \boldsymbol{Z}_{\mathrm{in}}^\top \boldsymbol{Z}_{\mathrm{in}} \Big( \boldsymbol{x}_i \boldsymbol{x}_i^\top - \tfrac{1}{V} \boldsymbol{I}_V \Big) \Big| \mathsf{Z}_{\mathrm{in}} \right] \boldsymbol{Z}_{\mathrm{in}}^\top \mathbf{z}_k \leq \frac{CNL}{d^2},$$

where we used Proposition 8. Therefore,

$$|\kappa_4| \lesssim \Big( \frac{1}{N \sqrt{V} L^{3/2} d} + \frac{1}{N L^{3/2} d^{3/2}} \Big)$$

Moreover, let

$$\gamma_i := \mathbf{z}_k^\top \boldsymbol{Z}_{\mathrm{in}} \Big( \boldsymbol{N}_i^\top \boldsymbol{N}_i - \tfrac{L-1}{V} \boldsymbol{I}_V \Big) \boldsymbol{Z}_{\mathrm{in}}^\top \boldsymbol{Z}_{\mathrm{in}} \big( \boldsymbol{N}_i^\top - \tfrac{1}{V} \mathbb{1}_V \mathbb{1}_{L-1}^\top \big) \mathbb{1}_{L-1}$$

$$- \mathbb{E} \left[ \mathbf{z}_k^\top \boldsymbol{Z}_{\mathrm{in}} \Big( \boldsymbol{N}_i^\top \boldsymbol{N}_i - \tfrac{L-1}{V} \boldsymbol{I}_V \Big) \boldsymbol{Z}_{\mathrm{in}}^\top \boldsymbol{Z}_{\mathrm{in}} \big( \boldsymbol{N}_i^\top - \tfrac{1}{V} \mathbb{1}_V \mathbb{1}_{L-1}^\top \big) \mathbb{1}_{L-1} \big| \mathsf{Z}_{\mathrm{in}} \right].$$

By Proposition 5, we have

$$\kappa_5 = \frac{1}{N^2 L^2} \Big( \sum_{j=1}^{N} (\boldsymbol{x}_j - \tfrac{1}{V} \mathbb{1}_V) \Big)^\top \Big( \sum_{i=1}^{N} (\boldsymbol{x}_i - \tfrac{1}{V} \mathbb{1}_V) \gamma_i \Big)$$

$$\pm \frac{\log^2 V}{N^2 L^2 \sqrt{d}} \Big\| \sum_{j=1}^{N} (\boldsymbol{x}_j - \tfrac{1}{V} \mathbb{1}_V) \Big\|_2 \Big\| \sum_{i=1}^{N} (\boldsymbol{x}_i - \tfrac{1}{V} \mathbb{1}_V) \gamma_i \Big\|_2$$

$$+ \frac{1}{N^2 L^2} \Big( \sum_{j=1}^{N} (\boldsymbol{x}_j - \tfrac{1}{V} \mathbb{1}_V) \Big)^\top$$

$$\times \Big( \sum_{i=1}^{N} (\boldsymbol{x}_i - \tfrac{1}{V} \mathbb{1}_V) \mathsf{z}_k^\top \boldsymbol{Z}_{\mathrm{in}} \mathbb{E}\Big[ \Big( \boldsymbol{N}_i^\top \boldsymbol{N}_i - \tfrac{L-1}{V} \boldsymbol{I}_V \Big) \boldsymbol{Z}_{\mathrm{in}}^\top \boldsymbol{Z}_{\mathrm{in}} \big( \boldsymbol{N}_i^\top - \tfrac{1}{V} \mathbb{1}_V \mathbb{1}_{L-1}^\top \big) \mathbb{1}_{L-1} | \mathsf{Z}_{\mathrm{in}} \Big] \Big)$$

$$\pm \frac{\log^2 V}{N^2 L^2 \sqrt{d}} \Big\| \sum_{j=1}^{N} (\boldsymbol{x}_j - \tfrac{1}{V} \mathbb{1}_V) \Big\|_2$$

$$\times \Big\| \sum_{i=1}^{N} (\boldsymbol{x}_i - \tfrac{1}{V} \mathbb{1}_V) \mathsf{z}_k^\top \boldsymbol{Z}_{\mathrm{in}} \mathbb{E}\Big[ \Big( \boldsymbol{N}_i^\top \boldsymbol{N}_i - \tfrac{L-1}{V} \boldsymbol{I}_V \Big) \boldsymbol{Z}_{\mathrm{in}}^\top \boldsymbol{Z}_{\mathrm{in}} \big( \boldsymbol{N}_i^\top - \tfrac{1}{V} \mathbb{1}_V \mathbb{1}_{L-1}^\top \big) \mathbb{1}_{L-1} | \mathsf{Z}_{\mathrm{in}} \Big] \Big\|_2.$$

By Proposition 2

$$\mathbb{E}\Big[ \Big( \sum_{i,j=1}^{N} (\mathbb{1}_{\boldsymbol{x}_i = \boldsymbol{x}_j} - \tfrac{1}{V}) \gamma_i \Big)^2 \Big] = \sum_{i=1}^{N} \mathbb{E}\Big[ \Big( \sum_{j=1}^{N} (\mathbb{1}_{\boldsymbol{x}_i = \boldsymbol{x}_j} - \tfrac{1}{V}) \gamma_i \Big)^2 \Big]$$

$$\le 2(1 - \tfrac{1}{V})^2 \sum_{i=1}^{N} \mathbb{E}[\gamma_i^2] + \frac{2(1 - \tfrac{1}{V})}{V} \sum_{i=1}^{N} \sum_{j \ne i}^{N} \mathbb{E}[\gamma_i^2]$$

$$\lesssim \frac{N^2}{V} \Big( \frac{L}{d} + \frac{L^2}{d^2} \Big).$$

Then,

$$\mathbb{E}\Big[ \Big\| \sum_{i=1}^{N} (\boldsymbol{x}_i - \frac{1}{V} \mathbb{1}_V) \gamma_i \Big\|_2^2 \Big] \le \sum_{i=1}^{N} \mathbb{E}[\gamma_i^2] \lesssim N \Big( \frac{L}{d} + \frac{L^2}{d^2} \Big).$$

Moreover, by Proposition 8, (E2.3) and (E2.7), we have

$$\Big\| \sum_{i=1}^{N} (\boldsymbol{x}_i - \tfrac{1}{V} \mathbb{1}_V) \mathsf{z}_k^\top \boldsymbol{Z}_{\mathrm{in}} \mathbb{E}\Big[ \Big( \boldsymbol{N}_i^\top \boldsymbol{N}_i - \tfrac{L-1}{V} \boldsymbol{I}_V \Big) \boldsymbol{Z}_{\mathrm{in}}^\top \boldsymbol{Z}_{\mathrm{in}} \big( \boldsymbol{N}_i^\top - \tfrac{1}{V} \mathbb{1}_V \mathbb{1}_{L-1}^\top \big) \mathbb{1}_{L-1} | \mathsf{Z}_{\mathrm{in}} \Big] \Big\|_2$$

$$\lesssim \frac{L\sqrt{N}}{\sqrt{Vd}}.$$

Therefore, by Chebyshev's inequality, we have

$$|\kappa_5| \lesssim \Big( \frac{1}{NL\sqrt{Vd}(L \wedge d)^{1/2}} + \frac{1}{NLd(L \wedge d)^{1/2}} + \frac{1}{NL\sqrt{Vd}} \Big)$$

Lastly, by Proposition 5,

$$\kappa_6 = \frac{\bar{\beta}}{N^2 LV} \sum_{i,j=1}^{N} (\mathbb{1}_{\boldsymbol{x}_i = \boldsymbol{x}_j} - \tfrac{1}{V}) \mathsf{z}_k^\top \boldsymbol{Z}_{\mathrm{in}} \boldsymbol{Z}_{\mathrm{in}}^\top \boldsymbol{Z}_{\mathrm{in}} \boldsymbol{X}_i^\top \mathbb{1}_L$$

$$\pm \frac{\bar{\beta}}{N^2 LV \sqrt{d}} \Big\| \sum_{i,j=1}^{N} (\boldsymbol{x}_j - \tfrac{1}{V} \mathbb{1}_V) \mathsf{z}_k^\top \boldsymbol{Z}_{\mathrm{in}} \boldsymbol{Z}_{\mathrm{in}}^\top \boldsymbol{Z}_{\mathrm{in}} \boldsymbol{X}_i^\top \mathbb{1}_L (\boldsymbol{x}_i - \tfrac{1}{V} \mathbb{1}_V)^\top \Big\|_F$$

$$= \frac{\bar{\beta}}{N^2 LV} \sum_{i,j=1}^{N} (\mathbb{1}_{\boldsymbol{x}_i = \boldsymbol{x}_j} - \tfrac{1}{V}) \mathsf{z}_k^\top \boldsymbol{Z}_{\mathrm{in}} \boldsymbol{Z}_{\mathrm{in}}^\top \boldsymbol{Z}_{\mathrm{in}} \boldsymbol{X}_i^\top \mathbb{1}_L$$

$$\pm \frac{\bar{\beta}}{V \sqrt{d}} \Big\| \frac{1}{N} \sum_{j=1}^{N} (\boldsymbol{x}_j - \tfrac{1}{V} \mathbb{1}_V) \Big\|_2 \Big\| \frac{1}{NL} \sum_{i=1}^{N} (\boldsymbol{x}_i - \tfrac{1}{V} \mathbb{1}_V) \mathsf{z}_k^\top \boldsymbol{Z}_{\mathrm{in}} \boldsymbol{Z}_{\mathrm{in}}^\top \boldsymbol{Z}_{\mathrm{in}} \boldsymbol{X}_i^\top \mathbb{1}_L \Big\|_2$$

We have

$$\left\| \frac{1}{NL} \sum_{i=1}^{N} \mathsf{z}_k^\top \boldsymbol{Z}_{\mathrm{in}} \boldsymbol{Z}_{\mathrm{in}}^\top \boldsymbol{Z}_{\mathrm{in}} \boldsymbol{X}_i^\top \mathbb{1}_L (\boldsymbol{x}_i - \tfrac{1}{V} \mathbb{1}_V)^\top \right\|_2$$

$$\leq \left\| \mathsf{z}_k^\top \boldsymbol{Z}_{\mathrm{in}} \boldsymbol{Z}_{\mathrm{in}}^\top \boldsymbol{Z}_{\mathrm{in}} \frac{1}{NL} \sum_{i=1}^{N} (\boldsymbol{X}_i^\top - \tfrac{1}{V} \mathbb{1}_V \mathbb{1}_L^\top) \mathbb{1}_L (\boldsymbol{x}_i - \tfrac{1}{V} \mathbb{1}_V)^\top \right\|_2$$

$$+ \frac{1}{V} |\mathsf{z}_k^\top \boldsymbol{Z}_{\mathrm{in}} \boldsymbol{Z}_{\mathrm{in}}^\top \boldsymbol{Z}_{\mathrm{in}} \mathbb{1}_V| \left\| \frac{1}{N} \sum_{i=1}^{N} (\boldsymbol{x}_i - \tfrac{1}{V} \mathbb{1}_V)^\top \right\|_2$$

$$\lesssim \frac{V}{d} \left\| \boldsymbol{Z}_{\mathrm{in}} \frac{1}{NL} \sum_{i=1}^{N} (\boldsymbol{X}_i^\top - \tfrac{1}{V} \mathbb{1}_V \mathbb{1}_L^\top) \mathbb{1}_L (\boldsymbol{x}_i - \tfrac{1}{V} \mathbb{1}_V)^\top \right\|_2 + \frac{\sqrt{V}}{d^{3/2}\sqrt{N}}$$

$$\lesssim \frac{CV}{\sqrt{NL}d^{3/2}}.$$

Moreover,

$$\mathbb{E}\left[ \left( \frac{1}{N^2 LV} \sum_{i,j=1}^{N} (\mathbb{1}_{\boldsymbol{x}_i = \boldsymbol{x}_j} - \tfrac{1}{V}) \mathsf{z}_k^\top \boldsymbol{Z}_{\mathrm{in}} \boldsymbol{Z}_{\mathrm{in}}^\top \boldsymbol{Z}_{\mathrm{in}} \boldsymbol{X}_i^\top \mathbb{1}_L \right)^2 | \mathsf{Z}_{\mathrm{in}} \right]$$

$$= \frac{1}{N^4 L^2 V^2} \sum_{j=1}^{N} \mathbb{E}\left[ \left( \sum_{i=1}^{N} (\mathbb{1}_{\boldsymbol{x}_i = \boldsymbol{x}_j} - \tfrac{1}{V}) \mathsf{z}_k^\top \boldsymbol{Z}_{\mathrm{in}} \boldsymbol{Z}_{\mathrm{in}}^\top \boldsymbol{Z}_{\mathrm{in}} \boldsymbol{X}_i^\top \mathbb{1}_L \right)^2 | \mathsf{Z}_{\mathrm{in}} \right]$$

$$\leq \frac{2}{N^4 L^2 V^2} \sum_{j=1}^{N} \mathbb{E}\left[ \left( \mathsf{z}_k^\top \boldsymbol{Z}_{\mathrm{in}} \boldsymbol{Z}_{\mathrm{in}}^\top \boldsymbol{Z}_{\mathrm{in}} \boldsymbol{X}_i^\top \mathbb{1}_L \right)^2 | \mathsf{Z}_{\mathrm{in}} \right]$$

$$+ \frac{2}{N^4 L^2 V^2} \sum_{j=1}^{N} \mathbb{E}\left[ \left( \sum_{\substack{i=1 \\ i \neq j}}^{N} (\mathbb{1}_{\boldsymbol{x}_i = \boldsymbol{x}_j} - \tfrac{1}{V}) \mathsf{z}_k^\top \boldsymbol{Z}_{\mathrm{in}} \boldsymbol{Z}_{\mathrm{in}}^\top \boldsymbol{Z}_{\mathrm{in}} \boldsymbol{X}_i^\top \mathbb{1}_L \right)^2 | \mathsf{Z}_{\mathrm{in}} \right]$$

$$\leq \frac{2}{N^4 L^2 V^2} \sum_{j=1}^{N} \mathbb{E}\left[ \left( \mathsf{z}_{\nu,\delta}^\top \boldsymbol{Z}_{\mathrm{in}} \boldsymbol{Z}_{\mathrm{in}}^\top \boldsymbol{Z}_{\mathrm{in}} \boldsymbol{X}_i^\top \mathbb{1}_L \right)^2 | \mathsf{Z}_{\mathrm{in}} \right]$$

$$+ \frac{2}{N^2 V^3} \sum_{j=1}^{N} \mathbb{E}\left[ \left\| \frac{1}{NL} \sum_{\substack{i=1 \\ i \neq j}}^{N} \mathsf{z}_k^\top \boldsymbol{Z}_{\mathrm{in}} \boldsymbol{Z}_{\mathrm{in}}^\top \boldsymbol{Z}_{\mathrm{in}} \boldsymbol{X}_i^\top \mathbb{1}_L (\boldsymbol{x}_i - \tfrac{1}{V} \mathbb{1}_V)^\top \right\|_2^2 | \mathsf{Z}_{\mathrm{in}} \right]$$

$$\leq \frac{2}{N^4 L^2 V^2} \sum_{j=1}^{N} \mathbb{E}\left[ \left( \mathsf{z}_k^\top \boldsymbol{Z}_{\mathrm{in}} \boldsymbol{Z}_{\mathrm{in}}^\top \boldsymbol{Z}_{\mathrm{in}} \boldsymbol{X}_i^\top \mathbb{1}_L \right)^2 | \mathsf{Z}_{\mathrm{in}} \right] + \frac{C}{N^2 V L d^3},$$

where we used D.2.1 in the last step. We have

$$\frac{1}{N^4 L^2 V^2} \sum_{j=1}^{N} \mathbb{E}\left[ \left( \mathsf{z}_k^\top \boldsymbol{Z}_{\mathrm{in}} \boldsymbol{Z}_{\mathrm{in}}^\top \boldsymbol{Z}_{\mathrm{in}} \boldsymbol{X}_i^\top \mathbb{1}_L \right)^2 | \mathsf{Z}_{\mathrm{in}} \right]$$

$$\leq \frac{1}{N^3 L^2 V^2} \mathsf{z}_k^\top \boldsymbol{Z}_{\mathrm{in}} \boldsymbol{Z}_{\mathrm{in}}^\top \boldsymbol{Z}_{\mathrm{in}} \left( \tfrac{L^2}{V^2} \mathbb{1}_V \mathbb{1}_V^\top + \tfrac{L}{V} \boldsymbol{I}_V \right) \boldsymbol{Z}_{\mathrm{in}}^\top \boldsymbol{Z}_{\mathrm{in}} \boldsymbol{Z}_{\mathrm{in}} \mathsf{z}_k \leq \frac{C \log^2 V}{N^3 L d^3}$$

Therefore, by Chebyshev's inequality, we have

$$|\kappa_6| \lesssim \frac{1}{N\sqrt{L}d^2}.$$

Overall, by using $N \ll VL$,

$$|\kappa| \leq \left( \frac{1}{N\sqrt{L}d(L \wedge d)} + \frac{1}{NLd(L \wedge d)^{1/2}} + \frac{1}{NL\sqrt{Vd}} + \frac{1}{\sqrt{N}VLd} + \frac{1}{\sqrt{N}VL^2\sqrt{d}} \right)$$

$$+ \left( \frac{1}{VL^2\sqrt{d}} \frac{1}{V^2\sqrt{L}d^{3/2}} \right).$$

## D.3 Concentration bound for $\mathbf{s_3}$

We have

$$
\boldsymbol{e}_l^\top \boldsymbol{s}_3 = \frac{1}{N^2 L} \sum_{i,j=1}^{N} \mathbf{z}_k^\top \boldsymbol{Z}_{\mathrm{in}} \boldsymbol{X}_i^\top \boldsymbol{X}_i \boldsymbol{Z}_{\mathrm{in}}^\top
$$

$$
\times \Big( \frac{1}{m} \sum_{k=1}^{m} \boldsymbol{w}_k \phi'\big(\tfrac{1}{L} \boldsymbol{w}_k^\top \boldsymbol{Z}_{\mathrm{in}} \boldsymbol{X}_i^\top \mathbb{1}_L\big) \phi\big(\tfrac{1}{L} \boldsymbol{w}_k^\top \boldsymbol{Z}_{\mathrm{in}} \boldsymbol{X}_j^\top \mathbb{1}_L\big)
$$

$$
- \mathbb{E}\Big[ \boldsymbol{w}_k \phi'\big(\tfrac{1}{L} \boldsymbol{w}_k^\top \boldsymbol{Z}_{\mathrm{in}} \boldsymbol{X}_i^\top \mathbb{1}_L\big) \phi\big(\tfrac{1}{L} \boldsymbol{w}_k^\top \boldsymbol{Z}_{\mathrm{in}} \boldsymbol{X}_j^\top \mathbb{1}_L\big) \Big] \Big) (\boldsymbol{x}_j - \tfrac{1}{V} \mathbb{1}_V)^\top \boldsymbol{Z}_{\mathrm{out}}^\top \boldsymbol{Z}_{\mathrm{out}} (\boldsymbol{x}_i - \tfrac{1}{V} \mathbb{1}_V)
$$

$$
- \frac{1}{N^2 L^2} \sum_{i,j=1}^{N} \mathbf{z}_k^\top \boldsymbol{Z}_{\mathrm{in}} \boldsymbol{X}_i^\top \mathbb{1}_L \mathbb{1}_L^\top \boldsymbol{X}_i \boldsymbol{Z}_{\mathrm{in}}^\top
$$

$$
\times \Big( \frac{1}{m} \sum_{k=1}^{m} \boldsymbol{w}_k \phi'\big(\tfrac{1}{L} \boldsymbol{w}_k^\top \boldsymbol{Z}_{\mathrm{in}} \boldsymbol{X}_i^\top \mathbb{1}_L\big) \phi\big(\tfrac{1}{L} \boldsymbol{w}_k^\top \boldsymbol{Z}_{\mathrm{in}} \boldsymbol{X}_j^\top \mathbb{1}_L\big)
$$

$$
- \mathbb{E}\Big[ \boldsymbol{w}_k \phi'\big(\tfrac{1}{L} \boldsymbol{w}_k^\top \boldsymbol{Z}_{\mathrm{in}} \boldsymbol{X}_i^\top \mathbb{1}_L\big) \phi\big(\tfrac{1}{L} \boldsymbol{w}_k^\top \boldsymbol{Z}_{\mathrm{in}} \boldsymbol{X}_j^\top \mathbb{1}_L\big) \Big] \Big) (\boldsymbol{x}_j - \tfrac{1}{V} \mathbb{1}_V)^\top \boldsymbol{Z}_{\mathrm{out}}^\top \boldsymbol{Z}_{\mathrm{out}} (\boldsymbol{x}_i - \tfrac{1}{V} \mathbb{1}_V)
$$

$$
+ \frac{\mu_{kl}}{N^2 L} \sum_{i,j=1}^{N} \big( \boldsymbol{e}_1 - \tfrac{1}{L} \mathbb{1}_L \big)^\top \boldsymbol{X}_i \boldsymbol{Z}_{\mathrm{in}}^\top
$$

$$
\times \Big( \frac{1}{m} \sum_{k=1}^{m} \boldsymbol{w}_k \phi'\big(\tfrac{1}{L} \boldsymbol{w}_k^\top \boldsymbol{Z}_{\mathrm{in}} \boldsymbol{X}_i^\top \mathbb{1}_L\big) \phi\big(\tfrac{1}{L} \boldsymbol{w}_k^\top \boldsymbol{Z}_{\mathrm{in}} \boldsymbol{X}_j^\top \mathbb{1}_L\big)
$$

$$
=: \nu + \text{negligible terms.}
$$

### D.3.1 Concentration bound for $\nu$

We define

$$
\tilde{\nu} := \mathrm{tr}\Big( \frac{1}{NL} \sum_{i=1}^{N} (\boldsymbol{x}_i - \tfrac{1}{V} \mathbb{1}_V) \mathbf{z}_k^\top \boldsymbol{Z}_{\mathrm{in}} \boldsymbol{X}_i^\top \boldsymbol{X}_i \boldsymbol{Z}_{\mathrm{in}}^\top \boldsymbol{w}_k \phi'\big(\tfrac{1}{L} \boldsymbol{w}_k^\top \boldsymbol{Z}_{\mathrm{in}} \boldsymbol{X}_i^\top \mathbb{1}_L\big)
$$

$$
\times \frac{1}{N} \sum_{j=1}^{N} \phi\big(\tfrac{1}{L} \boldsymbol{w}_k^\top \boldsymbol{Z}_{\mathrm{in}} \boldsymbol{X}_j^\top \mathbb{1}_L\big) (\boldsymbol{x}_j - \tfrac{1}{V} \mathbb{1}_V)^\top \boldsymbol{Z}_{\mathrm{out}}^\top \boldsymbol{Z}_{\mathrm{out}} \Big)
$$

$$
= \mathrm{tr}\Big( \frac{1}{NL} \sum_{i=1}^{N} (\boldsymbol{x}_i - \tfrac{1}{V} \mathbb{1}_V) \mathbf{z}_k^\top \boldsymbol{Z}_{\mathrm{in}} \boldsymbol{X}_i^\top \boldsymbol{X}_i \boldsymbol{Z}_{\mathrm{in}}^\top \boldsymbol{w}_k \phi'\big(\tfrac{1}{L} \boldsymbol{w}_k^\top \boldsymbol{Z}_{\mathrm{in}} \boldsymbol{X}_i^\top \mathbb{1}_L\big)
$$

$$
\times \frac{1}{N} \sum_{j=1}^{N} \phi\big(\tfrac{1}{L} \boldsymbol{w}_k^\top \boldsymbol{Z}_{\mathrm{in}} \boldsymbol{X}_j^\top \mathbb{1}_L\big) (\boldsymbol{x}_j - \tfrac{1}{V} \mathbb{1}_V)^\top \Big)
$$

$$
\pm \frac{\log^2 V}{\sqrt{d}} \frac{1}{NL} \Big\| \sum_{i=1}^{N} (\boldsymbol{x}_i - \tfrac{1}{V} \mathbb{1}_V) \mathbf{z}_{\nu,\delta}^\top \boldsymbol{Z}_{\mathrm{in}} \boldsymbol{X}_i^\top \boldsymbol{X}_i \boldsymbol{Z}_{\mathrm{in}}^\top \boldsymbol{w}_k \phi'\big(\tfrac{1}{L} \boldsymbol{w}_k^\top \boldsymbol{Z}_{\mathrm{in}} \boldsymbol{X}_i^\top \mathbb{1}_L\big) \Big\|_2
$$

$$
\times \Big\| \frac{1}{N} \sum_{j=1}^{N} \phi\big(\tfrac{1}{L} \boldsymbol{w}_k^\top \boldsymbol{Z}_{\mathrm{in}} \boldsymbol{X}_j^\top \mathbb{1}_L\big) (\boldsymbol{x}_j - \tfrac{1}{V} \mathbb{1}_V) \Big\|_2
$$

$$
=: \tilde{\nu}_1 + \tilde{\nu}_2,
$$

where we used Proposition 5 for the second step. We define

$$
\phi(t) =: \phi(0) + t\psi(t) \quad \text{and} \quad \phi'(t) =: \phi(0) + t\psi_1(t) \quad \text{and} \quad \psi(t) =: \psi(0) + t\psi_2(t).
$$

and write

$$
\tilde{\nu}_1 = \phi(0)\phi'(0)\,\mathrm{tr}\Big( \frac{1}{NL} \sum_{i=1}^{N} \boldsymbol{x}_i \mathbf{z}_k^\top \boldsymbol{Z}_{\mathrm{in}} \boldsymbol{X}_i^\top \boldsymbol{X}_i \boldsymbol{Z}_{\mathrm{in}}^\top \boldsymbol{w}_k \frac{1}{N} \sum_{j=1}^{N} (\boldsymbol{x}_j - \tfrac{1}{V} \mathbb{1}_V)^\top \Big)
$$

$$+ \phi(0)\mathrm{tr}\Big( \frac{1}{NL^2}\sum_{i=1}^N \boldsymbol{x}_i \mathsf{z}_k^\top \boldsymbol{Z}_{\mathrm{in}} \boldsymbol{x}_i \boldsymbol{x}_i^\top \boldsymbol{Z}_{\mathrm{in}}^\top \boldsymbol{w}_k \boldsymbol{w}_k^\top$$

$$\times \boldsymbol{Z}_{\mathrm{in}} \boldsymbol{X}_i^\top \mathbb{1}_L \psi_1\big(\tfrac{1}{L}\boldsymbol{w}_k^\top \boldsymbol{Z}_{\mathrm{in}} \boldsymbol{X}_i^\top \mathbb{1}_L\big) \frac{1}{N}\sum_{j=1}^N (\boldsymbol{x}_j - \tfrac{1}{V}\mathbb{1}_V)^\top \Big)$$

$$+ \phi(0)\mathrm{tr}\Big( \frac{1}{NL^2}\sum_{i=1}^N \boldsymbol{x}_i \mathsf{z}_k^\top \boldsymbol{Z}_{\mathrm{in}} \boldsymbol{N}_i^\top \boldsymbol{N}_i \boldsymbol{Z}_{\mathrm{in}}^\top \boldsymbol{w}_k \boldsymbol{w}_k^\top$$

$$\times \boldsymbol{Z}_{\mathrm{in}} \boldsymbol{X}_i^\top \mathbb{1}_L \psi_1\big(\tfrac{1}{L}\boldsymbol{w}_k^\top \boldsymbol{Z}_{\mathrm{in}} \boldsymbol{X}_i^\top \mathbb{1}_L\big) \frac{1}{N}\sum_{j=1}^N (\boldsymbol{x}_j - \tfrac{1}{V}\mathbb{1}_V)^\top \Big)$$

$$+ \mathrm{tr}\Big( \frac{1}{NL}\sum_{i=1}^N (\boldsymbol{x}_i - \tfrac{1}{V}\mathbb{1}_V)\mathsf{z}_k^\top \boldsymbol{Z}_{\mathrm{in}} \boldsymbol{X}_i^\top \boldsymbol{X}_i \boldsymbol{Z}_{\mathrm{in}}^\top \boldsymbol{w}_k \phi'\big(\tfrac{1}{L}\boldsymbol{w}_k^\top \boldsymbol{Z}_{\mathrm{in}} \boldsymbol{X}_i^\top \mathbb{1}_L\big)$$

$$\times \frac{1}{N}\sum_{j=1}^N \psi\big(\tfrac{1}{L}\boldsymbol{w}_k^\top \boldsymbol{Z}_{\mathrm{in}} \boldsymbol{X}_j^\top \mathbb{1}_L\big) \frac{1}{L}\boldsymbol{w}_k^\top \boldsymbol{Z}_{\mathrm{in}} \boldsymbol{X}_j^\top \mathbb{1}_L (\boldsymbol{x}_j - \tfrac{1}{V}\mathbb{1}_V)^\top \Big)$$

$$=: \tilde{\nu}_{11} + \tilde{\nu}_{12} + \tilde{\nu}_{13} + \tilde{\nu}_{14}.$$

In the following, we bound each term separately. Let $n_w := |\{i \in [N] : \boldsymbol{x}_i = \boldsymbol{e}_w\}|$.

- We have

$$\tilde{\nu}_{11} = \frac{1}{L}\sum_{w=1}^V (\frac{n_w}{N} - \frac{1}{V})\frac{n_w}{N}\mathsf{z}_k^\top \boldsymbol{Z}_{\mathrm{in}}\Big( \boldsymbol{e}_w \boldsymbol{e}_w^\top + \tfrac{L-1}{V}\boldsymbol{I}_V \Big)\boldsymbol{Z}_{\mathrm{in}}^\top \boldsymbol{w}_k$$

$$+ \mathsf{z}_k^\top \boldsymbol{Z}_{\mathrm{in}} \frac{1}{NL}\sum_{w=1}^V (\frac{n_w}{N} - \frac{1}{V})\sum_{i\in\{i_1,\cdots,i_{n_w}\}} \Big( \boldsymbol{N}_i^\top \boldsymbol{N}_i - \tfrac{L-1}{V}\boldsymbol{I}_V \Big)\boldsymbol{Z}_{\mathrm{in}}^\top \boldsymbol{w}_k$$

We have by using Lemma 5 and Proposition 8,

$$\mathbb{E}\Big[ \Big(\mathsf{z}_k^\top \boldsymbol{Z}_{\mathrm{in}} \frac{1}{NL}\sum_{w=1}^V (\frac{n_w}{N} - \frac{1}{V})\sum_{i\in\{i_1,\cdots,i_{n_w}\}} \Big( \boldsymbol{N}_i^\top \boldsymbol{N}_i - \tfrac{L-1}{V}\boldsymbol{I}_V \Big)\boldsymbol{Z}_{\mathrm{in}}^\top \boldsymbol{w}_k \Big)^2 |\mathsf{Z}_{\mathrm{in}}\Big]$$

$$= \mathbb{E}\Big[ \Big\|\mathsf{z}_k^\top \boldsymbol{Z}_{\mathrm{in}} \frac{1}{NL}\sum_{w=1}^V (\frac{n_w}{N} - \frac{1}{V})\sum_{i\in\{i_1,\cdots,i_{n_w}\}} \Big( \boldsymbol{N}_i^\top \boldsymbol{N}_i - \tfrac{L-1}{V}\boldsymbol{I}_V \Big)\boldsymbol{Z}_{\mathrm{in}}^\top \Big\|_2^2 |\mathsf{Z}_{\mathrm{in}}\Big]$$

$$= \mathsf{z}_k^\top \boldsymbol{Z}_{\mathrm{in}} \frac{1}{N^2L^2}\sum_{w=1}^V \mathbb{E}\Big[ (\frac{n_w}{N} - \frac{1}{V})^2 n_w \Big]$$

$$\times \mathbb{E}\Big[ \Big( \boldsymbol{N}_1^\top \boldsymbol{N}_1 - \tfrac{L-1}{V}\boldsymbol{I}_V \Big)\boldsymbol{Z}_{\mathrm{in}}^\top \boldsymbol{Z}_{\mathrm{in}}\Big( \boldsymbol{N}_1^\top \boldsymbol{N}_1 - \tfrac{L-1}{V}\boldsymbol{I}_V \Big)|\mathsf{Z}_{\mathrm{in}}\Big] \boldsymbol{Z}_{\mathrm{in}}^\top \mathsf{z}_{\nu,\delta}$$

$$\leq \frac{C}{N^2L^2V}\mathsf{z}_k^\top \boldsymbol{Z}_{\mathrm{in}} \mathbb{E}\Big[ \Big( \boldsymbol{N}_1^\top \boldsymbol{N}_1 - \tfrac{L-1}{V}\boldsymbol{I}_V \Big)\boldsymbol{Z}_{\mathrm{in}}^\top \boldsymbol{Z}_{\mathrm{in}}\Big( \boldsymbol{N}_1^\top \boldsymbol{N}_1 - \tfrac{L-1}{V}\boldsymbol{I}_V \Big)|\mathsf{Z}_{\mathrm{in}}\Big] \boldsymbol{Z}_{\mathrm{in}}^\top \mathsf{z}_{\nu,\delta}$$

$$= \frac{C}{N^2V d(L \wedge d)}.$$

Moreover, by using (E1.1) and Proposition 8,

$$\mathbb{E}\Big[ \Big(\frac{1}{L}\sum_{w=1}^V (\frac{n_w}{N} - \frac{1}{V})\frac{n_w}{N}\mathsf{z}_k^\top \boldsymbol{Z}_{\mathrm{in}}\Big( \boldsymbol{e}_w \boldsymbol{e}_w^\top + \tfrac{L-1}{V}\boldsymbol{I}_V \Big)\boldsymbol{Z}_{\mathrm{in}}^\top \boldsymbol{w}_k \Big)^2 |\mathsf{Z}_{\mathrm{in}}\Big]$$

$$= \frac{1}{L^2}\mathbb{E}\Big[ \Big\|\mathsf{z}_{\nu,\delta}^\top \boldsymbol{Z}_{\mathrm{in}}\Big( \sum_{w=1}^V (\frac{n_w}{N} - \frac{1}{V})\frac{n_w}{N}\boldsymbol{e}_w \boldsymbol{e}_w^\top + (\frac{n_w}{N} - \frac{1}{V})^2\tfrac{L-1}{V}\boldsymbol{I}_V \Big)\boldsymbol{Z}_{\mathrm{in}}^\top \Big\|_2^2 |\mathsf{Z}_{\mathrm{in}}\Big]$$

$$\leq \frac{V^2}{L^2d^2}\mathbb{E}\Big[ \Big\| \sum_{w=1}^V (\frac{n_w}{N} - \frac{1}{V})\frac{n_w}{N}\boldsymbol{e}_w \boldsymbol{e}_w^\top + (\frac{n_w}{N} - \frac{1}{V})^2\tfrac{L-1}{V}\boldsymbol{I}_V \Big\|_2^2 |\mathsf{Z}_{\mathrm{in}}\Big]$$

$$\leq \frac{CV^2}{L^2 d^2} \mathbb{E}\Big[\sup_{w \in [N]} \Big|(\frac{n_w}{N} - \frac{1}{V})\frac{n_w}{N}\Big|^2\Big] + \frac{C}{d^2}\mathbb{E}\Big[\Big(\sum_{w=1}^{V}(\frac{n_w}{N} - \frac{1}{V})^2\Big)^2\Big] \leq \frac{C}{d^2 N^2}.$$

Therefore,

$$\mathbb{E}\Big[\tilde{\nu}_{11}^2 | \mathsf{Z}_{\mathrm{in}}\Big] \lesssim \frac{1}{d^2 N^2}.$$

- Moreover,

$$\tilde{\nu}_{12}^2 \leq \frac{C}{N}\Big\|\frac{1}{NL^2}\sum_{i=1}^{N} \boldsymbol{x}_i \mathsf{z}_k^\top \boldsymbol{Z}_{\mathrm{in}} \boldsymbol{x}_i \boldsymbol{x}_i^\top \boldsymbol{Z}_{\mathrm{in}}^\top \boldsymbol{w}_k \boldsymbol{w}_k^\top \boldsymbol{Z}_{\mathrm{in}} \boldsymbol{X}_i^\top \mathbb{1}_L \psi_1\big(\frac{1}{L}\boldsymbol{w}_k^\top \boldsymbol{Z}_{\mathrm{in}}\boldsymbol{X}_i^\top \mathbb{1}_L\big)\Big\|_2^2.$$

We have for any $i \in [N]$,

$$\Big|\mathsf{z}_k^\top \boldsymbol{Z}_{\mathrm{in}} \boldsymbol{x}_i \boldsymbol{x}_i^\top \boldsymbol{Z}_{\mathrm{in}}^\top \boldsymbol{w}_k \boldsymbol{w}_k^\top \boldsymbol{Z}_{\mathrm{in}} \boldsymbol{X}_i^\top \mathbb{1}_L \psi_1\big(\frac{1}{L}\boldsymbol{w}_k^\top \boldsymbol{Z}_{\mathrm{in}}\boldsymbol{X}_i^\top \mathbb{1}_L\big)\Big| \lesssim \sqrt{L}\Big(\mathbb{1}_{\boldsymbol{x}_i = \boldsymbol{e}_k} + \frac{1}{\sqrt{d}}\Big)$$

Then,

$$\Big\|\frac{1}{NL^2}\sum_{i=1}^{N} \boldsymbol{x}_i \mathsf{z}_k^\top \boldsymbol{Z}_{\mathrm{in}} \boldsymbol{x}_i \boldsymbol{x}_i^\top \boldsymbol{Z}_{\mathrm{in}}^\top \boldsymbol{w}_k \boldsymbol{w}_k^\top \boldsymbol{Z}_{\mathrm{in}} \boldsymbol{X}_i^\top \mathbb{1}_L \psi_1\big(\frac{1}{L}\boldsymbol{w}_k^\top \boldsymbol{Z}_{\mathrm{in}}\boldsymbol{X}_i^\top \mathbb{1}_L\big)\Big\|_2^2$$

$$\lesssim \frac{1}{L^3}\Big\|\frac{1}{N}\sum_{i=1}^{N}\boldsymbol{x}_i\Big(\mathbb{1}_{\boldsymbol{x}_i = \boldsymbol{e}_k} + \frac{1}{\sqrt{d}}\Big)\Big\|^2 \lesssim \frac{1}{VdL^3}$$

Then,

$$\mathbb{E}[\tilde{\nu}_{12}^2 | \mathsf{Z}_{\mathrm{in}}] \lesssim \frac{1}{NVdL^3}.$$

- Moreover,

$$\tilde{\nu}_{13}^2 \leq \frac{C}{N}\Big\|\frac{1}{NL^2}\sum_{i=1}^{N} \boldsymbol{x}_i \mathsf{z}_k^\top \boldsymbol{Z}_{\mathrm{in}} \boldsymbol{N}_i^\top \boldsymbol{N}_i \boldsymbol{Z}_{\mathrm{in}}^\top \boldsymbol{w}_k \boldsymbol{w}_k^\top \boldsymbol{Z}_{\mathrm{in}} \boldsymbol{X}_i^\top \mathbb{1}_L \psi_1\big(\frac{1}{L}\boldsymbol{w}_k^\top \boldsymbol{Z}_{\mathrm{in}}\boldsymbol{X}_i^\top \mathbb{1}_L\big)\Big\|_2^2.$$

We have for any $i \in [N]$,

$$\Big|\mathsf{z}_k^\top \boldsymbol{Z}_{\mathrm{in}} \boldsymbol{N}_i^\top \boldsymbol{N}_i \boldsymbol{Z}_{\mathrm{in}}^\top \boldsymbol{w}_k \boldsymbol{w}_k^\top \boldsymbol{Z}_{\mathrm{in}} \boldsymbol{X}_i^\top \mathbb{1}_L \psi_1\big(\frac{1}{L}\boldsymbol{w}_k^\top \boldsymbol{Z}_{\mathrm{in}}\boldsymbol{X}_i^\top \mathbb{1}_L\big)\Big|$$

$$\lesssim \sqrt{L}\|\boldsymbol{Z}_{\mathrm{in}}\boldsymbol{N}_i^\top \boldsymbol{N}_i \boldsymbol{Z}_{\mathrm{in}}^\top \mathsf{z}_k\|_2 \lesssim \sqrt{L}\big(\boldsymbol{e}_k^\top \boldsymbol{N}_i^\top \mathbb{1}_{L-1} + 1 + \frac{L}{d}\big)$$

Then,

$$\Big\|\frac{1}{NL^2}\sum_{i=1}^{N} \boldsymbol{x}_i \mathsf{z}_k^\top \boldsymbol{Z}_{\mathrm{in}} \boldsymbol{N}_i^\top \boldsymbol{N}_i \boldsymbol{Z}_{\mathrm{in}}^\top \boldsymbol{w}_k \boldsymbol{w}_k^\top \boldsymbol{Z}_{\mathrm{in}} \boldsymbol{X}_i^\top \mathbb{1}_L \psi_1\big(\frac{1}{L}\boldsymbol{w}_k^\top \boldsymbol{Z}_{\mathrm{in}}\boldsymbol{X}_i^\top \mathbb{1}_L\big)\Big\|_2^2$$

$$\lesssim \frac{1}{N^2 L^3}\Big\|\sum_{i=1}^{N}\boldsymbol{x}_i\Big(\mathbb{1}_{L-1}^\top \boldsymbol{N}_i \boldsymbol{e}_k + 1 + \frac{L}{d}\Big)\Big\|_2^2 \lesssim \frac{1}{VLd(L \wedge d)} + \frac{1}{N^2 L^3}\Big\|\sum_{i=1}^{N}\boldsymbol{x}_i\mathbb{1}_{L-1}^\top \boldsymbol{N}_i \boldsymbol{e}_k\Big\|_2^2$$

We have

$$\frac{1}{N^2 L^3}\mathbb{E}\Big[\Big\|\sum_{i=1}^{N}\boldsymbol{x}_i\mathbb{1}_{L-1}^\top \boldsymbol{N}_i \boldsymbol{e}_k\Big\|_2^2\Big] \lesssim \frac{1}{V^3 L} + \frac{1}{NVL^2}.$$

Then,

$$\mathbb{E}[\tilde{\nu}_{13}^2 | \mathsf{Z}_{\mathrm{in}}] \lesssim \frac{1}{NVLd(L \wedge d)}.$$

- Lastly, we have

$$|\tilde{\nu}_{14}| \leq \Big\| \frac{1}{NL} \sum_{i=1}^{N} (\boldsymbol{x}_i - \tfrac{1}{V} \mathbb{1}_V) \mathbf{z}_k^\top \boldsymbol{Z}_{\mathrm{in}} \boldsymbol{X}_i^\top \boldsymbol{X}_i \boldsymbol{Z}_{\mathrm{in}}^\top \boldsymbol{w}_k \phi'\big(\tfrac{1}{L} \boldsymbol{w}_k^\top \boldsymbol{Z}_{\mathrm{in}} \boldsymbol{X}_i^\top \mathbb{1}_L \big) \Big\|_2$$

$$\times \Big\| \frac{1}{N} \sum_{j=1}^{N} \psi\big(\tfrac{1}{L} \boldsymbol{w}_k^\top \boldsymbol{Z}_{\mathrm{in}} \boldsymbol{X}_j^\top \mathbb{1}_L \big) \tfrac{1}{L} \boldsymbol{w}_k^\top \boldsymbol{Z}_{\mathrm{in}} \boldsymbol{X}_j^\top \mathbb{1}_L (\boldsymbol{x}_j - \tfrac{1}{V} \mathbb{1}_V)^\top \Big\|_2.$$

By using the derivations in the two previous items, we have

$$\Big\| \frac{1}{NL} \sum_{i=1}^{N} (\boldsymbol{x}_i - \tfrac{1}{V} \mathbb{1}_V)^\top \mathbf{z}_k^\top \boldsymbol{Z}_{\mathrm{in}} \boldsymbol{X}_i^\top \boldsymbol{X}_i \boldsymbol{Z}_{\mathrm{in}}^\top \boldsymbol{w}_k \phi'\big(\tfrac{1}{L} \boldsymbol{w}_k^\top \boldsymbol{Z}_{\mathrm{in}} \boldsymbol{X}_i^\top \mathbb{1}_L \big) \Big\|_2$$

$$\leq \Big\| \frac{1}{NL} \sum_{i=1}^{N} \boldsymbol{x}_i \mathbf{z}_k^\top \boldsymbol{Z}_{\mathrm{in}} \boldsymbol{x}_i \boldsymbol{x}_i^\top \boldsymbol{Z}_{\mathrm{in}}^\top \boldsymbol{w}_k \phi'\big(\tfrac{1}{L} \boldsymbol{w}_k^\top \boldsymbol{Z}_{\mathrm{in}} \boldsymbol{X}_i^\top \mathbb{1}_L \big) \Big\|_2$$

$$+ \Big\| \frac{1}{NL^2} \sum_{i=1}^{N} \boldsymbol{x}_i \mathbf{z}_k^\top \boldsymbol{Z}_{\mathrm{in}} \boldsymbol{N}_i^\top \boldsymbol{N}_i \boldsymbol{Z}_{\mathrm{in}}^\top \boldsymbol{w}_k \boldsymbol{w}_k^\top \boldsymbol{Z}_{\mathrm{in}} \boldsymbol{X}_i^\top \mathbb{1}_L \psi_1\big(\tfrac{1}{L} \boldsymbol{w}_k^\top \boldsymbol{Z}_{\mathrm{in}} \boldsymbol{X}_i^\top \mathbb{1}_L \big) \Big\|_2$$

$$+ |\phi'(0)| \Big\| \frac{1}{NL} \sum_{i=1}^{N} (\boldsymbol{x}_i - \tfrac{1}{V} \mathbb{1}_V) \mathbf{z}_k^\top \boldsymbol{Z}_{\mathrm{in}} \boldsymbol{N}_i^\top \boldsymbol{N}_i \boldsymbol{Z}_{\mathrm{in}}^\top \boldsymbol{w}_k \Big\|_2$$

$$\lesssim \frac{1}{\sqrt{VLd(L \wedge d)}} + \phi'(0) \Big\| \frac{1}{NL} \sum_{i=1}^{N} (\boldsymbol{x}_i - \tfrac{1}{V} \mathbb{1}_V) \mathbf{z}_k^\top \boldsymbol{Z}_{\mathrm{in}} \boldsymbol{N}_i^\top \boldsymbol{N}_i \boldsymbol{Z}_{\mathrm{in}}^\top \boldsymbol{w}_k \Big\|_2$$

We have

$$\mathbb{E}\Big[ \Big\| \frac{1}{NL} \sum_{i=1}^{N} (\boldsymbol{x}_i - \tfrac{1}{V} \mathbb{1}_V) \mathbf{z}_k^\top \boldsymbol{Z}_{\mathrm{in}} \boldsymbol{N}_i^\top \boldsymbol{N}_i \boldsymbol{Z}_{\mathrm{in}}^\top \boldsymbol{w}_k \Big\|_2^2 \Big| \mathbf{Z}_{\mathrm{in}} \Big]$$

$$\leq \frac{1}{N^2 L^2} \mathbb{E}\Big[ \sum_{i,j=1}^{N} (\mathbb{1}_{\boldsymbol{x}_i = \boldsymbol{x}_j} - \tfrac{1}{V}) \mathbf{z}_k^\top \boldsymbol{Z}_{\mathrm{in}} \boldsymbol{N}_i^\top \boldsymbol{N}_i \boldsymbol{Z}_{\mathrm{in}}^\top \boldsymbol{Z}_{\mathrm{in}} \boldsymbol{N}_i^\top \boldsymbol{N}_i \boldsymbol{Z}_{\mathrm{in}}^\top \mathbf{z}_k \Big| \mathbf{Z}_{\mathrm{in}} \Big]$$

$$\leq \frac{1}{NL^2} \frac{L}{V} \mathbf{z}_k^\top \boldsymbol{Z}_{\mathrm{in}} \mathrm{diag}(\boldsymbol{Z}_{\mathrm{in}}^\top \boldsymbol{Z}_{\mathrm{in}}) \boldsymbol{Z}_{\mathrm{in}}^\top \mathbf{z}_k + \frac{1}{NL^2} \frac{L^2}{V^2} \mathbf{z}_k^\top \boldsymbol{Z}_{\mathrm{in}} \boldsymbol{Z}_{\mathrm{in}}^\top \boldsymbol{Z}_{\mathrm{in}} \boldsymbol{Z}_{\mathrm{in}}^\top \mathbf{z}_k = \frac{1}{Nd(L \wedge d)} \quad \text{(D.25)}$$

Moreover,

$$\Big\| \frac{1}{NL} \sum_{j=1}^{N} \psi\big(\tfrac{1}{L} \boldsymbol{w}_k^\top \boldsymbol{Z}_{\mathrm{in}} \boldsymbol{X}_j^\top \mathbb{1}_L \big) \boldsymbol{w}_k^\top \boldsymbol{Z}_{\mathrm{in}} \boldsymbol{X}_j^\top \mathbb{1}_L (\boldsymbol{x}_j - \tfrac{1}{V} \mathbb{1}_V)^\top \Big\|_2$$

$$= |\psi(0)| \Big\| \frac{1}{NL} \sum_{j=1}^{N} \boldsymbol{w}_k^\top \boldsymbol{Z}_{\mathrm{in}} \boldsymbol{X}_j^\top \mathbb{1}_L (\boldsymbol{x}_j - \tfrac{1}{V} \mathbb{1}_V)^\top \Big\|_2$$

$$+ \Big\| \frac{1}{NL^2} \sum_{j=1}^{N} \boldsymbol{x}_j \psi_2\big(\tfrac{1}{L} \boldsymbol{w}_k^\top \boldsymbol{Z}_{\mathrm{in}} \boldsymbol{X}_j^\top \mathbb{1}_L \big) (\boldsymbol{w}_k^\top \boldsymbol{Z}_{\mathrm{in}} \boldsymbol{X}_j^\top \mathbb{1}_L)^2 \Big\|_2.$$

We have

$$\Big| \psi_2\big(\tfrac{1}{L} \boldsymbol{w}_k^\top \boldsymbol{Z}_{\mathrm{in}} \boldsymbol{X}_j^\top \mathbb{1}_L \big) (\boldsymbol{w}_k^\top \boldsymbol{Z}_{\mathrm{in}} \boldsymbol{X}_j^\top \mathbb{1}_L)^2 \Big| \lesssim L.$$

Therefore,

$$\Big\| \frac{1}{NL^2} \sum_{j=1}^{N} \boldsymbol{x}_j \psi_2\big(\tfrac{1}{L} \boldsymbol{w}_k^\top \boldsymbol{Z}_{\mathrm{in}} \boldsymbol{X}_j^\top \mathbb{1}_L \big) (\boldsymbol{w}_k^\top \boldsymbol{Z}_{\mathrm{in}} \boldsymbol{X}_j^\top \mathbb{1}_L)^2 \Big\|_2 \lesssim \frac{1}{\sqrt{V}L}.$$

Moreover,

$$\Big\| \frac{1}{NL} \sum_{j=1}^{N} \boldsymbol{w}_k^\top \boldsymbol{Z}_{\mathrm{in}} \boldsymbol{X}_j^\top \mathbb{1}_L (\boldsymbol{x}_j - \tfrac{1}{V} \mathbb{1}_V)^\top \Big\|_2$$

$$\leq \Big\| \frac{1}{NL} \sum_{j=1}^{N} \boldsymbol{w}_k^\top \boldsymbol{Z}_{\mathrm{in}} (\boldsymbol{X}_j^\top - \tfrac{1}{V} \mathbb{1}_V \mathbb{1}_L^\top) \mathbb{1}_L (\boldsymbol{x}_j - \tfrac{1}{V} \mathbb{1}_V)^\top \Big\|_2$$

$$+ \frac{1}{V} |\boldsymbol{w}_k^\top \boldsymbol{Z}_{\mathrm{in}} \mathbb{1}_V| \Big\| \frac{1}{N} \sum_{j=1}^{N} (\boldsymbol{x}_j - \tfrac{1}{V} \mathbb{1}_V)^\top \Big\|_2 .$$

We have

- $\frac{1}{V} |\boldsymbol{w}_k^\top \boldsymbol{Z}_{\mathrm{in}} \mathbb{1}_V| \big\| \frac{1}{N} \sum_{j=1}^{N} (\boldsymbol{x}_j - \tfrac{1}{V} \mathbb{1}_V)^\top \big\|_2 \leq \frac{C \log^2 V}{\sqrt{VN}}$

- Moreover,

$$\Big\| \frac{1}{NL} \sum_{j=1}^{N} \boldsymbol{w}_k^\top \boldsymbol{Z}_{\mathrm{in}} (\boldsymbol{X}_j^\top - \tfrac{1}{V} \mathbb{1}_V \mathbb{1}_L^\top) \mathbb{1}_L (\boldsymbol{x}_j - \tfrac{1}{V} \mathbb{1}_V)^\top \Big\|_2^2$$

$$= \boldsymbol{w}_k^\top \boldsymbol{Z}_{\mathrm{in}} \Big( \frac{1}{NL} \sum_{j=1}^{N} (\boldsymbol{X}_j^\top - \tfrac{1}{V} \mathbb{1}_V \mathbb{1}_L^\top) \mathbb{1}_L (\boldsymbol{x}_j - \tfrac{1}{V} \mathbb{1}_V)^\top \Big)$$

$$\times \Big( \frac{1}{NL} \sum_{j=1}^{N} (\boldsymbol{X}_j^\top - \tfrac{1}{V} \mathbb{1}_V \mathbb{1}_L^\top) \mathbb{1}_L (\boldsymbol{x}_j - \tfrac{1}{V} \mathbb{1}_V)^\top \Big)^\top \boldsymbol{Z}_{\mathrm{in}}^\top \boldsymbol{w}_k \lesssim \frac{1}{NL} .$$

Then, for $N \ll VL$

$$\mathbb{E}[\bar{\nu}_{14}^2 | \mathsf{Z}_{\mathrm{in}}] \leq \frac{C \log^{16} V}{N^2 L d (L \wedge d)}$$

- On the other hand, we have

$$|\nu_2| \leq \frac{1}{\sqrt{d}} \Big\| \frac{1}{NL} \sum_{i=1}^{N} (\boldsymbol{x}_i - \tfrac{1}{V} \mathbb{1}_V) z_k^\top \boldsymbol{Z}_{\mathrm{in}} \boldsymbol{X}_i^\top \boldsymbol{X}_i \boldsymbol{Z}_{\mathrm{in}}^\top \boldsymbol{w}_k \phi'\big( \tfrac{1}{L} \boldsymbol{w}_k^\top \boldsymbol{Z}_{\mathrm{in}} \boldsymbol{X}_i^\top \mathbb{1}_L \big) \Big\|_2$$

$$\times \Big\| \frac{1}{N} \sum_{j=1}^{N} \phi\big( \tfrac{1}{L} \boldsymbol{w}_k^\top \boldsymbol{Z}_{\mathrm{in}} \boldsymbol{X}_j^\top \mathbb{1}_L \big) (\boldsymbol{x}_j - \tfrac{1}{V} \mathbb{1}_V) \Big\|_2$$

Note that

$$\Big\| \frac{1}{N} \sum_{j=1}^{N} \phi\big( \tfrac{1}{L} \boldsymbol{w}_k^\top \boldsymbol{Z}_{\mathrm{in}} \boldsymbol{X}_j^\top \mathbb{1}_L \big) (\boldsymbol{x}_j - \tfrac{1}{V} \mathbb{1}_V) \Big\|_2 = |\phi(0)| \Big\| \frac{1}{N} \sum_{j=1}^{N} (\boldsymbol{x}_j - \frac{1}{V} \mathbb{1}_V) \Big\|_2$$

$$+ \Big\| \frac{1}{N} \sum_{j=1}^{N} \psi\big( \tfrac{1}{L} \boldsymbol{w}_k^\top \boldsymbol{Z}_{\mathrm{in}} \boldsymbol{X}_j^\top \mathbb{1}_L \big) \tfrac{1}{L} \boldsymbol{w}_k^\top \boldsymbol{Z}_{\mathrm{in}} \boldsymbol{X}_j^\top \mathbb{1}_L (\boldsymbol{x}_j - \tfrac{1}{V} \mathbb{1}_V) \Big\|_2 \lesssim \frac{1}{\sqrt{N}} .$$

Therefore by (D.25), we have

$$\mathbb{E}\Big[ \nu_2^2 | \mathsf{Z}_{\mathrm{in}} \Big] \lesssim \frac{1}{N^2 d^2 (L \wedge d)}$$

Therefore, we have

$$\mathbb{E}[\nu | \mathsf{Z}_{\mathrm{in}}] = 0 \quad \text{and} \quad \mathrm{Variance}(\nu | \mathsf{Z}_{\mathrm{in}}) \lesssim \frac{1}{N^2 d^2 m} .$$

# E    LOWER BOUND

To prove a lower bound, we construct a Bayesian setting with the same likelihood distribution in our setting. In particular, the ground truth permutation is chosen from the set of permutation matrices:

$$\mathcal{H} := \{ \boldsymbol{P} \in \{0, 1\}^{V \times V} \mid \boldsymbol{\Pi} \text{ is a permutation matrix} \}.$$

We describe our Bayesian setting as a game between `Environment` and `Learner` as follows:

- At the beginning, `Environment` samples $\boldsymbol{P}_* \sim \text{Unif}(\mathcal{H})$, probability vectors without revealing them to the learner.

- `Learner` observes $L + 1$ channel that generates words from the set $\mathcal{V} = \{\boldsymbol{e}_1, \boldsymbol{e}_2, \cdots, \boldsymbol{e}_V\}$ sequentially for $t = 1, 2, \cdots, N$ with distributions:

    - At every round, `Environment` randomly picks a channel $\ell_t$
    - *Label:* Channel 0 generates $\boldsymbol{p}_t \sim_{iid} \text{Unif}(\mathcal{V})$
    - *Input:* Given $\ell_t$ and $\boldsymbol{p}_t$, Channel $\ell_t$ generates $\boldsymbol{X}_{\ell_t, t} = \boldsymbol{P}_* \boldsymbol{p}_t$
    - *Noise distribution:* Channel $j \in [L] \setminus \{\ell_t\}$ generate $\boldsymbol{X}_{j,t} \sim \text{Unif}(\mathcal{V})$ independent of Channel 0.

- Let $\mathcal{D} := \{(\boldsymbol{X}_t, \boldsymbol{p}_t)\}_{t \leq N}$ be the dataset. We study the Bayes estimator with $0 - 1$ loss given the representation of the past: $S = f(\mathcal{D}, \ell_{1:N})$:

$$\hat{\boldsymbol{P}} = \underset{\boldsymbol{P} \in \mathcal{H}}{\arg \max} \, \mathbb{P}[\boldsymbol{P} = \boldsymbol{P}_* | S, Z_{\text{in}}]. \tag{E.1}$$

In the following we consider the empirical mean and covariance of embedded words as the given data, i.e., $S := \{(\boldsymbol{\mu}_t, \boldsymbol{\Sigma}_t, \boldsymbol{p}_t)\}_{t \leq N}$, where

$$\boldsymbol{\mu}_t := \frac{1}{L} \boldsymbol{Z}_{\text{in}} \boldsymbol{X}_t^\top \mathbb{1}_L + \frac{\sigma_{\boldsymbol{\mu}}}{\sqrt{L}} \boldsymbol{g}_t \ \text{ and } \ \boldsymbol{\Sigma}_t := \frac{1}{L} \boldsymbol{Z}_{\text{in}} \boldsymbol{X}_t^\top \boldsymbol{X}_t \boldsymbol{Z}_{\text{in}}^\top + \frac{\sigma_{\boldsymbol{\Sigma}}}{\sqrt{dL}} \boldsymbol{G}_t.$$

where $\{(\boldsymbol{g}_t, \boldsymbol{G}_t)\}_{t \leq N}$ are i.i.d. measurement noise with distributions $\boldsymbol{g}_t \sim \mathcal{N}(0, \frac{1}{d} \boldsymbol{I}_d)$ and $\boldsymbol{G}_{t,ij} = \boldsymbol{G}_{t,ji}$ with $\boldsymbol{G}_{t,ij} \sim \mathcal{N}\big(0, \frac{(1+\delta_{ij})}{d}\big)$ i.i.d. for $i < j$.

**Theorem 4.** *The following lower bound holds:*

$$\mathbb{P}[\hat{\boldsymbol{P}} \neq \boldsymbol{P}_* | Z_{\text{in}}] \geq 1 - o_V(1) - \frac{\Omega(N)}{V} \left( 1 \wedge \Big( \frac{1}{\sigma_{\boldsymbol{\mu}}^2} \frac{d}{L \log V} + \frac{C}{\sigma_{\boldsymbol{\Sigma}}^2} \frac{d^2}{L \log V} \Big) \right)$$

We use an information-theoretic argument to prove Theorem 4. For the proof, let $H(A)$ and $H(A|C)$ denote the entropy and conditional entropy of $A$ given $C$; let $I(A; B) = H(A) - H(A|B)$ and $I(A; B|C) = H(A|C) - H(A|B, C)$ denote the mutual information between random variables $A$ and $B$ and the conditional mutual given $C$, respectively. We let $D_{\text{KL}}$ denote the Kullback-Leibler (KL) divergence. We start with an auxiliary statement for the proof.

**Lemma 2.** *Let $A, B, C, D$ be discrete random variables defined on the same probability space. The following statements hold:*

- *In general, $H(A|B, C) \leq H(A|B)$. The equality is satisfied if and only if $A \perp\!\!\!\perp C|B$.*

- *If $B \perp\!\!\!\perp D \mid (A, C)$, we have $I(A, B|C, D) \leq I(A, B|C)$.*

- *Let $S = g(A, C)$ be a measurable function of $(A, C)$. If $B \perp\!\!\!\perp A|(S, C, D)$, then $I(A; B|C, D) = I(S; B|C, D)$.*

- *Given, $\boldsymbol{\mu}, \boldsymbol{\mu}' \in \mathbb{R}^d$, positive definite $\boldsymbol{\Sigma} \in \mathbb{R}^{d \times d}$ and $\text{supp}(A) \subseteq \mathbb{R}^d$, we have*

$$D_{\text{KL}}(\mathcal{N}(\boldsymbol{\mu} + A, \boldsymbol{\Sigma}) \| \mathcal{N}(\boldsymbol{\mu}' + A, \boldsymbol{\Sigma})) \leq \frac{1}{2} (\boldsymbol{\mu} - \boldsymbol{\mu}')^\top \boldsymbol{\Sigma}^{-1} (\boldsymbol{\mu} - \boldsymbol{\mu}').$$

*Proof.* We have

$$H(A|B) - H(A|B, C) = \mathbb{E}\left[ \log \frac{\mathbb{P}(A|B, C)}{\mathbb{P}(A|B)} \right] = \mathbb{E}\left[ \log \frac{\mathbb{P}(A, C|B)}{\mathbb{P}(A|B)\mathbb{P}(C|B)} \right] = I(A, C|B).$$

Since the mutual information is non-negative, the first item follows. Moreover, since $I(A, C|B) = 0$ if and only if $A \perp\!\!\!\perp C|B$. For the second item, by using the first item,

$$I(A, B|C, D) = H(B|C, D) - H(B|A, C, D) \leq H(B|C) - H(B|A, C) = I(A, B|C).$$

For the third item, since $S$ is a function of $(A, C)$, we have

$$I(A; B|C, D) = I((A, S); B|C, D) = H(B|C, D) - H(B|A, S, C, D)$$

$$= H(B|C, D) - H(B|S, C, D) = I(S; B|C, D).$$

Let $f$ denotes the Gaussian pdf with 0 and covariance $\boldsymbol{\Sigma}$. For any $\boldsymbol{x} \in \mathbb{R}^d$, since $t \to t \log t$ is convex

$$\Big( \sum_{\boldsymbol{a} \in \mathrm{supp}(A)} p(\boldsymbol{a}) f(\boldsymbol{x} - \boldsymbol{\mu} - \boldsymbol{a}) \Big) \log \frac{\Big( \sum_{\boldsymbol{a} \in \mathrm{supp}(A)} p(\boldsymbol{a}) f(\boldsymbol{x} - \boldsymbol{\mu} - \boldsymbol{a}) \Big)}{\Big( \sum_{\boldsymbol{a} \in \mathrm{supp}(A)} p(a) f(\boldsymbol{x} - \boldsymbol{\mu}' - \boldsymbol{a}) \Big)}$$

$$\leq \sum_{\boldsymbol{a} \in \mathrm{supp}(A)} p(\boldsymbol{a}) f(\boldsymbol{x} - \boldsymbol{\mu} - \boldsymbol{a}) \log \frac{f(\boldsymbol{x} - \boldsymbol{\mu} - \boldsymbol{a})}{f(\boldsymbol{x} - \boldsymbol{\mu}' - \boldsymbol{a})}.$$

Therefore, we have

$$D_{\mathrm{KL}}(\mathcal{N}(\boldsymbol{\mu} + A, \boldsymbol{\Sigma})||\mathcal{N}(\boldsymbol{\mu}' + A, \boldsymbol{\Sigma})) \leq \sum_{\boldsymbol{a} \in \mathrm{supp}(A)} p(\boldsymbol{a}) D_{\mathrm{KL}}(\mathcal{N}(\boldsymbol{\mu} + \boldsymbol{a}, \boldsymbol{\Sigma})||\mathcal{N}(\boldsymbol{\mu}' + \boldsymbol{a}, \boldsymbol{\Sigma}))$$

$$= D_{\mathrm{KL}}(\mathcal{N}(\boldsymbol{\mu}, \boldsymbol{\Sigma})||\mathcal{N}(\boldsymbol{\mu}', \boldsymbol{\Sigma})),$$

where the last inequality follows the invariance of KL divergence in the second line to constant shifts. The final bound follows the known formula for the KL divergence between Gaussian distributions. □

The proof of Theorem 4 is given in the following:

*Proof of Theorem 4.* Since we assume $\mathsf{Z}_{\mathrm{in}}$ is known by the learner, we will fix it in the following without explicitly conditioning thte terms on it. Note that we consider the Bayes decision rule in (E.1) and use Fano's inequality (Scarlett & Cevher, 2019) to lower bound its error probability:

$$\mathbb{P}[\hat{\boldsymbol{P}} \neq \boldsymbol{P}_*|\mathsf{Z}_{\mathrm{in}}] \geq 1 - \frac{I(\boldsymbol{P}_*; S) + \log 2}{\log|\mathcal{H}|}. \tag{E.2}$$

We have

$$I(\boldsymbol{P}_*; S) = I(\boldsymbol{P}_*; \{(\boldsymbol{\mu}_t, \boldsymbol{\Sigma}_t, \boldsymbol{p}_t)\}_{t \leq N}) = I(\boldsymbol{P}_*; \{\boldsymbol{p}_t\}_{t \leq N}) + I(\boldsymbol{P}_*; \{(\boldsymbol{\mu}_t, \boldsymbol{\Sigma}_t)\}_{t \leq N}|\{\boldsymbol{p}_t\}_{t \leq N},)$$

$$\overset{(a)}{=} I(\boldsymbol{P}_*; \{(\boldsymbol{\mu}_t, \boldsymbol{\Sigma}_t)\}_{t \leq N}|\{\boldsymbol{p}_t\}_{t \leq N})$$

$$= \sum_{t=1}^N I(\boldsymbol{P}_*; (\boldsymbol{\mu}_t, \boldsymbol{\Sigma}_t)|\{(\boldsymbol{\mu}_u, \boldsymbol{\Sigma}_u)\}_{u<t}, \{\boldsymbol{p}_t\}_{t \leq N})$$

Given fixed $\mathsf{Z}_{\mathrm{in}}$, we observe that $(\boldsymbol{\mu}_t, \boldsymbol{\Sigma}_t) \perp\!\!\!\perp \{(\boldsymbol{\mu}_u, \boldsymbol{\Sigma}_u)\}_{u<t} \mid \boldsymbol{P}_*, \{\boldsymbol{p}_t\}_{t \leq N}$ and $(\boldsymbol{\mu}_t, \boldsymbol{\Sigma}_t) \perp\!\!\!\perp \{\boldsymbol{p}_u\}_{u \neq t}|\boldsymbol{P}_*,$.Therefore, by Lemma 2,

$$I(\boldsymbol{P}_*; S) \leq \sum_{t=1}^N I(\boldsymbol{P}_*; (\boldsymbol{\mu}_t, \boldsymbol{\Sigma}_t)|\{\boldsymbol{p}_t\}_{t \leq N}) \leq \sum_{t=1}^N I(\boldsymbol{P}_*; (\boldsymbol{\mu}_t, \boldsymbol{\Sigma}_t)|\boldsymbol{p}_t).$$

Moreover, we have $\boldsymbol{P}_* \perp\!\!\!\perp (\boldsymbol{\mu}_t, \boldsymbol{\Sigma}_t) \mid \boldsymbol{X}_{\ell_t, t}, \boldsymbol{p}_t$, where $\boldsymbol{X}_{\ell_t, t}$ is a function of $(\boldsymbol{P}_*, \boldsymbol{p}_t)$. Therefore, by Lemma 2,

$$I(\boldsymbol{P}_*; S) \leq \sum_{t=1}^N I(\boldsymbol{X}_{\ell_t, t}; (\boldsymbol{\mu}_t, \boldsymbol{\Sigma}_t)|\boldsymbol{p}_t).$$

We have

$$I(\boldsymbol{X}_{\ell_t, t}; (\boldsymbol{\mu}_t, \boldsymbol{\Sigma}_t)|\boldsymbol{p}_t, \mathsf{Z}_{\mathrm{in}}) = \frac{1}{V} \sum_{k=1}^V D_{\mathrm{KL}}(\mathbb{P}^k_{(\boldsymbol{\mu}_t, \boldsymbol{\Sigma}_t)}||\mathbb{P}_0) \overset{(b)}{\leq} \frac{1}{V^2} \sum_{j,k=1}^V D_{\mathrm{KL}}(\mathbb{P}^k_{(\boldsymbol{\mu}_t, \boldsymbol{\Sigma}_t)}||\mathbb{P}^j_{(\boldsymbol{\mu}_t, \boldsymbol{\Sigma}_t)})$$

where $\mathbb{P}^k_{(\boldsymbol{\mu}_t, \boldsymbol{\Sigma}_t)}$ denotes the distribution of $(\boldsymbol{s}_t, \boldsymbol{\Sigma}_t)|\boldsymbol{X}_{\ell_t, t} = \boldsymbol{e}_k$, $\mathbb{P}_0$ denotes $\mathbb{P}_0 = \frac{1}{V} \sum_{k=1}^V \mathbb{P}^k_{(\boldsymbol{\mu}_t, \boldsymbol{\Sigma}_t)}$, and (b) follows the convexity of KL divergence in its second argument. For $k \neq j$, by the last item of Lemma 2, we have

$$D_{\mathrm{KL}}(\mathbb{P}^k_{(\boldsymbol{\mu}_t, \boldsymbol{\Sigma}_t)}||\mathbb{P}^j_{(\boldsymbol{\mu}_t, \boldsymbol{\Sigma}_t)}) \leq \frac{C}{\sigma^2_{\boldsymbol{\mu}}} \frac{d}{L} \|\mathsf{z}_k - \mathsf{z}_j\|^2_2 + \frac{C}{\sigma^2_{\boldsymbol{\Sigma}}} \frac{d^2}{L} \|\mathsf{z}_k \mathsf{z}_k^\top - \mathsf{z}_j \mathsf{z}_j^\top\|^2_F \leq \frac{C}{\sigma^2_{\boldsymbol{\mu}}} \frac{d}{L} + \frac{C}{\sigma^2_{\boldsymbol{\Sigma}}} \frac{d^2}{L}.$$

Therefore, we have

$$I(\boldsymbol{P}_*; S) \leq N\Big(\frac{C}{\sigma_{\boldsymbol{\mu}}^2}\frac{d}{L} + \frac{C}{\sigma_{\boldsymbol{\Sigma}}^2}\frac{d^2}{L}\Big).$$

Moreover, we can write

$$
\begin{aligned}
I(\boldsymbol{P}_*; S) \leq I(\boldsymbol{P}_*; \mathcal{D}, \ell_{1:N}) &= I(\boldsymbol{P}_*; \{\boldsymbol{X}_t\}_{t\leq N} | \{\boldsymbol{p}_t\}_{t\leq N}, \ell_{1:N}) \\
&\leq \sum_{t=1}^{N} I(\boldsymbol{P}_*; \boldsymbol{X}_{\ell_t, t} | \{\boldsymbol{p}_t, \ell_t\}_{t\leq N}) \\
&\leq \sum_{t=1}^{N} I(\boldsymbol{P}_*; \boldsymbol{X}_{\ell_t, t} | \boldsymbol{p}_t, \ell_t)
\end{aligned}
$$

where the first inequality follows data processing inequality, third and fourth inequalities follow the first and second items in Lemma 2. We have

$$I(\boldsymbol{P}_*; \boldsymbol{X}_{\ell_t, t} | \boldsymbol{p}_t, \ell_t) = \underbrace{H(\boldsymbol{X}_{\ell_t, t} | \boldsymbol{p}_t, \ell_t)}_{\log V} - \underbrace{H(\boldsymbol{X}_{\ell_t, t} | \boldsymbol{p}_t, \ell_t, \boldsymbol{P}_*)}_{=0} = \log V.$$

Therefore, we have $I(\boldsymbol{P}_*; S) \leq N \log V$. Finally, we have

$$I(\boldsymbol{P}_*; S) \leq N\left( \log V \wedge \Big(\frac{C}{\sigma_{\boldsymbol{\mu}}^2}\frac{d}{L} + \frac{C}{\sigma_{\boldsymbol{\Sigma}}^2}\frac{d^2}{L}\Big)\right).$$

The result follows from (E.2). $\qquad \square$

# F  AUXILIARY STATEMENTS

## F.1  GAUSSIAN MATRICES AND RELATED STATEMENTS

**Lemma 3.** *Let $\boldsymbol{z} \sim \mathcal{N}(0, \boldsymbol{I}_d)$. We have $\mathbb{E}[\|\boldsymbol{z}\|_2^{2k}] = d(d+2)\cdots(d+2k-2)$.*

*Proof.* We observe that $\|\boldsymbol{z}\|_2 \sim \chi_d^2$. By using the moment formula for chi-squared distribution, we have the result. $\qquad \square$

**Lemma 4.** *Let $\boldsymbol{z} \sim \mathcal{N}(0, \boldsymbol{I}_d)$ and $\boldsymbol{S} \in \mathbb{R}^{d\times d}$ be a symmetric matrix. For $u > 0$,*

$$\mathbb{P}\left[|\boldsymbol{z}^\top \boldsymbol{S}\boldsymbol{z} - \mathrm{tr}(\boldsymbol{S})| \geq 2\|\boldsymbol{S}\|_F u + 2\|\boldsymbol{S}\|_2 u^2\right] \leq 2e^{-u^2}.$$

*Proof.* We note that $\boldsymbol{z}^\top \boldsymbol{S}\boldsymbol{z} - \mathrm{tr}(\boldsymbol{S})$ has the same distribution with $\sum_{i=1}^{d}\lambda_i(\boldsymbol{S})(Z_i^2 - 1)$, where $Z_i \sim_{iid} \mathcal{N}(0,1)$. By using the Laurent-Massart lemma, we have the result. $\qquad \square$

**Proposition 4.** *Let $\boldsymbol{S} \in \mathbb{R}^{V\times V}$ be a symmetric positive semidefinite matrix. Let*

$$\boldsymbol{M} = \boldsymbol{Z}_{\mathrm{in}}\boldsymbol{S}\boldsymbol{Z}_{\mathrm{in}}^\top.$$

*For $\mathrm{poly}(d) \gg V \gg d$, We have*

$$\mathbb{P}\left[\Big\|\boldsymbol{M} - \frac{\mathrm{tr}(\boldsymbol{S})}{d}\boldsymbol{I}_d\Big\|_2 \geq \max\Big\{\frac{\|\boldsymbol{S}\|_F}{\sqrt{d}}\log V, \|\boldsymbol{S}\|_2 \log^2 V\Big\}\right] \leq \exp(-c\log^2 V).$$

*Proof.* Without loss of generality, we can assume that $\boldsymbol{S}$ is diagonal, i.e., $\boldsymbol{S} = \mathrm{diag}(s_1, \cdots, s_V)$. We have

$$\boldsymbol{M} - \frac{\mathrm{tr}(\boldsymbol{S})}{d}\boldsymbol{I}_d = \sum_{i=1}^{V} s_i\big(\boldsymbol{z}_i\boldsymbol{z}_i^\top - \frac{1}{d}\boldsymbol{I}_d\big).$$

We have

$$\mathbb{E}\Big[\Big(\sum_{i=1}^{V} s_i\big(\boldsymbol{z}_i\boldsymbol{z}_i^\top - \frac{1}{d}\boldsymbol{I}_d\big)\Big)^2\Big] = \frac{1}{d}(1+\frac{1}{d})\|\boldsymbol{S}\|_F^2\boldsymbol{I}_d$$

Moreover, for $p \leq \frac{d}{2}$

$$\mathbb{E}\Big[\|\boldsymbol{z}_i\boldsymbol{z}_i^\top - \frac{1}{d}\boldsymbol{I}_d\|_2^p\Big] \leq \mathbb{E}[\|\boldsymbol{z}_i\|_2^{2p}] \leq 2^p.$$

By Proposition 15, we have $2 \leq p \leq \frac{d}{2}$

$$\mathbb{E}\Big[\|\boldsymbol{M} - \frac{\mathrm{tr}(\boldsymbol{S})}{d}\|_2^p\Big] \leq C\Big(\sqrt{p \vee \log d}\frac{\|\boldsymbol{S}\|_F}{\sqrt{d}} + (p \vee \log d)V^{\frac{1}{p}}\|\boldsymbol{S}\|_2\Big).$$

For $p = \frac{1}{e^2C^2}\log^2 V$, we have the result.

$\square$

**Proposition 5.** *Let $\boldsymbol{S} \in \mathbb{R}^{V \times V}$ be a square matrix and let $\boldsymbol{M} = \boldsymbol{Z}_{\mathrm{in}}\boldsymbol{S}\boldsymbol{Z}_{\mathrm{in}}^\top$. For $\mathrm{poly}(d) \gg V \gg d$, We have*

$$\mathbb{P}\big[|\mathrm{tr}(\boldsymbol{M}) - \mathrm{tr}(\boldsymbol{S})| \geq \log^2 V\frac{\|\boldsymbol{S}\|_F}{\sqrt{d}}\big] \leq \exp(-c\log^2 V).$$

*Proof.* Without loss of generality, we can assume that $\boldsymbol{S}$ is diagonal, i.e., $\boldsymbol{S} = \mathrm{diag}(s_1, \cdots, s_V)$. We have

$$\mathrm{tr}(\boldsymbol{M}) - \mathrm{tr}(\boldsymbol{S}) = \sum_{i=1}^{V} s_i\big(\|\boldsymbol{z}_i\|_2^2 - 1\big).$$

We have

$$\mathbb{E}\big[\exp(\lambda s_i\big(\|\boldsymbol{z}_i\|_2^2 - 1)\big)\big] \leq \exp\Big(\frac{4\lambda^2 s_i^2}{d}\Big), \ |\lambda| \leq \frac{d}{4|s_i|}.$$

Then,

$$\mathbb{E}\big[\exp(\lambda\big(\mathrm{tr}(\boldsymbol{M}) - \mathrm{tr}(\boldsymbol{S})\big))\big] \leq \exp\Big(\frac{4\lambda^2\|\boldsymbol{S}\|_F^2}{d}\Big), \ |\lambda| \leq \frac{d}{4\|\boldsymbol{S}\|_2}$$

We have

$$\mathbb{P}\Big[|\mathrm{tr}(\boldsymbol{M}) - \mathrm{tr}(\boldsymbol{S})| \geq \log^2 V\frac{\|\boldsymbol{S}\|_F}{\sqrt{d}}\Big] \leq \exp(-c\log^2 V).$$

$\square$

**Proposition 6.** *Let $\boldsymbol{S} \in \mathbb{R}^{V \times V}$ be a square matrix. For $\boldsymbol{u}, \boldsymbol{v} \in S^{d-1}$ and $\boldsymbol{M} = \boldsymbol{Z}_{\mathrm{in}}\boldsymbol{S}\boldsymbol{Z}_{\mathrm{in}}^\top$, we have*

$$\mathbb{P}\Big[\big|\big(\boldsymbol{v}^\top\boldsymbol{M}\boldsymbol{u} - \frac{\mathrm{tr}(\boldsymbol{S})}{d}\boldsymbol{v}^\top\boldsymbol{u}\big)\big| \geq \frac{\|\boldsymbol{u}\|_2\|\boldsymbol{v}\|_2}{d}\max\big\{\|\mathrm{sym}(\boldsymbol{S})\|_F t, \|\mathrm{sym}(\boldsymbol{S})\|_2 t^2\big\}\Big]$$
$$\leq 2\exp(-ct^2).$$

*Proof.* Consider $\boldsymbol{g} = \sqrt{d}\mathrm{vec}(\boldsymbol{Z})$, where $\boldsymbol{g} \sim \mathcal{N}(0, \boldsymbol{I}_{dV})$. We have

$$\boldsymbol{v}^\top\boldsymbol{M}\boldsymbol{u} = \frac{1}{d}\boldsymbol{g}^\top(\boldsymbol{u}\boldsymbol{v}^\top) \otimes \boldsymbol{S}\boldsymbol{g} = \frac{1}{d}\boldsymbol{g}^\top\mathrm{sym}(\boldsymbol{u}\boldsymbol{v}^\top) \otimes \mathrm{sym}(\boldsymbol{S})\boldsymbol{g}$$

By using Proposition 11, we have

$$\mathbb{E}[\boldsymbol{g}^\top\mathrm{sym}(\boldsymbol{u}\boldsymbol{v}^\top) \otimes \mathrm{sym}(\boldsymbol{S})\boldsymbol{g}] = \mathrm{tr}(\boldsymbol{S})\boldsymbol{u}^\top\boldsymbol{v}.$$

Moreover,

$$\big(\boldsymbol{g}^\top\mathrm{sym}(\boldsymbol{u}\boldsymbol{v}^\top) \otimes \mathrm{sym}(\boldsymbol{S})\boldsymbol{g} - \mathrm{tr}(\boldsymbol{S})\boldsymbol{u}^\top\boldsymbol{v}\big) =_d \sum_{i=1}^{dV} \lambda_i(g_i^2 - 1)$$

where $g_i \sim N(0, 1)$. By using the subexponential concentration, we have the result. $\square$

**Proposition 7.** *For $\boldsymbol{u}, \boldsymbol{v} \in \mathbb{R}^V$, we have*

$$\mathbb{P}\Big[\Big|\boldsymbol{v}^\top \boldsymbol{Z}_{\text{in}}^\top \boldsymbol{Z}_{\text{in}} \boldsymbol{Z}_{\text{in}}^\top \boldsymbol{Z}_{\text{in}} \boldsymbol{u} - \boldsymbol{u}^\top \boldsymbol{v}\Big(1 + \frac{V-1}{d}\Big)\Big| \geq C \|\boldsymbol{u}\|_2 \|\boldsymbol{v}\|_2 \log V\Big(\frac{\sqrt{V}}{d} + \frac{V}{d^{3/2}}\Big)\Big]$$
$$\leq 10 \exp(-c \log^2 V).$$

*Proof.* Without loss of generality, we assume that $\boldsymbol{u}$ and $\boldsymbol{v}$ have a unit norm. Let

$$\boldsymbol{v}_\perp := \frac{1}{\sqrt{1 - (\boldsymbol{u}^\top \boldsymbol{v})^2}} (\boldsymbol{I}_V - \boldsymbol{v}\boldsymbol{v}^\top)\boldsymbol{u}.$$

We have

$$\boldsymbol{v}^\top \boldsymbol{Z}_{\text{in}}^\top \boldsymbol{Z}_{\text{in}} \boldsymbol{Z}_{\text{in}}^\top \boldsymbol{Z}_{\text{in}} \boldsymbol{u} = (\boldsymbol{u}^\top \boldsymbol{v}) \boldsymbol{v}^\top \boldsymbol{Z}_{\text{in}}^\top \boldsymbol{Z}_{\text{in}} \boldsymbol{Z}_{\text{in}}^\top \boldsymbol{Z}_{\text{in}} \boldsymbol{v} + \sqrt{1 - (\boldsymbol{u}^\top \boldsymbol{v})^2}\, \boldsymbol{v}^\top \boldsymbol{Z}_{\text{in}}^\top \boldsymbol{Z}_{\text{in}} \boldsymbol{Z}_{\text{in}}^\top \boldsymbol{Z}_{\text{in}} \boldsymbol{v}_\perp.$$

Without loss of generality, we consider $\boldsymbol{v} = \boldsymbol{e}_1$ and $\boldsymbol{v}_\perp = \boldsymbol{e}_2$. For the second term, we write $\boldsymbol{z}_i := \boldsymbol{Z}_{\text{in}} \boldsymbol{e}_i$ and let $\tilde{\boldsymbol{Z}} := \{\boldsymbol{z}_i\}_{i=3}^V$ and $\boldsymbol{g} = \sqrt{d}\mathrm{vec}(\tilde{\boldsymbol{Z}})$.

$$\boldsymbol{e}_1^\top \boldsymbol{Z}_{\text{in}}^\top \boldsymbol{Z}_{\text{in}} \boldsymbol{Z}_{\text{in}}^\top \boldsymbol{Z}_{\text{in}} \boldsymbol{e}_2 = (\|\boldsymbol{z}_1\|_2^2 + \|\boldsymbol{z}_2\|_2^2)\boldsymbol{z}_1^\top \boldsymbol{z}_2 + \boldsymbol{z}_1^\top \tilde{\boldsymbol{Z}}\tilde{\boldsymbol{Z}}^\top \boldsymbol{z}_2$$
$$= (\|\boldsymbol{z}_1\|_2^2 + \|\boldsymbol{z}_2\|_2^2)\boldsymbol{z}_1^\top \boldsymbol{z}_2 + \frac{1}{d}\boldsymbol{g}^\top \mathrm{sym}(\boldsymbol{z}_1 \boldsymbol{z}_2^\top) \otimes \boldsymbol{I}_{V-2} \boldsymbol{g}.$$

We have

- By Lemma 4, and Proposition 6

$$\mathbb{P}\Big[\big|\|\boldsymbol{z}_1\|_2^2 - 1\big| \leq \frac{5 \log V}{\sqrt{d}} \text{ and } \big|\|\boldsymbol{z}_2\|_2^2 - 1\big| \leq \frac{5 \log V}{\sqrt{d}} \text{ and } |\boldsymbol{z}_1^\top \boldsymbol{z}_2| \leq \frac{\log V}{\sqrt{d}}\Big]$$
$$\leq 1 - 6 \exp(-c \log^2 V).$$

- By Proposition 11, we have

  - $\|\mathrm{sym}(\boldsymbol{z}_1 \boldsymbol{z}_2^\top) \otimes \boldsymbol{I}_{V-2}\|_2 \leq \|\boldsymbol{z}_1\|_2 \|\boldsymbol{z}_2\|_2$
  - $\|\mathrm{sym}(\boldsymbol{z}_1 \boldsymbol{z}_2^\top) \otimes \boldsymbol{I}_{V-2}\|_F \leq \sqrt{V} \|\boldsymbol{z}_1\|_2 \|\boldsymbol{z}_2\|_2$
  - $\mathrm{tr}\big(\mathrm{sym}(\boldsymbol{z}_1 \boldsymbol{z}_2^\top) \otimes \boldsymbol{I}_{V-2}\big) = (V-2)\boldsymbol{z}_1^\top \boldsymbol{z}_2$.

  Therefore, by Lemma 4, we have

$$\mathbb{P}\Big[\Big|\frac{1}{d}\boldsymbol{g}^\top \mathrm{sym}(\boldsymbol{z}_1 \boldsymbol{z}_2^\top) \otimes \boldsymbol{I}_{V-2} \boldsymbol{g} - \frac{(V-2)}{d}\boldsymbol{z}_1^\top \boldsymbol{z}_2\Big| \leq 2\|\boldsymbol{z}_1\|_2 \|\boldsymbol{z}_2\|_2 \Big(\frac{\log V}{d}\sqrt{V} + \frac{\log^2 V}{d}\Big)\Big]$$
$$\leq 1 - 2 \exp(-c \log^2 V).$$

By union bound of the precious two items, we have

$$\mathbb{P}\Big[\Big|\boldsymbol{e}_1^\top \boldsymbol{Z}_{\text{in}}^\top \boldsymbol{Z}_{\text{in}} \boldsymbol{Z}_{\text{in}}^\top \boldsymbol{Z}_{\text{in}} \boldsymbol{e}_2\Big| \leq 2 \log V\Big(\frac{V}{d^{3/2}} + \frac{\sqrt{V}}{d}\Big)\Big] \geq 1 - 8 \exp(-c \log^2 V). \tag{F.1}$$

Next, we redefine the notation: $\tilde{\boldsymbol{Z}} := \{\boldsymbol{z}_i\}_{i=2}^V$. We write

$$\boldsymbol{z}_1^\top \boldsymbol{Z}_{\text{in}} \boldsymbol{Z}_{\text{in}}^\top \boldsymbol{z}_1 - 1 - \frac{V-1}{d} = \|\boldsymbol{z}_1\|_2^4 - 1 + \boldsymbol{z}_1^\top \Big(\tilde{\boldsymbol{Z}}\tilde{\boldsymbol{Z}}^\top - \frac{V-1}{d}\boldsymbol{I}_d\Big)\boldsymbol{z}_1 - \frac{V-1}{d}(\|\boldsymbol{z}_1\|_2^2 - 1)$$

By Proposition bla, we have

$$\mathbb{P}\Big[\boldsymbol{z}_1^\top \Big(\tilde{\boldsymbol{Z}}\tilde{\boldsymbol{Z}}^\top - \frac{V-1}{d}\boldsymbol{I}_d\Big)\boldsymbol{z}_1 \leq \log V \|\boldsymbol{z}_1\|_2^2 \frac{\sqrt{V}}{d}\Big] \leq 1 - 2 \exp(-c \log^2 V)$$

By using the first item above, we have

$$\mathbb{P}\Big[\Big|\boldsymbol{z}_1^\top \boldsymbol{Z}_{\text{in}} \boldsymbol{Z}_{\text{in}}^\top \boldsymbol{z}_1 - 1 - \frac{V-1}{d}\Big| \geq 6 \log V\Big(\frac{\sqrt{V}}{d} + \frac{V}{d^{3/2}}\Big)\Big] \leq 1 - 2 \exp(-c \log^2 V). \tag{F.2}$$

The result follows (F.1) and (F.2). $\qquad\square$

### F.2 MULTINOMIAL DISTRIBUTION AND RELATED STATEMENTS

**Lemma 5.** *Let* $(n_1, \cdots, n_V) \in \mathrm{Mult}\big(N; (p_1, \cdots, p_V)\big)$. *For* $\boldsymbol{t} \in \mathbb{R}^V$,

$$\mathbb{E}\Big[\exp\Big(\sum_{i=1}^{V} t_i n_i\Big)\Big] = \Big(\sum_{i=1}^{V} p_i e^{t_i}\Big)^N.$$

*Then, if* $p_i = \frac{1}{V}$, $i \in [V]$,

- *We have*

  - $\mathbb{E}\Big[\prod_{i=1}^{V} n_i(n_i-1)\cdots(n_i - k_i + 1)\Big] = \frac{N(N-1)\cdots(N-K+1)}{V^K}$, *where* $K := \sum_{i=1}^{V} k_i$.

- *By the previous item, we can write*

  - $\mathbb{E}\big[n_i^2\big] = \frac{N}{V} + \frac{N(N-1)}{V^2}$
  - $\mathbb{E}\big[\big(\frac{n_i}{N} - \frac{1}{V}\big)^2 n_i\big] = \frac{(V-1)(N+V-2)}{NV^3}$.
  - $\mathbb{E}\big[n_i^3\big] = \frac{N}{V} + \frac{3N(N-1)}{V^2} + \frac{N(N-1)(N-2)}{V^3}$
  - $\mathbb{E}\big[n_i^4\big] = \frac{N}{V} + \frac{7N(N-1)}{V^2} + \frac{6N(N-1)(N-2)}{V^3} + \frac{N(N-1)(N-2)(N-3)}{V^4}$
  - *For* $i \neq i'$, $\mathbb{E}\big[n_i^2 n_{i'}^2\big] = \frac{N(N-1)}{V^2} + \frac{2N(N-1)(N-2)}{V^3} + \frac{N(N-1)(N-2)(N-3)}{V^4}$.
  - $\mathbb{E}\big[\big(\sum_{i=1}^{V} n_i^2\big)^2\big] = N^2 + \frac{2(N+1)N(N-1)}{V} + \frac{(N+1)N(N-1)(N-2)}{V^2}$

*Proof.* Let $\boldsymbol{x}_j$ sampled from $\{\boldsymbol{e}_1, \cdots, \boldsymbol{e}_V\}$ with $(p_1, \cdots, p_V)$. We have $n_i = \sum_{j=1}^{N} \boldsymbol{e}_i^\top \boldsymbol{x}_i$. We have

$$\mathbb{E}\Big[\exp\Big(\sum_{i=1}^{V} t_i n_i\Big)\Big] = \mathbb{E}\Big[\exp\Big(\sum_{j=1}^{N} \langle \boldsymbol{t}, \boldsymbol{x}_j \rangle\Big)\Big] = \Big(\mathbb{E}\big[\exp\big(\langle \boldsymbol{t}, \boldsymbol{x}_1 \rangle\big)\big]\Big)^N = \Big(\sum_{i=1}^{V} p_i e^{t_i}\Big)^N.$$

The later statements can be derived by using $z_i = e^{t_i}$ and taking derivatives of both sides with respect $(z_1, \cdots, z_V)$. $\qquad\square$

**Proposition 8.** *Let* $\boldsymbol{n} := (n_1, \cdots, n_V) \sim \mathrm{Mult}\big(L, \frac{1}{V}\mathbb{1}_V\big)$ *and* $\boldsymbol{S} \in \mathbb{R}^{V \times V}$ *be a symmetric matrix. The following statements hold:*

- *We have*

  - $\mathbb{E}[\mathrm{diag}(\boldsymbol{n})\boldsymbol{S}\,\mathrm{diag}(\boldsymbol{n})] = L\,\mathbb{E}[\boldsymbol{x}_1^\top \boldsymbol{S}\boldsymbol{x}_1 \boldsymbol{x}_1 \boldsymbol{x}_1^\top] + \frac{L(L-1)}{V^2}\boldsymbol{S}$
  - $\mathbb{E}[\mathrm{diag}(\boldsymbol{n} - \frac{L}{V}\mathbb{1}_V)\boldsymbol{S}\,\mathrm{diag}(\boldsymbol{n} - \frac{L}{V}\mathbb{1}_V)] = L\,\mathbb{E}[\boldsymbol{x}_1^\top \boldsymbol{S}\boldsymbol{x}_1 \boldsymbol{x}_1 \boldsymbol{x}_1^\top] - \frac{L}{V^2}\boldsymbol{S}$.

- *We have*

  - $\mathbb{E}[\boldsymbol{n}\boldsymbol{n}^\top \boldsymbol{S}\boldsymbol{n}] = \frac{2L(L-1)}{V^2}\boldsymbol{S}\mathbb{1}_V + L\,\mathbb{E}\big[\boldsymbol{x}_1 \boldsymbol{x}_1^\top \boldsymbol{S}\boldsymbol{x}_1\big] + \Big(\frac{L(L-1)}{V^2}\mathrm{tr}(\boldsymbol{S}) + \frac{L(L-1)(L-2)}{V^3}\mathbb{1}_V^\top \boldsymbol{S}\mathbb{1}_V\Big)\mathbb{1}_V$.

- *We have*

$$\mathbb{E}\Big[\big((\boldsymbol{n} - \tfrac{L}{V}\mathbb{1}_V)^\top \boldsymbol{S}(\boldsymbol{n} - \tfrac{L}{V}\mathbb{1}_V)\big)^2\Big] = \frac{L}{V}\Big\|\mathrm{diag}(\boldsymbol{S}) - \frac{2}{V}\boldsymbol{S}\mathbb{1}_V + \frac{1}{V^2}\big(\mathbb{1}_V^\top \boldsymbol{S}\mathbb{1}_V\big)\mathbb{1}_V\Big\|_2^2$$
$$+ \frac{L(L-1)}{V^2}\mathrm{tr}\Big(\big(\boldsymbol{I}_V - \tfrac{1}{V}\mathbb{1}_V\mathbb{1}_V^\top\big)\boldsymbol{S}\Big)^2$$
$$+ \frac{L(L-1)}{V^2}\mathrm{tr}\Big(\big(\boldsymbol{I}_V - \tfrac{1}{V}\mathbb{1}_V\mathbb{1}_V^\top\big)\boldsymbol{S}\big(\boldsymbol{I}_V - \tfrac{1}{V}\mathbb{1}_V\mathbb{1}_V^\top\big)\boldsymbol{S}\Big)$$

*Proof.* For the first item, we observe that

$$\boldsymbol{e}_j^\top \mathbb{E}[\mathrm{diag}(\boldsymbol{n})\boldsymbol{S}\,\mathrm{diag}(\boldsymbol{n})]\boldsymbol{e}_i = \mathbb{E}[n_j n_i]\boldsymbol{S}_{ij} = \Big(\frac{L}{V}\delta_{ij} + \frac{L(L-1)}{V^2}\Big)\boldsymbol{S}_{ij},$$

from which the first equation follows. For the second equation,

$$\boldsymbol{e}_j^\top \mathbb{E}[\mathrm{diag}(\boldsymbol{n} - \tfrac{L}{V}\mathbb{1}_V)\boldsymbol{S}\mathrm{diag}(\boldsymbol{n} - \tfrac{L}{V}\mathbb{1}_V)]\boldsymbol{e}_i = \mathbb{E}[(n_j - \tfrac{L}{V})(n_i - \tfrac{L}{V})]\boldsymbol{S}_{ij} = \Big(\tfrac{L}{V}\delta_{ij} - \tfrac{L}{V^2}\Big)\boldsymbol{S}_{ij}.$$

For the second item, we have

$$(\mathbb{E}[\boldsymbol{n}\boldsymbol{n}^\top \boldsymbol{S}\boldsymbol{n}])_i = \sum_{jk} \boldsymbol{S}_{jk}\,\mathbb{E}[n_i n_j n_k]$$

$$= \frac{L(L-1)(L-2)}{V^3}\Big(\sum_{i\neq j\neq k}\boldsymbol{S}_{jk}\Big) + \Big(\frac{L(L-1)(L-2)}{V^3} + \frac{L(L-1)}{V^2}\Big)\Big(2\sum_{i\neq k}\boldsymbol{S}_{ik} + \sum_{i\neq k}\boldsymbol{S}_{kk}\Big)$$

$$+ \Big(\frac{L}{V} + \frac{3L(L-1)}{V^2} + \frac{L(L-1)(L-2)}{V^3}\Big)\boldsymbol{S}_{ii}$$

$$= \frac{L}{V}\boldsymbol{S}_{ii} + \frac{L(L-1)}{V}\mathrm{tr}(\boldsymbol{S}) + \frac{2L(L-1)}{V}\sum_k \boldsymbol{S}_{ik} + \frac{L(L-1)(L-2)}{V^3}\Big(\sum_{jk}\boldsymbol{S}_{jk}\Big).$$

For the third item, we have $\big(\boldsymbol{n} - \tfrac{L}{V}\mathbb{1}_V\big) = \sum_{i=1}^L(\boldsymbol{x}_i - \tfrac{1}{V}\mathbb{1}_V)$ in distribution. For notational convenience, let

$$s_{ij} := (\boldsymbol{x}_i - \frac{1}{V}\mathbb{1}_V)^\top \boldsymbol{S}(\boldsymbol{x}_j - \frac{1}{V}\mathbb{1}_V).$$

Then,

$$\mathbb{E}\Big[\Big((\boldsymbol{n} - \frac{L}{V}\mathbb{1}_V)^\top \boldsymbol{S}(\boldsymbol{n} - \frac{L}{V}\mathbb{1}_V)\Big)^2\Big] = \sum_{i,j,k,l=1}^L \mathbb{E}[s_{ij}s_{kl}]$$

By independence, only $(i,j,k,l)$ where each index occur even times contribute. The possible cases are:

- All four indices equal ($i = j = k = l$): There are $L$ many terms here with contribution

$$\mathbb{E}[s_{ii}^2] = \frac{1}{V}\Big\|\mathrm{diag}\Big((\boldsymbol{I}_V - \frac{1}{V}\mathbb{1}_V\mathbb{1}_V^\top)\boldsymbol{S}(\boldsymbol{I}_V - \frac{1}{V}\mathbb{1}_V\mathbb{1}_V^\top)\Big)\Big\|_2^2$$

$$= \frac{1}{V}\Big\|\mathrm{diag}(\boldsymbol{S}) - \frac{2}{V}\boldsymbol{S}\mathbb{1}_V + \frac{1}{V^2}(\mathbb{1}_V^\top \boldsymbol{S}\mathbb{1}_V)\mathbb{1}_V\Big\|_2^2.$$

- Two distinct indices, both pairs diagonal ($i = j$ and $k = l$ and $i \neq k$): There are $L(L-1)$ many terms here with contribution

$$\mathbb{E}[s_{ii}s_{kk}] = \mathbb{E}[s_{ii}]^2 = \frac{1}{V^2}\mathrm{tr}\Big((\boldsymbol{I}_V - \frac{1}{V}\mathbb{1}_V\mathbb{1}_V^\top)\boldsymbol{S}\Big)^2$$

- Two distinct indices, paired off-diagonal: ($i = k$ and $j = l$ and $i \neq j$): There are $L(L-1)$ many terms here with contribution

$$\mathbb{E}[s_{ij}^2] = \mathrm{tr}\Big(\mathbb{E}[(\boldsymbol{x}_1 - \frac{1}{V}\mathbb{1}_V)(\boldsymbol{x}_1 - \frac{1}{V}\mathbb{1}_V)^\top \boldsymbol{S}(\boldsymbol{x}_2 - \frac{1}{V}\mathbb{1}_V)(\boldsymbol{x}_2 - \frac{1}{V}\mathbb{1}_V)^\top]\boldsymbol{S}\Big)$$

$$= \frac{1}{V^2}\mathrm{tr}\Big((\boldsymbol{I}_V - \frac{1}{V}\mathbb{1}_V\mathbb{1}_V^\top)\boldsymbol{S}(\boldsymbol{I}_V - \frac{1}{V}\mathbb{1}_V\mathbb{1}_V^\top)\boldsymbol{S}\Big).$$

□

**Proposition 9.** *Let $V^3 \gg L$. There exists a universal $C > 0$ such that the following holds:*

- *Let $m_{ij} := (1 - \frac{1}{V})\mathbb{1}_{i=j} + \frac{L}{V}$. For $K > 0$ and $p \geq \log V$,*

  - $\mathbb{E}\Big[\Big|\frac{1}{L}\mathbb{1}_L^\top \boldsymbol{X}_i \boldsymbol{X}_j^\top \mathbb{1}_L - m_{ij}\Big|^p\Big]^{\frac{1}{p}} \leq C\Big(\frac{p^{\frac{3}{2}}}{\sqrt{V}} + \frac{p^2}{L}\Big)$

- $\mathbb{P}\Big[\big|\frac{1}{L}\mathbb{1}_L^\top \boldsymbol{X}_i \boldsymbol{X}_j^\top \mathbb{1}_L - m_{ij}\big| \geq CK^2 \frac{\log^2 V}{\sqrt{V}\wedge L}\Big] \leq \frac{1}{V^K}$

- *For $K > 0$ and $p \geq \log V$,*

  - $\mathbb{E}\Big[\big\|\frac{1}{NL}\sum_{i=1}^N (\boldsymbol{X}_i^\top - \frac{1}{V}\mathbb{1}_V\mathbb{1}_L^\top)\mathbb{1}_L\mathbb{1}_L^\top(\boldsymbol{X}_i^\top - \frac{1}{V}\mathbb{1}_V\mathbb{1}_L^\top)^\top - \frac{1}{V}(\boldsymbol{I} - \frac{1}{V}\mathbb{1}_V\mathbb{1}_V^\top)\big\|_2^p\Big]^{\frac{1}{p}} \leq C\Big(\sqrt{\frac{p}{NV}} +$

    $\frac{p}{N}\big(1+\frac{p^2}{\sqrt{V}\wedge L}\big)\Big)$

  - $\mathbb{P}\Big[\big\|\frac{1}{NL}\sum_{i=1}^N (\boldsymbol{X}_i^\top - \frac{1}{V}\mathbb{1}_V\mathbb{1}_L^\top)\mathbb{1}_L\mathbb{1}_L^\top(\boldsymbol{X}_i^\top - \frac{1}{V}\mathbb{1}_V\mathbb{1}_L^\top)^\top - \frac{1}{V}(\boldsymbol{I} - \frac{1}{V}\mathbb{1}_V\mathbb{1}_V^\top)\big\|_2 > CK\big(\frac{\log^2 V}{\sqrt{NV}} +$

    $\frac{\log^2 V}{N}\big(1+\frac{\log^2 V}{\sqrt{V}\wedge L}\big)\big)\Big] \leq \frac{1}{V^K}.$

*Proof.* Let $\boldsymbol{x}_{il}$ be i.i.d. copies of $\boldsymbol{x}_1$. We note that $\boldsymbol{X}_i\mathbb{1}_L = \sum_{l=1}^L \boldsymbol{x}_{il}$ in distribution. For $i = j$, we have

$$\frac{1}{L}\mathbb{1}_L^\top \boldsymbol{X}_i \boldsymbol{X}_i^\top \mathbb{1}_L = 1 + \frac{2}{L}\sum_{1\leq l<r\leq L}\mathbb{1}_{\boldsymbol{x}_{ir}=\boldsymbol{x}_{il}} = 1 + \frac{(L-1)}{V} + \frac{2}{L}\sum_{l=2}^L\sum_{r=1}^{l-1}\big(\mathbb{1}_{\boldsymbol{x}_{ir}=\boldsymbol{x}_{il}} - \frac{1}{V}\big)$$

Define

$$Y_l := \sum_{r=1}^{l-1}\big(\mathbb{1}_{\boldsymbol{x}_{ir}=\boldsymbol{x}_{il}} - \frac{1}{V}\big) \text{ and } \mathcal{F}_l := \sigma(Y_1,\cdots,Y_l).$$

Given that

$$\sum_{r=1}^{l-1}\mathbb{1}_{\boldsymbol{x}_{ir}=\boldsymbol{x}_{il}}|\boldsymbol{x}_{il} \sim \text{Binomial}(l-1,\frac{1}{V}) \Rightarrow \mathbb{E}[|Y_k|^p]^{\frac{1}{p}} \leq C(\sqrt{p}\sqrt{\frac{L}{V}}+p), \;\; p\geq\log V. \quad \text{(F.3)}$$

where we used Corollary 3. As for the quadratic variation

$$Q_L := \sum_{l=1}^L \mathbb{E}[Y_l^2|\mathcal{F}_{l-1}] = \sum_{l=1}^L \frac{1}{V}\big(\big\|\sum_{r=1}^{l-1}\boldsymbol{x}_{ir}\big\|_2^2 - \frac{(l-1)^2}{V}\big) = \frac{1}{V}\sum_{l=1}^L\big\|\sum_{r=1}^{l-1}\boldsymbol{x}_{ir} - \frac{l-1}{V}\mathbb{1}_V\big\|_2^2.$$

For $p \geq \log V$, by using triangle inequality,

$$\mathbb{E}[|Q_L|^{\frac{p}{2}}]^{\frac{2}{p}} \leq \frac{1}{V}\sum_{l=1}^L \mathbb{E}\Big[\big\|\sum_{r=1}^{l-1}\boldsymbol{x}_{ir} - \frac{l-1}{V}\mathbb{1}_V\big\|_2^p\Big]^{\frac{2}{p}}$$

$$\overset{(a)}{\leq} \frac{1}{V}\sum_{l=1}^V(l-1)\mathbb{E}\Big[\big\|\sum_{r=1}^{l-1}\boldsymbol{x}_{ir}\big\|_p^p\Big]^{\frac{2}{p}} + \sum_{l=V+1}^L \mathbb{E}\Big[\big\|\sum_{r=1}^{l-1}\boldsymbol{x}_{ir} - \frac{l-1}{V}\mathbb{1}_V\big\|_p^p\Big]^{\frac{2}{p}}$$

$$\overset{(b)}{\leq} Cp^2\frac{1}{V}\sum_{l=1}^L l$$

$$= Cp^2\frac{L^2}{V},$$

where we used Hölder's inequality in (a) and Corollary 3 in (b). By using (F.3) and Proposition 15, for $p \geq \log V$, we have

$$\mathbb{E}\Big[\big|\sum_{l=1}^L Y_k\big|^p\Big]^{\frac{1}{p}} \leq C\Big(p\sqrt{p}\frac{L}{\sqrt{V}} + p^2\Big).$$

By using $p = \log V$, we have

$$\mathbb{P}\Big[\big|\frac{1}{L}\sum_{l=1}^L Y_k\big| > CeK^2\frac{\log^2 V}{\sqrt{V}\wedge L}\Big] \leq \frac{1}{V^K}.$$

Hence, we have the $i = j$ case. For $i \neq j$, we have

$$\frac{1}{L}\mathbb{1}_L^\top \boldsymbol{X}_j \boldsymbol{X}_i^\top \mathbb{1}_L = \frac{L}{V} + \frac{1}{L}\sum_{l=1}^{L}\sum_{r=1}^{L}\mathbb{1}_{\boldsymbol{x}_{il}=\boldsymbol{x}_{jr}} - \frac{1}{V}$$

We redefine the martingale difference sequence as

$$Y_l := \sum_{r=1}^{L}\mathbb{1}_{\boldsymbol{x}_{il}=\boldsymbol{x}_{jr}} - \frac{1}{V}.$$

Conditioned on $\boldsymbol{X}_j$, we have $\{Y_1, \cdots, Y_L\}$ are i.i.d. and

$$\mathbb{E}[Y_k|\boldsymbol{X}_j] = 0 \;\text{ and }\; \mathbb{E}[Y_k^p|\boldsymbol{X}_j] = \frac{1}{V}\|(\boldsymbol{X}_j^\top - \frac{1}{V}\mathbb{1}_V\mathbb{1}_L^\top)\mathbb{1}_L\|_p^p$$

By Proposition 15, for $p \geq \log V$, we have

$$\mathbb{E}\Big[\Big|\frac{1}{L}\sum_{l=1}^{L}Y_l\Big|^p\Big]^{\frac{1}{p}} \leq C\Big(\frac{\sqrt{p}}{\sqrt{V}} + \frac{p^{\frac{3}{2}}}{\sqrt{LV}} + \frac{p^2}{L}\Big).$$

By using $p = \log V$, we have

$$\mathbb{P}\Big[\Big|\frac{1}{L}\mathbb{1}_L^\top \boldsymbol{X}_j \boldsymbol{X}_i^\top \mathbb{1}_L - \frac{L}{V}\Big| \geq \frac{CK^2\log^2 V}{\sqrt{V}\vee L}\Big] \leq \frac{1}{V^K}.$$

For the second item, we define

$$\boldsymbol{Y}_i := \frac{1}{L}\big(\boldsymbol{X}_i^\top - \frac{1}{V}\mathbb{1}_V\mathbb{1}_L^\top\big)\mathbb{1}_L\mathbb{1}_L^\top\big(\boldsymbol{X}_i^\top - \frac{1}{V}\mathbb{1}_V\mathbb{1}_L^\top\big)^\top - \frac{1}{V}\big(\boldsymbol{I}_V - \frac{1}{V}\mathbb{1}_V\mathbb{1}_V^\top\big)$$

and $\boldsymbol{Q}_N := N\mathbb{E}[\boldsymbol{Y}_1^2]$. We have

$$\boldsymbol{Q}_N \preceq N\mathbb{E}\Big[\Big\|\frac{1}{\sqrt{L}}\big(\boldsymbol{X}_1^\top - \tfrac{1}{V}\mathbb{1}_V\mathbb{1}_L^\top\big)\mathbb{1}_L\Big\|_2^2\frac{1}{L}\big(\boldsymbol{X}_1^\top - \tfrac{1}{V}\mathbb{1}_V\mathbb{1}_L^\top\big)\mathbb{1}_L\mathbb{1}_L^\top\big(\boldsymbol{X}_1^\top - \tfrac{1}{V}\mathbb{1}_V\mathbb{1}_L^\top\big)^\top\Big]$$

$$= N\mathbb{E}\Big[(1 - \tfrac{1}{V})\frac{1}{L}\big(\boldsymbol{X}_1^\top - \tfrac{1}{V}\mathbb{1}_V\mathbb{1}_L^\top\big)\mathbb{1}_L\mathbb{1}_L^\top\big(\boldsymbol{X}_1^\top - \tfrac{1}{V}\mathbb{1}_V\mathbb{1}_L^\top\big)^\top\Big]$$

$$+ N\mathbb{E}\Big[\Big(\Big\|\frac{1}{\sqrt{L}}\big(\boldsymbol{X}_1^\top - \tfrac{1}{V}\mathbb{1}_V\mathbb{1}_L^\top\big)\mathbb{1}_L\Big\|_2^2 - (1 - \tfrac{1}{V})\Big)$$

$$\times \Big(\frac{1}{L}\big(\boldsymbol{X}_1^\top - \tfrac{1}{V}\mathbb{1}_V\mathbb{1}_L^\top\big)\mathbb{1}_L\mathbb{1}_L^\top\big(\boldsymbol{X}_1^\top - \tfrac{1}{V}\mathbb{1}_V\mathbb{1}_L^\top\big)^\top - \frac{1}{V}\big(\boldsymbol{I}_V - \tfrac{1}{V}\mathbb{1}_V\mathbb{1}_V^\top\big)\Big)\Big]$$

$$\overset{(c)}{\preceq} \frac{CN}{V}\boldsymbol{I}_V + \frac{1}{2}\boldsymbol{Q}_N,$$

where we use Proposition 14 in (c). Therefore, we have $\|\boldsymbol{Q}_N\|_2 \leq \frac{CN}{V}$. Moreover, we observe that

$$\|\boldsymbol{Y}_i\|_2 \leq \frac{1}{V} + \Big\|\frac{1}{\sqrt{L}}\big(\boldsymbol{X}_i^\top - \frac{1}{V}\mathbb{1}_V\mathbb{1}_L^\top\big)\mathbb{1}_L\Big\|_2^2.$$

By using the first item,

$$\mathbb{E}[\|\boldsymbol{Y}_i\|_2^p]^{\frac{1}{p}} \leq \frac{1}{V} + \mathbb{E}\Big[\Big\|\frac{1}{\sqrt{L}}\big(\boldsymbol{X}_i^\top - \frac{1}{V}\mathbb{1}_V\mathbb{1}_L^\top\big)\mathbb{1}_L\Big\|_2^{2p}\Big]^{\frac{1}{p}} \leq 1 + C\Big(\frac{p^{\frac{3}{2}}}{\sqrt{V}} + \frac{p^2}{L}\Big).$$

Therefore, by using Proposition 15, we have

$$\mathbb{E}\Big[\Big\|\frac{1}{N}\sum_{i=1}^{N}\boldsymbol{Y}_i\Big\|_2^p\Big] \leq C\Bigg(\sqrt{p\vee \log V}\sqrt{\frac{1}{NV}} + (p\vee \log V)N^{\frac{1}{p}-1}\Big(1 + \frac{p^{\frac{3}{2}}}{\sqrt{V}} + \frac{p^2}{L}\Big)\Bigg).$$

By using $p = \log V$, we have

$$\mathbb{P}\Big[\Big\|\frac{1}{N}\sum_{i=1}^{N}\boldsymbol{Y}_i\Big\|_2 > CK\log^2 V\Big(\frac{1}{\sqrt{NV}} + \frac{1}{N}\Big(1 + \frac{\log^2 V}{\sqrt{V}\wedge L}\Big)\Big)\Big] \leq \frac{1}{V^K}.$$

$\square$

**Proposition 10.** *We consider $S_1$, $S_2$ and $S_3$ defined in (B.5), (B.6) and (B.7) in the regime $V^3 \gg N \gg V$ and $L \asymp V^\varepsilon$, $\varepsilon \in (0,1)$. For any $K > 0$ and $V \geq \Omega_{K,\varepsilon}(1)$, the following holds:*

*1. We have*

$$\mathbb{P}\left[\left|\operatorname{tr}(\boldsymbol{S}_1) - \frac{1-1/V}{L^2}\left(\frac{1}{V} + (1-\frac{2}{V})\frac{1}{N}\right)\right| > CK^2\frac{\log^2 V}{L^2 N\sqrt{V}} \ \ or \ \ \|\boldsymbol{S}_1\|_2 > \frac{e^2}{L^2 V^2}\right] \leq \frac{2}{V^K}.$$

*2. We have*

$$\mathbb{P}\left[\left|\operatorname{tr}(\boldsymbol{S}_2) - (1-\frac{1}{V})^2\frac{L-1}{L^2 N}\right| > C\frac{K^{\frac{3}{2}}\log^3 V}{LNV} \ \ or \ \ \|\boldsymbol{S}_2\|_2 > C\frac{K^{\frac{3}{2}}\log^2 V}{NLV}\right] \leq \frac{4}{V^K}.$$

*3. We have*

$$\mathbb{P}\left[\frac{-CK^2\log^2 V}{N\sqrt{V}}\frac{1}{V^2 L^2}\mathbb{1}_V\mathbb{1}_V^\top \preceq \boldsymbol{S}_3 - \frac{1}{N}\frac{1}{V^2 L^2}\mathbb{1}_V\mathbb{1}_V^\top \preceq \frac{CK^2\log^2 V}{N\sqrt{V}}\frac{1}{V^2 L^2}\mathbb{1}_V\mathbb{1}_V^\top\right] \leq \frac{1}{V^K}.$$

*Proof.* We define $n_i := |\{j \leq N \mid \boldsymbol{x}_j = \boldsymbol{e}_i\}|$. We observe that

$$\operatorname{tr}(\boldsymbol{S}_1) = (1-\frac{2}{V})\frac{1}{L^2 N^2}\sum_{i=1}^V n_i^2 + \frac{1}{V^2 L^2} \ \ \text{and} \ \ \|\boldsymbol{S}_1\|_2 \leq \sup_{i \leq N}\frac{n_i^2}{L^2 N^2}.$$

By using Proposition 9 and Corollary 3, we have the first item. For the second item, we write

$$\boldsymbol{S}_2 = \frac{(1-\frac{1}{V})}{L^2 N^2}\sum_{j=1}^N(\boldsymbol{N}_j^\top - \frac{1}{V}\mathbb{1}_V\mathbb{1}_{L-1}^\top)\mathbb{1}_{L-1}\mathbb{1}_{L-1}^\top(\boldsymbol{N}_j^\top - \frac{1}{V}\mathbb{1}_V\mathbb{1}_{L-1}^\top)^\top$$

$$+ \frac{2}{L^2 N^2}\sum_{j<k}\left(\mathbb{1}_{\boldsymbol{x}_j=\boldsymbol{x}_k} - \frac{1}{V}\right)\operatorname{sym}\left((\boldsymbol{N}_j^\top - \frac{1}{V}\mathbb{1}_V\mathbb{1}_{L-1}^\top)\mathbb{1}_{L-1}\mathbb{1}_{L-1}^\top(\boldsymbol{N}_k^\top - \frac{1}{V}\mathbb{1}_V\mathbb{1}_{L-1}^\top)^\top\right)$$

$$=: \boldsymbol{S}_{21} + \boldsymbol{S}_{22}$$

We will analyze $\boldsymbol{S}_{21}$ and $\boldsymbol{S}_{22}$ separately.

**Bounds for $\boldsymbol{S}_{21}$:**  We have

$$\operatorname{tr}(\boldsymbol{S}_{21}) - (1-\frac{1}{V})^2\frac{L-1}{L^2 N}$$

$$= (1-\frac{1}{V})\frac{L-1}{L^2 N^2}\sum_{j=1}^N\underbrace{\|\frac{1}{\sqrt{L-1}}(\boldsymbol{N}_j^\top - \frac{1}{V}\mathbb{1}_V\mathbb{1}_{L-1}^\top)\mathbb{1}_{L-1}\|_2^2 - (1-\frac{1}{V})}_{:=Y_{1,j}}.$$

We have $\mathbb{E}[Y_{1,j}^2] \leq \frac{2}{V}$ and by Proposition 9,

$$\mathbb{E}[|Y_{1,j}|^p]^{\frac{1}{p}} \leq \frac{Cp^2}{\sqrt{V}\wedge L}.$$

Therefore, by Proposition 15,

$$\mathbb{E}\left[\left|\operatorname{tr}(\boldsymbol{S}_{21}) - (1-\frac{1}{V})^2\frac{L-1}{L^2 N}\right|^p\right]^{\frac{1}{p}} \leq \frac{C}{LN^2}\left(\sqrt{\frac{pN}{V}} + pN^{\frac{1}{p}}\frac{p^2}{\sqrt{V}\wedge L}\right)$$

By using $p = \log V$, we have

$$\mathbb{P}\left[\left|\operatorname{tr}(\boldsymbol{S}_{21}) - (1-\frac{1}{V})^2\frac{L-1}{L^2 N}\right| > C\frac{K\log^3 V}{LN\sqrt{NV}}\right] \leq \frac{1}{V^K}. \tag{F.4}$$

Moreover, by Proposition 9, we have

$$\mathbb{P}\left[\left\|\boldsymbol{S}_{21} - (1-\frac{1}{V})\frac{L-1}{L^2 N}\frac{1}{V}(\boldsymbol{I}_V - \mathbb{1}_V\mathbb{1}_V^\top)\right\|_2 > C\frac{K\log^2 V}{LN}\left(\frac{1}{\sqrt{NV}} + \frac{1}{N}\left(1 + \frac{\log^2 V}{\sqrt{V}\wedge L}\right)\right)\right]$$

$$\leq \frac{1}{V^K}. \tag{F.5}$$

**Bounds for $S_{22}$:** We write

$$\text{tr}(S_{22}) = \frac{2}{L^2 N^2} \sum_{k=2}^{N} \sum_{j=1}^{k-1} \left( \mathbb{1}_{x_j = x_k} - \frac{1}{V} \right) \mathbb{1}_{L-1}^\top (N_j^\top - \frac{1}{V} \mathbb{1}_V \mathbb{1}_{L-1}^\top)^\top (N_k^\top - \frac{1}{V} \mathbb{1}_V \mathbb{1}_{L-1}^\top) \mathbb{1}_{L-1}$$

$$=: \sum_{k=2}^{N} Y_{2,k}.$$

Let $\mathcal{F}_k := \sigma(N_{1:k})$ and $Y_{2,1} = 0$. We have

$$\mathbb{E}[Y_{2,k}^2 | \mathcal{F}_{k-1}] = \frac{4(L-1)}{L^4 N^4} \frac{1}{V} \mathbb{E} \left[ \left\| \sum_{j=1}^{k-1} \left( \mathbb{1}_{x_j = x_k} - \frac{1}{V} \right)(N_j^\top - \frac{1}{V} \mathbb{1}_V \mathbb{1}_{L-1}^\top) \mathbb{1}_{L-1} \right\|_2^2 \Big| \mathcal{F}_{k-1} \right]$$

$$= (1 - \frac{1}{V}) \frac{4(L-1)}{L^4 N^4} \frac{1}{V^2} \sum_{j=1}^{k-1} \|(N_j^\top - \frac{1}{V} \mathbb{1}_V \mathbb{1}_{L-1}^\top) \mathbb{1}_{L-1}\|_2^2$$

Then,

$$Q_N = \sum_{k=1}^{N} \mathbb{E}[Y_{2,k}^2 | \mathcal{F}_{k-1}] = (1 - \frac{1}{V}) \frac{4(L-1)}{L^4 N^4 V^2} \sum_{k=2}^{N} \sum_{j=1}^{k-1} \|(N_j^\top - \frac{1}{V} \mathbb{1}_V \mathbb{1}_{L-1}^\top) \mathbb{1}_{L-1}\|_2^2$$

$$= (1 - \frac{1}{V}) \frac{4(L-1)}{L^4 N^4 V^2} \sum_{k=1}^{N-1} (N-k) \|(N_k^\top - \frac{1}{V} \mathbb{1}_V \mathbb{1}_{L-1}^\top) \mathbb{1}_{L-1}\|_2^2.$$

Then, for $p \geq \log V$,

$$\mathbb{E}\left[ |Q_N|^{\frac{p}{2}} \right]^{\frac{2}{p}} \leq \frac{5}{L^3 N^3 V^2} \sum_{k=1}^{N} \mathbb{E}\left[ \|N_k^\top \mathbb{1}_{L-1}\|_2^p \right]^{\frac{2}{p}} \overset{(a)}{\leq} \frac{5 L^{1-\frac{2}{p}}}{L^3 N^3 V^2} \sum_{k=1}^{N} \mathbb{E}\left[ \|N_k^\top \mathbb{1}_{L-1}\|_p^p \right]^{\frac{2}{p}}$$

$$\overset{(b)}{\leq} \frac{5 p^2}{L^2 N^2 V^2}, \tag{F.6}$$

where we used Hölder's inequality in (a) and Corollary 3 in (b). By using Proposition 15, we show the following:

- To bound $\mathbb{E}[|Y_{2,k}|^p]^{\frac{1}{p}}$ for $p \geq \log V$, by using the conditional independence of $\{x_j\}_{j=1}^{k-1}$, we write

$$\mathbb{E}[|Y_{2,k}|^p | N_{1:k}, x_k]^{\frac{1}{p}}$$

$$\leq \frac{C}{LN^2} \frac{\sqrt{p}}{\sqrt{V}} \left( \sum_{j=1}^{k-1} \left| \frac{1}{L-1} \left\langle (N_k^\top - \frac{1}{V} \mathbb{1}_V \mathbb{1}_{L-1}^\top) \mathbb{1}_{L-1}, (N_j^\top - \frac{1}{V} \mathbb{1}_V \mathbb{1}_{L-1}^\top) \mathbb{1}_{L-1} \right\rangle \right|^2 \right)^{\frac{1}{2}}$$

$$+ \frac{C p k^{\frac{1}{p}}}{LN^2} \left( \sum_{j=1}^{k-1} \left| \frac{1}{L-1} \left\langle (N_k^\top - \frac{1}{V} \mathbb{1}_V \mathbb{1}_{L-1}^\top) \mathbb{1}_{L-1}, (N_j^\top - \frac{1}{V} \mathbb{1}_V \mathbb{1}_{L-1}^\top) \mathbb{1}_{L-1} \right\rangle \right|^p \right)^{\frac{1}{p}}$$

Therefore,

$$\mathbb{E}[|Y_{2,k}|^p]^{\frac{1}{p}} \leq \frac{C}{LN^2} \left( \frac{\sqrt{p}\sqrt{k}}{\sqrt{V}} + p k^{\frac{2}{p}} \right) \mathbb{E}\left[ \left| \frac{1}{L-1} \left\langle (N_k^\top - \frac{1}{V} \mathbb{1}_V \mathbb{1}_{L-1}^\top) \mathbb{1}_{L-1}, (N_1^\top - \frac{1}{V} \mathbb{1}_V \mathbb{1}_{L-1}^\top) \mathbb{1}_{L-1} \right\rangle \right|^p \right]^{\frac{1}{p}}$$

$$\leq \frac{C}{LN^2} \left( \frac{\sqrt{p}\sqrt{k}}{\sqrt{V}} + p k^{\frac{2}{p}} \right) \mathbb{E}\left[ \left| \frac{1}{L-1} \left\langle N_k^\top \mathbb{1}_{L-1}, N_1^\top \mathbb{1}_{L-1} \right\rangle - \frac{L-1}{V} \right|^p \right]^{\frac{1}{p}}$$

$$\overset{(c)}{\leq} \frac{C p^2}{LN^2} \left( \frac{\sqrt{p}\sqrt{k}}{\sqrt{V}} + p k^{\frac{2}{p}} \right) \frac{1}{\sqrt{V} \wedge L}, \tag{F.7}$$

where we used Proposition 9 in (c).

- Then by using (F.6) and (F.7), we have for $p = \log V$

$$\mathbb{E}[|\mathrm{tr}(\boldsymbol{S}_{22})|^p]^{\frac{1}{p}}$$

$$\leq C\Big(\frac{p^{\frac{3}{2}}}{NLV} + \frac{p^4 N^{\frac{1}{p}}}{LN^{\frac{3}{2}}\sqrt{V}} \frac{1}{\sqrt{V} \wedge L}\Big) \leq \frac{Cp^{\frac{3}{2}}}{NLV}.$$

Then, we have

$$\mathbb{P}\left[|\mathrm{tr}(\boldsymbol{S}_{22})| > \frac{CK^{\frac{3}{2}}\log^{\frac{3}{2}}V}{NLV}\right] \leq \frac{1}{V^K}. \tag{F.8}$$

To bound $\|\boldsymbol{S}_{22}\|_2$, we define

$$\boldsymbol{Y}_k := \sum_{j=1}^{k-1}\big(\mathbb{1}_{\boldsymbol{x}_j = \boldsymbol{x}_k} - \frac{1}{V}\big)\mathrm{sym}\Big((\boldsymbol{N}_j^\top - \frac{1}{V}\mathbb{1}_V\mathbb{1}_{L-1}^\top)\mathbb{1}_{L-1}\mathbb{1}_{L-1}^\top(\boldsymbol{N}_k^\top - \frac{1}{V}\mathbb{1}_V\mathbb{1}_{L-1}^\top)^\top\Big).$$

We have

$$\mathbb{E}[\boldsymbol{Y}_k^2|\mathcal{F}_{k-1}]$$

$$\preceq \frac{2}{V}\sum_{j=1}^{k-1}\mathbb{E}\Big[(\boldsymbol{N}_k^\top - \frac{1}{V}\mathbb{1}_V\mathbb{1}_{L-1}^\top)\mathbb{1}_{L-1}\mathbb{1}_{L-1}^\top(\boldsymbol{N}_k^\top - \frac{1}{V}\mathbb{1}_V\mathbb{1}_{L-1}^\top)^\top$$

$$\times \Big\|(\boldsymbol{N}_j^\top - \frac{1}{V}\mathbb{1}_V\mathbb{1}_{L-1}^\top)\mathbb{1}_{L-1}\Big\|_2^2\Big|\mathcal{F}_{k-1}\Big]$$

$$+ \frac{2}{V}\sum_{j=1}^{k-1}\mathbb{E}\Big[\Big\|(\boldsymbol{N}_k^\top - \frac{1}{V}\mathbb{1}_V\mathbb{1}_{L-1}^\top)\mathbb{1}_{L-1}\Big\|_2^2$$

$$\times (\boldsymbol{N}_j^\top - \frac{1}{V}\mathbb{1}_V\mathbb{1}_{L-1}^\top)\mathbb{1}_{L-1}\mathbb{1}_{L-1}^\top(\boldsymbol{N}_j^\top - \frac{1}{V}\mathbb{1}_V\mathbb{1}_{L-1}^\top)^\top\Big|\mathcal{F}_{k-1}\Big]$$

$$\preceq \frac{2L}{V^2}\sum_{j=1}^{k-1}\Big\|(\boldsymbol{N}_j^\top - \frac{1}{V}\mathbb{1}_V\mathbb{1}_{L-1}^\top)\mathbb{1}_{L-1}\Big\|_2^2\boldsymbol{I}_V$$

$$+ \frac{2L}{V}\sum_{j=1}^{k-1}(\boldsymbol{N}_j^\top - \frac{1}{V}\mathbb{1}_V\mathbb{1}_{L-1}^\top)\mathbb{1}_{L-1}\mathbb{1}_{L-1}^\top(\boldsymbol{N}_j^\top - \frac{1}{V}\mathbb{1}_V\mathbb{1}_{L-1}^\top)^\top$$

Therefore, we have

$$\boldsymbol{Q}_N := \sum_{k=1}^N \mathbb{E}[\boldsymbol{Y}_k^2|\mathcal{F}_{k-1}]$$

$$\preceq \frac{2L}{V^2}\sum_{k=1}^{N-1}(N-k)\Big\|(\boldsymbol{N}_k^\top - \frac{1}{V}\mathbb{1}_V\mathbb{1}_{L-1}^\top)\mathbb{1}_{L-1}\Big\|_2^2\boldsymbol{I}_V$$

$$+ \frac{2L^2 N}{V}\frac{1}{LN}\sum_{k=1}^{N-1}(N-k)(\boldsymbol{N}_k^\top - \frac{1}{V}\mathbb{1}_V\mathbb{1}_{L-1}^\top)\mathbb{1}_{L-1}\mathbb{1}_{L-1}^\top(\boldsymbol{N}_k^\top - \frac{1}{V}\mathbb{1}_V\mathbb{1}_{L-1}^\top)^\top$$

$$\preceq \frac{2NL}{V^2}\sum_{k=1}^{N-1}\Big\|(\boldsymbol{N}_k^\top - \frac{1}{V}\mathbb{1}_V\mathbb{1}_{L-1}^\top)\mathbb{1}_{L-1}\Big\|_2^2\boldsymbol{I}_V$$

$$+ \frac{2L^2 N^2}{V}\frac{1}{LN}\sum_{k=1}^{N-1}(\boldsymbol{N}_k^\top - \frac{1}{V}\mathbb{1}_V\mathbb{1}_{L-1}^\top)\mathbb{1}_{L-1}\mathbb{1}_{L-1}^\top(\boldsymbol{N}_k^\top - \frac{1}{V}\mathbb{1}_V\mathbb{1}_{L-1}^\top)^\top.$$

Then,

$$\mathbb{E}[\|\boldsymbol{Q}_N\|_2^{\frac{p}{2}}]^{\frac{2}{p}} \leq \frac{2NL}{V^2}\mathbb{E}\Big[\Big(\sum_{k=1}^{N-1}\Big\|(\boldsymbol{N}_k^\top - \frac{1}{V}\mathbb{1}_V\mathbb{1}_{L-1}^\top)\mathbb{1}_{L-1}\Big\|_2^2\Big)^{\frac{p}{2}}\Big]^{\frac{2}{p}}$$

$$+ \frac{2L^2N^2}{V} \mathbb{E}\Big[\Big\| \frac{1}{N(L-1)} \sum_{k=1}^{N-1} (\boldsymbol{N}_k^\top - \tfrac{1}{V}\mathbb{1}_V\mathbb{1}_{L-1}^\top)\mathbb{1}_{L-1}\mathbb{1}_{L-1}^\top(\boldsymbol{N}_k^\top - \tfrac{1}{V}\mathbb{1}_V\mathbb{1}_{L-1}^\top)^\top \Big\|_2^{\frac{p}{2}}\Big]^{\frac{2}{p}}$$

$$\leq \frac{2N^2L^2}{V^2} \mathbb{E}\Big[\Big\| \frac{1}{\sqrt{L-1}}(\boldsymbol{N}_1^\top - \tfrac{1}{V}\mathbb{1}_V\mathbb{1}_{L-1}^\top)\mathbb{1}_{L-1}\Big\|_2^p\Big]^{\frac{2}{p}}$$

$$+ \frac{2L^2N^2}{V} \mathbb{E}\Big[\Big\| \frac{1}{N(L-1)} \sum_{k=1}^{N-1} (\boldsymbol{N}_k^\top - \tfrac{1}{V}\mathbb{1}_V\mathbb{1}_{L-1}^\top)\mathbb{1}_{L-1}\mathbb{1}_{L-1}^\top(\boldsymbol{N}_k^\top - \tfrac{1}{V}\mathbb{1}_V\mathbb{1}_{L-1}^\top)^\top \Big\|_2^{\frac{p}{2}}\Big]^{\frac{2}{p}}$$

$$\leq \frac{CN^2L^2}{V^2}\Big(1 + \frac{p^2}{\sqrt{V}\vee L}\Big) + \frac{CL^2N^2}{V}\Big(\frac{1}{V} + \sqrt{\frac{p}{NV}} + \frac{p}{N}\Big(1 + \frac{p^2}{\sqrt{V}\wedge L}\Big)\Big)$$

$$\leq \frac{CN^2L^2}{V^2}\Big(1 + \frac{p}{N/V} + \frac{p^2}{\sqrt{V}\vee L}\Big).$$

To bound $\mathbb{E}[\|\boldsymbol{Y}_k\|_2^p]$, we observe that

- We have

$$\mathbb{E}\Big[\Big((\mathbb{1}_{\boldsymbol{x}_j=\boldsymbol{x}_k} - \tfrac{1}{V})\mathrm{sym}\big((\boldsymbol{N}_j^\top - \tfrac{1}{V}\mathbb{1}_V\mathbb{1}_{L-1}^\top)\mathbb{1}_{L-1}\mathbb{1}_{L-1}^\top(\boldsymbol{N}_k^\top - \tfrac{1}{V}\mathbb{1}_V\mathbb{1}_{L-1}^\top)^\top\big)\Big)^2\Big|\boldsymbol{x}_k, \boldsymbol{N}_k\Big]$$

$$\preceq \frac{L}{V^2}\|(\boldsymbol{N}_k^\top - \tfrac{1}{V}\mathbb{1}_V\mathbb{1}_{L-1}^\top)\mathbb{1}_{L-1}\|_2^2\boldsymbol{I}_V + \frac{L}{V}(\boldsymbol{N}_k^\top - \tfrac{1}{V}\mathbb{1}_V\mathbb{1}_{L-1}^\top)\mathbb{1}_{L-1}\mathbb{1}_{L-1}^\top(\boldsymbol{N}_k^\top - \tfrac{1}{V}\mathbb{1}_V\mathbb{1}_{L-1}^\top)^\top.$$

Moreover,

$$\mathbb{E}\Big[\Big\|(\mathbb{1}_{\boldsymbol{x}_j=\boldsymbol{x}_k} - \tfrac{1}{V})\mathrm{sym}\big((\boldsymbol{N}_j^\top - \tfrac{1}{V}\mathbb{1}_V\mathbb{1}_{L-1}^\top)\mathbb{1}_{L-1}\mathbb{1}_{L-1}^\top(\boldsymbol{N}_k^\top - \tfrac{1}{V}\mathbb{1}_V\mathbb{1}_{L-1}^\top)^\top\big)\Big\|_2^p\Big|\boldsymbol{x}_k, \boldsymbol{N}_k\Big]^{\frac{1}{p}}$$

$$\leq 2\mathbb{E}\Big[\Big|(\mathbb{1}_{\boldsymbol{x}_j=\boldsymbol{x}_k} - \tfrac{1}{V})\Big|^p\Big|\boldsymbol{x}_k\Big]^{\frac{1}{p}}\|(\boldsymbol{N}_k^\top - \tfrac{1}{V}\mathbb{1}_V\mathbb{1}_{L-1}^\top)\mathbb{1}_{L-1}\|_2 \mathbb{E}\Big[\|(\boldsymbol{N}_j^\top - \tfrac{1}{V}\mathbb{1}_V\mathbb{1}_{L-1}^\top)\mathbb{1}_{L-1}\|_2^p\Big]^{\frac{1}{p}}$$

$$\leq C\sqrt{L}\|(\boldsymbol{N}_k^\top - \tfrac{1}{V}\mathbb{1}_V\mathbb{1}_{L-1}^\top)\mathbb{1}_{L-1}\|_2\Big(\sqrt{\frac{p}{V}} + p\Big)\Big(1 + \frac{p^2}{\sqrt{V}\wedge L}\Big).$$

- By Proposition 15, we have for $p = \log V$,

$$\mathbb{E}[\|\boldsymbol{Y}_k\|_2^p|\boldsymbol{x}_k, \boldsymbol{N}_k]^{\frac{1}{p}} \leq C\|(\boldsymbol{N}_k^\top - \tfrac{1}{V}\mathbb{1}_V\mathbb{1}_{L-1}^\top)\mathbb{1}_{L-1}\|_2\Big(\sqrt{p}\sqrt{\frac{Lk}{V}} + p^2\sqrt{L}k^{\frac{1}{p}}\Big),$$

which implies

$$\mathbb{E}[\|\boldsymbol{Y}_k\|_2^p]^{\frac{1}{p}} \leq CLp\Big(\sqrt{p}\sqrt{\frac{k}{V}} + p^2k^{\frac{1}{p}}\Big).$$

Therefore, for $p = \log V$, we have

$$\mathbb{E}[\|\boldsymbol{S}_{22}\|_2^p] \leq C\Big(\frac{\sqrt{p}}{NLV} + \frac{p^{5/2}}{LN\sqrt{NV}} + \frac{p^4}{LN^2}\Big)$$

Therefore, we have

$$\mathbb{P}\Big[\|\boldsymbol{S}_{22}\|_2 > C\frac{K^{3/2}\log^{3/2}V}{NLV}\Big] \leq \frac{1}{V^K}. \tag{F.9}$$

By (F.4), (F.5), (F.8), and (F.9), we have the second item. For the last item, we have

$$\boldsymbol{S}_3 - \frac{1}{N}\frac{1}{V^2L^2}\mathbb{1}_V\mathbb{1}_V^\top = \frac{1}{V^2L^2}\mathbb{1}_V\mathbb{1}_V^\top\Big(\Big\|\frac{1}{N}\sum_{j=1}^N(\boldsymbol{x}_j - \tfrac{1}{V}\mathbb{1}_V)\Big\|_2^2 - \frac{1}{N}\Big)$$

By Proposition 9,

$$\mathbb{P}\Big[\Big|\Big\|\frac{1}{N}\sum_{j=1}^N(\boldsymbol{x}_j - \tfrac{1}{V}\mathbb{1}_V)\Big\|_2^2 - \frac{1}{N}\Big| > \frac{CK^2\log^2 V}{N\sqrt{V}}\Big] \leq \frac{1}{V^K}.$$

The displayed equation implies the third item. $\qquad\square$

## G  MISCELLANEOUS

**Proposition 11.** *Let $A \in \mathbb{R}^{d \times d}$ and $B \in \mathbb{R}^{V \times V}$. Let $M := A \otimes B$. We have*
$$\|M\|_2 = \|A\|_2 \|B\|_2 \ \ and \ \ \|M\|_F = \|A\|_F \|B\|_F \ \ and \ \ \mathrm{tr}(M) = \mathrm{tr}(A)\mathrm{tr}(B).$$

*Proof.* The Frobenius norm and trace are straightforward. For the $\ell_2$ norm, let $A =: \sum_{i=1}^d \sigma_i u_i v_i^\top$ and $B =: \sum_{j=1}^V \tilde{\sigma}_j \tilde{u}_j \tilde{v}_j^\top$. We have
$$M = \sum_{i=1}^d \sum_{j=1}^V \sigma_i \tilde{\sigma}_j (u_i v_i^\top) \otimes (\tilde{u}_j \tilde{v}_j^\top) = \sum_{i=1}^d \sum_{j=1}^V \sigma_i \tilde{\sigma}_j (u_i \otimes \tilde{u}_j)(v_i \otimes \tilde{v}_j)^\top.$$
For any $(i, j) \neq (i', j')$, we have
$$(u_i \otimes \tilde{u}_j)^\top (u_{i'} \otimes \tilde{u}_{j'}) = (v_i \otimes \tilde{v}_j)^\top (v_{i'} \otimes \tilde{v}_{j'}) = 0.$$
Therefore,
$$\|M\|_2 = \max_{i,j} \sigma_i \tilde{\sigma}_j = \max_i \sigma_i \max_j \tilde{\sigma}_j.$$

$\square$

**Proposition 12.** *Let $z \sim \mathcal{N}(0, I_d)$ and $P_k : \mathbb{R}^d \to [0, \infty)$ denotes a degree $k$ polynomial which takes nonnegative values. For $p \geq 1$, we have*
$$\mathbb{E}[|P_k(z)|^p]^{\frac{1}{p}} \leq (8(p-1))^{\frac{k}{2}} \mathbb{E}[P_k(z)].$$

*Proof.* By hypercontractivity, it is sufficient to prove that $\frac{\mathbb{E}[|P_k(z)^2]^{\frac{1}{2}}}{\mathbb{E}[P_k(z)]} \leq 8^{\frac{k}{2}}$. We have
$$\mathbb{E}[|P_k(z)^2]^2 \leq \mathbb{E}[|P_k(z)] \mathbb{E}[|P_k(z)^3] \leq 2^{\frac{3k}{2}} \mathbb{E}[|P_k(z)] \mathbb{E}[|P_k(z)^2]^{\frac{3}{2}}$$
which proves the result.

$\square$

**Proposition 13.** *Let $k \in \mathbb{N}$ and $w \sim N(0, I_d)$. For $L > 0$ and $u, v \in S^{d-1}$, we have*
$$\mathbb{E}\left[H_{e_k}\left(\frac{1}{\sqrt{L}} w^\top u\right) H_{e_k}\left(\frac{1}{\sqrt{L}} w^\top v\right)\right] = \frac{k!}{L^k} \sum_{i=0}^{\lfloor k/2 \rfloor} \frac{(2i-1)!!}{2i!!} \binom{k}{2i} (L-1)^{2i} \langle u, v \rangle^{k-2i}$$

*Proof.* For $a \in \mathbb{R}$, we have
$$H_{e_k}(ax) = \sum_{i=0}^{\lfloor k/2 \rfloor} \frac{k!}{2^i i!(k-2i)!} (a^2-1)^i a^{k-2i} H_{e_{k-2i}}(x)$$
Therefore, for $a = 1/\sqrt{L}$, we have
$$\mathbb{E}\left[H_{e_k}\left(\frac{1}{\sqrt{L}} w^\top u\right) H_{e_k}\left(\frac{1}{\sqrt{L}} w^\top v\right)\right]$$
$$= \mathbb{E}\left[\left(\sum_{i=0}^{\lfloor k/2 \rfloor} \frac{k!}{2^i i!(k-2i)!} (a^2-1)^i a^{k-2i} H_{e_{k-2i}}(w^\top u)\right)\right.$$
$$\left. \times \left(\sum_{i=0}^{\lfloor k/2 \rfloor} \frac{k!}{2^i i!(k-2i)!} (a^2-1)^i a^{k-2i} H_{e_{k-2i}}(w^\top v)\right)\right]$$
$$= \sum_{i=0}^{\lfloor k/2 \rfloor} \left(\frac{k!}{2^i i!(k-2i)!}\right)^2 (a^2-1)^{2i} a^{2(k-2i)} (k-2i)! \langle u, v \rangle^{k-2i}$$
$$= \frac{k!}{L^k} \sum_{i=0}^{\lfloor k/2 \rfloor} \frac{(2i-1)!!}{2i!!} \binom{k}{2i} (L-1)^{2i} \langle u, v \rangle^{k-2i}.$$

$\square$

**Proposition 14.** *Let $Z$ be a random variable and $\boldsymbol{X}$ be a $d \times d$ symmetric matrix valued random matrix. We have*

$$-\mathbb{E}[Z^2]^{\frac{1}{2}}\mathbb{E}[\boldsymbol{X}^2]^{\frac{1}{2}} \preceq \mathbb{E}[Z\boldsymbol{X}] \preceq \mathbb{E}[Z^2]^{\frac{1}{2}}\mathbb{E}[\boldsymbol{X}^2]^{\frac{1}{2}}.$$

*Proof.* We observe that

$$\begin{bmatrix} Z\boldsymbol{I}_d \\ \boldsymbol{X} \end{bmatrix} [Z\boldsymbol{I}_d \quad \boldsymbol{X}] = \begin{bmatrix} Z^2\boldsymbol{I}_d & Z\boldsymbol{X} \\ Z\boldsymbol{X} & \boldsymbol{X}^2 \end{bmatrix} \succeq 0 \Rightarrow \begin{bmatrix} \mathbb{E}[Z^2]\boldsymbol{I}_d & \mathbb{E}[Z\boldsymbol{X}] \\ \mathbb{E}[Z\boldsymbol{X}] & \mathbb{E}[\boldsymbol{X}^2] \end{bmatrix} \succeq 0. \tag{G.1}$$

By (Ben Arous et al., 2025, Proposition 24), we know that (G.1) is equivalent to $\mathbb{E}[Z\boldsymbol{X}]^2 \preceq \mathbb{E}[Z^2]\mathbb{E}[\boldsymbol{X}^2]$. Since $\boldsymbol{X} \to \sqrt{\boldsymbol{X}}$ is monotone in matrix order, we have the result. $\qquad\square$

### G.1 ROSENTHAL-BURKHOLDER INEQUALITY AND COROLLARIES

We will rely on the following inequality:

**Proposition 15** ((Peng et al., 2025, Theorem 2.1))**.** *Let $\{\boldsymbol{M}_k\}_{k=1}^{N}$ be a d-dimensional symmetric matrix valued martingale adapted to the filtration $\{\mathcal{F}_k\}_{k=0}^{N}$. Let $\boldsymbol{Y}_k := \boldsymbol{M}_k - \boldsymbol{M}_{k-1}$ be its corresponding difference sequence and the quadratic variation is defined as*

$$\boldsymbol{Q}_N := \sum_{k=1}^{N} \mathbb{E}[\boldsymbol{Y}_k^2|\mathcal{F}_{k-1}].$$

*For any $p \geq 2$, suppose*

$$\mathbb{E}\left[\|\boldsymbol{Q}_N\|_2^{\frac{p}{2}}\right]^{\frac{1}{p}} < \infty \ \text{ and } \ \sup_{k\in[N]} \mathbb{E}\left[\|\boldsymbol{Y}_k\|_2^{p}\right]^{\frac{1}{p}} < \infty.$$

*Then it holds that*

$$\mathbb{E}\left[\|\boldsymbol{M}_N\|_2^{p}\right]^{\frac{1}{p}} \leq C\left(\sqrt{p \vee \log d}\ \mathbb{E}\left[\|\boldsymbol{Q}_N\|_2^{\frac{p}{2}}\right]^{\frac{1}{p}} + (p \vee \log d)N^{\frac{1}{p}} \sup_{k\in[N]} \mathbb{E}\left[\|\boldsymbol{Y}_k\|_2^{p}\right]^{\frac{1}{p}}\right).$$

We have the following corollaries:

**Corollary 3.** *The following statements holds for general $L, V > 0$:*

1. *For $X \sim \text{Binomial}(L, \frac{1}{V})$, we have*

$$\mathbb{E}[|X - kq|^p]^{\frac{1}{p}} \leq C\left(\sqrt{p}\sqrt{\frac{L}{V}} + p\left(\frac{L}{V}\right)^{\frac{1}{p}}\right).$$

2. *Let $\boldsymbol{c} = (c_1, \cdots, c_V) \sim \text{Multinomial}(L, \frac{1}{V}\mathbb{1}_V)$. For $p \geq 1$, we have*

$$\mathbb{E}[\|\boldsymbol{c}\|_p^p] \leq C^p V\left(\left(\frac{L}{V}\right)^p + \left(\frac{pL}{V}\right)^{\frac{p}{2}} + p^p\frac{L}{V}\right).$$

3. *By following the notation in the second item,*

   - *If $V \gg L$, we have for $L \geq e^{2e} + 1$,*

   $$\mathbb{P}\left[\|\boldsymbol{c}\|_\infty \geq \log L\right] \leq \left(\frac{2e}{\log L - 1}\right)^{\frac{\log L - 1}{2}}\left(\frac{L}{V}\right)^{\log L - 2}$$

   - *If $L \gg V$, we have*

   $$\mathbb{P}\left[\|\boldsymbol{c}\|_\infty \geq \frac{eL}{V}\right] \leq 2Ve^{-L/V}.$$

*Proof.* The first two items are direct consequence of Proposition 15. For the third item, using $\mathbb{1}_{c_w \geq k} \leq \frac{c_w(c_w-1)\cdots(c_w-k+1)}{k!}$ and linearity of expectation

$$\mathbb{P}[\|\boldsymbol{c}\|_\infty \geq k] \leq \sum_{w=1}^{V} \mathbb{P}[c_w \geq k] \leq \sum_{w=1}^{V} \frac{\mathbb{E}[c_w(c_w-1)\cdots(c_w-k+1)]}{k!}$$
$$= \frac{L(L-1)\cdots(L-k+1)}{k!V^{k-1}}.$$

For $V \gg L$, by choosing $k = \lfloor \log L \rfloor$, the result follows. For $L \gg V$, by choosing $k = \lfloor \frac{eL}{V} \rfloor$, the result follows. $\qquad\square$

