# OpenReview forum: "Learning to Recall with Transformers Beyond Orthogonal Embeddings"
_ICLR.cc/2026/Conference — ICLR 2026 Poster_

### Official Review · Reviewer_yVha · 2025-10-24

**Soundness:** 4
**Presentation:** 4
**Contribution:** 3
**Rating:** 6
**Confidence:** 3

**Summary:**

This work establishes a learnability statement on a specific factual recall task. Within a sequence, there is a token position which is marked, and the goal is to retrieve this token with the attention mechanism and then use the MLP to recall the correct label of the input, which is assigned randomly. This task is analysed in the first few steps of learning dynamics in the large limit of tokens, and under various (scaling) parameters such as embedding dimension, sample complexity and hidden layer size. The authors provide both a learnability criterion under these hyperparameters and a statistical lower bound based on the accessibility of attention queries that holds for more general algorithms (but is not very strong or specific).
The main takeaway is that denoising the correct item and memorisation are more difficult for longer sequences, where a larger embedding size and/or more samples are needed to reach a similar performance as for small sequences.

**Strengths:**

- The presentation and clarity of the writing make it easy to follow the work.
- The dynamical analysis of the first three steps already gives a good insight into the general behaviour and the type of solution that is learned to fulfil the recall task.
- The illustration of Thm1 in terms of comparing the signal with the noise at various network components is helpful and intuitive.
- The empirical findings validate the theoretical results closely, even when the theory does not hold anymore for longer time dynamics.
- This work offers plenty of interesting directions to follow up on for the community.

**Weaknesses:**

- The model is not very close to large-scale models, but it has some resemblance to factual recall tasks and comes with the advantage of being analyzable.
- see questions

**Questions:**

- Do you have any insights in your model on how strictly non-orthogonal vectors would do, and how close to non-orthogonality your embeddings actually are? Since they are random and not learned, I would expect that they are still "quite" orthogonal. What would change in your experiments if you could actually learn them too, and therefore impose better-than-random, learned, possibly non-orthogonal (or more orthogonal...) structure? Do you have an intuition for that? I think an empirical ablation like this would help contextualise the result a bit better.


Some suggestions/clarifications on intuition:
- In the description of the problem setting, you say "can then recall the correct label via an associative memory mechanism." (L.126). Later on, L.144, you say the goal is to "learn the target function (permutation)". I understand that the VxV matrix is essentially the random input class->output class mapping, but it took me a moment to see that this is where the memorisation comes in; you might want to make this more explicit.
- Do you have an intuition about how increasing the sequence length L and the vocabulary length makes the task harder? Longer lengths make detection harder, and larger V memorisation - but if intuitively the two functionalities are located in different models (L.125), why do they both scale linearly in all of the other parameters, and not independently?
- Do you observe that $W_{KQ}$ learns $z_{trig}$?
- L.219 "during the second gradient step," -> "after the first step" (?) (the object reference is a bit ambiguous)
- What do the different alpha (transparency) values of the regressed information mean? It is there in almost all figures.

---

> ### Author Response · Authors · 2025-11-22
> **Response to Reviewer yVha**
>
> We thank our reviewer for constructive evaluation of our work, and address the reviewer's concerns below:
>
> ---
> * **Theory vs. practice and motivation of our work:** As suggested of our reviewer, we aim to understand how the storage capacity of GD-trained transformers scales with model and sample size in an *analytically tractable* setting, which allows us to isolate the contribution of different hyperparameters—such as embedding dimension, network width, and activation function—to the storage scaling of a single transformer unit. To the best of our knowledge, no prior theoretical or empirical work provides a comparable characterization of storage capacity (quantitatively tracking the dependence on all hyperparameters) in transformers with non-orthogonal embeddings. At the same time, our setup is generic enough that the qualitative behavior predicted by our theory also appears in more realistic setups, including multi-step training with Adam (see the new experiments added in the revised manuscript).
> * **Random embeddings and learned embeddings:** We can show that the inner products of random embeddings satisfy $\max_{i \not = j}  \lvert \langle z_i, z_j \rangle \rvert = \tilde O(1)/\sqrt{d} $ with high probability. As the reviewer points out, the embeddings become increasingly orthogonal as $d$ grows; however, the signal relevant to the hidden permutation also vanishes asymptotically. Therefore, it is crucial how fast $d$ grows relative to the other hyperparameters $(N, V, L, m)$, which is precisely what we demonstrate in our three-step gradient descent analysis.
>     For intuition about learned embeddings, since the tokens follow a uniform distribution, the pairwise inner products should be uniformly small, a property that random embeddings asymptotically satisfy. Therefore, learned embeddings are also expected to have similarly small pairwise inner products. A potential benefit of using learned embeddings is that we require much less than $\sqrt{V}$ (in fact $\Theta(\log V)$) to represent the permutation in the output layer, thereby increasing capacity. On the other hand, for the first steps of gradient descent, our lower bound should still apply, suggesting that the task of distinguishing informative tokens becomes statistically harder for smaller embedding dimension.  It is an interesting open question whether these two effects always trade off, and whether there is room for improvement by training the embeddings. We are currently running experiments to assess this trade-off.
> * **Scaling with sequence length $L$ and vocabulary size $V$:** Although the attention and MLP components serve distinct roles, they are learned jointly through backpropagation. As a result, noise in the MLP weights affects the signal-to-noise ratio of the gradient with respect to the attention parameters, which in turn appears as a multiplicative interaction between the corresponding hyperparameters in Theorem 1. While we analyze an asymptotic regime where both $V$ and $L$ grow, their relationship is not restricted to be proportional (or linear): we allow a polynomial relation  $V = L^c$  with  $c \in (0,1)$. Our results can be extended to the finite-$L$ case (i.e.,  $c = 0$), at the cost of additional technical derivations. We assume $c < 1$ for two reasons: (i) in many practical large-language-model pretraining setups, the context length is smaller than the vocabulary size, and (ii) the condition $L \ll V$  simplifies several terms in the proofs. We have clarified these points in the revised manuscript.
> * **Does $W_{KQ}$ learn $z_{\text{trig}}$:**  In our setting, we can show that $W_{\mathrm{KQ}} z_{\mathrm{EOS}}$ aligns with a vector $z_{\mathrm{trig}}$, up to an additive bias term that arises from the nonzero empirical mean of the embeddings. This is made precise in Section 5 of the updated manuscript, where we overview the proof. In the orthogonal case, we show that the attention pre-softmax scores in Eq. (7) aligns with the trigger vector up to a vector of ones (as seen in the limits of the Informative and Non-informative terms in Eqs. (8)–(9)). In the non-orthogonal case, the same phenomenon holds: the score aligns with a vector $z_{\mathrm{trig}}$, again up to an additive bias term.
> * **"during the second gradient step" $\to$ "after the first step":** Thank you for the close reading; we have updated the wording.
> * **Different alpha (transparency) values in figures:** In the heatmaps, we identify the smallest embedding dimension that achieves accuracies in $\{0.10, 0.125, 0.15\}$ and plot these as data points in Fig 1, Figs 2-5(c). The different alpha values arise from overlapping points: for example, if the accuracy jumps from below $ 0.10$ to above $0.15$, multiple accuracy thresholds select the same  (param size, vocab size)  pair, leading to repeated points that appear less transparent. We clarified this point in the revised manuscript.
>
> ---
> We would be happy to answer any follow-up questions in the discussion period.

---

### Official Review · Reviewer_nvK4 · 2025-10-31

**Soundness:** 3
**Presentation:** 3
**Contribution:** 2
**Rating:** 6
**Confidence:** 3

**Summary:**

This paper investigates the gradient-based learning of the Transformer architecture for identifying informative tokens within the context. The theoretical result provides conditions on hyperparameters for successful learning, and the empirical result validates this.

**Strengths:**

1. This paper addresses an important problem of the learning dynamics for Transformer models.

2. The paper complements its theoretical predictions through experiments.

**Weaknesses:**

1. The results in the paper are limited to a simple toy model (easy task, single layer, 3 simplified gradient steps on a fixed batch). Although additional experiments exist beyond this toy setting, it remains unclear how the results will generalize to more realistic settings.

2. The setting assumes that the input contains a trigger embedding that marks the informative token, which makes this toy setting less practical. It is difficult to imagine a realistic task where this assumption holds.

**Questions:**

1. What are the useful implications of the theoretical results? Specifically, what is the takeaway for practitioners who train Transformer language models? What is the main contribution to the deep learning theory or language model theory that is not available from prior works?

2. Does adding an MLP layer on top of the attention layer hurt? It seems that this makes the theoretical condition for successful learning more difficult to satisfy.

---

> ### Author Response · Authors · 2025-11-22
> **Response to Reviewer nvK4**
>
> We thank our reviewer for constructive evaluation of our work, and address the reviewer's concerns below:
>
> ---
> * **Motivation for our setting and scope of our work:** In this work, our goal is to understand how the storage capacity of transformers scales with model and sample size. Due to the non-convex landscape and highly nontrivial optimization dynamics, we employ theoretical simplifications pioneered in (Bietti et al. 2023, Oymak et al, 2023) to introduce an *analytically tractable* model, which has two main advantages:
>   * First, analytical tractability allows us to isolate the contribution of each hyperparameter—such as sample size, embedding dimension, network width, and activation function—to storage. To the best of our knowledge, no prior work provides a comparable characterization of how different hyperparameters affect storage capacity in transformers.
>   * Second, our setting considers a natural transformer architecture in which the attention block must distinguish informative tokens from non-informative ones by aligning with a low-rank structure spanned by the embeddings of the informative tokens, as observed in many realistic setups~(Elhage et al. 2023). In particular, our theoretical characterization quantifies storage in terms of generic quantities such as the signal-to-noise ratio and the correlation (induced by non-orthogonality) between this low-rank structure and the remaining embeddings, and describes how these interactions evolve over a few gradient step.
>
>   All these mechanisms are generic enough to apply to transformer blocks trained with gradient-based methods. Consequently, we can verify that the qualitative behavior predicted by our theory also appears in more practical setups, including multi-step training with Adam.  (please see the new experiments added in Section 4.3).
>
> * **Motivation for the trigger model and its relevance to practice:**
> The “trigger’’ model we study is a simplified version of how attention often  operates in practice, where the key-query (QK) part of attention identifies certain relevant directions to separate informative tokens from non-informative ones, while the output-value (OV) matrix copies or modifies a different feature. Such mechanisms appear in several well-studied circuits, such as induction heads (Bietti et al., 2023; Elhage et al., 2023), where the QK matrix detects one or more trigger directions and the OV matrix copies the relevant embedding, and the “opposite head’’ circuit identified in a recent Anthropic blog post  in [this blog post by Anthropic](https://transformer-circuits.pub/2025/attention-qk/index.html), where the QK matrix isolates an “opposite’’ feature and the OV part maps a word to its opposite (e.g."small" to "large").
>
>   Our trigger model corresponds to the special case where this relevant direction is a fixed vector, a theoretical model that has been used in several works for its tractability (Oymak et al., 2023; Nichani et al., 2024; Marion et al., 2025).
>
> * **Contributions and implications of our work:** Our work goes beyond earlier papers that study factual recall mechanisms by allowing both finite sample size and non-orthogonal embeddings. This enables us to capture the difficulty of learning in terms of the overlap between embeddings and, in turn, to characterize how each hyperparameter
> $(V, N,d,L,m)$ contributes to storage scaling, making the trade-offs between them explicit in a natural theoretical setup.
>   * From a *technical* perspective, establishing these results required lengthy and involved derivations. We hope this analysis can serve as a starting point for future work that moves even closer to fully realistic settings.
>   * From a *practical* perspective, our analysis disentangles the different contributors to the scaling properties of a transformer block, which can help practitioners better interpret empirical scaling behaviour and initiate a line of work that enables more principled hyperparameter choices when designing or scaling transformer models.
>
> *  **Pros/Cons of the MLP layer:** The benefit of including an MLP layer depends on the noise level in the sequence, which is controlled by the sequence length $L$, since each input sequence contains a single informative token. In the small-$L$ regime, adding an MLP layer allows us to reduce the embedding dimension while still achieving perfect storage (this is consistent with [Nichani et al. 2024]). In contrast, in the high-noise regime (large $L$), if the embedding dimension is too small, the overlap between embeddings of informative and non-informative tokens is too large and makes it difficult to learn. In this sense, our analysis highlights a trade-off: the nonlinearity in the MLP layer is beneficial when sequences are relatively short (low noise), whereas for long, noisy sequences, larger embeddings are necessary regardless of the presence of an MLP layer.
>
> ---
> We would be happy to answer any follow-up questions in the discussion period.

---

> > ### Comment · Reviewer_nvK4 · 2025-11-25
> >
> > I acknowledge that the rebuttal addresses all of my concerns. I updated my score accordingly.

---

### Official Review · Reviewer_BKC3 · 2025-11-04

**Soundness:** 3
**Presentation:** 4
**Contribution:** 3
**Rating:** 8
**Confidence:** 3

**Summary:**

This work presents a theoretical analysis of the training dynamics of transformers on a factual recall task. Specifically, it considers two model architectures: (1) a single attention-only layer and (2) one attention layer followed by an MLP layer. The study examines how these transformers learn to map an output token to an informative position within a sequence. The authors quantify model accuracy after three steps of gradient descent and establish learnability guarantees as functions of vocabulary size, sequence length, model size, and the number of training samples. From an empirical perspective, the results provide practical guidance on selecting the hidden dimension and sample size, while also elucidating the role of the MLP component in recall tasks.

**Strengths:**

+ The paper is clearly written, with precise notations and formal language, and includes detailed technical assumptions.

+ The theoretical results appear sound and solid. To the best of my knowledge, this work provides the first theoretical analysis of training transformers to perform recall tasks, particularly in the regime where the embedding dimension can be smaller than the vocabulary size.

+ The theoretical findings are further validated through numerical experiments conducted in controlled settings, which support the correctness of the theory and demonstrate its potential generalization to more practical scenarios.

**Weaknesses:**

- Some assumptions may require further justification. For instance, in Assumption 1, it is not entirely clear why it is necessary to set $L = V^c, c \in (0, 1)$? Under this assumption, the sequence length is necessarily smaller than the vocabulary size, and the motivation for this specific scaling is not well explained.

- It is also suggested that the authors present their results in a more comparative manner and provide readers with additional background context. My understanding is that the main contribution of this work lies in analyzing the learning dynamics of transformers on factual recall tasks in the regime where $d < V$. However, it is somewhat difficult to determine whether the theoretical settings in this paper are consistent with prior works. For example, regarding the choice of model architectures, learning tasks, training algorithms, assumptions, and the final results. Without such contextualization, it becomes challenging for the general audience to assess the true contribution of the paper and identify whether the new result stems from new assumptions or from novel proof techniques.

**Questions:**

It appears that the considered factual recall task can be learned exactly after just three steps of gradient descent, provided that the embedding dimension or the sample size is sufficiently large. Could the authors offer some intuition for why this occurs? For example, even in the orthogonal regime, $d > V$, why can the model achieve perfect recall accuracy with only three-step GD?

---

> ### Author Response · Authors · 2025-11-22
> **Response to Reviewer BKC3**
>
> We thank the reviewer for the positive evaluation and constructive feedback. We address the reviewer’s concerns point by point below.
>
> ---
> *  **Vocabulary size ($V$) and sequence length ($L$):** We consider the regime $V = L^c$ with $c \in (0,1)$ to study an asymptotic setting where both $L$ and $V$ can be large. Our results can be extended to the finite-$L$ case (i.e., $c = 0$), at the cost of some additional technical work. We assume $c < 1$ for two reasons: (i) in many practical large-language-model pretraining setups, the context length is smaller than the vocabulary size, and (ii) the condition $L \ll V$ simplifies several terms in the proofs. We have clarified these points in the revised manuscript.
> * **Comparing with prior work and additional background:**
>     We are not aware of any prior work that studies exactly the same statistical model as in our paper; in particular, one of our main contributions is to study optimization under non-orthogonal embeddings, which is largely absent in prior theoretical works on recall taks. As a result, it is not easy to make a direct quantitative comparison; for a brief quantitative discussion of the closest works, please see the **Quantitative summary of prior works**  part in our response to Reviewer 1 (Reviewer iRpb).
>   * On the other hand, our model and problem setting are very much in line with prior works  (Oymak et al., 2023; Nichani et al. 2024; Marion et al., 2025): we study learning a statistical relationship between informative tokens and labels, while simultaneously learning to ignore non-informative tokens. We assume that informative tokens carry a low-dimensional “marking’’ that represents context, a structure that has been used in several theoretical works to model how attention aligns with relevant embeddings in practice~ (Elhage et al. 2023; Oymak et al., 2023; Marion et al., 2025). Finally, we focus on three gradient descent steps as in (Oymak et al., 2023), which is similar in spirit to prior work that analyzes few-step training as a lens on the early phase of feature learning (e.g., Bietti et al., 2023; Wang et al., 2025). Compared to these works, our contribution is to allow non-orthogonal embeddings and to characterize storage capacity in terms of the overlap between informative and non-informative embeddings, in addition to its dependence on sample size and sequence length.
>
>   We will emphasize these points further in the revised manuscript.
> * **Intuition for how three step GD learns:**  Following our reviewers' suggestion, in the revised manuscript, we have added an intuition section in the main text (see Section 5), where we explain the intuition for how three gradient steps learn the underlying model in the orthogonal-embedding setting. We also refer the reader to the proof-overview section in the appendix for a proof sketch of the non-orthogonal embedding case, where we highlight the key arguments.
>
> ---
> We would be happy to answer any follow-up questions in the discussion period.

---

### Official Review · Reviewer_iRpb · 2025-11-08

**Soundness:** 4
**Presentation:** 4
**Contribution:** 4
**Rating:** 6
**Confidence:** 4

**Summary:**

This paper presents a theoretical framework for understanding how transformers acquire factual recall when trained on finite data with non-orthogonal embeddings. Focusing on single-layer attention and attention-plus-MLP architectures, the authors derive precise conditions under which gradient descent can identify and retrieve informative tokens within a sequence. They show that learning efficiency follows a multiplicative relationship among vocabulary size, sequence length, embedding dimension, MLP width, and sample size. Unlike prior analyses, this work establishes learnability even when the vocabulary size is larger than the embedding dimension. Empirical results corroborate the theoretical scaling laws and demonstrate relevance to practical settings.

**Strengths:**

This work provides a clear problem formulation and rigorous proofs supporting its theoretical claims. The technical results are overall a good contribution . Beyond theory, it offers numerical experiments that closely match the analytical predictions, showing the precision and tightness of the derived results. The findings also yield practical insights into transformer training, key architectural parameter choice, the importance of the MLP component.

**Weaknesses:**

- The clarity of some parts needs improvement (see Questions). It is recommended to include a brief intuition or proof sketch to better convey the key insights underlying the proof. The current presentation may be less accessible to a broader ML audience.
-  The gap between theory and practice remains substantial. The definition of the factual recall task requires further clarification. In the current setup, the model identifies a marked token within a sequence and maps it to another token through a ground-truth permutation $\Pi^*$. However, this formulation appears disconnected from practical scenarios, particularly regarding how the key information is represented or “marked” in the token features. It also seems misaligned with the setting described in the cited work [1].
- As for the theoretical aspects, it is recommended to include a quantitative summary of prior works. For example, specifying what learnability results/bounds have been established in those studies that analyze orthogonal embeddings.

[1] Nichani et al., Understanding Factual Recall in Transformers via Associative Memories

**Questions:**

- The clarity of some assumptions needs to be improved. For example, what do $\eta \approx 0$ and $\gamma \gg 0$ specifically mean? Why are these conditions not expressed using big-O notation?
- Similarly, it may be non-standard to write $\Omega(V \log V) \le N$ and $V \ge \Omega(1)$. When using big-O or related asymptotic notations, should readers assume that these are taken with respect to $V$?
- In addition, how did the authors name or interpret each term in Eq. (6)? It is recommended to provide readers with more intuition or explanation for these terms to enhance understanding.

---

> ### Author Response · Authors · 2025-11-22
> **Response to Reviewer iRpb (1/2)**
>
> We thank the reviewer for the positive evaluation and constructive feedback. We address the reviewer’s concerns point by point below.
>
> ---
> * **Proof overview:** Thank you for the suggestion. The current manuscript had already included a proof overview in Appendix C.1, right before the detailed proofs. In the revised manuscript, we have also added an intuition section in the main text (see Section 5), where we explain the intuition for how three gradient steps learn the underlying model in the orthogonal-embedding setting, and we refer the reader to the proof-overview section in the appendix for a proof sketch of the non-orthogonal embedding case, where we highlight the key arguments.
> * **Motivation for our setting and scope of our work:** In this work, our goal is to understand how the storage capacity of transformers scales with model and sample size. Due to the non-convex landscape and highly nontrivial optimization dynamics, we employ theoretical simplifications pioneered in (Bietti et al. 2023, Oymak et al, 2023) to introduce an *analytically tractable* model, which has two main advantages:
>   * First, analytical tractability allows us to isolate the contribution of each hyperparameter—such as sample size, embedding dimension, network width, and activation function—to storage. To the best of our knowledge, no prior work provides a comparable characterization of how different hyperparameters affect storage capacity in transformers.
>   * Second, our setting considers a natural transformer architecture in which the attention block must distinguish informative tokens from non-informative ones by aligning with a low-rank structure spanned by the embeddings of the informative tokens, as observed in many realistic setups~(Elhage et al. 2023). In particular, our theoretical characterization quantifies storage in terms of generic quantities such as the signal-to-noise ratio and the correlation (induced by non-orthogonality) between this low-rank structure and the remaining embeddings, and describes how these interactions evolve over a few gradient step.
>
>   All these mechanisms are generic enough to apply to transformer blocks trained with gradient-based methods. Consequently, we can verify that the qualitative behavior predicted by our theory also appears in more practical setups, including multi-step training with Adam.  (please see the new experiments added in Section 4.3).
>
> *  **Motivation for the trigger model and its relation to Nichani et al.:** The “trigger’’ model we study is a simplified version of how attention often  operates in practice, where the key-query (QK) part of attention identifies certain relevant directions to separate informative tokens from non-informative ones, while the output-value (OV) matrix copies or modifies a different feature. Such mechanisms appear in several well-studied circuits, such as induction heads (Bietti et al., 2023; Elhage et al., 2023), where the QK matrix detects one or more trigger directions and the OV matrix copies the relevant embedding, and the “opposite head’’ circuit identified in a recent Anthropic blog post  in [this blog post by Anthropic](https://transformer-circuits.pub/2025/attention-qk/index.html), where the QK matrix isolates an “opposite’’ feature (which acts as the trigger direction) and the OV part maps a word to its opposite (e.g. "small" to "large"), similar to our target permutation.
>
>     In Nichani et al., informative and non-informative tokens are assumed to have disjoint support, and the attention mechanism aligns with a one-dimensional vector given by the mean of the informative embeddings, which then allows the QK circuit to identify any informative token. This avoids the need for an explicit trigger embedding, but requires a much larger embedding dimension to ensure that all informative tokens can be attended to, unless multiple heads are used. We chose to use a separate trigger to avoid the need for multiple heads, which would make the analysis significantly harder.

---

> ### Author Response · Authors · 2025-11-23
> **Response to Reviewer iRpb (2/2)**
>
> * **Quantitative summary of prior works:** We are not aware of any prior work that studies exactly the same statistical model as in our paper. The closest works are Oymak et al.  (2023) and Nichani et al. (2024), which both consider sequence models where informative and non-informative tokens coexist.
>   * Oymak et al.  study learning a linearly separable model with a constant context vector (similar to our marking vector) and analyze three steps of gradient descent. Their relevant result in Section 4.3 is roughly  $N \geq O(V / L^4)$, which has a similar flavor to our scaling, but their setting does not address learning under non-orthogonal embeddings.
>   * On the other hand, the factual-recall setup in Section 4 of Nichani et al. is closer to ours: they consider one-to-one recall with non-orthogonal embeddings, but only at the level of the population minimizer, and show an $O(V)$ parameter requirement to learn the model. In the low-noise (small-$L$) regime, our parameter requirement is consistent with this lower bound, while also achieving minimax sample complexity of learning a permutation $O(V \log V)$.
>
>    We will update the related-work discussion in the revised manuscript to make these quantitative comparisons explicit.
>
> *  **Asymptotics in $V$:** We state our results asymptotically   $V \to \infty$, and the big-Oh and big-Omega notations are taken with respect to $V$. However, by Assumption 1 all hyperparameters are polynomially related, so the same statements can be equivalently rephrased as asymptotics in other parameters (e.g., width or sequence length).
> * **Assumptions on learning rates $(\eta, \gamma)$:** Our assumptions on the learning rates mean that there exist   $\eta$ and $\gamma$ that vanish/diverge at suitable rates as $V \to \infty$ such that our guarantees hold. We know that such regime of learning rate has also appeared in many prior works (e.g., Oymak et al. (2023)). We clarified this point in the revised manuscript.
> *  **Interpretation of the terms in Eq (6):** The current manuscript contains a short summary of where each term in Eq. (6) arises right after Theorem 1, and more intuition is deferred to the proof-overview section in the Appendix C.1.  We clarified this point in the revised manuscript.
> ---
> We would be happy to answer any follow-up questions in the discussion period.

---

### Author Response · Authors · 2025-11-22
**General response**

We sincerely thank all the reviewers for their time and effort in evaluating our work. We appreciate the constructive feedback and have revised the manuscript accordingly. The main changes in the revised version are summarized below.

* We added new experiments in Section 4.3, where we train our Attention-only model with additional layer normalization and Adam using mini-batch gradient descent. The resulting performance qualitatively matches the behavior predicted by our theory.
* We added a new proof-sketch section in Section 5 to build intuition for how gradient descent learns the underlying mechanism by studying population dynamics with orthogonal embeddings.
* We addressed minor wording issues and expanded the related-work section as requested by the reviewers.

We hope that these additions address the questions raised by the reviewers, and we believe that the revisions have improved the clarity and quality of the submission.

---

### Author Response · Authors · 2025-12-04
**Summary of the Work and Rebuttal Discussion**

Dear (New) AC,

Thank you for taking over our submission. For your convenience, we briefly summarize our work and the main points addressed during the rebuttal period.

### **Summary of the Work**

In our work, we study the information storage capacity of a two-layer transformer with non-orthogonal embeddings trained with finite dataset and gradient descent. Our work is motivated both by practice and by earlier work showing that non-orthogonal embeddings are necessary to achieve optimal capacity (Nichani et al., 2024). To our knowledge, this is the first theoretical characterization of storage capacity of transformers with non-orthogonal embeddings when trained with gradient-based methods. The analysis requires substantial technical work to control overlap between embeddings in the higher-order polynomial terms that arise in the transformer gradients.

To analyze storage capacity, we consider a simplified setting where the transformer must locate an informative token in each sequence and learn the one-to-one mapping between  informative tokens and labels. Our main contribution is to characterize when the transformer can learn this mapping in terms of the sample size $N$, sequence length $L$, embedding dimension $d$, and network width $m$ (see Theorem 1). The result is proven by balancing the noise introduced by overlap between non-orthogonal embeddings, finite samples, and non-informative tokens, and it reveals how each of   $(N,d,L,m)$ contribute to the storage capacity of the transformer. To our knowledge, no prior work provides a comparable description of these trade-offs. Finally, we verify our theoretical findings with experiments.

### **Summary of the Rebuttal Discussion**

During the rebuttal phase, the reviewers mainly raised the following points, which we addressed as follows:
* **Motivation for our setting and scope of our work.**  We clarified the following:
  * We adopt theoretical simplifications, following Bietti et al. (2023) and Oymak et al. (2023), to obtain an analytically tractable model that allows us to characterize the contribution of hyperparameters to storage capacity, while preserving a natural attention mechanism.
  * The condition in Theorem 1 depends on quantities such as the correlation between embedding vectors (induced by non-orthogonality) and the strength of the low-rank structure in the informative tokens, which are generic enough to serve as a theoretical guide for studying transformers in realistic models.
  * Finally, to further support this point, we added new experiments in Section 4.3, where we train our attention-only model with additional layer normalization and Adam, using mini-batch gradient descent. The resulting performance qualitatively matches the behavior predicted by our theory.
* **Motivation for the trigger model and its relevance to practice.** We emphasized that the trigger model we study is a simplified version of how attention often operates in practice, where the key–query part identifies certain relevant directions to separate informative tokens from non-informative ones, while the output–value matrix copies or modifies a different feature. To provide examples of circuits with such a mechanism, we highlighted the “induction head’’ circuits as well as “opposite head’’ circuits found in the mechanistic interpretability literature (see, e.g., [this Anthropic post](https://transformer-circuits.pub/2025/attention-qk/index.html)), where a single direction is used in key–query attention and the value matrix copies a token to the following position or converts a word to its antonym.
* **On the learning mechanism and intuition for gradient descent.**  To clarify how gradient descent learns the model, we added a new proof-sketch section in Section 5. In this section, we study our setting with orthogonal embeddings and explain the key mechanisms behind the learning dynamics.
* **On related work and presentation.** We expanded the related-work section as requested by the reviewers and addressed several minor wording and clarity issues throughout the manuscript.


We would also like to note that, based on our rebuttal, Reviewer nvK4 increased their score from 6 to 8. Because the discussion phase ended early, we did not receive responses from  other reviewers. Finally, we uploaded a revised manuscript in which all updated parts are highlighted in red for ease of reference.

Thank you very much for your time and consideration.

---

### Meta-Review · Area_Chair_vyS5 · 2025-12-27

**Summary:**

The reviewers were generally quite positive about the technical contributions of this work. They raised some questions on the technical details of the theoretical analysis, and the authors have addressed them properly during the rebuttal.

**Reviewer Concerns:**

The reviewers had concerns regarding the clarity of the technical analysis, such as the simplified model and some assumptions, as well as the technical contributions compared with existing works. The authors have clarified those questions satisfactorily.

Another concern is the gap between the theory and practice, which is quite common in this area. Nevertheless, this work takes an initial attempt to understand the information storage capacity of transformers with non-orthogonal embeddings when trained with gradient-based methods, which is meaningful and technically non-trivial..

**Reviewer Scores:**

The reviewers were all positive (6/8/6/6) at the initial round, with a few questions regarding the clarity of the technical details. The authors have addressed those questions properly during the rebuttal, and I believe that the reviewers would remain their positive ratings. One reviewer indicated that he/she has updated the rating during the rebuttal, which most likely would bump the rating from 6 to 8.

---

### Decision · Program_Chairs · 2026-01-26

Accept (Poster)